# Understanding Multi-phase Optimization Dynamics and Rich Nonlinear Behaviors of ReLU Networks

**Mingze Wang**
School of Mathematical Sciences
Peking University
Beijing, 100081, P.R. China
mingzewang@stu.pku.edu.cn

**Chao Ma**
Department of Mathematics
Stanford University
Stanford, CA 94305
chaoma@stanford.edu

## Abstract

The training process of ReLU neural networks often exhibits complicated nonlinear phenomena. The nonlinearity of models and non-convexity of loss pose significant challenges for theoretical analysis. Therefore, most previous theoretical works on the optimization dynamics of neural networks focus either on local analysis (like the end of training) or approximate linear models (like Neural Tangent Kernel). In this work, we conduct a complete theoretical characterization of the training process of a two-layer ReLU network trained by Gradient Flow on a linearly separable data. In this specific setting, our analysis captures the whole optimization process starting from random initialization to final convergence. Despite the relatively simple model and data that we studied, we reveal four different phases from the whole training process showing a general simplifying-to-complicating learning trend. Specific nonlinear behaviors can also be precisely identified and captured theoretically, such as initial condensation, saddle-to-plateau dynamics, plateau escape, changes of activation patterns, learning with increasing complexity, etc.

## 1 Introduction

Deep learning shows its remarkable capabilities across various fields of applications. However, the theoretical understanding of its great success still has a long way to go. Among all theoretical topics, one of the most crucial aspect is the understanding of the optimization dynamics of deep neural network (NN), particularly the dynamics produced by Gradient Descent (GD) and its variants. This topic is highly challenging due to the highly non-convex loss landscape and existing works usually work with settings that do not align well with realistic practices. For instance, the extensived studied Neural Tangent Kernel (NTK) theory (Jacot et al., 2018; Du et al., 2018; 2019; Zou et al., 2018; Allen-Zhu et al., 2019) proves the global convergence of Stochastic gradient descent (SGD) to zero training error for highly over-parameterized neural networks; however, the optimization behaviors are similar to kernel methods and do not exhibit nonlinear behaviors, because neurons remain close to their initialization throughout training.

In reality, however, the training of practical networks can exhibit plenty of nonlinear behaviors (Chizat and Bach, 2018; Mei et al., 2019; Woodworth et al., 2020). In the initial stage of the training, a prevalent nonlinear phenomenon induced by small initialization is *initial condensation* (Maennel et al., 2018; Luo et al., 2021), where neurons condense onto a few isolated orientations. At the end of training, NNs trained by GD can *directionally converge* to the KKT points of some constrained max-margin problem (Nacson et al., 2019; Lyu and Li, 2019; Ji and Telgarsky, 2020). However, KKT points are not generally unique, and determining which direction GD converges to can be challenging. Nonlinear training behaviors besides initial and terminating stages of optimization are also numerous. For example, for square loss, Jacot et al. (2021) investigates the *saddle-to-saddle* dynamics where GD traverses a sequence of saddles during training, but it is unclear whether similar behavior can

occur for classification tasks using exp-tailed loss. Moreover, while in lazy regime, most activation patterns do not change during training ReLU networks, it remains uncertain when and how *activation patterns evolve* beyond lazy regime. Additionally, while it is generally conjectured that GD *learns functions of increasing complexity* (Nakkiran et al., 2019), this perspective has yet to be proven.

As reviewed in Section 2, works have been done to analyze and explain the nonlinear training behaviors listed above. However, due to the complexity of the training dynamics, most existing works only focus on one phenomenon and conduct analysis on a certain stage of the training process. Few attempts have been done to derive a full characterization of the whole training dynamics from the initialization to convergence, and the settings adopted by these works are usually too simple to capture many nonlinear behaviors (Phuong and Lampert, 2021; Lyu et al., 2021; Wang and Ma, 2022; Boursier et al., 2022).

**In this work**, we make an attempt to theoretically describe the whole neural network training dynamics beyond the linear regime, in a setting that many nonlinear behaviors manifest. Specifically, We analyze the training process of a two-layer ReLU network trained by Gradient Flow (GF) on a linearly separable data. In this setting, our analysis captures the whole optimization process starting from random initialization to final convergence. Despite the relatively simple model and data that we studied, we reveal multiple phases in training process, and show a general simplifying-to-complicating learning trend by detailed analysis of each phase. Specifically, by our meticulous theoretical analysis of the whole training process, we precisely identify **four different phases** that exhibit **numerous nonlinear behaviors**. In Phase I, *initial condensation and simplification* occur as living neurons rapidly condense in two different directions. Meanwhile, GF *escapes from the saddle* around initialization. In Phase II, GF *gets stuck into the plateau* of training accuracy for a long time, then *escapes*. In Phase III, a significant number of neurons are *deactivated*, leading to *self-simplification* of the network, then GF tries to learn using the almost simplest network. In Phase IV, a considerable number of neurons are *reactivated*, causing *self-complication* of the network. Finally, GF *converges towards an initialization-dependent direction*, and this direction is *not even a local max margin direction*. Overall, the whole training process exhibits a remarkable *simplifying-to-complicating* behavior.

## 2 Other Related Works

Initial condensation phenomenon are studied in (Maennel et al., 2018; Luo et al., 2021; Zhou et al., 2022a;b; Abbe et al., 2022a;b; Chen et al., 2023). Theoretically, Lyu et al. (2021); Boursier et al. (2022) analyze the condensation directions under their settings, which are some types of data average. Additionally, Atanasov et al. (2022) demonstrates that NNs in the rich feature learning regime learn a kernel machine due to the silent alignment phenomenon, similar to the initial condensation.

The end of training is extensively studied for classification tasks. Specifically, for classification with exponentially-tailed loss functions, if all the training data can be classified correctly, NNs trained by GD converge to the KKT directions of some constrained max-margin problem (Nacson et al., 2019; Lyu and Li, 2019; Chizat and Bach, 2020; Ji and Telgarsky, 2020; Kunin et al., 2023). In (Phuong and Lampert, 2021; Lyu et al., 2021), they analyze entire training dynamics and derive specific convergent directions that only depend on the data. Furthermore, another famous phenomenon in the end of training is the neural collapse (Papyan et al., 2020; Fang et al., 2021; Zhu et al., 2021; Han et al., 2021), which says the features represented by over-parameterized neural networks for data in a same class will collapse to one point, and such points for all classes converge to a simplex equiangular tight frame.

Saddle-to-saddle dynamics are explored for square loss in (Jacot et al., 2021; Zhang et al., 2022; Boursier et al., 2022; Pesme and Flammarion, 2023; Abbe et al., 2023). Furthermore, learning of increasing complexity, also called simplifying-to-complicating or frequency-principle, is investigated in (Arpit et al., 2017; Nakkiran et al., 2019; Xu et al., 2019; Rahaman et al., 2019).

Beyond lazy regime and local analysis, Phuong and Lampert (2021); Lyu et al. (2021); Wang and Ma (2022); Boursier et al. (2022) also characterize the whole training dynamics and exhibit a few of nonlinear behaviors. Specifically, Lyu et al. (2021) studies the training dynamics of GF on Leaky ReLU networks, which differ from ReLU networks because Leaky ReLU is always activated on any data. In (Safran et al., 2022), they studies the dynamics of GF on one dimensional dataset, and characterizes the effective number of linear regions. In (Brutzkus et al., 2017), they studies the

dynamics of SGD on Leaky ReLU networks and linearly separable dataset. Moreover, Boursier et al. (2022) characterizes the dynamics on orthogonally data for square loss. The studies closest to our work are Phuong and Lampert (2021); Wang and Ma (2022), exploring the complete dynamics on classifying orthogonally separable data. However, this data is easy to learn, and all the features can be learned rapidly (accuracy=$100\%$) in initial training, followed by lazy training (activation patterns do not change). Unfortunately, this simplicity does not hold true for actual tasks on much more complex data, and NNs can only learn some features in initial training, which complicates the overall learning process. Furthermore, we provide a detailed comparison between our results and these works in Section 5. Another related work (Saxe et al., 2022) introduces a novel bias of learning dynamics: toward shared representations. This idea and the view of gating networks are enlightening for extending our two-layer theory to deep ReLU neural networks.

Our work also investigates the max-margin implicit bias of ReLU neural networks, and related works have been listed above. Although in homogenized neural networks such as ReLU, GD implicitly converges to a KKT point of the max-margin problem, it is still unclear where it is an actual optimum. A recent work (Vardi et al., 2022) showed that in many cases, the converged KKT point is not even a local optimum of the max margin problem. Besides, there are many other attempts to explain the implicit bias of deep learning (Vardi, 2023). Another popular implicit bias is the flat minima bias (Hochreiter and Schmidhuber, 1997; Keskar et al., 2016). Recent studies (Wu et al., 2018; Blanc et al., 2020; Ma and Ying, 2021; Li et al., 2021; Mulayoff et al., 2021; Wu et al., 2022; Wu and Su, 2023) provided explanations for why SGD favors flat minima and flat minima generalize well.

## 3 Preliminaries

**Basic Notations.** We use bold letters for vectors or matrices and lowercase letters for scalars, e.g. $\boldsymbol{x} = (x_1, \cdots, x_d)^\top \in \mathbb{R}^d$ and $\boldsymbol{P} = (P_{ij})_{m_1 \times m_2} \in \mathbb{R}^{m_1 \times m_2}$. We use $\langle \cdot, \cdot \rangle$ for the standard Euclidean inner product between two vectors, and $\|\cdot\|$ for the $l_2$ norm of a vector or the spectral norm of a matrix. We use progressive representation $\mathcal{O}, \Omega, \Theta$ to hide absolute positive constants. For any positive integer $n$, let $[n] = \{1, \cdots, n\}$. Denote by $\mathcal{N}(\boldsymbol{\mu}, \boldsymbol{\Sigma})$ the Gaussian distribution with mean $\boldsymbol{\mu}$ and covariance matrix $\boldsymbol{\Sigma}$, $\mathbb{U}(\mathcal{S})$ the uniform distribution on a set $\mathcal{S}$. Denote by $\mathbb{I}\{E\}$ the indicator function for an event $E$.

### 3.1 Binary Classification with Two-layer ReLU Networks

**Binary classification.** In this paper, we consider the binary classification problem. We are given $n$ training data $\mathcal{S} = \{(\boldsymbol{x}_i, y_i)\}_{i=1}^n \subset \mathbb{R}^d \times \{\pm 1\}$. Let $f(\cdot; \boldsymbol{\theta})$ be a neural network model parameterized by $\boldsymbol{\theta}$, and aim to minimize the empirical risk given by:

$$\mathcal{L}(\boldsymbol{\theta}) = \frac{1}{n} \sum_{i=1}^n \ell(y_i f(\boldsymbol{x}_i; \boldsymbol{\theta})), \tag{1}$$

where $\ell(\cdot) : \mathbb{R} \to \mathbb{R}$ is the exponential-type loss function (Soudry et al., 2018; Lyu and Li, 2019) for classification tasks, including the most popular classification losses: exponential loss, logistic loss, and cross-entropy loss. Our analysis focuses on the exponential loss $\ell(z) = e^{-z}$, while our method can be extended to logistic loss and cross-entropy loss.

**Two-layer ReLU Network.** Throughout the following sections, we consider two-layer ReLU neural networks comprising $m$ neurons defined as

$$f(\boldsymbol{x}; \boldsymbol{\theta}) = \sum_{k=1}^m a_k \sigma(\boldsymbol{b}_k^\top \boldsymbol{x}),$$

where $\sigma(z) = \max\{z, 0\}$ is the ReLU activation function, $\boldsymbol{b}_1 \cdots, \boldsymbol{b}_m \in \mathbb{R}^d$ are the weights in the first layer, $a_1, \cdots, a_m$ are the weights in the second layer. And we consider the case that the weights in the second layer are fixed, which is a common setting used in previous studies (Arora et al., 2019; Chatterji et al., 2021). We use $\boldsymbol{\theta} = (\boldsymbol{b}_1^\top, \cdots, \boldsymbol{b}_m^\top)^\top \in \mathbb{R}^{md}$ to denote the concatenation of all trainable weights.

## 3.2 Gradient Flow Starting from Random Initialization

**Gradient Flow.** As the limiting dynamics of (Stochastic) Gradient Descent with infinitesimal learning rate (Li et al., 2017; 2019), we study the following Gradient Flow (GF) on the objective function (1):

$$\frac{\mathrm{d}\boldsymbol{\theta}(t)}{\mathrm{d}t} \in -\partial^{\circ}\mathcal{L}(\boldsymbol{\theta}(t)), \quad t \geq 0. \tag{2}$$

Notice that the ReLU is not differentiable at 0, and therefore, the dynamics is defined as a subgradient inclusion flow (Bolte et al., 2010). Here, $\partial^{\circ}$ denotes the Clarke subdifferential, which is a generalization of the derivative for non-differentiable functions. Additionally, to address the potential non-uniqueness of gradient flow trajectories, we adopt the definition of solutions for discontinuous systems (Filippov, 2013). For formal definitions, please refer to Appendix B, G, and H.

**Random Initialization.** We consider GF (2) starting from the following initialization:

$$\boldsymbol{b}_k(0) \overset{\text{i.i.d.}}{\sim} \frac{\kappa_1}{\sqrt{m}}\mathbb{U}(\mathbb{S}^{d-1}) \text{ and } a_k = \mathrm{s}_k\frac{\kappa_2}{\sqrt{m}} \text{ for } k \in [m];$$

$$\mathrm{s}_k = 1 \text{ for } k \in [m/2]; \ \mathrm{s}_k = -1 \text{ for } k \in [m] - [m/2].$$

Here, $0 < \kappa_1 < \kappa_2 \leq 1$ control the initialization scale. It is worth noting that since the distribution $\mathcal{N}(\mathbf{0}, \boldsymbol{I}_d/d)$ is close to $\mathbb{U}(\mathbb{S}^{d-1})$ in high-dimensional settings, our result can be extended to the initialization $\boldsymbol{b}_k \overset{\text{i.i.d.}}{\sim} \mathcal{N}(\mathbf{0}, \kappa_1^2\boldsymbol{I}_d/md)$ with high probability guarantees.

## 3.3 Linearly Separable Data beyond Orthogonally Separable

In previous works (Phuong and Lampert, 2021; Wang and Ma, 2022), a special case of the linearly separable dataset was investigated, namely "orthogonally separable". A training dataset is orthogonally separable when $\langle \boldsymbol{x}_i, \boldsymbol{x}_j \rangle \geq 0$ for $i, j$ in the same class, and $\langle \boldsymbol{x}_i, \boldsymbol{x}_j \rangle \leq 0$ for $i, j$ in different classes. As mentioned in Section 2, in this case, GF can learn all features and achieve $100\%$ training accuracy quickly, followed by lazy training. In this work, we consider data that is more difficult to learn, which leads to more complicated optimization dynamics. Specifically, we consider the following data.

**Assumption 3.1.** Consider the linearly separable dataset $\mathcal{S} = \{(\boldsymbol{x}_i, y_i)\}_{i \in [n]} \subset \mathbb{R}^d \times \mathbb{R}$ such that $(\boldsymbol{x}_i, y_i) = \begin{cases} (\boldsymbol{x}_+, 1), & i \in [n_+] \\ (\boldsymbol{x}_-, -1), & i \in [n] - [n_+] \end{cases}$ , where $\boldsymbol{x}_+, \boldsymbol{x}_- \in \mathbb{S}^{d-1}$ are two data points with a small angle $\Delta \in (0, \pi/2)$, and $n_+, n_-$ are the numbers of positive and negative samples, respectively, with $n = n_+ + n_-$. We also use $p := n_+/n_-$ to denote the ratio of $n_+$ and $n_-$, which measures the class imbalance. Furthermore, we assume $p\cos\Delta > 1$.

**Remark 3.2.** We focus on the training dataset satisfying Assumption 3.1 with a small $\Delta \ll 1$. The margin of the dataset is $\sin(\Delta/2)$, which implies that the separability of this data is much weaker than that of orthogonal separable data. Additionally, the condition $p\cos\Delta > 1$ merely requires a slight imbalance in the data. These two properties work together to produce rich nonlinear behaviors during training.

# 4 Characterization of Four-phase Optimization Dynamics

In this section, we study the whole optimization dynamics of GF (2) starting from random initialization when training the two-layer ReLU network on linearly separable dataset satisfying Assumption 3.1 and using the loss function (1). To begin with, we introduce some additional notations.

**Additional Notations.** First, we identify several crucial data-dependent directions under Assumption 3.1. These include two directions that are orthogonal to the data, defined as $\boldsymbol{x}_+^{\perp} := \frac{\boldsymbol{x}_- - \langle \boldsymbol{x}_-, \boldsymbol{x}_+ \rangle \boldsymbol{x}_+}{\|\boldsymbol{x}_- - \langle \boldsymbol{x}_-, \boldsymbol{x}_+ \rangle \boldsymbol{x}_+\|}$ and $\boldsymbol{x}_-^{\perp} := \frac{\boldsymbol{x}_+ - \langle \boldsymbol{x}_+, \boldsymbol{x}_- \rangle \boldsymbol{x}_-}{\|\boldsymbol{x}_+ - \langle \boldsymbol{x}_+, \boldsymbol{x}_- \rangle \boldsymbol{x}_-\|}$, which satisfy $\langle \boldsymbol{x}_+, \boldsymbol{x}_+^{\perp} \rangle = \langle \boldsymbol{x}_-, \boldsymbol{x}_-^{\perp} \rangle = 0$. Additionally, we define the label-average data direction as $\boldsymbol{\mu} := \frac{\boldsymbol{z}}{\|\boldsymbol{z}\|}$ where $\boldsymbol{z} = \frac{1}{n}\sum_{i=1}^n y_i\boldsymbol{x}_i$. One can verify that $\langle \boldsymbol{\mu}, \boldsymbol{x}_+ \rangle > 0$ and $\langle \boldsymbol{\mu}, \boldsymbol{x}_- \rangle > 0$ under the condition $p\cos\Delta > 1$. In Figure 3, we visualize these directions.

Second, we use the following notations to denote important quantities during the GF training process. We denote the prediction on $\boldsymbol{x}_+$ and $\boldsymbol{x}_-$ by $f_+(t) := f(\boldsymbol{x}_+; \boldsymbol{\theta}(t)), f_-(t) := f(\boldsymbol{x}_-; \boldsymbol{\theta}(t))$. We use

$\text{Acc}(t) := \frac{1}{n} \sum_{i=1}^{n} \mathbb{I}\{y_i f(\boldsymbol{x}_i; \boldsymbol{\theta}(t)) > 0\}$ to denote the training accuracy at time $t$. For each neuron $k \in [m]$, we use $\boldsymbol{w}_k(t) := \boldsymbol{b}(t)/\|\boldsymbol{b}(t)\|$ and $\rho_k(t) := \|\boldsymbol{b}(t)\|$ to denote its direction and norm, respectively. To capture the activation dynamics of each neuron $k \in [m]$ on each data, we use $\text{sgn}_k^+(t) := \text{sgn}(\langle \boldsymbol{b}_k(t), \boldsymbol{x}_+ \rangle)$ to record whether the $k$-th neuron is activated with respect to $\boldsymbol{x}_+$, and $\text{sgn}_k^-(t) := \text{sgn}(\langle \boldsymbol{b}_k(t), \boldsymbol{x}_- \rangle)$ defined similarly, which we call ReLU activation patterns.

## 4.1 A Brief Overview of four-phase Optimization Dynamics

We illustrate different phases in the training dynamics by a numerical example. Specifically, we train a network on the dataset that satisfies Assumption 3.1 with $p = 4$ and $\Delta = \pi/15$. The directions and magnitudes of the neurons at some important times are shown in Figure 1, reflecting four different phases on the training behavior and activation patterns. More experiment details and results can be found in Appendix A.1.

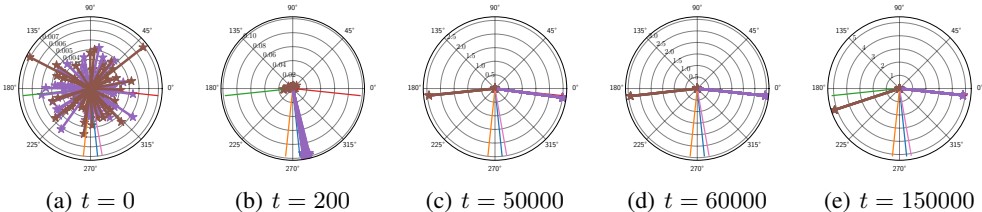

| (a) $t = 0$ | (b) $t = 200$ | (c) $t = 50000$ | (d) $t = 60000$ | (e) $t = 150000$ |

Figure 1: These figures visualize (in polar coordinates) the projections of all neurons $\{\boldsymbol{b}_k(t)\}_{k \in [m]}$ onto the 2d subspace $\text{span}\{\boldsymbol{x}_+, \boldsymbol{x}_-\}$ during training. Each purple star represents a positive neuron ($k \in [m/2]$), while each brown star represents a negative neuron ($k \in [m] - [m/2]$). Additionally, the directions of $\boldsymbol{x}_+, \boldsymbol{x}_-, \boldsymbol{x}_+^\perp, \boldsymbol{x}_-^\perp, \boldsymbol{\mu}$ are plotted in blue, orange, green, red and pink colors, respectively. The complete version of these figures is Figure 4 in Appendix A.1.

From Fig 1(a) to (b) is the **Phase I** of the dynamics, marked by a condensation of neurons. Although the initial directions are random, we see that all neurons are rapidly divided into three categories: living positive neurons ($k \in \mathcal{K}_+$) and living negative neurons ($k \in \mathcal{K}_-$) condense in one direction each ($\boldsymbol{\mu}$ and $\boldsymbol{x}_+^\perp$), while other neurons ($k \notin \mathcal{K}_+ \cup \mathcal{K}_-$) are deactivated forever. From the perspective of loss landscape, GF rapidly escapes from the saddle near $\boldsymbol{0}$ where the loss gradient vanishes.

From Fig 1(b) to (c) is the **Phase II** of the dynamics, in which GF gets stuck into a plateau with training accuracy $\frac{p}{1+p}$ for a long time $T_{\text{plat}}$ before escaping. Once the dynamics escapes from the plateau, the training accuracy rises to a perfect $100\%$. Moreover, activation patterns do not change in this phase.

From Fig 1(c) to (d) is the **Phase III** of the dynamics. The phase transition from phase II to phase III sees a rapid deactivation of all the living positive neurons $k \in \mathcal{K}_+$ on $\boldsymbol{x}_-$ rapidly, while other activation patterns are unchanged. This leads to a simpler network in phase III, in which only living positive neurons (in $\mathcal{K}_+$) predict $\boldsymbol{x}_+$, and only living negative neurons (in $\mathcal{K}_-$) predict $\boldsymbol{x}_-$. Hence, in this phase the GF tries to learn the training data using almost the simplest network by only changing the norms of the neurons.

Finally, Fig 1(d) to (e) shows **Phase IV**, starting from another "phase transition" when all the living negative neurons ($k \in \mathcal{K}_-$) reactivate simultaneously on $\boldsymbol{x}_+$. This leads to a more complicated network. After the phase transition, the activation patterns no longer change, and the neurons eventually converges towards some specific directions dependent on both data and initialization. Additionally, this direction is not even the local optimal max margin direction.

**Overall**, the whole dynamics exhibit a simplifying-to-complicating learning trend.

In the following four subsections, we present a meticulously detailed and comprehensive depiction of the whole optimization dynamics and nonlinear behaviors. For clarity, in Figure 2, we first display the timeline of our dynamics and some nonlinear behaviors.

In Appendix A.2, we further validate our theoretical bounds on the key time points in Figure 2 numerically. Additionally, in Appendix A.3, we relax the data Assumption 3.1 by perturbing the

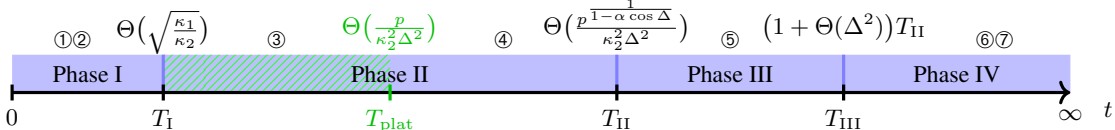

Figure 2: Timeline of the four-phase optimization dynamics, containing some key time points $T_{\text{I}}, T_{\text{II}}, T_{\text{III}}, T_{\text{plat}}$ and their theoretical estimates, and some basic nonlinear behaviors: ① initial condensation, ② saddle escape, ③ getting stuck in plateau, ④ plateau escape, ⑤ neuron deactivation, ⑥ neuron reactivation, ⑦ initialization-dependent directional convergence. Notice ①∼⑦ are only some basic nonlinear behaviors. Moreover, ②+③ is saddle-to-plateau, ①+⑤+⑥ is simplifying-to-complicating.

data with random noise, and our experimental results illustrate that similar four-phase dynamics and nonlinear behaviors persist.

## 4.2 Phase I. Initial Condensation and Saddle Escape

Let $T_{\text{I}} = 10\sqrt{\frac{\kappa_1}{\kappa_2}}$, and we call $t \in [0, T_{\text{I}}]$ Phase I. The theorem below is our main result in Phase I.

**Theorem 4.1** (Initial Condensation). *Let the width $m = \Omega\left(\log(1/\delta)\right)$, the initialization $\kappa_1, \kappa_2 = \mathcal{O}(1)$ and $\kappa_1/\kappa_2 = \mathcal{O}(\Delta^8)$. Then with probability at least $1 - \delta$, the following results hold at $T_{\text{I}}$:*

**(S1)** *Let $\mathcal{K}_+$ be the index set of living positive neurons at $T_{\text{I}}$, i.e. $\mathcal{K}_+ := \{k \in [m/2] : \text{sgn}_k^+(T_{\text{I}}) = 1 \text{ or } \text{sgn}_k^-(T_{\text{I}}) = 1\}$. Then, **(i)** $0.21m \leq |\mathcal{K}_+(T_{\text{I}})| \leq 0.29m$. Moreover, for any $k \in \mathcal{K}_+$, **(ii)** its norm is small but significant: $\rho_k(T_{\text{I}}) = \Theta\left(\sqrt{\frac{\kappa_1\kappa_2}{m}}\right)$; **(iii)** Its direction is strongly aligned with $\boldsymbol{\mu}$: $\langle \boldsymbol{w}_k(T_{\text{I}}), \boldsymbol{\mu} \rangle \geq 1 - \mathcal{O}\left(\sqrt{\kappa_1\kappa_2}\right) - \mathcal{O}\left((\kappa_1/\kappa_2)^{0.55}\right)$; **(iv)** $\text{sgn}_k^+(T_{\text{I}}) = \text{sgn}_k^-(T_{\text{I}}) = 1$.*

**(S2)** *Let $\mathcal{K}_-$ be the index set of living negative neurons at $T_{\text{I}}$, i.e. $\mathcal{K}_- := \{k \in [m] - [m/2] : \text{sgn}_k^+(T_{\text{I}}) = 1 \text{ or } \text{sgn}_k^-(T_{\text{I}}) = 1\}$. Then, **(i)** $0.075m \leq |\mathcal{K}_-| \leq 0.205m$. Moreover, for any $k \in \mathcal{K}_-$, **(ii)** its norm is tiny: $\rho_k(T_{\text{I}}) = \mathcal{O}\left(\frac{\sqrt{\kappa_1\kappa_2}}{\sqrt{m}}\left(\sqrt{\frac{\kappa_1}{\kappa_2}} + \frac{\Delta}{p}\right)\right)$; **(iii)** its direction is aligned with $\boldsymbol{x}_+^\perp$: $\langle \boldsymbol{w}_k(T_{\text{I}}), \boldsymbol{x}_+^\perp \rangle \geq 1 - \mathcal{O}\left(\left(\sqrt{\frac{\kappa_1}{\kappa_2}}\frac{p}{\Delta}\right)^{1.6}\right)$; **(iv)** $\text{sgn}_k^-(T_{\text{I}}) = 1$, but $\text{sgn}_k^+(T_{\text{I}}) = 0$.*

**(S3)** *For other neuron $k \notin \mathcal{K}_+ \cup \mathcal{K}_-$, it dies and remains unchanged during the remaining training process: $\text{sgn}_k^+(t) \leq 0, \text{sgn}_k^-(t) \leq 0, \boldsymbol{b}_k(t) \equiv \boldsymbol{b}_k(T_{\text{I}}), \forall t \geq T_{\text{I}}$.*

**(S4)**. *$f_+(T_{\text{I}}) = \Theta\left(\kappa_2\sqrt{\kappa_1\kappa_2}\right), f_-(T_{\text{I}}) = \Theta\left(\kappa_2\sqrt{\kappa_1\kappa_2}\right)$, and $\text{Acc}(T_{\text{I}}) = \frac{p}{1+p}$.*

**Initial condensation and simplification.** Theorem 4.1 (S1)(S2)(S3) show that, after a short time $T_{\text{I}} = \Theta(\sqrt{\kappa_1/\kappa_2})$, all neurons are implicitly simplified to three categories: $\mathcal{K}_+, \mathcal{K}_-$ and others. The living positive neurons $k \in \mathcal{K}_+$ align strongly with $\boldsymbol{\mu}$, and the living negative neurons $k \in \mathcal{K}_-$ align with $\boldsymbol{x}_+^\perp$ and lie on the manifold orthogonal to $\boldsymbol{x}_+$. Other neurons die and remain unchanged during the remaining training process. Moreover, we also estimate tight bounds for $|\mathcal{K}_+|$ and $|\mathcal{K}_+|$. Actually, $\kappa_1/\kappa_2 = \mathcal{O}(1)$ can ensure Theorem 4.1 and initial condensation hold (please refer to Appendix C), and we write $\kappa_1/\kappa_2 = \mathcal{O}(\Delta^8)$ here to ensure that the dynamics of later phases hold. In Phase I, the dynamics exhibit a fast condensation phenomenon, i.e., in addition to dead neurons, living positive and negative neurons condense in one direction each.

**Saddle-to-Plateau.** The network is initially close to the saddle point at $\boldsymbol{0}$ (where the loss gradient vanishes). However, Theorem 4.1 (S1) reveals that despite being small, there is a significant growth in the norm of living positive neuron $k \in \mathcal{K}_+$ from $\Theta(\kappa_1/\sqrt{m})$ to $\Theta(\sqrt{\kappa_1\kappa_2}/\sqrt{m})$ and the predictions also experience substantial growth (S4). This means that GF escapes from this saddle rapidly. Furthermore, it is worth noting that initial training accuracy can randomly be $0, \frac{1}{1+p}, \frac{p}{1+p}$, or $1$. However, after Phase I, the training accuracy reaches $\text{Acc}(T_{\text{I}}) = \frac{p}{1+p}$ which we will prove as a plateau in the next subsection. Therefore, Phase I exhibits saddle-to-plateau dynamics.

**Remark 4.2.** Throughout the following subsections, we call the neuron $k \in \mathcal{K}_+$ the "living positive neuron", the neuron $k \in \mathcal{K}_-$ the "living negative neuron", and the neuron $k \notin \mathcal{K}_+ \cup \mathcal{K}_-$ the

"dead neuron". Moreover, we denote $m_+ := |\mathcal{K}_+|$, $m_- := |\mathcal{K}_-|$, and $\alpha := \frac{m_-}{m_+}$. Notice that Theorem 4.1(S1)(S2) guarantee that $0 < \frac{0.075}{0.29} \leq \alpha \leq \frac{0.205}{0.21} < 1$.

**Remark 4.3.** The results in the following subsections are all based on the occurrence of the events in Theorem 4.1 and with the same settings as Theorem 4.1. So they all hold with probability at least $1 - \delta$.

Please refer to Appendix C for the proof of Phase I.

### 4.3 Phase II. Getting Stuck in and Escaping from Plateau

In this phase, we study the dynamics before the patterns of living neurons change again after Phase I. Specifically, we define

$$T_{\mathrm{II}} := \inf\{t > T_{\mathrm{I}} : \exists k \in \mathcal{K}_+ \cup \mathcal{K}_-, \mathtt{sgn}_k^+(t) \neq \mathtt{sgn}_k^+(T_{\mathrm{I}}) \text{ or } \mathtt{sgn}_k^-(t) \neq \mathtt{sgn}_k^-(T_{\mathrm{I}})\},$$

and call $t \in (T_{\mathrm{I}}, T_{\mathrm{II}}]$ Phase II.

**Theorem 4.4** (End of Phase II). **(S1)** $T_{\mathrm{II}} = \Theta\left(\frac{p^{\frac{1}{1-\alpha\cos\Delta}}}{\kappa_2^2\Delta^2}\right)$. **(S2)** $\mathcal{L}(\boldsymbol{\theta}(T_{\mathrm{II}})) = \Theta\left(p^{-\frac{1}{1-\alpha\cos\Delta}}\right)$.
**(S3)** *One of living positive neuron* $k_0 \in \mathcal{K}_+$ *precisely changes its pattern on* $\boldsymbol{x}_-$ *at* $T_{\mathrm{II}}$: $\lim\limits_{t \to T_{\mathrm{II}}^-} \mathtt{sgn}_{k_0}^-(t) = 1$ *and* $\lim\limits_{t \to T_{\mathrm{II}}^+} \mathtt{sgn}_{k_0}^-(t) = 0$, *while all other activation patterns remain unchanged.*

Recalling the results in Theorem 4.1, during Phase II, the activation patterns do not change with $\mathtt{sgn}_k^+(t) = \mathtt{sgn}_k^-(t) = 1$ for $k \in \mathcal{K}_+$ and $\mathtt{sgn}_k^+(t) = 0, \mathtt{sgn}_k^-(t) = 1$ for $k \in \mathcal{K}_-$. Theorem 4.4 demonstrates that at the end of Phase II, except for one of living positive neuron $k_0 \in \mathcal{K}_+$ precisely changes its pattern on $\boldsymbol{x}_-$, all other activation patterns remain unchanged.

**Theorem 4.5** (Plateau). *We define the hitting time* $T_{\mathrm{plat}} := \inf\{t \in [T_{\mathrm{I}}, T_{\mathrm{II}}] : \mathrm{Acc}(t) = 1\}$. *Then,*
**(S1)** $T_{\mathrm{plat}} = \Theta\left(\frac{p}{\kappa_2^2\Delta^2}\right)$; **(S2)** $\forall t \in [T_{\mathrm{I}}, T_{\mathrm{plat}}]$, $\mathrm{Acc}(t) \equiv \frac{p}{1+p}$; **(S3)** $\forall t \in (T_{\mathrm{plat}}, T_{\mathrm{II}}]$, $\mathrm{Acc}(t) \equiv 1$.

**Plateau of training accuracy.** According to Theorem 4.5, during Phase II, the training accuracy gets stuck in a long plateau $\frac{p}{1+p}$, which lasts for $\Theta\left(\frac{p}{\kappa_2^2\Delta^2}\right)$ time. However, once escaping from this plateau, the training accuracy rises to $100\%$. It is worth noting that this plateau is essentially induced by the dataset. All that's required is only mild imbalance ($p$ is slightly greater than 1 such that $p\cos\Delta > 1$) and a small margin $\sin(\Delta/2)$ of two data classes. Notably, if the dataset has an extremely tiny margin ($\Delta \to 0$), then the length of this plateau will be significantly prolonged ($T_{\mathrm{plat}} \to +\infty$), which implies how the data separation can affect the training dynamics. Additionally, using a smaller initialization scale $\kappa_1$ of the input layers cannot avoid this plateau.

Please refer to Appendix D for the proof of Phase II.

### 4.4 Phase III. Simplifying by Neuron Deactivation, and Trying to Learn by Simplest Network

Building upon Phase II, we demonstrate that within a short time, all the living positive neurons $\mathcal{K}_+$ change their activation patterns, corresponding to a "phase transition". Specifically, we define

$$T_{\mathrm{II}}^{\mathrm{PT}} := \inf\{t > T_{\mathrm{II}} : \forall k \in \mathcal{K}_+, \mathtt{sgn}_k^-(t) = 0\},$$

and we call $t \in (T_{\mathrm{II}}, T_{\mathrm{II}}^{\mathrm{PT}}]$ the phase transition from Phase II to Phase III.

**Theorem 4.6** (Phase Transition). **(S1)** $T_{\mathrm{II}}^{\mathrm{PT}} = \left(1 + \mathcal{O}\left(\sqrt{\kappa_1\kappa_2^3}\right)\right)T_{\mathrm{II}}$; **(S2)** $\mathtt{sgn}_k^+(T_{\mathrm{II}}^{\mathrm{PT}}) = 1$ *and* $\mathtt{sgn}_k^-(T_{\mathrm{II}}^{\mathrm{PT}}) = 0$ *for any* $k \in \mathcal{K}_+$; $\mathtt{sgn}_k^+(T_{\mathrm{II}}^{\mathrm{PT}}) = 0$ *and* $\mathtt{sgn}_k^-(T_{\mathrm{II}}^{\mathrm{PT}}) = 1$ *for any* $k \in \mathcal{K}_-$.

**Neuron deactivation.** As shown in Theorem 4.6 (S2), after the phase transition, *all* the living positive neurons $k \in \mathcal{K}_+$ undergo deactivation for $\boldsymbol{x}_-$, i.e., $\mathtt{sgn}_k^-(t)$ changes from 1 to 0, while other activation patterns remain unchanged. Furthermore, Theorem 4.6 (S1) reveals that the phase transition is completed quite quickly by using sufficiently small initialization value $\kappa_1, \kappa_2$. A smaller initialization value leads to a more precise initial condensation $\boldsymbol{w}_k(T_{\mathrm{I}}) \approx \boldsymbol{\mu}$, causing all living positive neurons to remain closer together before $T_{\mathrm{II}}$ and thus changing their patterns nearly simultaneously.

**Self-simplifying.** As a result, the network implicitly simplifies itself through the deactivation behavior. At $T_{\mathrm{III}}^{\mathrm{PT}}$, only living negative neurons $k \in \mathcal{K}_+$ are used for predicting on $\boldsymbol{x}_-$, i.e., $f_-(T_{\mathrm{II}}^{\mathrm{PT}}) = \frac{\kappa_2}{\sqrt{m}} \sum_{k \in \mathcal{K}_-} \sigma(\langle \boldsymbol{b}_k(T_{\mathrm{II}}^{\mathrm{PT}}), \boldsymbol{x}_- \rangle)$. In contrast, during Phase II, both living positive and living negative neurons jointly predict on $\boldsymbol{x}_-$, i.e., $f_-(t) = \frac{\kappa_2}{\sqrt{m}} \sum_{k \in \mathcal{K}_+} \sigma(\langle \boldsymbol{b}_k(t), \boldsymbol{x}_- \rangle) - \frac{\kappa_2}{\sqrt{m}} \sum_{k \in \mathcal{K}_-} \sigma(\langle \boldsymbol{b}_k(t), \boldsymbol{x}_- \rangle)$. As indicated in Table 1, two classes of activation patterns are simplified from $(1, 0)$ to $(0, 0)$, while others do not change.

After this phase transition, we study the dynamics before the patterns of living neurons change again. Specifically, we define

$$T_{\mathrm{III}} := \inf\{t > T_{\mathrm{II}}^{\mathrm{PT}} : \exists k \in \mathcal{K}_+ \cup \mathcal{K}_-, \mathtt{sgn}_k^+(t) \neq \mathtt{sgn}_k^+(T_{\mathrm{II}}^{\mathrm{PT}}) \text{ or } \mathtt{sgn}_k^-(t) \neq \mathtt{sgn}_k^-(T_{\mathrm{II}}^{\mathrm{PT}})\},$$

and call $t \in (T_{\mathrm{II}}, T_{\mathrm{III}}]$ Phase III.

**Theorem 4.7** (End of Phase III). $T_{\mathrm{III}} = \left(1 + \Theta(\Delta^2)\right)T_{\mathrm{II}}$.

**Learning by simplest network.** During $t \in (T_{\mathrm{II}}^{\mathrm{PT}}, T_{\mathrm{III}})$, all activation patterns do not change. This ensures that $f_+(t) = \frac{\kappa_2}{\sqrt{m}} \sum_{k \in \mathcal{K}_+} \sigma(\langle \boldsymbol{b}_k(t), \boldsymbol{x}_+ \rangle)$, while $f_-(t) = -\frac{\kappa_2}{\sqrt{m}} \sum_{k \in \mathcal{K}_-} \sigma(\langle \boldsymbol{b}_k(t), \boldsymbol{x}_- \rangle)$. Additionally, by using sufficiently small $\kappa_1$, the neurons in $\mathcal{K}_+$ and $\mathcal{K}_-$ keep close together respectively before $T_{\mathrm{III}}$, making the network close to a simple two-neuron network consisting of one positive neuron and one negative neuron. Please refer to Appendix E for more details. Furthermore, this pattern scheme is almost the "simplest" way to ensure binary classification: the living positive neurons only predict positive data $\boldsymbol{x}_+$ while the living negative neurons only predict negative data $\boldsymbol{x}_-$. Therefore, GF tries to learn by this almost simplest network in this phase.

Please refer to Appendix E for the proof of Phase III.

## 4.5 Phase IV. Complicating by Neuron Reactivation, and Directional Convergence

Phase IV begins with an instantaneous phase transition at time $T_{\mathrm{III}}$.

**Theorem 4.8** (Phase Transition). *All living negative neuron $k \in \mathcal{K}_-$ simultaneously change their patterns on $\boldsymbol{x}_+$ at $T_{\mathrm{III}}$:* $\lim_{t \to T_{\mathrm{III}}^-} \mathtt{sgn}_k^+(t) = 0$, $\lim_{t \to T_{\mathrm{III}}^+} \mathtt{sgn}_k^+(t) = 1$, *while others remain unchanged.*

**Neuron reactivation.** According to Theorem 4.8, *all* of living negative neurons $k \in \mathcal{K}_-$ reactivate *simultaneously* on $\boldsymbol{x}_+$ at $T_{\mathrm{III}}$: $\mathtt{sgn}_k^+(t)$ changes from 0 to 1, while other activation patterns remain unchanged.

**Self-Complicating.** Along with the reactivation behavior, GF implicitly complicates itself. In Phase III, only living negative neurons $k \in \mathcal{K}_+$ are used to predict on $\boldsymbol{x}_+$, i.e., $f_+(t) = \frac{\kappa_2}{\sqrt{m}} \sum_{k \in \mathcal{K}_+} \sigma(\langle \boldsymbol{b}_k(t), \boldsymbol{x}_+ \rangle)$. In contrast, after the phase transition at $T_{\mathrm{III}}$, both living positive and living negative neurons jointly predict on $\boldsymbol{x}_+$, i.e. $f_+(t) = \frac{\kappa_2}{\sqrt{m}} \sum_{k \in \mathcal{K}_+} \sigma(\langle \boldsymbol{b}_k(t), \boldsymbol{x}_+ \rangle) - \frac{\kappa_2}{\sqrt{m}} \sum_{k \in \mathcal{K}_-} \sigma(\langle \boldsymbol{b}_k(t), \boldsymbol{x}_+ \rangle)$. As indicated in Table 1, two classes of activation patterns are complicated from $(0, 0)$ to $(0, 1)$, while others do not change.

In this phase, we study the dynamics before activation patterns change again after the phase transition in Theorem 4.8. We define the hitting time:

$$T_{\mathrm{IV}} := \inf\{t > T_{\mathrm{III}} : \exists k \in \mathcal{K}_+ \cup \mathcal{K}_-, \mathtt{sgn}_k^+(t) \neq \lim_{s \to T_{\mathrm{III}}^+} \mathtt{sgn}_k^+(s) \text{ or } \mathtt{sgn}_k^-(t) \neq \lim_{s \to T_{\mathrm{III}}^+} \mathtt{sgn}_k^-(s)\},$$

and we call $t \in (T_{\mathrm{III}}, T_{\mathrm{IV}}]$ Phase IV.

**Theorem 4.9** (Phase IV). **(S1)** $T_{\mathrm{IV}} = +\infty$. *Moreover, for any $t > T_{\mathrm{III}}$,* **(S2)** *all activation patterns do not change;* **(S3)** *the loss converges with* $\mathcal{L}(\boldsymbol{\theta}(t)) = \Theta\left(\frac{1}{p^{\frac{1}{1 - \alpha \cos \Delta}} + \kappa_2^2 \Delta^2(t - T_{\mathrm{III}})}\right)$.

Theorem 4.9 illustrates that all activation patterns never change again after the phase transition at $T_{\mathrm{III}}$ with $\mathtt{sgn}_k^+(t) = 1, \mathtt{sgn}_k^-(t) = 0$ for any $k \in \mathcal{K}_+$ and $\mathtt{sgn}_k^+(t) = \mathtt{sgn}_k^-(t) = 1$ for any $k \in \mathcal{K}_-$. Additionally, the loss converges with the polynomial rate $\Theta(1/\kappa_2^2 \Delta^2 t)$. Furthermore, we present the following theorem about the convergent direction of each neuron.

**Theorem 4.10** (Directional Convergence). *The limit* $\lim_{t \to +\infty} \frac{\boldsymbol{\theta}(t)}{\|\boldsymbol{\theta}(t)\|_2}$ *exists and denoted by* $\overline{\boldsymbol{\theta}} = (\overline{\boldsymbol{b}}_1^\top, \cdots, \overline{\boldsymbol{b}}_m^\top)^\top \in \mathbb{S}^{md-1}$. *Moreover, (i) for any* $k \notin \mathcal{K}_+ \cup \mathcal{K}_-, \overline{\boldsymbol{b}}_k = \mathbf{0}$; *(ii) for any* $k \in \mathcal{K}_+, \overline{\boldsymbol{b}}_k \equiv \boldsymbol{v}_+ = C(\boldsymbol{x}_+ - \boldsymbol{x}_- \cos \Delta)$; *(iii) for any* $k \in \mathcal{K}_-, \overline{\boldsymbol{b}}_k \equiv \boldsymbol{v}_- = C\big((1 + \frac{\sin^2 \Delta}{\alpha(1 + \cos \Delta)})\boldsymbol{x}_- - \boldsymbol{x}_+\big)$, *where* $C > 0$ *is a scaling constant such that* $\|\overline{\boldsymbol{\theta}}\|_2 = 1$. *(iv) Additionally,* $f_-(\overline{\boldsymbol{\theta}}) = -f_+(\overline{\boldsymbol{\theta}}) > 0$.

**Initialization-dependent Directional Convergence.** As an asymptotic result, Theorem 4.10 provides the final convergent direction of GF. All living positive neurons ($k \in \mathcal{K}_+$) directionally converge to $\boldsymbol{v}_+ \parallel \boldsymbol{x}_-^\perp$ with $\langle \boldsymbol{v}_+, \boldsymbol{x}_+ \rangle > 0$ and $\langle \boldsymbol{v}_+, \boldsymbol{x}_- \rangle = 0$, while all living negative neurons ($k \in \mathcal{K}_-$) directionally converge to $\boldsymbol{v}_- \in \text{span}\{\boldsymbol{x}_+, \boldsymbol{x}_-\}$ with $\langle \boldsymbol{v}_-, \boldsymbol{x}_+ \rangle > 0$ and $\langle \boldsymbol{v}_-, \boldsymbol{x}_- \rangle > 0$. It is worth noting that $\boldsymbol{v}_-$ directly depends not only on the data but also on the ratio $\alpha = |\mathcal{K}_-|/|\mathcal{K}_+|$ (defined in Remark 4.2). Recalling the results in Phase I, $\alpha$ lies in a certain range with high probability; but it is still a random variable due to its dependence on random initialization. Different initializations may lead to different values $|\mathcal{K}_-|/|\mathcal{K}_+|$ at the end of Phase I, eventually causing different convergent directions in Phase IV.

**Non-(Local)-Max-Margin Direction.** Lastly, we study the implicit bias of the final convergence rate. According to Lyu and Li (2019); Ji and Telgarsky (2020) and our results above, $\overline{\boldsymbol{\theta}}$ in Theorem 4.10 must be a KKT direction of some max-margin optimization problem. However, it is not clear whether the direction $\overline{\boldsymbol{\theta}}$ is actually an actual optimum of this problem. Surprisingly, in next Theorem, we demonstrate that the final convergent direction is **not even a local optimal** direction of this problem, which enlightens us to rethink the max margin bias of ReLU neural networks.

**Theorem 4.11** (Implicit Bias). *The final convergent direction* $\overline{\boldsymbol{\theta}}$ *(in Theorem 4.10) is a KKT direction of the max-margin problem* $\min : \frac{1}{2} \|\boldsymbol{\theta}\|^2$ *s.t.* $y_i f(\boldsymbol{x}_i; \boldsymbol{\theta}) \geq 1, i \in [n]$. *However,* $\overline{\boldsymbol{\theta}}$ *is not even a local optimal direction of this problem.*

Please refer to Appendix F for the proof of Phase IV.

## 5  Discussion and Comparison on Nonlinear Behaviors

Throughout the whole training process in Section 4, we divide the phases based on the evolution of ReLU activation patterns. During Phase I, as well as the beginning of Phase II and III, numerous activation patterns undergo rapid changes. Table 1 summarizes the evolution of activation patterns for all living neurons after Phase I. These results are also numerically validated in Figure 1.

Table 1: The evolution of two classes of activation patterns of living neurons after Phase I. As for other two classes, $\text{sgn}_k^+(t)$ ($k \in \mathcal{K}_+$) and $\text{sgn}_k^-(t)$ ($k \in \mathcal{K}_-$), they remain equal to 1 after Phase I.

| | $t \in (T_{\text{I}}, T_{\text{II}})$ | $t \in (T_{\text{II}}, T_{\text{II}}^{\text{PT}})$ | $t \in (T_{\text{II}}^{\text{PT}}, T_{\text{III}})$ | $t \in (T_{\text{III}}, +\infty)$ |
|---|---|---|---|---|
| $\text{sgn}_k^-(t)$ ($k \in \mathcal{K}_+$) | 1 | 1 or 0 | 0 | 0 |
| $\text{sgn}_k^+(t)$ ($k \in \mathcal{K}_-$) | 0 | 0 | 0 | 1 |

**Simplifying-to-Complicating.** In phase I, GF simplifies all the neurons from random directions into three categories: living positive neurons $\mathcal{K}_+$ and living negative neurons $\mathcal{K}_-$ condense in one direction each, which other neurons are deactivated forever. After Phase I, as shown in Table 1, the two classes of activation patterns change from $(1, 0) \overset{\text{simplify}}{\to} (0, 0) \overset{\text{complicate}}{\to} (0, 1)$, while other patterns remain unchanged. Therefore, the evolution of activation patterns exhibits a simplifying-to-complicating learning trend, which also implies that the network trained by GF learn features in increasing complexity.

**Comparison with NTK.** In the lazy regime such as NTK, most neurons keep close to the initialization and most activation patterns do not change during training. Specifically, for any training data $\boldsymbol{x}_i$, $\frac{1}{m} \sum_{k \in [m]} \mathbb{I}\{\text{sgn}(\langle \boldsymbol{b}_k(t), \boldsymbol{x}_i \rangle) \neq \text{sgn}(\langle \boldsymbol{b}_k(0), \boldsymbol{x}_i \rangle)\} = o(1), \forall t > 0$ (Du et al., 2018). However, our work stands out from lazy regime as activation patterns undergo numerous changes during training. In Phase I, initial condensation causes substantial changes in activation patterns, which is similarly observed in (Phuong and Lampert, 2021). Furthermore, even after Phase I, there are notable modifications in activation patterns. As shown in Table 1, the proportion of changes in

activation patterns for any given training data is the $\Theta(1)$, as compared with the $o(1)$ in NTK regime. Specifically, at any $t > T_{\mathrm{III}}$, $\frac{1}{m}\sum_{k\in[m]}\mathbb{I}\{\mathtt{sgn}_k^+(t) \neq \mathtt{sgn}_k^+(T_{\mathrm{I}})\} = \frac{1}{m}|\mathcal{K}_-| = \Theta(1)$ and $\frac{1}{m}\sum_{k\in[m]}\mathbb{I}\{\mathtt{sgn}_k^-(t) \neq \mathtt{sgn}_k^-(T_{\mathrm{I}})\} = \frac{1}{m}|\mathcal{K}_+| = \Theta(1)$. On the other hand, in our analysis, the requirement on the network's width $m$ is only $m = \Omega(\log(1/\delta))$ (Theorem 4.1), regardless of data parameters $p, \Delta$, while NTK regime requires a much larger width $m = \Omega(\log(p/\delta)/\Delta^6)$ (Ji and Telgarsky, 2019).

**Comparison with** Phuong and Lampert (2021); Lyu et al. (2021); Wang and Ma (2022); Boursier et al. (2022). Beyond lazy regime and local analysis, these works also characterize the entire training dynamics and analyze a few nonlinear behaviors. Now we compare our results with these works in detail. (i) While Lyu et al. (2021) focuses on training Leaky ReLU NNs, our work and the other three papers study ReLU NNs. It is worth noting that the dynamics of Leaky ReLU NNs differ from ReLU due to the permanent activation of Leaky ReLU ($\sigma'(\cdot) \geq \alpha > 0$). (ii) Initial condensation is also proven in (Lyu et al., 2021; Boursier et al., 2022), and the condensation directions are some types of data averages. In our work, neurons can aggregate towards not only the average direction $\boldsymbol{\mu}$, but also another direction $\boldsymbol{x}_+^\perp$. Moreover, we also estimate the number of neurons that condense into two directions. (iii) Saddle-to-saddle dynamics are proven in (Phuong and Lampert, 2021) for square loss, where the second saddle is about training loss and caused by incomplete fitting. However, our work focus on classification with exponential loss and exhibit a similar saddle-to-plateau dynamics, where the plateau is about training accuracy, caused by incomplete feature learning. (iv) Phased feature learning. In (Phuong and Lampert, 2021; Wang and Ma, 2022), all features can be rapidly learned in Phase I (accuracy= $100\%$), followed by lazy training. However, for practical tasks on more complex data, NNs can hardly learn all features in such short time. In our work, the data is more difficult to learn, resulting in incomplete feature learning in Phase I (accuracy< $100\%$). Subsequently, NNs experience a long time to learn other features completely. Such multi-phase feature leaning dynamics are closer to practical training process. (v) Neuron reactivation and deactivation. For ReLU NNs, The evolution of activation patterns is one of the essential causes of nonlinear dynamics. In (Phuong and Lampert, 2021; Wang and Ma, 2022), activation patterns only change rapidly in Phase I, after which they remain unchanged. In (Boursier et al., 2022), their lemma 6 shows that their dynamics lack neuron reactivation. However, in our dynamics, even after Phase I, our dynamics exhibit significant neuron deactivation and reactivation as discussed in Table 1. (vi) The final convergent directions are also derived in (Phuong and Lampert, 2021; Lyu et al., 2021; Boursier et al., 2022), which only depend on the data. However, in our setting, the convergent direction is more complicated, determined by both data and random initialization. (vii) Furthermore, our four-phase dynamics demonstrate the whole evolution of activation patterns during training and reveal a general simplifying-to-complicating learning trend.

In summary, our whole four-phase optimization dynamics capture more nonlinear behaviors than these works. Furthermore, we conduct a more thorough and detailed theoretical analysis of these nonlinear behaviors, providing a more systematic and comprehensive understanding.

# 6   Conclusion and Future Work

In this work, we study the optimization dynamics of ReLU neural networks trained by GF on a linearly separable data. Our analysis captures the whole optimization process starting from random initialization to final convergence. Throughout the whole training process, we reveal four different phases and identify rich nonlinear behaviors theoretically. However, theoretical understanding of the training of NNs still has a long way to go. For instance, although we conduct a fine-grained analysis of GF, the dynamics of GD are more complex and exhibit other nonlinear behaviors such as progressive sharpening and edge of stability (Wu et al., 2018; Jastrzębski et al., 2019; Cohen et al., 2021; Ma et al., 2022; Li et al., 2022; Damian et al., 2022; Zhu et al., 2022; Ahn et al., 2022a;b). Additionally, unlike GD, SGD uses only mini-batches of data and injects noise (Zhu et al., 2019; Thomas et al., 2020; Feng and Tu, 2021; Liu et al., 2021; Ziyin et al., 2022; Wu et al., 2022; Wojtowytsch, 2023; Wang and Wu, 2023) in each iteration, which can have a pronounced impact on the optimization dynamics and nonlinear behaviors. Better understanding of the nonlinear behaviors during GD or SGD training is an important direction of future work.

## Acknowledgments and Disclosure of Funding

We thank Prof. Weinan E, Prof. Lei Wu, Prof. Zhi-Qin John Xu and anonymous reviewers for helpful suggestions. Mingze Wang is supported in part by the National Key Basic Research Program of China: 2015CB856000.

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

# Appendix

## Contents

# A  Experimental Details

All experiments are conducted on a MacBook pro 13 (M2) only using CPU. See the code at https://github.com/wmz9/Understanding_Multi-phase_Optimization_NeurIPS2023.

## A.1  Experiments on standard Dataset

We train the two-layer network on the dataset that satisfies Assumption 3.1 with $d = 20$, $p = 4$ and $\Delta = \pi/15$. Specifically, we choose the network width $m = 100$; the initialization scale $\kappa_1 = 0.1, \kappa_2 = 1$; the small learning rate $\eta = 0.01$.

In Figure 3, we show some key data directions in this dataset, as well as the training accuracy, which contains a long plateau. Furthermore, in Figure 4, we provide the evolution of all neurons during training from $t = 0$ to $t = 150000$, which is a more complete version of Figure 1.

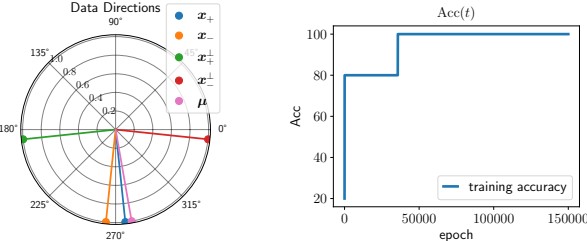

Figure 3: (left) Some key data directions: the directions of $\boldsymbol{x}_+, \boldsymbol{x}_-, \boldsymbol{x}_+^\perp, \boldsymbol{x}_-^\perp, \boldsymbol{\mu}$ are plotted in blue, orange, green, red and pink colors, respectively; (right) The training accuracy.

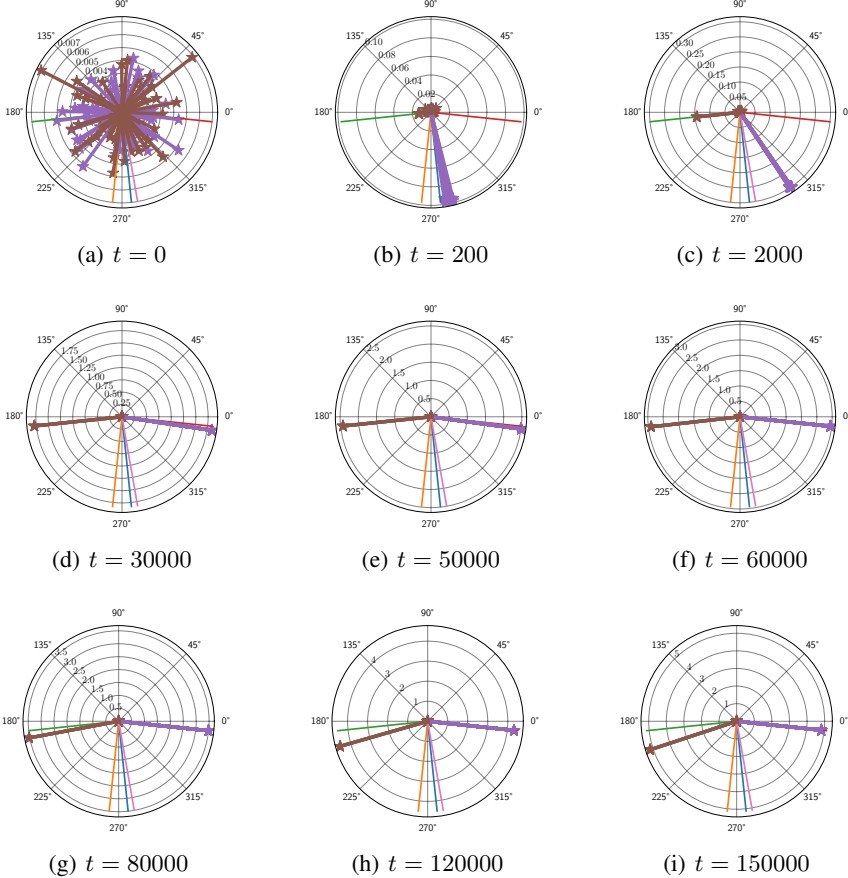

(a) $t = 0$     (b) $t = 200$     (c) $t = 2000$

(d) $t = 30000$     (e) $t = 50000$     (f) $t = 60000$

(g) $t = 80000$     (h) $t = 120000$     (i) $t = 150000$

Figure 4: (A more complete version of Figure 1) These figures visualize (in polar coordinates) the projections of all neurons $\{\boldsymbol{b}_k(t)\}_{k\in[m]}$ onto the 2d subspace $\mathrm{span}\{\boldsymbol{x}_+, \boldsymbol{x}_-\}$ during training (from $t = 0$ to $t = 150000$). Each purple star represents a positive neuron ($k \in [m/2]$), while each brown star represents a negative neuron ($k \in [m] - [m/2]$).

## A.2 Numerical validation on our theoretical bounds

We conduct experiments to validate our theoretical bounds on $T_{\text{plat}}$ and $T_{\text{III}}$ under different $p$ and $\Delta$, and the results are shown in  2.

In the first experiment (1st and 2nd subtable), we fix $p = 4$ for change $\Delta$; in the second experiment (3rd and 4th subtable), we fix $\Delta = \pi/15$ and change $p$. As for other hyperparameters (such as $\kappa_1, \kappa_2, d, m$), we keep the same scales as our setups in Appendix A.1.

We have two main conclusions: (1) four training phases in our theory persistently exist; (the same as Fig 1 in Appendix A, and be omitted due to the limited space) (2) our theoretical estimates on $T_{\text{plat}}$ and $T_{\text{III}}$ are relatively tight, basically in the same magnitude as the realistic time.

Table 2: The change of our theoretical bounds on $T_{\text{plat}}$ and $T_{\text{III}}$ under different $p$ and $\Delta$.

| $\Delta$ | $\frac{4\pi}{45}$ | $\frac{4\pi}{45} \cdot \frac{3}{4} = \frac{\pi}{15}$ | $\frac{4\pi}{45} \cdot \left(\frac{3}{4}\right)^2$ | $\frac{4\pi}{45} \cdot \left(\frac{3}{4}\right)^3$ |
|---|---|---|---|---|
| Realistic $T_{\text{plat}}$ | $1.96 \times 10^4$ | $3.68 \times 10^4$ | $7.25 \times 10^4$ | $12.87 \times 10^4$ |
| Our estimate $\Theta(1/\Delta^2)$: $950/\Delta^2 + 1520/\Delta - 9943$ | $1.90 \times 10^4$ | $3.82 \times 10^4$ | $7.14 \times 10^4$ | $12.89 \times 10^4$ |

| $\Delta$ | $\frac{4\pi}{45}$ | $\frac{4\pi}{45} \cdot \frac{3}{4} = \frac{\pi}{15}$ | $\frac{4\pi}{45} \cdot \left(\frac{3}{4}\right)^2$ | $\frac{4\pi}{45} \cdot \left(\frac{3}{4}\right)^3$ |
|---|---|---|---|---|
| Realistic $T_{\text{III}}$ | $4.98 \times 10^4$ | $6.14 \times 10^4$ | $9.18 \times 10^4$ | $15.63 \times 10^4$ |
| Our estimate $\Theta(1/\Delta^2)$: $1772/\Delta^2 - 12218/\Delta + 67621$ | $4.97 \times 10^4$ | $6.17 \times 10^4$ | $9.16 \times 10^4$ | $15.63 \times 10^4$ |

| $p$ | 6 | 8 | 10 | 12 |
|---|---|---|---|---|
| Realistic $T_{\text{plat}}$ | $6.14 \times 10^4$ | $9.57 \times 10^4$ | $13.96 \times 10^4$ | $17.61 \times 10^4$ |
| Our estimate $\Theta(p)$: $19400p - 56400$ | $6.00 \times 10^4$ | $9.88 \times 10^4$ | $13.76 \times 10^4$ | $17.64 \times 10^4$ |

| $p$ | 6 | 8 | 10 | 12 |
|---|---|---|---|---|
| Realistic $T_{\text{III}}$ | $8.92 \times 10^4$ | $13.40 \times 10^4$ | $19.72 \times 10^4$ | $27.68 \times 10^4$ |
| Our estimate $\Theta(p^{\frac{1}{1-\alpha\cos\Delta}})$: $6912p^{1.5} - 15897$ | $8.59 \times 10^4$ | $14.05 \times 10^4$ | $20.27 \times 10^4$ | $27.14 \times 10^4$ |

## A.3 Experiments on Noisy Dataset

We conduct numerical experiments on the setting of adding small stochastic noise on top of $\boldsymbol{x}_+$ and $\boldsymbol{x}_-$, a little bit more realistic setting. Specifically, in $\text{span}\{\boldsymbol{x}_+, \boldsymbol{x}_-\}$, we perturb the angles of $n_+ - 1$ instances of $\boldsymbol{x}_+$ and $n_- - 1$ instances of $\boldsymbol{x}_-$ using stochastic noise $\xi \sim \text{Unif}([0, \Delta/4])$.

In Figure 5, we visualize (i) the evolution of each neuron throughout the training process; (ii) some key data directions; (iii) the evolution of training accuracy.

From the numerical results in Figure 5, we have two main conclusions: (1) we ascertain that the same four-phase optimization dynamics and nonlinear behaviors persist, even for our dataset with small stochastic noise; (2) a slight difference is that there is more than one plateau of training accuracy in Phase II. The reason is that for noisy data, GF needs to learn negative data one by one in Phase II. For example, three distinct negative data are employed in this experiment, so three plateaus of training accuracy emerge ($12/15$, $13/15$, and $14/15$).

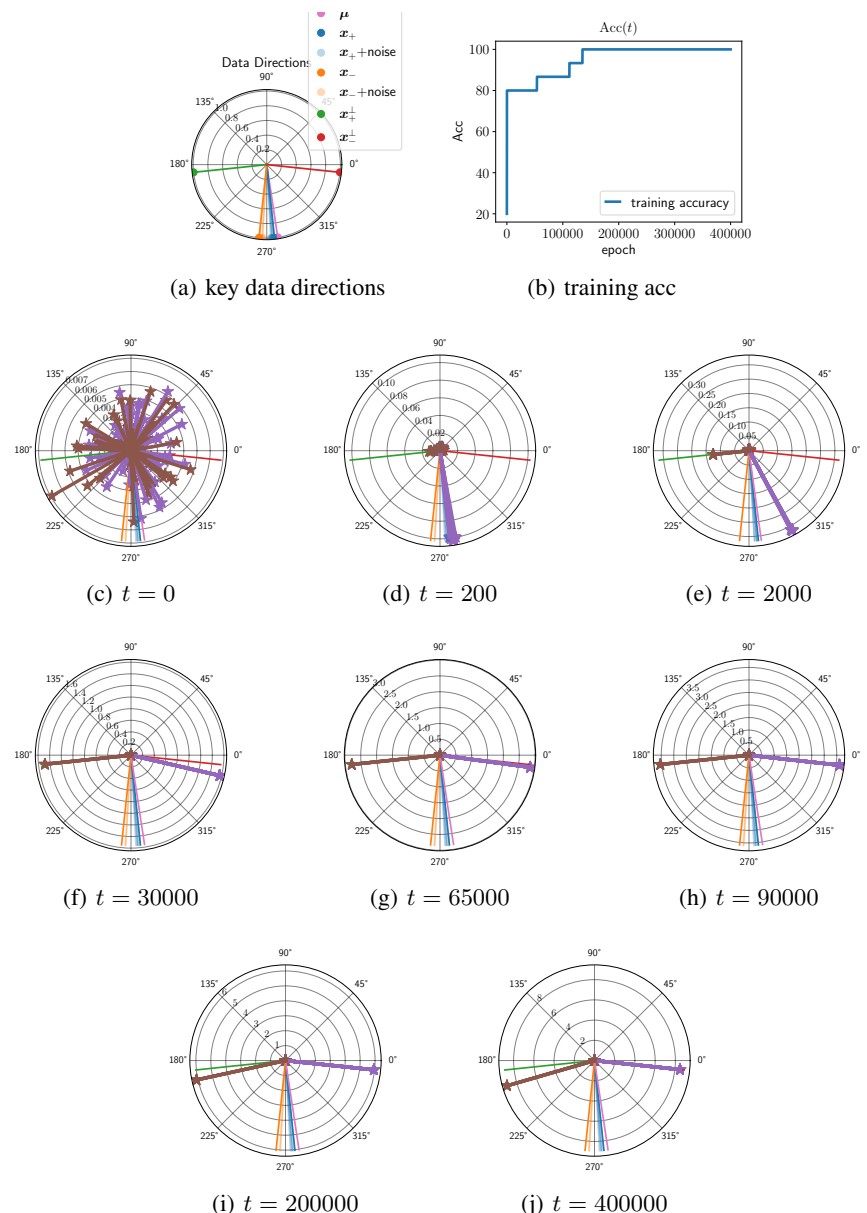

Figure 5: The experimental results for noisy data with $\Delta = \pi/15$, $n_+ = 12$, $n_- = 3$. Fig (a): Some key data directions, including $\boldsymbol{x}_+$, $\boldsymbol{x}_+$+noise, $\boldsymbol{x}_-$, $\boldsymbol{x}_-$+noise, $\boldsymbol{\mu}$, $\boldsymbol{x}_+^\perp$, and $\boldsymbol{x}_-^\perp$. Fig (b): The evolution of training accuracy. Figs (c)~(j): the evolution of the projections of all neurons $\{\boldsymbol{b}_k(t)\}_{k\in[m]}$ onto the 2d subspace $\mathrm{span}\{\boldsymbol{x}_+, \boldsymbol{x}_-\}$ during training (from $t = 0$ to $t = 400000$). Each purple star represents a positive neuron ($k \in [m/2]$), while each brown star represents a negative neuron ($k \in [m] - [m/2]$). **Four-phase dynamics**: from Fig (c) to (d) is Phase I; from Fig (d) to (g) is Phase II; from Fig (g) to (h) is Phase III; from Fig (h) to (j) is Phase IV. To compare these results with noiseless data, please refer to Figures 3 and 4.

# B Proof Preparation

**Selection of initialization parameters.**

For the data satisfying Assumption 3.1, we consider the regime that $\Delta \ll 1$ with $p \cos \Delta > 1$. During the entire proof, we select the initialization scale $\kappa_1, \kappa_2$ as follows:

$$\kappa_2 = \mathcal{O}(1), \quad \frac{\kappa_1}{\kappa_2} = \mathcal{O}\left(\Delta^8\right). \tag{3}$$

**Gradient Flow.** In general, for any $k \in [m]$, the GF dynamics of $\boldsymbol{b}_k(t)$ can be written as

$$\begin{aligned}
\frac{\mathrm{d}\boldsymbol{b}_k(t)}{\mathrm{d}t} &\in \frac{\partial^\circ \mathcal{L}(\boldsymbol{\theta})}{\partial \boldsymbol{b}_k} = \frac{\mathrm{s}_k \kappa_2}{\sqrt{m}} \frac{1}{n} \sum_{i=1}^n e^{-y_i f_i(t)} \partial^\circ \sigma(\langle \boldsymbol{b}_k(t), \boldsymbol{x}_i \rangle) y_i \boldsymbol{x}_i \\
&= \frac{\mathrm{s}_k \kappa_2}{\sqrt{m}} \left( \frac{p}{1+p} e^{-f_+(t)} \partial^\circ \sigma(\langle \boldsymbol{b}_k(t), \boldsymbol{x}_+ \rangle) \boldsymbol{x}_+ - \frac{1}{1+p} e^{f_-(t)} \partial^\circ \sigma(\langle \boldsymbol{b}_k(t), \boldsymbol{x}_- \rangle) \boldsymbol{x}_- \right),
\end{aligned} \tag{4}$$

where $\partial^\circ$ is Clarke's subdifferential defined in Defnition G.1. Notice that if $\mathcal{L}$ is continuously differentiable at $\boldsymbol{\theta}$, then $\partial^\circ \mathcal{L}(\boldsymbol{\theta}) = \{\nabla \mathcal{L}(\boldsymbol{\theta})\}$ is unique.

However, for discontinuous differentiable points of $\mathcal{L}$, the differential inclusion flow $\frac{\mathrm{d}\boldsymbol{\theta}}{\mathrm{d}t} \in \partial^\circ \mathcal{L}(\boldsymbol{\theta})$ may not be unique. To study a more specific dynamics, we also utilize Definition H.1 to determine GF at some of such points, which overcomes non-uniqueness of GF trajectories to some extent. It is worth noting that Definition H.1 and Definition G.1 are compatible and specifically, the dynamics defined in Definition H.1(Case I, III) lie in the convex hull defined in Definition G.1.

**Remark B.1.** In (Lyu et al., 2021), the non-branching starting point Assumption is employed to address the technical challenge of non-uniqueness in GF trajectories. By comparison, in this work, we do not need this assumption. We adopt Definition H.1 to uniquely determine the Gradient Flow trajectories theoretically near some discontinuous differential regions, such as "Ridge", "Valley", and "Refraction edge" discussed in Section I.2 in (Lyu et al., 2021).

Additionally, in the following sections, we may rewrite this dynamics accordingly, such as specific forms and dynamics decomposition.

**Additional Notations.** As a similar description to $\mathrm{sgn}_k^+(\cdot)$ and $\mathrm{sgn}_k^-(\cdot)$, we also employ the following six manifolds to characterize activation patterns (by judging which manifold the neuron $\boldsymbol{w}_k$ belongs to):

$$\begin{aligned}
\mathcal{M}_+^+ &:= \{\boldsymbol{w} \in \mathbb{S}^{d-1} : \langle \boldsymbol{w}, \boldsymbol{x}_+ \rangle > 0\}, \quad \mathcal{M}_-^+ := \{\boldsymbol{w} \in \mathbb{S}^{d-1} : \langle \boldsymbol{w}, \boldsymbol{x}_- \rangle > 0\}, \\
\mathcal{M}_+^0 &:= \{\boldsymbol{w} \in \mathbb{S}^{d-1} : \langle \boldsymbol{w}, \boldsymbol{x}_+ \rangle = 0\}, \quad \mathcal{M}_-^0 := \{\boldsymbol{w} \in \mathbb{S}^{d-1} : \langle \boldsymbol{w}, \boldsymbol{x}_- \rangle = 0\}, \\
\mathcal{M}_+^- &:= \{\boldsymbol{w} \in \mathbb{S}^{d-1} : \langle \boldsymbol{w}, \boldsymbol{x}_+ \rangle \leq 0\}, \quad \mathcal{M}_-^- := \{\boldsymbol{w} \in \mathbb{S}^{d-1} : \langle \boldsymbol{w}, \boldsymbol{x}_- \rangle \leq 0\}.
\end{aligned}$$

As one of our interested manifolds, the border $\partial(\mathcal{M}_+^+ \cap \mathcal{M}_-^+)$ can be divided into

$$\begin{aligned}
\partial(\mathcal{M}_+^+ \cap \mathcal{M}_-^+) =& \{\boldsymbol{w} \in \mathbb{S}^{d-1} : \langle \boldsymbol{w}, \boldsymbol{x}_+ \rangle = 0 \text{ or } \langle \boldsymbol{w}, \boldsymbol{x}_- \rangle = 0\} \\
=& \{\boldsymbol{w} \in \mathbb{S}^{d-1} : \langle \boldsymbol{w}, \boldsymbol{x}_+ \rangle = 0, \langle \boldsymbol{w}, \boldsymbol{x}_- \rangle > 0\} \bigcup \{\boldsymbol{w} \in \mathbb{S}^{d-1} : \langle \boldsymbol{w}, \boldsymbol{x}_+ \rangle > 0, \langle \boldsymbol{w}, \boldsymbol{x}_- \rangle = 0\} \\
& \bigcup \{\boldsymbol{w} \in \mathbb{S}^{d-1} : \langle \boldsymbol{w}, \boldsymbol{x}_+ \rangle = 0, \langle \boldsymbol{w}, \boldsymbol{x}_- \rangle = 0\} \\
=& \left(\mathcal{M}_+^0 \cap \mathcal{M}_-^+\right) \cup \left(\mathcal{M}_-^0 \cap \mathcal{M}_+^+\right) \cup \left(\mathcal{M}_+^0 \cap \mathcal{M}_-^0\right).
\end{aligned}$$

Furthermore, we also utilize the following notations:

$$\begin{aligned}
\mathcal{P}_+^+ &:= \{\boldsymbol{b} \in \mathbb{R}^d : \langle \boldsymbol{b}, \boldsymbol{x}_+ \rangle > 0\}, \quad \mathcal{P}_-^+ := \{\boldsymbol{b} \in \mathbb{R}^d : \langle \boldsymbol{b}, \boldsymbol{x}_- \rangle > 0\}, \\
\mathcal{P}_+^0 &:= \{\boldsymbol{b} \in \mathbb{R}^d : \langle \boldsymbol{b}, \boldsymbol{x}_+ \rangle = 0\}, \quad \mathcal{P}_-^0 := \{\boldsymbol{b} \in \mathbb{R}^{d-1} : \langle \boldsymbol{b}, \boldsymbol{x}_- \rangle = 0\}, \\
\mathcal{P}_+^- &:= \{\boldsymbol{b} \in \mathbb{R}^d : \langle \boldsymbol{b}, \boldsymbol{x}_+ \rangle \leq 0\}, \quad \mathcal{P}_-^- := \{\boldsymbol{b} \in \mathbb{R}^d : \langle \boldsymbol{b}, \boldsymbol{x}_- \rangle \leq 0\}.
\end{aligned}$$

Notice that for the direction of a neuron $\boldsymbol{b}_k \neq \boldsymbol{0}$, using $\mathcal{P}_+^+, \mathcal{P}_+^0, \mathcal{P}_+^-, \mathcal{P}_-^+, \mathcal{P}_-^0, \mathcal{P}_-^-$ to describe $\boldsymbol{b}_k$ is equivalent to using $\mathcal{M}_+^+, \mathcal{M}_+^0, \mathcal{M}_+^-, \mathcal{M}_-^+, \mathcal{M}_-^0, \mathcal{M}_-^-$ to describe $\boldsymbol{w}_k$.

**Lemma B.2** (Dead neurons keep dead). *For the $k$-th neuron, if there exists a $t_0 \geq 0$, s.t. $\boldsymbol{w}_k(t_0) \in \mathcal{M}_+^- \cap \mathcal{M}_-^-$, then it dies and remains unchanged during the remaining training: $\boldsymbol{w}_k(t) \in \mathcal{M}_+^- \cap \mathcal{M}_-^-$ and $\boldsymbol{b}_k(t) \equiv \boldsymbol{b}_k(t_0)$ for any $t \geq t_0$.*

*Proof of Lemma B.2.* A straightforward calculation. $\qquad\qquad\qquad\qquad\qquad\qquad\qquad\qquad$ □

The above fact is a basic fact in our setting, which illustrates that if the neuron $\boldsymbol{b}_k$ is deactivated for both data $\boldsymbol{x}_+$ and $\boldsymbol{x}_-$ at some time, then it remains "dead" forever.

It is worth mentioning that if the neuron $\boldsymbol{b}_k$ is deactivated on $\boldsymbol{x}_+$ but activated on $\boldsymbol{x}_-$ at some time, it can still reactivate on $\boldsymbol{x}_+$ later, which is one of the important reasons why our ReLU optimization dynamics are complicated.

# C   Proofs of Optimization Dynamics in Phase I

In this section, we conduct a detailed analysis of the training dynamics of each neuron in Phase I. The main proof idea is to decompose neurons' dynamics into tangential and radial dynamics. For small initialization, in Phase I, the radial increasing of neurons is much slower than their tangential velocity, which can result in condensation. However, the main challenges arise from the initial direction's randomness and ReLU's discontinuous derivative, leading to eight categories of neuron dynamics. Moreover, it is also nontrivial and requires meticulous analysis to estimate the number of two classes of living neurons at the end of Phase I.

We define the time

$$T_{\mathrm{I}} := 10\sqrt{\frac{\kappa_1}{\kappa_2}}, \tag{5}$$

and call $t \in [0, T_{\mathrm{I}}]$ "Phase I".

Recall that the selection (3) about initialization parameters can guarantee the **whole** four-phase optimization dynamics. Nevertheless, when we focus on Phase I, $\kappa_2 = \mathcal{O}(1)$ and $\kappa_1/\kappa_2 = \mathcal{O}(1)$ suffice to ensure the dynamics in Phase I. Specifically, during the proof of Phase I, we can use the following selection (6) on $\kappa_1, \kappa_2$, which is much weaker than (3):

$$\kappa_2 = \mathcal{O}(1), \quad \frac{\kappa_1}{\kappa_2} = \mathcal{O}(1). \tag{6}$$

For simplicity, we assume $\Delta \leq \frac{1}{5}$. And for convenience, we assume $p \geq 5$. It is worth mentioning that our proof approach also applies for $p = 2, 3, 4$, with at most one absolute constant difference.

Prepared for the analysis in Phase I, we decompose the dynamics of $\boldsymbol{b}_k(t)$ into the radial movement $\rho_k(t) \in \mathbb{R}$ and the tangential movement $\boldsymbol{w}_k(t) \in \mathbb{S}^{d-1}$ satisfied to $\boldsymbol{b}_k(t) = \rho_k(t)\boldsymbol{w}_k(t)$.

**Lemma C.1** (Dynamics decomposition). *For any $k \in [m]$, the dynamics of $\boldsymbol{b}_k(t)$ can be decomposed into the radial movement $\rho_k(t) \in \mathbb{R}$ and the tangential movement $\boldsymbol{w}_k(t) \in \mathbb{S}^{d-1}$:*

$$\begin{aligned}
\frac{\mathrm{d}\boldsymbol{w}_k(t)}{\mathrm{d}t} &\in \frac{\mathrm{s}_k\kappa_2}{\sqrt{m}\rho_k(t)}\Big(\boldsymbol{F}_k(t) - \langle \boldsymbol{F}_k(t), \boldsymbol{w}_k(t)\rangle\,\boldsymbol{w}_k(t)\Big), \\
\frac{\mathrm{d}\rho_k(t)}{\mathrm{d}t} &\in \frac{\mathrm{s}_k\kappa_2}{\sqrt{m}}\,\langle \boldsymbol{F}_k(t), \boldsymbol{w}_k(t)\rangle,
\end{aligned} \tag{7}$$

*where $\rho_k(t) = \|\boldsymbol{b}_k(t)\|$, $\boldsymbol{w}_k(t) = \boldsymbol{b}_k(t)/\|\boldsymbol{b}_k(t)\|$ and*

$$\boldsymbol{F}_k(t) = \frac{1}{n}\sum_{i=1}^n e^{-y_i f_i(t)}\partial^\circ \sigma\left(\langle \boldsymbol{w}_k(t), \boldsymbol{x}_i\rangle\right)y_i\boldsymbol{x}_i. \tag{8}$$

*Proof of Lemma C.1.*
From the dynamics of $\boldsymbol{b}_k(t)$:

$$\frac{\mathrm{d}\boldsymbol{b}_k(t)}{\mathrm{d}t} \in \frac{\mathrm{s}_k\kappa_2}{\sqrt{m}}\boldsymbol{F}_k(t) = \frac{\mathrm{s}_k\kappa_2}{\sqrt{m}}\frac{1}{n}\sum_{i=1}^n e^{-y_i f_i(t)}\partial^\circ \sigma\left(\langle \boldsymbol{w}_k(t), \boldsymbol{x}_i\rangle\right)y_i\boldsymbol{x}_i,$$

we have

$$\begin{aligned}
\frac{\mathrm{d}\rho_k(t)}{\mathrm{d}t} &= \frac{1}{2\|\boldsymbol{b}_k(t)\|}\frac{\mathrm{d}\|\boldsymbol{b}_k(t)\|^2}{\mathrm{d}t} = \frac{1}{\|\boldsymbol{b}_k(t)\|}\left\langle \boldsymbol{b}_k(t), \frac{\mathrm{d}\boldsymbol{b}_k(t)}{\mathrm{d}t}\right\rangle \\
&\in \frac{1}{\|\boldsymbol{b}_k(t)\|}\left\langle \boldsymbol{b}_k(t), \frac{\mathrm{s}_k\kappa_2}{\sqrt{m}}\boldsymbol{F}_k(t)\right\rangle = \frac{\mathrm{s}_k\kappa_2}{\sqrt{m}}\,\langle \boldsymbol{F}_k(t), \boldsymbol{w}_k(t)\rangle,
\end{aligned}$$

$$\begin{aligned}
\frac{\mathrm{d}\boldsymbol{w}_k(t)}{\mathrm{d}t} &= \frac{\|\boldsymbol{b}_k(t)\|\frac{\mathrm{d}\boldsymbol{b}_k(t)}{\mathrm{d}t} - \frac{\mathrm{d}\|\boldsymbol{b}_k(t)\|}{\mathrm{d}t}\boldsymbol{b}_k(t)}{\|\boldsymbol{b}_k(t)\|^2} = \frac{1}{\rho_k(t)}\left(\frac{\mathrm{d}\boldsymbol{b}_k(t)}{\mathrm{d}t} - \frac{\mathrm{d}\rho_k(t)}{\mathrm{d}t}\boldsymbol{w}_k(t)\right) \\
&\in \frac{1}{\rho_k(t)}\left(\frac{\mathrm{s}_k\kappa_2}{\sqrt{m}}\boldsymbol{F}_k(t) - \frac{\mathrm{s}_k\kappa_2}{\sqrt{m}}\langle \boldsymbol{F}_k(t), \boldsymbol{w}_k(t)\rangle\boldsymbol{w}_k(t)\right) = \frac{\mathrm{s}_k\kappa_2}{\sqrt{m}\rho_k(t)}\Big(\boldsymbol{F}_k(t) - \langle \boldsymbol{F}_k(t), \boldsymbol{w}_k(t)\rangle\,\boldsymbol{w}_k(t)\Big).
\end{aligned}$$

$\square$

Prepared for the analysis of the neurons' dynamics, we establish a rough estimate about the norm and prediction growth of each neuron in Phase I, and we will improve it later.

**Lemma C.2** (A Rough Estimate of Norm and Prediction in Phase I)**.**
*For any $t \leq T_{\mathrm{I}}$, $k \in [m]$, and $i \in [n]$, we have the following estimates:*

$$\rho_k(t) \leq \frac{\kappa_1 + 1.1\kappa_2 t}{\sqrt{m}},$$
$$|f_i(t)| \leq \sqrt{\kappa_1 \kappa_2},$$
$$\left| e^{-y_i f_i(t)} - 1 \right| \leq 1.1\sqrt{\kappa_1 \kappa_2}.$$

*Proof of Lemma C.2.*
First, we define the hitting time

$$T_{\sqrt{\kappa_1 \kappa_2}} := \inf \left\{ t > 0 : \max_{i \in [n]} |f_i(t)| > \sqrt{\kappa_1 \kappa_2} \right\}.$$

From $|f_i(0)| \leq m \frac{\kappa_2}{\sqrt{m}} \frac{\kappa_1}{\sqrt{m}} = \kappa_1 \kappa_2 < \sqrt{\kappa_1 \kappa_2}$ and the continuity of $f_i(\cdot)$, we know $T_{\sqrt{\kappa_1 \kappa_2}} > 0$. Then we will prove $T_{\mathrm{I}} = 10\sqrt{\frac{\kappa_1}{\kappa_2}} \leq T_{\sqrt{\kappa_1 \kappa_2}}$.

For any $k \in [m]$, $i \in [n]$, and $t \leq T_{\sqrt{\kappa_1 \kappa_2}}$, we have the following estimates.

Recalling the definition of $\boldsymbol{F}_k(t)$ (8), we have

$$\|\boldsymbol{F}_k(t)\| = \left\| \frac{1}{n} \sum_{i=1}^{n} e^{-y_i f_i(t)} \partial^{\circ}\sigma\left(\langle \boldsymbol{w}_k(t), \boldsymbol{x}_i \rangle\right) y_i \boldsymbol{x}_i \right\| \leq \max_{i \in [n]} \left\| e^{-y_i f_i(t)} \right\|$$
$$\leq e^{\max_{i \in [n]} |f_i(t)|} \leq e^{\sqrt{\kappa_1 \kappa_2}} \leq 1.1.$$

Recalling the dynamics (7), for any $t \leq T_{\sqrt{\kappa_1 \kappa_2}}$,

$$\frac{\mathrm{d}\rho_k(t)}{\mathrm{d}t} \leq \frac{\kappa_2}{\sqrt{m}} \|\langle \boldsymbol{F}_k(t), \boldsymbol{w}_k(t) \rangle\| \leq \frac{\kappa_2}{\sqrt{m}} \|\boldsymbol{F}_k(t)\| \leq \frac{1.1\kappa_2}{\sqrt{m}}.$$

Then combining $\rho_k(0) = \frac{\kappa_1}{\sqrt{m}}$, for any $t \leq T_{\sqrt{\kappa_1 \kappa_2}}$,

$$\rho_k(t) \leq \frac{\kappa_1 + 1.1\kappa_2 t}{\sqrt{m}},$$
$$|f_i(t)| \leq \sum_{k=1}^{m} |a_k| \rho_k(t) \leq m \frac{\kappa_2}{\sqrt{m}} \frac{\kappa_1 + 1.1\kappa_2 t}{\sqrt{m}} \leq \kappa_2\left(\kappa_1 + 1.1\kappa_2 t\right). \tag{9}$$

So for any $t \leq T_{\mathrm{I}}$, we have

$$|f_i(t)| \leq \kappa_2\left(\kappa_1 + 1.1\kappa_2 T_{\mathrm{I}}\right) \leq \kappa_2\left(\kappa_1 + 11\sqrt{\kappa_1 \kappa_2}\right) \leq 12\kappa_2\sqrt{\kappa_1 \kappa_2} \leq \sqrt{\kappa_1 \kappa_2}. \tag{10}$$

From the definition of $T_{\sqrt{\kappa_1 \kappa_2}}$, we have proved $T_{\mathrm{I}} \leq T_{\sqrt{\kappa_1 \kappa_2}}$.

Moreover, from this proof, we know (9) holds for any $t \leq T_{\mathrm{I}}$. Moreover, by Mean Value Theorem,

$$\left| e^{-y_i f_i(t)} - 1 \right| \leq \left( \max_{z \in [0, \sqrt{\kappa_1 \kappa_2}]} e^z \right) | - y_i f_i(t)| \leq 1.1|f_i(t)| \leq 1.1\sqrt{\kappa_1 \kappa_2}.$$

$\square$

Now we delve into the optimization dynamics of all neurons in Phase I in the following Section C.1 and C.2.

## C.1 The Dynamics of Positive Neurons

According to the initial direction, all positive neurons ($s_k = 1$) can be divided into the following four classes.

$$[m/2] = \left\{ k \in [m/2] : \boldsymbol{w}_k(0) \in \mathcal{M}_+^+ \cap \mathcal{M}_-^+ \right\} \bigcup \left\{ k \in [m/2] : \boldsymbol{w}_k(0) \in \mathcal{M}_+^- \cap \mathcal{M}_-^+ \right\}$$
$$\bigcup \left\{ k \in [m/2] : \boldsymbol{w}_k(0) \in \mathcal{M}_+^+ \cap \mathcal{M}_-^- \right\} \bigcup \left\{ k \in [m/2] : \boldsymbol{w}_k(0) \in \mathcal{M}_+^- \cap \mathcal{M}_-^- \right\}.$$

In the following four lemmas, we will prove the dynamics for these four classes of positive neurons. In summary, in Phase I ($t < T_{\mathrm{I}}$), some of positive neurons align well with the direction $\boldsymbol{\mu}$, and their norms experiment a small but significant increase, while other positive neurons go dead.

**Lemma C.3** (Positive, $\mathcal{M}_+^+ \cap \mathcal{M}_-^+$).
*For positive neuron $k \in \{ k \in [m/2] : \boldsymbol{w}_k(0) \in \mathcal{M}_+^+ \cap \mathcal{M}_-^+ \}$, at the end of Phase I, it holds that*

**(Direction).** *It is aligned with $\boldsymbol{\mu}$ :* $\langle \boldsymbol{w}_k(T_{\mathrm{I}}), \boldsymbol{\mu} \rangle \geq \left( 1 - \mathcal{O}(\sqrt{\kappa_1 \kappa_2}) \right) \left( 1 - \mathcal{O}\left( (\frac{\kappa_1}{\kappa_2})^{0.55} \right) \right)$ ;

**(Norm).** *It has a small but significant norm :* $\rho_k(T_{\mathrm{I}}) = \Theta\left( \frac{\sqrt{\kappa_1 \kappa_2}}{\sqrt{m}} \right)$.

*Proof of Lemma C.3.*
We do the following analysis for any $k \in \{ k \in [m/2] : \boldsymbol{w}_k(0) \in \mathcal{M}_+^+ \cap \mathcal{M}_-^+ \}$, i.e. $s_k = 1$, $\langle \boldsymbol{w}_k(0), \boldsymbol{x}_+ \rangle > 0$, and $\langle \boldsymbol{w}_k(0), \boldsymbol{x}_- \rangle > 0$.

**Step I.** The neuron stays in $\mathcal{M}_+^+ \cap \mathcal{M}_-^+$ for any $t \leq T_{\mathrm{I}}$.

First, we define the hitting time

$$T_{\mathrm{hit}} := \inf \left\{ t \in (0, T_{\mathrm{I}}] : \boldsymbol{w}_k(t) \notin \partial(\mathcal{M}_+^+ \cap \mathcal{M}_-^+) \right\},$$

and we aim to prove that $T_{\mathrm{hit}}$ does not exist. From the definition of $T_{\mathrm{hit}}$ and (4), the dynamics of the neuron is:

$$\frac{\mathrm{d}\boldsymbol{b}_k(t)}{\mathrm{d}t} = \frac{\kappa_2}{\sqrt{m}} \left( \frac{p}{1+p} e^{-f_+(t)} \boldsymbol{x}_+ - \frac{1}{1+p} e^{f_-(t)} \boldsymbol{x}_- \right), \ t \leq T_{\mathrm{hit}}.$$

From $T_{\mathrm{hit}} \leq T_{\mathrm{I}}$ and Lemma C.2, we have $\left| e^{-y_i f_i(t)} - 1 \right| \leq 1.1\sqrt{\kappa_1 \kappa_2}$ for any $t \leq T_{\mathrm{hit}}$. Then we have:

$$\frac{\mathrm{d} \langle \boldsymbol{b}_k(t), \boldsymbol{x}_+ \rangle}{\mathrm{d}t} = \frac{\kappa_2}{\sqrt{m}} \left( \frac{p e^{-f_+(t)}}{1+p} - \frac{e^{f_-(t)}}{1+p} \cos \Delta \right)$$
$$\geq \frac{\kappa_2}{\sqrt{m}} \left( \frac{p(1 - 1.1\sqrt{\kappa_1 \kappa_2})}{1+p} - \frac{(1 + 1.1\sqrt{\kappa_1 \kappa_2}) \cos \Delta}{1+p} \right) > 0,$$

$$\frac{\mathrm{d} \langle \boldsymbol{b}_k(t), \boldsymbol{x}_- \rangle}{\mathrm{d}t} = \frac{\kappa_2}{\sqrt{m}} \left( \frac{p e^{-f_+(t)} \cos \Delta}{1+p} - \frac{e^{f_-(t)}}{1+p} \right)$$
$$\geq \frac{\kappa_2}{\sqrt{m}} \left( \frac{p(1 - 1.1\sqrt{\kappa_1 \kappa_2}) \cos \Delta}{1+p} - \frac{1 + 1.1\sqrt{\kappa_1 \kappa_2}}{1+p} \right) > 0.$$

Hence, for any $t \leq T_{\mathrm{hit}}$, $\langle \boldsymbol{b}_k(t), \boldsymbol{x}_+ \rangle > \langle \boldsymbol{b}_k(0), \boldsymbol{x}_+ \rangle > 0$ and $\langle \boldsymbol{b}_k(t), \boldsymbol{x}_- \rangle > \langle \boldsymbol{b}_k(0), \boldsymbol{x}_- \rangle > 0$. According to the definition of $T_{\mathrm{hit}}$, we have proved that $T_{\mathrm{hit}}$ does not exist, which means the neuron stays in $\mathcal{M}_+^+ \cap \mathcal{M}_-^+$ for any $t \leq T_{\mathrm{I}}$.

**Step II.** Estimate the evolution of $\langle \boldsymbol{w}_k(t), \boldsymbol{\mu} \rangle$.

With the help of **Step I**, we were able to determine the dynamics for $t \leq T_{\mathrm{I}}$. For any $t \leq T_{\mathrm{I}}$, we can do the following estimate.

From (7), the tangential dynamics of the neuron is

$$\frac{\mathrm{d}\boldsymbol{w}_k(t)}{\mathrm{d}t} = \frac{\kappa_2}{\sqrt{m}\rho_k(t)}\Big(\boldsymbol{F}_k(t) - \langle \boldsymbol{F}_k(t), \boldsymbol{w}_k(t)\rangle\,\boldsymbol{w}_k(t)\Big),$$

$$\text{where}\quad \boldsymbol{F}_k(t) := \frac{p}{1+p}e^{-f_+(t)}\boldsymbol{x}_+ - \frac{1}{1+p}e^{f_-(t)}\boldsymbol{x}_-.$$

Recalling the definitions of $\boldsymbol{\mu}$ and $\boldsymbol{z}$, we can estimate the difference between $\boldsymbol{F}_k(t)$ and $\boldsymbol{z}$:

$$\langle \boldsymbol{F}_k(t), \boldsymbol{z}\rangle = \Big\langle \boldsymbol{z} + \frac{p}{1+p}\big(e^{-f_+(t)}-1\big)\boldsymbol{x}_+ - \frac{1}{1+p}\big(e^{f_-(t)}-1\big)\boldsymbol{x}_-, \boldsymbol{z}\Big\rangle$$

$$= \|\boldsymbol{z}\|^2 + \Big\langle \frac{p}{1+p}\big(e^{-f_+(t)}-1\big)\boldsymbol{x}_+ - \frac{1}{1+p}\big(e^{f_-(t)}-1\big)\boldsymbol{x}_-, \frac{p}{1+p}\boldsymbol{x}_+ - \frac{1}{1+p}\boldsymbol{x}_-\Big\rangle$$

$$\geq \|\boldsymbol{z}\|^2 - \frac{p^2}{(1+p)^2}\Big|e^{-f_+(t)}-1\Big| - \frac{1}{(1+p)^2}\Big|e^{f_-(t)}-1\Big| - \frac{p\cos\Delta}{(1+p)^2}\Big(\Big|e^{-f_+(t)}-1\Big| + \Big|e^{f_-(t)}-1\Big|\Big)$$

$$\overset{\text{Lemma C.2}}{\geq} \|\boldsymbol{z}\|^2 - 1.1\sqrt{\kappa_1\kappa_2}\Big(\frac{p^2}{(1+p)^2} + \frac{1}{(1+p)^2} + \frac{2p\cos\Delta}{(1+p)^2}\Big) \geq \|\boldsymbol{z}\|^2 - 1.1\sqrt{\kappa_1\kappa_2},$$

$$\|\boldsymbol{F}_k(t) - \boldsymbol{z}\| = \Big\|\frac{p}{1+p}\big(e^{-f_+(t)}-1\big)\boldsymbol{x}_+ - \frac{1}{1+p}\big(e^{f_-(t)}-1\big)\boldsymbol{x}_-\Big\|$$

$$\leq \Big\|\frac{p}{1+p}\big(e^{-f_+(t)}-1\big)\boldsymbol{x}_+\Big\| + \Big\|\frac{1}{1+p}\big(e^{f_-(t)}-1\big)\boldsymbol{x}_-\Big\|$$

$$\overset{\text{Lemma C.2}}{\leq} 1.1\sqrt{\kappa_1\kappa_2}\frac{p}{1+p} + 1.1\sqrt{\kappa_1\kappa_2}\frac{1}{1+p} = 1.1\sqrt{\kappa_1\kappa_2}.$$

The dynamics of $\langle \boldsymbol{w}_k(t), \boldsymbol{\mu}\rangle$ is:

$$\frac{\mathrm{d}\langle \boldsymbol{w}_k(t), \boldsymbol{\mu}\rangle}{\mathrm{d}t} = \frac{1}{\|\boldsymbol{z}\|}\frac{\mathrm{d}\langle \boldsymbol{w}_k(t), \boldsymbol{z}\rangle}{\mathrm{d}t} = \frac{\kappa_2}{\|\boldsymbol{z}\|\sqrt{m}\rho_k(t)}\Big(\langle \boldsymbol{F}_k(t), \boldsymbol{z}\rangle - \langle \boldsymbol{F}_k(t), \boldsymbol{w}_k(t)\rangle\langle \boldsymbol{w}_k(t), \boldsymbol{z}\rangle\Big).$$

And it can be estimated by:

$$\frac{\mathrm{d}\langle \boldsymbol{w}_k(t), \boldsymbol{\mu}\rangle}{\mathrm{d}t}$$

$$= \frac{\kappa_2}{\|\boldsymbol{z}\|\sqrt{m}\rho_k(t)}\Big(\langle \boldsymbol{F}_k(t), \boldsymbol{z}\rangle - \langle \boldsymbol{z}, \boldsymbol{w}_k(t)\rangle\langle \boldsymbol{w}_k(t), \boldsymbol{z}\rangle - \langle \boldsymbol{F}_k(t)-\boldsymbol{z}, \boldsymbol{w}_k(t)\rangle\langle \boldsymbol{w}_k(t), \boldsymbol{z}\rangle\Big)$$

$$\geq \frac{\kappa_2}{\|\boldsymbol{z}\|\sqrt{m}\rho_k(t)}\Big(\langle \boldsymbol{F}_k(t), \boldsymbol{z}\rangle - \langle \boldsymbol{w}_k(t), \boldsymbol{z}\rangle^2 - \|\boldsymbol{F}_k(t)-\boldsymbol{z}\|\,\|\boldsymbol{z}\|\Big)$$

$$\geq \frac{\kappa_2}{\|\boldsymbol{z}\|\sqrt{m}\rho_k(t)}\Big(\|\boldsymbol{z}\|^2 - 1.1\sqrt{\kappa_1\kappa_2} - \langle \boldsymbol{w}_k(t), \boldsymbol{z}\rangle^2 - 1.1\sqrt{\kappa_1\kappa_2}\|\boldsymbol{z}\|\Big)$$

$$= \frac{\kappa_2}{\|\boldsymbol{z}\|\sqrt{m}\rho_k(t)}\Big(\|\boldsymbol{z}\|^2 - \|\boldsymbol{z}\|^2\langle \boldsymbol{w}_k(t), \boldsymbol{\mu}\rangle^2 - 1.1\sqrt{\kappa_1\kappa_2}\big(1 + \|\boldsymbol{z}\|\big)\Big)$$

$$= \frac{\kappa_2\|\boldsymbol{z}\|}{\sqrt{m}\rho_k(t)}\Big(1 - \langle \boldsymbol{w}_k(t), \boldsymbol{\mu}\rangle^2 - \frac{1.1\sqrt{\kappa_1\kappa_2}\big(1 + \|\boldsymbol{z}\|\big)}{\|\boldsymbol{z}\|^2}\Big)$$

$$\overset{\text{Lemma I.3}}{\geq} \frac{2\kappa_2}{3\sqrt{m}\rho_k(t)}\Big(1 - \langle \boldsymbol{w}_k(t), \boldsymbol{\mu}\rangle^2 - 1.1\sqrt{\kappa_1\kappa_2}\big(1.5 + 1.5^2\big)\Big) > \frac{2\kappa_2}{3\sqrt{m}\rho_k(t)}\Big(1 - 4.2\sqrt{\kappa_1\kappa_2} - \langle \boldsymbol{w}_k(t), \boldsymbol{\mu}\rangle^2\Big)$$

$$\overset{\text{Lemma C.2}}{\geq} \frac{2\kappa_2}{3\sqrt{m}\frac{\kappa_1+1.1\kappa_2 t}{\sqrt{m}}}\Big(1 - 4.2\sqrt{\kappa_1\kappa_2} - \langle \boldsymbol{w}_k(t), \boldsymbol{\mu}\rangle^2\Big) = \frac{2}{3\big(\frac{\kappa_1}{\kappa_2}+1.1t\big)}\Big(1 - 4.2\sqrt{\kappa_1\kappa_2} - \langle \boldsymbol{w}_k(t), \boldsymbol{\mu}\rangle^2\Big).$$

$$\tag{11}$$

Noticing $\boldsymbol{w}_k(0) \in \mathcal{M}_+^+ \cap \mathcal{M}_-^+$, we have:

$$\langle \boldsymbol{w}_k(0), \boldsymbol{\mu}\rangle = \Big\langle \boldsymbol{w}_k(0), \frac{p}{1+p}\boldsymbol{x}_+ - \frac{1}{1+p}\boldsymbol{x}_-\Big\rangle > -\Big\langle \boldsymbol{w}_k(0), \frac{1}{1+p}\boldsymbol{x}_-\Big\rangle \geq -\frac{1}{1+p}. \tag{12}$$

Now we consider the following auxiliary ODE:

$$\begin{cases} \frac{dU(t)}{dt} = \frac{2}{3\left(\frac{\kappa_1}{\kappa_2}+1.1t\right)}\left(1 - 4.2\sqrt{\kappa_1\kappa_2} - U^2(t)\right) \\ U(0) = -\frac{1}{1+p} \end{cases}, \tag{13}$$

and let $U(t)$ is the solution of (13). Due to (11) (12), we know that $\langle \boldsymbol{w}_k(t), \boldsymbol{\mu} \rangle$ is an upper solution of ODE (13). From the Comparison Principle of ODEs, we know this means:

$$\langle \boldsymbol{w}_k(t), \boldsymbol{\mu} \rangle > U(t), \text{ for any } t \leq T_{\mathrm{I}}.$$

Hence, in order to estimate $\langle \boldsymbol{w}_k(t), \boldsymbol{\mu} \rangle$, we only need to study the solution of ODE (13). It is easy to verify that the solution of (13) satisfies

$$\log\left(\frac{1 - 4.2\sqrt{\kappa_1\kappa_2} + U(t)}{1 - 4.2\sqrt{\kappa_1\kappa_2} - U(t)}\right) - \log\left(\frac{1 - 4.2\sqrt{\kappa_1\kappa_2} - \frac{1}{1+p}}{1 - 4.2\sqrt{\kappa_1\kappa_2} + \frac{1}{1+p}}\right) = \frac{4(1 - 4.2\sqrt{\kappa_1\kappa_2})}{3.3}\log\left(1 + \frac{1.1\kappa_2}{\kappa_1}t\right).$$

Then we have:

$$\log\left(\frac{1 - 4.2\sqrt{\kappa_1\kappa_2} + U(t)}{1 - 4.2\sqrt{\kappa_1\kappa_2} - U(t)}\right) \geq \log\left(\frac{1 - 4.2\sqrt{\kappa_1\kappa_2} - \frac{1}{6}}{1 - 4.2\sqrt{\kappa_1\kappa_2} + \frac{1}{6}}\right) + \frac{4(1 - 4.2\sqrt{\kappa_1\kappa_2})}{3.3}\log\left(1 + \frac{1.1\kappa_2}{\kappa_1}t\right)$$

$$\geq \log\left(0.7\left(1 + \frac{1.1\kappa_2}{\kappa_1}t\right)^{1.15}\right) \geq \log\left(0.7\left(1 + \frac{1.1\kappa_2}{\kappa_1}t\right)^{1.15}\right),$$

which means

$$U(t) \geq \left(1 - 4.2\sqrt{\kappa_1\kappa_2}\right)\left(1 - \frac{2}{1 + 0.7\left(1 + \frac{1.1\kappa_2}{\kappa_1}t\right)^{1.15}}\right).$$

Hence, we have the estimate of $\langle \boldsymbol{w}_k(t), \boldsymbol{\mu} \rangle$:

$$\langle \boldsymbol{w}_k(t), \boldsymbol{\mu} \rangle \geq \left(1 - 4.2\sqrt{\kappa_1\kappa_2}\right)\left(1 - \frac{2}{1 + 0.7\left(1 + \frac{1.1\kappa_2}{\kappa_1}t\right)^{1.15}}\right). \tag{14}$$

Specifically, we have:

$$\langle \boldsymbol{w}_k(T_{\mathrm{I}}), \boldsymbol{\mu} \rangle \geq \left(1 - 4.2\sqrt{\kappa_1\kappa_2}\right)\left(1 - \frac{2}{1 + 0.7\left(1 + 11\sqrt{\frac{\kappa_2}{\kappa_1}}\right)^{1.15}}\right).$$

**Step III.** A finer estimate of $\rho_k(T_{\mathrm{I}})$.

In this step, we will estimate the lower bound and upper bound for $\rho_k(T_{\mathrm{I}})$.

First, lemma C.2 gives us the upper bound for $\rho_k(T_{\mathrm{I}})$:

$$\rho_k(T_{\mathrm{I}}) \leq \frac{\kappa_1 + 1.1\kappa_2 T_{\mathrm{I}}}{\sqrt{m}} \leq \frac{\kappa_1 + 11\sqrt{\kappa_1\kappa_2}}{\sqrt{m}} \leq \frac{12\sqrt{\kappa_1\kappa_2}}{\sqrt{m}}.$$

Now we focus on the estimate of the lower bound. Recalling the dynamics of $\rho_k(t)$ (7), for any $t \leq T_{\mathrm{I}}$,

$$\frac{d\rho_k(t)}{dt} = \frac{\kappa_2}{\sqrt{m}}\langle \boldsymbol{F}_k(t), \boldsymbol{w}_k(t) \rangle = \frac{\kappa_2}{\sqrt{m}}\left(\langle \boldsymbol{z}, \boldsymbol{w}_k(t) \rangle + \langle \boldsymbol{F}_k(t) - \boldsymbol{z}, \boldsymbol{w}_k(t) \rangle\right)$$

$$\geq \frac{\kappa_2}{\sqrt{m}}\left(\langle \boldsymbol{z}, \boldsymbol{w}_k(t) \rangle - \|\boldsymbol{F}_k(t) - \boldsymbol{z}\|\right) \geq \frac{\kappa_2}{\sqrt{m}}\left(\|\boldsymbol{z}\|\langle \boldsymbol{w}_k(t), \boldsymbol{\mu} \rangle - 1.1\sqrt{\kappa_1\kappa_2}\right)$$

$$\overset{(14)}{\geq} \frac{\kappa_2}{\sqrt{m}}\left(\frac{2\left(1 - 4.2\sqrt{\kappa_1\kappa_2}\right)}{3}\left(1 - \frac{2}{1 + 0.7\left(1 + \frac{1.1\kappa_2}{\kappa_1}t\right)^{1.15}}\right) - 1.1\sqrt{\kappa_1\kappa_2}\right)$$

$$\geq \frac{\kappa_2}{\sqrt{m}}\left(0.627 - \frac{1.278}{1 + 0.7\left(1 + \frac{1.1\kappa_2}{\kappa_1}t\right)^{1.15}}\right).$$

We denote $T_0 = \frac{10\kappa_1}{\kappa_2}$, and it is easy to verify:

$$0.627 - \frac{1.278}{1 + 0.7\left(1 + \frac{1.1\kappa_2}{\kappa_1}t\right)^{1.15}} > 0.53, \text{ for any } t \in [T_0, T_\mathrm{I}].$$

So we have:

$$\rho_k(T_\mathrm{I}) \geq \rho_k(T_0) + \int_{T_0}^{T_\mathrm{I}} \frac{\kappa_2}{\sqrt{m}}\left(0.627 - \frac{1.278}{1 + 0.7\left(1 + \frac{1.1\kappa_2}{\kappa_1}t\right)^{1.15}}\right)\mathrm{d}t$$

$$> 0 + \int_{\frac{10\kappa_1}{\kappa_2}}^{T_\mathrm{I}} \frac{0.53\kappa_2}{\sqrt{m}}\mathrm{d}t = \frac{0.53(10\sqrt{\frac{\kappa_1}{\kappa_2}} - \frac{10\kappa_1}{\kappa_2})\kappa_2}{\sqrt{m}} \overset{(6)}{\geq} \frac{5.3 \cdot 0.9\sqrt{\kappa_1\kappa_2}}{\sqrt{m}} \geq \frac{4.77\sqrt{\kappa_1\kappa_2}}{\sqrt{m}}.$$

$\square$

**Lemma C.4** (Positive, $\mathcal{M}_+^+ \cap \mathcal{M}_-^-$).
*For positive neuron $k \in \{k \in [m/2] : \boldsymbol{w}_k(0) \in \mathcal{M}_+^+ \cap \mathcal{M}_-^-\}$, at the end of Phase I, it holds that*

**(Direction).** *It is aligned with $\boldsymbol{\mu}$ : $\langle \boldsymbol{w}_k(T_\mathrm{I}), \boldsymbol{\mu}\rangle \geq \left(1 - \mathcal{O}(\sqrt{\kappa_1\kappa_2})\right)\left(1 - \mathcal{O}\left((\frac{\kappa_1}{\kappa_2})^{0.55}\right)\right)$;*

**(Norm).** *It has a small but significant norm : $\rho_k(T_\mathrm{I}) = \Theta\left(\frac{\sqrt{\kappa_1\kappa_2}}{\sqrt{m}}\right)$.*

*Proof of Lemma C.4.*
We do the following analysis for any $k \in \{k \in [m/2] : \boldsymbol{w}_k(0) \in \mathcal{M}_+^+ \cap \mathcal{M}_-^-\}$, i.e. $\mathrm{s}_k = 1$, $\langle \boldsymbol{w}_k(0), \boldsymbol{x}_+\rangle > 0$, and $\langle \boldsymbol{w}_k(0), \boldsymbol{x}_-\rangle \leq 0$.

**Step I.** The neuron must arrives in $\mathcal{M}_+^+ \cap \mathcal{M}_-^0$ in $\mathcal{O}\left(\frac{\kappa_1\Delta}{\kappa_2}\right)$ time.

The case $\langle \boldsymbol{w}_k(0), \boldsymbol{x}_-\rangle = 0$ is trivial. Then we only need to consider the case $\langle \boldsymbol{w}_k(0), \boldsymbol{x}_-\rangle < 0$.

First, we define the hitting time

$$T_\mathrm{hit} := \inf\left\{t \in (0, T_\mathrm{I}] : \boldsymbol{w}_k(t) \notin \mathcal{M}_+^+ \cap \mathcal{M}_-^-\right\},$$

and we aim to estimate $T_\mathrm{hit}$ and prove $\boldsymbol{w}_k(T_\mathrm{hit}) \in \mathcal{M}_+^+ \cap \mathcal{M}_-^0$.

We focus on the dynamics of $\langle \boldsymbol{b}_k(t), \boldsymbol{x}_+\rangle$ and $\langle \boldsymbol{b}_k(t), \boldsymbol{x}_-\rangle$.

From the definition of $T_\mathrm{hit}$ and (4), the dynamics of the neuron is:

$$\frac{\mathrm{d}\boldsymbol{b}_k(t)}{\mathrm{d}t} = \frac{\kappa_2}{\sqrt{m}}\frac{p}{1+p}e^{-f_+(t)}\boldsymbol{x}_+, \ t \leq T_\mathrm{hit}.$$

Then we have

$$\frac{\mathrm{d}\langle \boldsymbol{b}_k(t), \boldsymbol{x}_+\rangle}{\mathrm{d}t} = \left\langle \frac{\kappa_2}{\sqrt{m}}\frac{p}{1+p}e^{-f_+(t)}\boldsymbol{x}_+, \boldsymbol{x}_+\right\rangle = \frac{\kappa_2 p}{\sqrt{m}(1+p)}e^{-f_+(t)},$$

$$\frac{\mathrm{d}\langle \boldsymbol{b}_k(t), \boldsymbol{x}_-\rangle}{\mathrm{d}t} = \left\langle \frac{\kappa_2}{\sqrt{m}}\frac{p}{1+p}e^{-f_+(t)}\boldsymbol{x}_+, \boldsymbol{x}_-\right\rangle = \frac{\kappa_2 p \cos\Delta}{\sqrt{m}(1+p)}e^{-f_+(t)}.$$

It is clear $\frac{\mathrm{d}\langle \boldsymbol{b}_k(t), \boldsymbol{x}_+\rangle}{\mathrm{d}t} > 0$, so $\langle \boldsymbol{b}_k(t), \boldsymbol{x}_+\rangle > \langle \boldsymbol{b}_k(0), \boldsymbol{x}_+\rangle > 0$ for any $t \leq T_\mathrm{hit}$. If we denote

$$T_{\mathrm{hit},-} := \inf\left\{t \in (0, T_\mathrm{I}] : \langle \boldsymbol{w}_k(t), \boldsymbol{x}_-\rangle \geq 0\right\},$$

then it holds:
$$T_{\text{hit}} = T_{\text{hit},-}.$$

So we only need to estimate $T_{\text{hit},-}$. Due to $T_{\text{hit}} \leq T_{\text{I}} \leq T_{\text{init}}$ and Lemma C.2, for any $t \leq T_{\text{hit}}$, we have $\left|e^{-y_i f_i(t)} - 1\right| \leq 0.11$. Then for any $t \leq T_{\text{hit}}$, we have:

$$\frac{\mathrm{d} \langle \boldsymbol{b}_k(t), \boldsymbol{x}_- \rangle}{\mathrm{d}t} \geq \frac{0.89 \kappa_2 p \cos \Delta}{\sqrt{m}(1+p)}.$$

Recalling $\langle \boldsymbol{w}_k(0), \boldsymbol{x}_+ \rangle > 0$ and $\langle \boldsymbol{w}_k(0), \boldsymbol{x}_- \rangle < 0$, with the help of Lemma I.2, we have $\langle \boldsymbol{w}_k(0), \boldsymbol{x}_- \rangle > -\sin \Delta$. Combining the two estimate, we have:

$$\langle \boldsymbol{b}_k(t), \boldsymbol{x}_- \rangle \geq \langle \boldsymbol{b}_k(0), \boldsymbol{x}_- \rangle + \int_0^t \frac{0.89 \kappa_2 p \cos \Delta}{\sqrt{m}(1+p)} \mathrm{d}t$$

$$> -\rho_k(0) \sin \Delta + \frac{0.89 \kappa_2 p \cos \Delta}{\sqrt{m}(1+p)}t = -\frac{\kappa_1 \sin \Delta}{\sqrt{m}} + \frac{0.89 \kappa_2 p \cos \Delta}{\sqrt{m}(1+p)}t.$$

Hence,
$$T_{\text{hit}} = T_{\text{hit},-} \leq \frac{(1+p)\tan \Delta}{0.89p} \frac{\kappa_1}{\kappa_2} \leq 2\Delta \frac{\kappa_1}{\kappa_2}.$$

**Step II.** Dynamics after arriving in the manifold $\mathcal{M}_+^+ \cap \mathcal{M}_-^0$.

In this step, we analyze the training dynamics after $\boldsymbol{w}_k(T_{\text{hit}}) \in \mathcal{M}_+^+ \cap \mathcal{M}_-^0$.

First, we will prove $\boldsymbol{w}_k(t)$ passes immediately from one side of the surface $\mathcal{M}_+^+ \cap \mathcal{M}_-^0$ to the other, i.e. $\boldsymbol{w}_k(t)$ enters into $\mathcal{M}_+^+ \cap \mathcal{M}_-^+$ at time $T_{\text{hit}}$. Equivalently, we only need to prove $\boldsymbol{b}_k(t)$ passes immediately from one side of the surface $\mathcal{P}_+^+ \cap \mathcal{P}_-^0$ to the other, i.e. $\boldsymbol{b}_k(t)$ enters into $\mathcal{P}_+^+ \cap \mathcal{P}_-^+$ at time $T_{\text{hit}}$.

For any $\tilde{\boldsymbol{b}} \in \mathcal{P}_+^+ \cap \mathcal{P}_-^0$ and $0 < \delta_0 \ll 1$, we know that $\mathcal{P}_+^+ \cap \mathcal{P}_-^0$ separates its neighborhood $\mathcal{B}(\tilde{\boldsymbol{b}}, \delta_0)$ into two domains $\mathcal{G}_- = \{\boldsymbol{b} \in \mathcal{B}(\tilde{\boldsymbol{b}}, \delta_0) : \langle \boldsymbol{b}, \boldsymbol{x}_- \rangle < 0\}$ and $\mathcal{G}_+ = \{\boldsymbol{b} \in \mathcal{B}(\tilde{\boldsymbol{b}}, \delta_0) : \langle \boldsymbol{b}, \boldsymbol{x}_- \rangle > 0\}$. Following Definition H.1, we calculate the limited vector field on $\tilde{\boldsymbol{b}}$ from $\mathcal{G}_-$ and $\mathcal{G}_+$.

(i) The limited vector field $\boldsymbol{F}^-$ on $\tilde{\boldsymbol{b}}$ (from $\mathcal{G}_-$):

$$\frac{\mathrm{d}\boldsymbol{b}}{\mathrm{d}t} = \boldsymbol{F}^-, \text{ where } \boldsymbol{F}^- = \frac{\kappa_2}{\sqrt{m}} \left( \frac{p}{1+p} e^{-f_+(t)} \boldsymbol{x}_+ \right).$$

(ii) The limited vector field $\boldsymbol{F}^+$ on $\tilde{\boldsymbol{b}}$ (from $\mathcal{G}_+$):

$$\frac{\mathrm{d}\boldsymbol{b}}{\mathrm{d}t} = \boldsymbol{F}^+, \text{ where } \boldsymbol{F}^+ = \frac{\kappa_2}{\sqrt{m}} \left( \frac{pe^{-f_+(t)}}{1+p} \boldsymbol{x}_+ - \frac{e^{f_-(t)}}{1+p} \boldsymbol{x}_- \right).$$

(iii) Then we calculate the projections of $\boldsymbol{F}^-$ and $\boldsymbol{F}^+$ onto $\boldsymbol{x}_-$ (the normal to the surface $\mathcal{P}_+^+ \cap \mathcal{P}_-^0$):

$$F_N^- = \langle \boldsymbol{F}^-, \boldsymbol{x}_- \rangle = \frac{\kappa_2 pe^{-f_+(t)}}{\sqrt{m}(1+p)} \cos \Delta,$$

$$F_N^+ = \langle \boldsymbol{F}^+, \boldsymbol{x}_- \rangle = \frac{\kappa_2 pe^{-f_+(t)}}{\sqrt{m}(1+p)} \cos \Delta - \frac{\kappa_2 e^{f_-(t)}}{\sqrt{m}(1+p)}.$$

From $T_{\text{I}} < T_{\text{init}}$ and Lemma C.2, we know $\left|e^{-y_i f_i(t)} - 1\right| \leq 0.11$, so $pe^{-f_+(t)} \cos \Delta - e^{f_-(t)} \geq 0.89p \cos \Delta - 1.11 > 0$, which means $F_N^+ > 0$. And it is clear that $F_N^- > 0$. Hence, the dynamics corresponds to Case (II) in Definition H.1 ($F_N^- > 0$ and $F_N^+ > 0$).

(iv) Hence, $\boldsymbol{b}_k(t)$ passes immediately from one side of the surface $\mathcal{P}_+^+ \cap \mathcal{P}_-^0$ to the other, i.e. $\boldsymbol{b}_k(t)$ enters into $\mathcal{P}_+^+ \cap \mathcal{P}_-^+$ at time $T_{\text{hit}}$.

Second, proceeding as in the proof of **Step I$\sim$III** of the Proof of Theorem C.3, we have the results:

$$\langle \boldsymbol{w}_k(T_{\text{I}}), \boldsymbol{\mu} \rangle \geq \left(1 - 4.2\sqrt{\kappa_1 \kappa_2}\right) \left(1 - \frac{2}{1 + 0.7\left(1 + 1.1(T_{\text{I}} - T_{\text{hit}})\right)^{1.15}}\right)$$

$$\geq \left(1 - 4.2\sqrt{\kappa_1\kappa_2}\right)\left(1 - \frac{2}{1 + 0.7\left(1 + 1.1\frac{\kappa_2}{\kappa_1}\left(10\sqrt{\frac{\kappa_1}{\kappa_2}} - 2\Delta\frac{\kappa_1}{\kappa_2}\right)\right)^{1.15}}\right)$$

$$\geq \left(1 - 4.2\sqrt{\kappa_1\kappa_2}\right)\left(1 - \frac{2}{1 + 0.7\left(1 + 9.9\sqrt{\frac{\kappa_2}{\kappa_1}}\right)^{1.15}}\right).$$

$$\rho_k(T_{\mathrm{I}}) \leq \frac{\kappa_1 + 1.1\kappa_2 T_{\mathrm{I}}}{\sqrt{m}} \leq \frac{\kappa_1 + 11\sqrt{\kappa_1\kappa_2}}{\sqrt{m}} \leq \frac{12\sqrt{\kappa_1\kappa_2}}{\sqrt{m}}.$$

$$\rho_k(T_{\mathrm{I}}) \geq \rho_k(T_0) + \int_{T_{\mathrm{hit}}+\frac{10\kappa_1}{\kappa_2}}^{T_{\mathrm{I}}} \frac{\kappa_2}{\sqrt{m}}\left(0.627 - \frac{1.278}{1 + 0.7\left(1 + \frac{1.1\kappa_2}{\kappa_1}t\right)^{1.15}}\right)\mathrm{d}t$$

$$> 0 + \int_{T_{\mathrm{hit}}+\frac{10\kappa_1}{\kappa_2}}^{T_{\mathrm{I}}} \frac{0.53\kappa_2}{\sqrt{m}}\mathrm{d}t = \frac{0.53\left(10\sqrt{\frac{\kappa_1}{\kappa_2}} - \frac{(10+2\Delta)\kappa_1}{\kappa_2}\right)\kappa_2}{\sqrt{m}} \overset{(6)}{\geq} \frac{5.3 \cdot 0.88\sqrt{\kappa_1\kappa_2}}{\sqrt{m}} \geq \frac{4.66\sqrt{\kappa_1\kappa_2}}{\sqrt{m}}.$$

$$\square$$

**Lemma C.5** (Positive, $\mathcal{M}_+^- \cap \mathcal{M}_-^+$).

*For positive neuron $k \in \{k \in [m/2] : \boldsymbol{w}_k(0) \in \mathcal{M}_+^- \cap \mathcal{M}_-^+\}$, after $\mathcal{O}\left(\frac{\kappa_1 p\Delta}{\kappa_2}\right)$ time, it goes dead:*

$$\boldsymbol{w}_k(t) \in \mathcal{M}_+^- \cap \mathcal{M}_-^-, \text{ for any } t \geq T_{\mathrm{I}} > \mathcal{O}\left(\frac{\kappa_1 p\Delta}{\kappa_2}\right).$$

*Proof of Lemma C.9.*
We do the following analysis for any $k \in \{k \in [m/2] : \boldsymbol{w}_k(0) \in \mathcal{M}_+^+ \cap \mathcal{M}_-^-\}$, i.e. $\mathrm{s}_k = 1$, $\langle \boldsymbol{w}_k(0), \boldsymbol{x}_+ \rangle > 0$, and $\langle \boldsymbol{w}_k(0), \boldsymbol{x}_- \rangle \leq 0$.

First, we define the hitting time

$$T_{\mathrm{hit}} := \inf\left\{t \in (0, T_{\mathrm{I}}] : \boldsymbol{w}_k(t) \notin \mathcal{M}_+^- \cap \mathcal{M}_-^+\right\},$$

and we aim to estimate $T_{\mathrm{hit}}$ and prove $\boldsymbol{w}_k(T_{\mathrm{hit}}) \in \mathcal{M}_+^- \cap \mathcal{M}_-^-$.

From the definition of $T_{\mathrm{hit}}$ and (4), the dynamics of the neuron is:

$$\frac{\mathrm{d}\boldsymbol{b}_k(t)}{\mathrm{d}t} = -\frac{\kappa_2}{\sqrt{m}}\frac{1}{1+p}e^{f_-(t)}\boldsymbol{x}_-, \ t \leq T_{\mathrm{hit}}.$$

Then we have

$$\frac{\mathrm{d}\langle \boldsymbol{b}_k(t), \boldsymbol{x}_+ \rangle}{\mathrm{d}t} = \left\langle -\frac{\kappa_2}{\sqrt{m}}\frac{1}{1+p}e^{f_-(t)}\boldsymbol{x}_-, \boldsymbol{x}_+ \right\rangle = -\frac{\kappa_2}{\sqrt{m}}\frac{1}{1+p}e^{f_-(t)}\cos\Delta,$$

$$\frac{\mathrm{d}\langle \boldsymbol{b}_k(t), \boldsymbol{x}_- \rangle}{\mathrm{d}t} = \left\langle -\frac{\kappa_2}{\sqrt{m}}\frac{1}{1+p}e^{f_-(t)}\boldsymbol{x}_-, \boldsymbol{x}_- \right\rangle = -\frac{\kappa_2}{\sqrt{m}}\frac{1}{1+p}e^{f_-(t)}.$$

It is clear $\frac{\mathrm{d}\langle \boldsymbol{b}_k(t), \boldsymbol{x}_+ \rangle}{\mathrm{d}t} < 0$, so $\langle \boldsymbol{b}_k(t), \boldsymbol{x}_+ \rangle < \langle \boldsymbol{b}_k(0), \boldsymbol{x}_+ \rangle \leq 0$ for any $t \leq T_{\mathrm{hit}}$. If we denote

$$T_{\mathrm{hit},-} := \inf\left\{t \in (0, T_{\mathrm{I}}] : \langle \boldsymbol{w}_k(t), \boldsymbol{x}_- \rangle \leq 0\right\},$$

then it holds:

$$T_{\mathrm{hit}} = T_{\mathrm{hit},-}.$$

So we only need to estimate $T_{\mathrm{hit},-}$. Due to $T_{\mathrm{hit}} \leq T_{\mathrm{I}} \leq T_{\mathrm{init}}$ and Lemma C.2, for any $t \leq T_{\mathrm{hit}}$, we have $\left|e^{-y_i f_i(t)} - 1\right| \leq 0.11$. Then for any $t \leq T_{\mathrm{hit}}$, we have:

$$\frac{\mathrm{d}\langle \boldsymbol{b}_k(t), \boldsymbol{x}_- \rangle}{\mathrm{d}t} \leq -\frac{0.89\kappa_2}{\sqrt{m}(1+p)}.$$

Recalling $\langle \boldsymbol{w}_k(0), \boldsymbol{x}_+ \rangle \leq 0$ and $\langle \boldsymbol{w}_k(0), \boldsymbol{x}_- \rangle > 0$, with the help of Lemma I.2, we have $\langle \boldsymbol{w}_k(0), \boldsymbol{x}_- \rangle \leq \sin \Delta$. Combining the two estimate, we have:

$$\langle \boldsymbol{b}_k(t), \boldsymbol{x}_- \rangle \leq \langle \boldsymbol{b}_k(0), \boldsymbol{x}_- \rangle - \int_0^t \frac{0.89\kappa_2}{\sqrt{m}(1+p)} \mathrm{d}t$$

$$\leq \rho_k(0) \sin \Delta - \frac{0.89\kappa_2}{\sqrt{m}(1+p)} t = \frac{\kappa_1 \sin \Delta}{\sqrt{m}} - \frac{0.89\kappa_2}{\sqrt{m}(1+p)} t.$$

Hence,

$$T_{\mathrm{hit}} = T_{\mathrm{hit},+} \leq \frac{(1+p)\sin \Delta}{0.89} \frac{\kappa_1}{\kappa_2} < T_{\mathrm{I}} = 10 \sqrt{\frac{\kappa_1}{\kappa_2}}.$$

Moreover, the analysis gives us $\boldsymbol{w}_k(T_{\mathrm{hit}}) \in \mathcal{M}_+^- \cap \mathcal{M}_-^-$. By Lemma B.2, we obtain:

$$\boldsymbol{w}_k(t) \in \mathcal{M}_+^- \cap \mathcal{M}_-^-, \text{ for any } t \geq T_{\mathrm{hit}}.$$

$\square$

**Lemma C.6** (Positive, $\mathcal{M}_+^- \cap \mathcal{M}_-^-$)**.**
*For positive neuron $k \in \{k \in [m/2] : \boldsymbol{w}_k(0) \in \mathcal{M}_+^- \cap \mathcal{M}_-^-\}$, it keeps dead forever.*

*Proof of Lemma C.6.* Due to Lemma B.2, this lemma is trivial. $\square$

## C.2 The Dynamics of Negative Neurons

According to the initial direction, all negative neurons ($\mathrm{d}_k = -1$) can be divided into the following four classes.

$$[m] - [m/2]$$
$$= \left\{ k \in [m] - [m/2] : \boldsymbol{w}_k(0) \in \mathcal{M}_+^+ \cap \mathcal{M}_-^+ \right\} \bigcup \left\{ k \in [m] - [m/2] : \boldsymbol{w}_k(0) \in \mathcal{M}_+^- \cap \mathcal{M}_-^+ \right\}$$
$$\bigcup \left\{ k \in [m] - [m/2] : \boldsymbol{w}_k(0) \in \mathcal{M}_+^+ \cap \mathcal{M}_-^- \right\} \bigcup \left\{ k \in [m] - [m/2] : \boldsymbol{w}_k(0) \in \mathcal{M}_+^- \cap \mathcal{M}_-^- \right\}.$$

In the following four lemmas, we will prove the dynamics of these four classes of negative neurons. In summary, in Phase I ($t < T_{\mathrm{I}}$), some of the negative neurons move to the manifold $\mathcal{M}_+^0 \cap \mathcal{M}_-^+$ in a shorter time and then remain on this manifold, and their norms grow slowly, while other negative neurons go dead.

**Lemma C.7** (Negative, $\mathcal{M}_+^+ \cap \mathcal{M}_-^+$)**.**
*For negative neuron $k \in \{k \in [m] - [m/2] : \boldsymbol{w}_k(0) \in \mathcal{M}_+^+ \cap \mathcal{M}_-^+\}$, in Phase I ($t \leq T_{\mathrm{I}}$), it's dynamics must belong to one of the following two cases:*

**(i. Living)**. *(S1).* $\boldsymbol{w}_k(t) \in \mathcal{M}_+^0 \cap \mathcal{M}_-^+$ *for any* $t \geq \mathcal{O}(\frac{\kappa_1}{\kappa_2})$,

   *(S2). It has a small norm :* $\rho_k(T_{\mathrm{I}}) = \mathcal{O}\left( \frac{\sqrt{\kappa_1 \kappa_2}}{\sqrt{m}} \left( \sqrt{\frac{\kappa_1}{\kappa_2}} + \frac{\Delta}{p} \right) \right)$,

   *(S3). It is weakly aligned with* $\boldsymbol{x}_+^\perp$ *:* $\langle \boldsymbol{w}_k(T_{\mathrm{I}}), \boldsymbol{x}_+^\perp \rangle \geq 1 - \mathcal{O}\left( \left( \sqrt{\frac{\kappa_1}{\kappa_2}} \frac{p}{\Delta} \right)^{1.6} \right)$;

**(ii. Dead)**. $\boldsymbol{w}_k(t) \in \mathcal{M}_+^- \cap \mathcal{M}_-^-$ *for any* $t \geq \mathcal{O}(\frac{\kappa_1}{\kappa_2})$.

**Moreover**, *if* $\langle \boldsymbol{w}_k(0), \boldsymbol{x}_- \rangle > \frac{(1+\mathcal{O}(\kappa_1 \kappa_2))p \cos \Delta - (1 - \mathcal{O}(\kappa_1 \kappa_2))}{(1 - \mathcal{O}(\kappa_1 \kappa_2))p - (1 + \mathcal{O}(\kappa_1 \kappa_2)) \cos \Delta} \langle \boldsymbol{w}_k(0), \boldsymbol{x}_+ \rangle$, *it must belongs to Case* (i)*; if* $\langle \boldsymbol{w}_k(0), \boldsymbol{x}_+ \rangle > \frac{(1+\mathcal{O}(\kappa_1 \kappa_2))p - (1 - \mathcal{O}(\kappa_1 \kappa_2)) \cos \Delta}{(1 - \mathcal{O}(\kappa_1 \kappa_2))p \cos \Delta - (1 + \mathcal{O}(\kappa_1 \kappa_2))} \langle \boldsymbol{w}_k(0), \boldsymbol{x}_- \rangle$, *it must belongs to Case* (ii).

*Proof of Lemma C.7.*
We do the following analysis for any $k \in \{k \in [m] - [m/2] : \boldsymbol{w}_k(0) \in \mathcal{M}_+^+ \cap \mathcal{M}_-^+\}$, i.e. $\mathrm{s}_k = -1$, $\langle \boldsymbol{w}_k(0), \boldsymbol{x}_+ \rangle > 0$, and $\langle \boldsymbol{w}_k(0), \boldsymbol{x}_- \rangle > 0$.

**Step I.** Neuron must arrives in the border $\partial(\mathcal{M}_+^+ \cap \mathcal{M}_-^+)$ in $\mathcal{O}(\frac{\kappa_1}{\kappa_2})$ time.

First, we define the hitting time

$$T_{\mathrm{hit}} := \inf\left\{ t \in (0, T_{\mathrm{I}}] : \boldsymbol{w}_k(t) \in \partial(\mathcal{M}_+^+ \cap \mathcal{M}_-^+)\right\},$$

and we aim to prove $T_{\mathrm{hit}}$ exists and estimate $T_{\mathrm{hit}}$.

Recalling the decoupling $\partial(\mathcal{M}_+^+ \cap \mathcal{M}_-^+) = \left(\mathcal{M}_+^0 \cap \mathcal{M}_-^+\right) \cup \left(\mathcal{M}_-^0 \cap \mathcal{M}_+^+\right) \cup \left(\mathcal{M}_+^0 \cap \mathcal{M}_-^0\right)$, we only need to focus on the dynamics of $\langle \boldsymbol{b}_k(t), \boldsymbol{x}_+\rangle$ and $\langle \boldsymbol{b}_k(t), \boldsymbol{x}_-\rangle$.

From the definition of $T_{\mathrm{hit}}$ and (4), the dynamics of the neuron is:

$$\frac{\mathrm{d}\boldsymbol{b}_k(t)}{\mathrm{d}t} = -\frac{\kappa_2}{\sqrt{m}}\frac{1}{n}\sum_{i=1}^n e^{-y_i f_i(t)} y_i \boldsymbol{x}_i = -\frac{\kappa_2}{\sqrt{m}}\left(\frac{p}{1+p}e^{-f_+(t)}\boldsymbol{x}_+ - \frac{1}{1+p}e^{f_-(t)}\boldsymbol{x}_-\right), \ t \le T_{\mathrm{hit}}.$$

Then we have

$$\frac{\mathrm{d}\langle \boldsymbol{b}_k(t), \boldsymbol{x}_+\rangle}{\mathrm{d}t} = \left\langle -\frac{\kappa_2}{\sqrt{m}}\left(\frac{p}{1+p}e^{-f_+(t)}\boldsymbol{x}_+ - \frac{1}{1+p}e^{f_-(t)}\boldsymbol{x}_-\right), \boldsymbol{x}_+\right\rangle$$

$$= -\frac{\kappa_2}{\sqrt{m}}\left(\frac{p}{1+p}e^{-f_+(t)} - \frac{1}{1+p}e^{f_-(t)}\cos\Delta\right),$$

$$\frac{\mathrm{d}\langle \boldsymbol{b}_k(t), \boldsymbol{x}_-\rangle}{\mathrm{d}t} = \left\langle -\frac{\kappa_2}{\sqrt{m}}\left(\frac{p}{1+p}e^{-f_+(t)}\boldsymbol{x}_+ - \frac{1}{1+p}e^{f_-(t)}\boldsymbol{x}_-\right), \boldsymbol{x}_-\right\rangle$$

$$= -\frac{\kappa_2}{\sqrt{m}}\left(\frac{p}{1+p}e^{-f_+(t)}\cos\Delta - \frac{1}{1+p}e^{f_-(t)}\right).$$

Due to $T_{\mathrm{hit}} \le T_{\mathrm{I}} \le T_{\mathrm{init}}$ and Lemma C.2, for any $t \le T_{\mathrm{hit}}$, we have $\left|e^{-y_i f_i(t)} - 1\right| \le 1.1\sqrt{\kappa_1\kappa_2}$. Then for any $t \le T_{\mathrm{hit}}$, we have:

$$\frac{\mathrm{d}\langle \boldsymbol{b}_k(t), \boldsymbol{x}_+\rangle}{\mathrm{d}t} \le -\frac{\kappa_2}{\sqrt{m}}\left(\frac{(1 - 1.1\sqrt{\kappa_1\kappa_2})p}{1+p} - \frac{1 + 1.1\sqrt{\kappa_1\kappa_2}}{1+p}\cos\Delta\right)$$

$$\le -\frac{\kappa_2\left((1 - 1.1\sqrt{\kappa_1\kappa_2})p - (1 + 1.1\sqrt{\kappa_1\kappa_2})\cos\Delta\right)}{\sqrt{m}(1+p)} \le -\frac{\kappa_2(0.98p - 1.02)}{\sqrt{m}(1+p)},$$

$$\frac{\mathrm{d}\langle \boldsymbol{b}_k(t), \boldsymbol{x}_-\rangle}{\mathrm{d}t} \le -\frac{\kappa_2}{\sqrt{m}}\left(\frac{(1 - 1.1\sqrt{\kappa_1\kappa_2})p}{1+p}\cos\Delta - \frac{1 + 1.1\sqrt{\kappa_1\kappa_2}}{1+p}\right)$$

$$\le -\frac{\kappa_2\left((1 - 1.1\sqrt{\kappa_1\kappa_2})p\cos\Delta - (1 + 1.1\sqrt{\kappa_1\kappa_2})\right)}{\sqrt{m}(1+p)} \le -\frac{0.98\kappa_2(p\cos\Delta - 1)}{2\sqrt{m}(1+p)}.$$

Now we consider the time

$$T_{\mathrm{test}} := \frac{3\kappa_1}{\kappa_2}.$$

If we assume $T_{\mathrm{test}} < T_{\mathrm{hit}}$, then we have the estimate:

$$\langle \boldsymbol{b}_k(T_{\mathrm{test}}), \boldsymbol{x}_+\rangle \le \langle \boldsymbol{b}_k(0), \boldsymbol{x}_+\rangle - \int_0^{T_{\mathrm{test}}} \frac{\kappa_2(0.98p - 1.02)}{\sqrt{m}(1+p)}\mathrm{d}t$$

$$\le \frac{\kappa_1}{\sqrt{m}} - \frac{\kappa_2(0.98p - 1.02)}{\sqrt{m}(1+p)}\frac{3\kappa_1}{\kappa_2} < 0,$$

which is contradict to the definition of $T_{\mathrm{hit}}$. Hence, we have:

$$T_{\mathrm{hit}} \le T_{\mathrm{test}} \le \frac{3\kappa_1}{\kappa_2},$$

which means neurons must arrive in the border $\partial(\mathcal{M}_+^+ \cap \mathcal{M}_-^+)$ in $\mathcal{O}(\frac{\kappa_1}{\kappa_2})$ time.

Because $\partial(\mathcal{M}_+^+ \cap \mathcal{M}_-^\pm) = \left(\mathcal{M}_+^0 \cap \mathcal{M}_-^\pm\right) \cup \left(\mathcal{M}_-^0 \cap \mathcal{M}_+^\pm\right) \cup \left(\mathcal{M}_+^0 \cap \mathcal{M}_-^0\right)$, the neuron must arrives in $\mathcal{M}_+^0 \cap \mathcal{M}_-^\pm$ or $\mathcal{M}_-^0 \cap \mathcal{M}_+^\pm$ or $\mathcal{M}_+^0 \cap \mathcal{M}_-^0$. If the neuron arrives in $\mathcal{M}_+^0 \cap \mathcal{M}_-^0$, it goes dead forever (Lemma B.2). We will analyze the training dynamics after arriving in $\mathcal{M}_+^0 \cap \mathcal{M}_-^\pm$ or $\mathcal{M}_-^0 \cap \mathcal{M}_+^\pm$ in the following Step II and Step III.

**Step II.** Dynamics after arriving in the manifold $\mathcal{M}_+^0 \cap \mathcal{M}_-^\pm$.

In this step, we will analyze the training dynamics after $\boldsymbol{w}_k(T_{\text{hit}}) \in \mathcal{M}_+^0 \cap \mathcal{M}_-^\pm$, i.e. after $\boldsymbol{b}_k(T_{\text{hit}}) \in \mathcal{P}_+^0 \cap \mathcal{P}_-^\pm$.

We first analysis the vector field around the manifold $\mathcal{P}_+^0 \cap \mathcal{P}_-^\pm$ for $T_{\text{hit}} \leq t \leq T_{\text{I}}$.

For any $\tilde{\boldsymbol{b}} \in \mathcal{P}_+^0 \cap \mathcal{P}_-^\pm$ and $0 < \delta_0 \ll 1$, we know that $\mathcal{P}_+^0 \cap \mathcal{P}_-^\pm$ separates its neighborhood $\mathcal{B}(\tilde{\boldsymbol{b}}, \delta_0)$ into two domains $\mathcal{G}_- = \{\boldsymbol{b} \in \mathcal{B}(\tilde{\boldsymbol{b}}, \delta_0) : \langle \boldsymbol{b}, \boldsymbol{x}_+ \rangle < 0\}$ and $\mathcal{G}_+ = \{\boldsymbol{b} \in \mathcal{B}(\tilde{\boldsymbol{b}}, \delta_0) : \langle \boldsymbol{b}, \boldsymbol{x}_+ \rangle > 0\}$. Following Definition H.1, we calculate the limited vector field on $\tilde{\boldsymbol{b}}$ from $\mathcal{G}_-$ and $\mathcal{G}_+$.

(i) The limited vector field $\boldsymbol{F}^-$ on $\tilde{\boldsymbol{b}}$ (from $\mathcal{G}_-$):

$$\frac{\mathrm{d}\boldsymbol{b}}{\mathrm{d}t} = \boldsymbol{F}^-, \text{ where } \boldsymbol{F}^- = \frac{\kappa_2}{\sqrt{m}} \frac{1}{1+p} e^{f_-(t)} \boldsymbol{x}_-.$$

(ii) The limited vector field $\boldsymbol{F}^+$ on $\tilde{\boldsymbol{b}}$ (from $\mathcal{G}_+$):

$$\frac{\mathrm{d}\boldsymbol{b}}{\mathrm{d}t} = \boldsymbol{F}^+, \text{ where } \boldsymbol{F}^+ = -\frac{\kappa_2}{\sqrt{m}} \left( \frac{pe^{-f_+(t)}}{1+p} \boldsymbol{x}_+ - \frac{e^{f_-(t)}}{1+p} \boldsymbol{x}_- \right).$$

(iii) Then we calculate the projections of $\boldsymbol{F}^-$ and $\boldsymbol{F}^+$ onto $\boldsymbol{x}_+$ (the normal to the surface $\mathcal{P}_+^0 \cap \mathcal{P}_-^\pm$):

$$F_N^- = \left\langle \boldsymbol{F}^-, \boldsymbol{x}_+ \right\rangle = \frac{\kappa_2 e^{f_-(t)}}{\sqrt{m}(1+p)} \cos \Delta,$$

$$F_N^+ = \left\langle \boldsymbol{F}^+, \boldsymbol{x}_+ \right\rangle = \frac{\kappa_2 e^{f_-(t)}}{\sqrt{m}(1+p)} \cos \Delta - \frac{\kappa_2 pe^{-f_+(t)}}{\sqrt{m}(1+p)}.$$

From $T_{\text{I}} < T_{\text{init}}$ and Lemma C.2, we know $|e^{-y_i f_i(t)} - 1| \leq 0.11$, so $pe^{-f_+(t)} \cos \Delta - e^{f_-(t)} \geq 0.89p \cos \Delta - 1.11 > 0$, which means $F_N^+ < 0$. And it is clear that $F_N^- > 0$. Hence, the dynamics corresponds to Case (I) in Definition H.1 ($F_N^- > 0$ and $F_N^+ < 0$).

(iv) Hence, $\boldsymbol{b}_k(t)$ can not leave $\mathcal{P}_+^0 \cap \mathcal{P}_-^\pm$ for $T_{\text{hit}} \leq t \leq T_{\text{I}}$.

(v) Moreover, the dynamics of $\boldsymbol{b}_k$ on $\mathcal{P}_+^0 \cap \mathcal{P}_-^\pm$ satisfies:

$$\frac{\mathrm{d}\boldsymbol{b}}{\mathrm{d}t} = \alpha \boldsymbol{F}^+ + (1-\alpha)\boldsymbol{F}^-, \quad \alpha = \frac{f_N^-}{f_N^- - f_N^+},$$

which is

$$\frac{\mathrm{d}\boldsymbol{b}_k(t)}{\mathrm{d}t} \frac{\kappa_2 e^{f_-(t)}}{\sqrt{m}(1+p)} \left( \boldsymbol{x}_- - \boldsymbol{x}_+ \cos \Delta \right).$$

By Lemma C.1, we know that the dynamics of $\boldsymbol{w}_k(t)$ on $\mathcal{M}_+^0 \cap \mathcal{M}_-^\pm$ and the dynamics of $\rho_k(t)$ are:

$$\frac{\mathrm{d}\boldsymbol{w}_k(t)}{\mathrm{d}t} = \frac{\kappa_2 e^{f_-(t)}}{\rho_k(t)\sqrt{m}(1+p)} \left( \boldsymbol{x}_- - \langle \boldsymbol{w}_k(t), \boldsymbol{x}_- \rangle \boldsymbol{w}_k - \boldsymbol{x}_+ \cos \Delta \right). \tag{15}$$

$$\frac{\mathrm{d}\rho_k(t)}{\mathrm{d}t} = \frac{\kappa_2 e^{f_-(t)}}{\sqrt{m}(1+p)} \left\langle \boldsymbol{w}_k(t), \boldsymbol{x}_- \right\rangle. \tag{16}$$

(vi) In this step, we aim to estimate $\rho_k(T_{\text{I}})$ and $\left\langle \boldsymbol{w}_k(t), \boldsymbol{x}_+^\perp \right\rangle$.

From $\boldsymbol{w}_k(T_{\text{hit}}) \in \mathcal{M}_+^0 \cap \mathcal{M}_-^\pm$, it holds that $\langle \boldsymbol{w}_k(T_{\text{hit}}), \boldsymbol{x}_+ \rangle = 0$ and $\langle \boldsymbol{w}_k(T_{\text{hit}}), \boldsymbol{x}_- \rangle > 0$. Using lemma I.2, we have $0 < \langle \boldsymbol{w}_k(T_{\text{hit}}), \boldsymbol{x}_- \rangle \leq \sin \Delta$.

Recalling Lemma C.2 and the estimate of $T_{\text{hit}}$ in **Step I**, we have:

$$0 \le \rho_k(T_{\text{hit}}) \le \frac{\kappa_1 + 1.1\kappa_2 T_{\text{hit}}}{\sqrt{m}},$$

and we can estimate the dynamics for $T_{\text{hit}} \le t \le T_{\text{I}}$ by (v)(vi):

$$0 \le \frac{\mathrm{d}\rho_k(t)}{\mathrm{d}t} = \frac{\kappa_2 e^{f_-(t)}}{\sqrt{m}(1+p)} \langle \boldsymbol{w}_k(t), \boldsymbol{x}_- \rangle \le \frac{\kappa_2 e^{f_-(t)}}{\sqrt{m}(1+p)} \sin \Delta,$$
$$e^{f_-(t)} \le 1 + 0.11 = 1.11.$$

Then we obtain the estimate of $\rho_k(t)$ for any $T_{\text{hit}} < t \le T_{\text{I}}$:

$$\rho_k(t) = \rho_k(T_{\text{hit}}) + \int_{T_{\text{hit}}}^{t} \frac{\mathrm{d}\rho_k(s)}{\mathrm{d}t}\mathrm{d}s \le \frac{\kappa_1 + 1.1\kappa_2 T_{\text{hit}}}{\sqrt{m}} + \frac{\kappa_2 e^{f_-(t)} \sin \Delta}{\sqrt{m}(1+p)}(t - T_{\text{hit}})$$

$$\le \frac{\kappa_1 + 1.1\kappa_2 T_{\text{hit}}}{\sqrt{m}} + \frac{1.11\kappa_2 \sin \Delta}{\sqrt{m}(1+p)}(T_{\text{I}} - T_{\text{hit}}) \le \frac{\kappa_1 + 1.1\kappa_2 \frac{3\kappa_1}{\kappa_2}}{\sqrt{m}} + \frac{1.11\kappa_2 \sin \Delta}{\sqrt{m}(1+p)}(T_{\text{I}} - \frac{3\kappa_1}{\kappa_2})$$

$$\le \frac{4.3\kappa_1}{\sqrt{m}} + \frac{1.11\kappa_2 \sin \Delta}{\sqrt{m}(1+p)}t.$$

Specifically, we have:

$$\rho_k(T_{\text{I}}) \le \frac{4.3\kappa_1}{\sqrt{m}} + \frac{1.11\kappa_2 \sin \Delta}{\sqrt{m}(1+p)}T_{\text{I}} \le \frac{4.3\kappa_1}{\sqrt{m}} + \frac{11.1\sqrt{\kappa_1\kappa_2} \sin \Delta}{\sqrt{m}(1+p)} = \frac{\sqrt{\kappa_1\kappa_2}}{\sqrt{m}}\left(4.3\sqrt{\frac{\kappa_1}{\kappa_2}} + \frac{11.1\sin \Delta}{1+p}\right).$$

For any $\boldsymbol{w} \in \mathcal{M}_+^0 \cap \mathcal{M}_-^+$, we have $\langle \boldsymbol{w}, \boldsymbol{x}_+ \rangle = 0$, so

$$\langle \boldsymbol{w}, \boldsymbol{x}_+^\perp \rangle = \left\langle \boldsymbol{w}, \frac{\boldsymbol{x}_- - \boldsymbol{x}_+ \cos \Delta}{\|\boldsymbol{x}_- - \boldsymbol{x}_+ \cos \Delta\|} \right\rangle = \frac{\langle \boldsymbol{w}, \boldsymbol{x}_- \rangle}{\|\boldsymbol{x}_- - \boldsymbol{x}_+ \cos \Delta\|} = \frac{1}{\sin \Delta} \langle \boldsymbol{w}, \boldsymbol{x}_- \rangle.$$

So we only need to focus on the dynamics of $\langle \boldsymbol{w}_k(t), \boldsymbol{x}_- \rangle$ to derive the dynamics of $\langle \boldsymbol{w}_k(t), \boldsymbol{x}_+^\perp \rangle$.

By (15) and the estimate of $\rho_k(t)$, for any $T_{\text{hit}} \le t \le T_{\text{I}}$ we have:

$$\frac{\mathrm{d}\langle \boldsymbol{w}_k(t), \boldsymbol{x}_- \rangle}{\mathrm{d}t} = \left\langle \frac{e^{f_-(t)}}{\rho_k(t)\sqrt{m}(1+p)}\left(\boldsymbol{x}_- - \langle \boldsymbol{w}_k(t), \boldsymbol{x}_- \rangle \boldsymbol{w}_k(t) - \boldsymbol{x}_+ \cos \Delta\right), \boldsymbol{x}_- \right\rangle$$

$$= \frac{\kappa_2 e^{f_-(t)}}{\rho_k(t)\sqrt{m}(1+p)}\left(\sin^2 \Delta - \langle \boldsymbol{w}_k(t), \boldsymbol{x}_- \rangle^2\right) \ge \frac{(1-0.11)\kappa_2}{(1+p)\left(4.3\kappa_1 + \frac{1.11\kappa_2 \sin \Delta}{(1+p)}t\right)}\left(\sin^2 \Delta - \langle \boldsymbol{w}_k(t), \boldsymbol{x}_- \rangle^2\right)$$

$$\ge \frac{0.89\kappa_2}{4.3(1+p)\kappa_1 + 1.11\kappa_2 t \sin \Delta}\left(\sin^2 \Delta - \langle \boldsymbol{w}_k(t), \boldsymbol{x}_- \rangle^2\right).$$

And we have $0 < \left\langle \boldsymbol{w}_k(\frac{3\kappa_1}{\kappa_2}), \boldsymbol{x}_- \right\rangle < \sin \Delta$.

Now we consider the following auxiliary ODE:

$$\begin{cases} \frac{\mathrm{d}U(t)}{\mathrm{d}t} = \frac{0.89\kappa_2}{4.3(1+p)\kappa_1 + 1.11\kappa_2 t \sin \Delta}\left(\sin^2 \Delta - U^2(t)\right) \\ U(0) = 0 \end{cases}, \tag{17}$$

and let $U(t)$ is the solution of (13). We know that $\langle \boldsymbol{w}_k(t), \boldsymbol{x}_- \rangle$ is an upper solution of ODE (17). From the Comparison Principle of ODEs, we know this means:

$$\langle \boldsymbol{w}_k(t), \boldsymbol{x}_- \rangle > U(t), \text{ for any } t \le T_{\text{I}}.$$

In order to estimate $\langle \boldsymbol{w}_k(t), \boldsymbol{x}_- \rangle$, we only need to study the solution of ODE (17). It is easy to verify that the solution of (17) satisfies

$$\log\left(\frac{\sin \Delta + U(t)}{\sin \Delta - U(t)}\right) - \log\left(\frac{\sin \Delta}{\sin \Delta}\right) = \frac{1.78\kappa_2 \Delta}{1.11\kappa_2 \sin \Delta} \log\left(\frac{4.3(1+p)\kappa_1 + 1.11\kappa_2 t \sin \Delta}{4.3(1+p)\kappa_1 + 3.33\kappa_1 \sin \Delta}\right)$$

Then we have:

$$\log\left(\frac{\sin\Delta + U(T_{\mathrm{I}})}{\sin\Delta - U(T_{\mathrm{I}})}\right) \geq \frac{1.78}{1.11}\log\left(\frac{4.3(1+p)\kappa_1 + 11.1\sqrt{\kappa_1\kappa_2}\sin\Delta}{4.3(1+p)\kappa_1 + 3.33\kappa_1\sin\Delta}\right)$$

$$> 1.6\log\left(1 + \frac{\left(11.1\sqrt{\frac{\kappa_2}{\kappa_1}} - 3.33\right)\frac{\sin\Delta}{1+p}}{4.3 + 3.33\frac{\sin\Delta}{1+p}}\right) > 1.6\log\left(1 + \frac{\left(11.1\sqrt{\frac{\kappa_2}{\kappa_1}} - 3.33\right)\frac{\sin\Delta}{1+p}}{4.6}\right)$$

$$> 1.6\log\left(1 + \frac{10.7}{4.6}\sqrt{\frac{\kappa_2}{\kappa_1}}\frac{\sin\Delta}{1+p}\right),$$

which means

$$U(T_{\mathrm{I}}) > \left(1 - \frac{2}{\left(1 + \frac{10.7}{4.6}\sqrt{\frac{\kappa_2}{\kappa_1}}\frac{\sin\Delta}{1+p}\right)^{1.6} + 1}\right)\sin\Delta.$$

Hence, we have the estimate of $\langle \boldsymbol{w}_k(t), \boldsymbol{x}_+^{\perp}\rangle$:

$$\langle \boldsymbol{w}_k(T_{\mathrm{I}}), \boldsymbol{x}_+^{\perp}\rangle = \frac{1}{\sin\Delta}\langle \boldsymbol{w}_k(T_{\mathrm{I}}), \boldsymbol{x}_-\rangle > \frac{1}{\sin\Delta}U(T_{\mathrm{I}}) > 1 - \frac{2}{\left(1 + 2.32\sqrt{\frac{\kappa_2}{\kappa_1}}\frac{\sin\Delta}{1+p}\right)^{1.6} + 1}.$$

**Step III.** Dynamics after arriving in the manifold $\mathcal{M}_+^+ \cap \mathcal{M}_-^0$.

In this step, we analyze the training dynamics after $\boldsymbol{w}_k(T_{\mathrm{hit}}) \in \mathcal{M}_+^+ \cap \mathcal{M}_-^0$, i.e. $\boldsymbol{b}_k(T_{\mathrm{hit}}) \in \mathcal{P}_+^+ \cap \mathcal{P}_-^0$.

For any $\tilde{\boldsymbol{b}} \in \mathcal{P}_+^+ \cap \mathcal{P}_-^0$ and $0 < \delta_0 \ll 1$, we know that $\mathcal{P}_+^+ \cap \mathcal{P}_-^0$ separates its neighborhood $\mathcal{B}(\tilde{\boldsymbol{b}}, \delta_0)$ into two domains $\mathcal{G}_- = \{\boldsymbol{b} \in \mathcal{B}(\tilde{\boldsymbol{b}}, \delta_0) : \langle\boldsymbol{b}, \boldsymbol{x}_-\rangle < 0\}$ and $\mathcal{G}_+ = \{\boldsymbol{b} \in \mathcal{B}(\tilde{\boldsymbol{b}}, \delta_0) : \langle\boldsymbol{b}, \boldsymbol{x}_-\rangle > 0\}$. Following Definition H.1, we calculate the limited vector field on $\tilde{\boldsymbol{b}}$ from $\mathcal{G}_-$ and $\mathcal{G}_+$.

For any $\tilde{\boldsymbol{b}} \in \mathcal{P}_+^+ \cap \mathcal{P}_-^0$ and $0 < \delta_0 \ll 1$, we know that $\mathcal{P}_+^+ \cap \mathcal{P}_-^0$ separates its neighborhood $\mathcal{B}(\tilde{\boldsymbol{b}}, \delta_0)$ into two domains $\mathcal{G}_- = \{\boldsymbol{b} \in \mathcal{B}(\tilde{\boldsymbol{b}}, \delta_0) : \langle\boldsymbol{b}, \boldsymbol{x}_-\rangle < 0\}$ and $\mathcal{G}_+ = \{\boldsymbol{b} \in \mathcal{B}(\tilde{\boldsymbol{b}}, \delta_0) : \langle\boldsymbol{b}, \boldsymbol{x}_-\rangle > 0\}$. Following Definition H.1, we calculate the limited vector field on $\tilde{\boldsymbol{b}}$ from $\mathcal{G}_-$ and $\mathcal{G}_+$.

(i) The limited vector field $\boldsymbol{F}^-$ on $\tilde{\boldsymbol{b}}$ (from $\mathcal{G}_-$):

$$\frac{\mathrm{d}\boldsymbol{b}}{\mathrm{d}t} = \boldsymbol{F}^-, \text{ where } \boldsymbol{F}^- = -\frac{\kappa_2}{\sqrt{m}}\frac{p}{1+p}e^{-f_+(t)}\boldsymbol{x}_+.$$

(ii) The limited vector field $\boldsymbol{F}^+$ on $\tilde{\boldsymbol{b}}$ (from $\mathcal{G}_+$):

$$\frac{\mathrm{d}\boldsymbol{b}}{\mathrm{d}t} = \boldsymbol{F}^+, \text{ where } \boldsymbol{F}^+ = -\frac{\kappa_2}{\sqrt{m}}\left(\frac{pe^{-f_+(t)}}{1+p}\boldsymbol{x}_+ - \frac{e^{f_-(t)}}{1+p}\boldsymbol{x}_-\right).$$

(iii) Then we calculate the projections of $\boldsymbol{F}^-$ and $\boldsymbol{F}^+$ onto $\boldsymbol{x}_-$ (the normal to the surface $\mathcal{P}_+^+ \cap \mathcal{P}_-^0$):

$$F_N^- = \langle\boldsymbol{F}^-, \boldsymbol{x}_-\rangle = -\frac{\kappa_2 pe^{-f_+(t)}}{\sqrt{m}(1+p)}\cos\Delta,$$

$$F_N^+ = \langle\boldsymbol{F}^+, \boldsymbol{x}_-\rangle = -\left(\frac{\kappa_2 pe^{-f_+(t)}}{\sqrt{m}(1+p)}\cos\Delta - \frac{\kappa_2 e^{f_-(t)}}{\sqrt{m}(1+p)}\right).$$

From $T_{\mathrm{I}} < T_{\mathrm{init}}$ and Lemma C.2, we know $|e^{-y_i f_i(t)} - 1| \leq 0.11$, so $pe^{-f_+(t)}\cos\Delta - e^{f_-(t)} \geq 0.89p\cos\Delta - 1.11 > 0$, which means $F_N^+ > 0$. And it is clear that $F_N^- > 0$. Hence, the dynamics corresponds to Case (II) in Definition H.1 ($F_N^- > 0$ and $F_N^+ > 0$).

(iv) Hence, $\boldsymbol{b}_k(t)$ passes immediately from one side of the surface $\mathcal{P}_+^+ \cap \mathcal{P}_-^0$ to the other, i.e. $\boldsymbol{b}_k(t)$ enters into $\mathcal{P}_+^+ \cap \mathcal{P}_-^+$ at time $T_{\mathrm{hit}}$.

Then the dynamics of $\boldsymbol{b}_k$ in $\mathcal{P}_+^+ \cap \mathcal{P}_-^-$ satisfies:

$$\frac{\mathrm{d}\boldsymbol{b}_k(t)}{\mathrm{d}t} = -\frac{\kappa_2}{\sqrt{m}} \frac{p}{1+p} e^{-f_+(t)} \boldsymbol{x}_+.$$

(v) We define the following time, and our aim is to estimate $T_{\text{test},1}$:

$$T_{\text{test},1} := \inf\left\{ t \in (T_{\text{hit}}, T_{\text{I}}] : \langle \boldsymbol{w}_k(t), \boldsymbol{x}_+ \rangle \le 0 \text{ or } \langle \boldsymbol{w}_k(t), \boldsymbol{x}_- \rangle \ge 0 \right\},$$

$$T_{\text{test},2} := \inf\left\{ t \in (T_{\text{hit}}, T_{\text{I}}] : \langle \boldsymbol{w}_k(t), \boldsymbol{x}_+ \rangle \le 0 \right\}$$

It is clear $T_{\text{test},1} \le T_{\text{test},2}$. Moreover, due to $\frac{\mathrm{d}\langle \boldsymbol{b}_k(t), \boldsymbol{x}_- \rangle}{\mathrm{d}t} = -\frac{\kappa_2}{\sqrt{m}} \frac{p}{1+p} e^{-f_+(t)} \cos \Delta < 0$ and $\langle \boldsymbol{b}_k(T_{\text{hit}}), \boldsymbol{x}_- \rangle = 0$, we know $\langle \boldsymbol{b}_k(t), \boldsymbol{x}_- \rangle < 0$ holds for any $t \le T_{\text{test},1}$. Hence, we have

$$T_{\text{test},1} = T_{\text{test},2}, \quad \langle \boldsymbol{w}_k(T_{\text{test},1}), \boldsymbol{x}_+ \rangle = 0, \quad \langle \boldsymbol{w}_k(T_{\text{test},1}), \boldsymbol{x}_- \rangle < 0.$$

And we only need to estimate $T_{\text{test},2}$. For any $T_{\text{hit}} < t \le T_{\text{test},1} = T_{\text{test},2}$, we have

$$\frac{\mathrm{d}\langle \boldsymbol{b}_k(t), \boldsymbol{x}_+ \rangle}{\mathrm{d}t} = -\frac{\kappa_2}{\sqrt{m}} \frac{p}{1+p} e^{-f_+(t)} \overset{\text{Lemma C.2}}{\le} -\frac{\kappa_2}{\sqrt{m}} \frac{p}{1+p}(1 - 0.11) = -\frac{0.89\kappa_2 p}{\sqrt{m}(1+p)}.$$

Recalling Lemma C.2 and the estimate of $T_{\text{hit}}$ in **Step I**, we have:

$$\langle \boldsymbol{b}_k(T_{\text{hit}}), \boldsymbol{x}_+ \rangle \le \|\rho_k(t)\| \le \frac{\kappa_1 + 1.1\kappa_2 T_{\text{hit}}}{\sqrt{m}} \le \|\rho_k(t)\| \le \frac{\kappa_1 + 1.1\kappa_2 \frac{3\kappa_1}{\kappa_2}}{\sqrt{m}} = \frac{4.3\kappa_1}{\sqrt{m}}.$$

Then for any $T_{\text{hit}} < t \le T_{\text{test},2}$, we have:

$$\langle \boldsymbol{b}_k(t), \boldsymbol{x}_+ \rangle \le \langle \boldsymbol{b}_k(T_{\text{hit}}), \boldsymbol{x}_+ \rangle - \int_{T_{\text{hit}}}^t \frac{0.89\kappa_2 p}{\sqrt{m}(1+p)}\mathrm{d}s \le \frac{4.3\kappa_1}{\sqrt{m}} - \frac{0.89\kappa_2 p(t - T_{\text{hit}})}{\sqrt{m}(1+p)}.$$

So we have the estimate

$$T_{\text{test},1} = T_{\text{test},2} \le T_{\text{hit}} + \frac{4.3\kappa_1(1+p)}{0.89\kappa_2 p} \le \left(3 + \frac{4.3 \cdot 6}{0.89 \cdot 5}\right)\frac{\kappa_1}{\kappa_2} \le \frac{9\kappa_1}{\kappa_2} < T_{\text{I}}.$$

Recalling $\boldsymbol{w}_k(T_{\text{test},1}) \in \mathcal{M}_+^- \cap \mathcal{M}_-^-$ and Lemma B.2, the neuron $\boldsymbol{b}_k(t)$ keeps dead for any $t \ge T_{\text{I}}$.

**Step IV.** Which subspace does the neuron select?

From **Step II**, we know that the neuron $\boldsymbol{w}_k(t)$ must arrives in $\mathcal{M}_+^0 \cap \mathcal{M}_-^+$ or $\mathcal{M}_-^0 \cap \mathcal{M}_-^+$ or $\mathcal{M}_+^0 \cap \mathcal{M}_-^0$. In this step, we will analyze which subspace does the neuron select.

We only need to compare the following two times:

$$T_{\text{hit},+} := \inf\left\{ t \in (0, T_{\text{I}}] : \langle \boldsymbol{b}_k(t), \boldsymbol{x}_+ \rangle \le 0 \right\},$$

$$T_{\text{hit},-} := \inf\left\{ t \in (0, T_{\text{I}}] : \langle \boldsymbol{b}_k(t), \boldsymbol{x}_- \rangle \le 0 \right\}.$$

From the definition of $T_{\text{hit}}$, we know $T_{\text{hit},+} = T_{\text{hit}}$ or $T_{\text{hit},-} = T_{\text{hit}}$.

Recalling the proof in **Step II**, we compare the following two dynamics for $t < T_{\text{hit}}$:

$$\frac{\mathrm{d}\langle \boldsymbol{b}_k(t), \boldsymbol{x}_+ \rangle}{\mathrm{d}t} = -\frac{\kappa_2}{\sqrt{m}}\left(\frac{p}{1+p}e^{-f_+(t)} - \frac{1}{1+p}e^{f_-(t)}\cos\Delta\right),$$

$$\frac{\mathrm{d}\langle \boldsymbol{b}_k(t), \boldsymbol{x}_- \rangle}{\mathrm{d}t} = -\frac{\kappa_2}{\sqrt{m}}\left(\frac{p}{1+p}e^{-f_+(t)}\cos\Delta - \frac{1}{1+p}e^{f_-(t)}\right).$$

With the help of Lemma C.2 and the estimate of $T_{\text{hit}}$, for any $t \le T_{\text{hit}}$,

$$|e^{-y_i f_i(t)} - 1| \le 1.1\kappa_2(\kappa_1 + 1.1\kappa_2 T_{\text{hit}}) \le 1.1\kappa_2(\kappa_1 + 3.3\kappa_1) = 4.73\kappa_1\kappa_2.$$

Hence, we have the estimate of the dynamics:

$$-\frac{\kappa_2\left((1 + 4.73\kappa_1\kappa_2)p - (1 - 4.73\kappa_1\kappa_2)\cos\Delta\right)}{\sqrt{m}(1+p)} \le \frac{\mathrm{d}\langle \boldsymbol{b}_k(t), \boldsymbol{x}_+ \rangle}{\mathrm{d}t} \le -\frac{\kappa_2\left((1 - 4.73\kappa_1\kappa_2)p - (1 + 4.73\kappa_1\kappa_2)\cos\Delta\right)}{\sqrt{m}(1+p)},$$

$$-\frac{\kappa_2\Big((1+4.73\kappa_1\kappa_2)p\cos\Delta-(1-4.73\kappa_1\kappa_2)\Big)}{\sqrt{m}(1+p)}\le\frac{\mathrm{d}\,\langle\boldsymbol{b}_k(t),\boldsymbol{x}_-\rangle}{\mathrm{d}t}\le-\frac{\kappa_2\Big((1-4.73\kappa_1\kappa_2)p\cos\Delta-(1+4.73\kappa_1\kappa_2)\Big)}{\sqrt{m}(1+p)}.$$

(i) If the initialization satisfies $\langle\boldsymbol{b}_k(0),\boldsymbol{x}_-\rangle>\frac{(1+4.73\kappa_1\kappa_2)p\cos\Delta-(1-4.73\kappa_1\kappa_2)}{(1-4.73\kappa_1\kappa_2)p-(1+4.73\kappa_1\kappa_2)\cos\Delta}\langle\boldsymbol{b}_k(0),\boldsymbol{x}_+\rangle$, we will prove that the neuron selects $\mathcal{M}_+^0\cap\mathcal{M}_-^+$ at $T_{\mathrm{hit}}$.

For any $t<T_{\mathrm{hit}}$, we have the estimate:

$$\langle\boldsymbol{b}_k(t),\boldsymbol{x}_+\rangle\le\langle\boldsymbol{b}_k(0),\boldsymbol{x}_+\rangle-\frac{\kappa_2\Big((1-4.73\kappa_1\kappa_2)p-(1+4.73\kappa_1\kappa_2)\cos\Delta\Big)}{\sqrt{m}(1+p)}t,$$

$$\langle\boldsymbol{b}_k(t),\boldsymbol{x}_-\rangle\ge\langle\boldsymbol{b}_k(0),\boldsymbol{x}_-\rangle-\frac{\kappa_2\Big((1+4.73\kappa_1\kappa_2)p\cos\Delta-(1-4.73\kappa_1\kappa_2)\Big)}{\sqrt{m}(1+p)}t$$

$$>\frac{(1+4.73\kappa_1\kappa_2)p\cos\Delta-(1-4.73\kappa_1\kappa_2)}{(1-4.73\kappa_1\kappa_2)p-(1+4.73\kappa_1\kappa_2)\cos\Delta}\left(\langle\boldsymbol{b}_k(0),\boldsymbol{x}_+\rangle-\frac{\kappa_2\Big((1-4.73\kappa_1\kappa_2)p-(1+4.73\kappa_1\kappa_2)\cos\Delta\Big)}{\sqrt{m}(1+p)}t\right),$$

Comparing these two inequalities, we have:

$$T_{\mathrm{hit},+}=T_{\mathrm{hit}}<T_{\mathrm{hit},-},$$

which means

$$\langle\boldsymbol{b}_k(T_{\mathrm{hit}}),\boldsymbol{x}_+\rangle=0,\quad\langle\boldsymbol{b}_k(T_{\mathrm{hit}}),\boldsymbol{x}_-\rangle>0.$$

So the neuron $\boldsymbol{w}_k(T_{\mathrm{hit}})\in\mathcal{M}_+^0\cap\mathcal{M}_-^+$.

(ii) If the initialization satisfies $\langle\boldsymbol{b}_k(0),\boldsymbol{x}_+\rangle>\frac{(1+4.73\kappa_1\kappa_2)p-(1-4.73\kappa_1\kappa_2)\cos\Delta}{(1-4.73\kappa_1\kappa_2)p\cos\Delta-(1+4.73\kappa_1\kappa_2)}\langle\boldsymbol{b}_k(0),\boldsymbol{x}_-\rangle$, we will prove that the neuron selects $\mathcal{M}_+^+\cap\mathcal{M}_-^-$ at $T_{\mathrm{hit}}$.

For any $t<T_{\mathrm{hit}}$, we have the estimate:

$$\langle\boldsymbol{b}_k(t),\boldsymbol{x}_-\rangle\le\langle\boldsymbol{b}_k(0),\boldsymbol{x}_-\rangle-\frac{\kappa_2\Big((1-4.73\kappa_1\kappa_2)p\cos\Delta-(1+4.73\kappa_1\kappa_2)\Big)}{\sqrt{m}(1+p)}t,$$

$$\langle\boldsymbol{b}_k(t),\boldsymbol{x}_+\rangle\ge\langle\boldsymbol{b}_k(0),\boldsymbol{x}_+\rangle-\frac{\kappa_2\Big((1+4.73\kappa_1\kappa_2)p-(1-4.73\kappa_1\kappa_2)\cos\Delta\Big)}{\sqrt{m}(1+p)}t$$

$$>\frac{(1+4.73\kappa_1\kappa_2)p-(1-4.73\kappa_1\kappa_2)\cos\Delta}{(1-4.73\kappa_1\kappa_2)p\cos\Delta-(1+4.73\kappa_1\kappa_2)}\left(\langle\boldsymbol{b}_k(0),\boldsymbol{x}_-\rangle-\frac{\kappa_2\Big((1-4.73\kappa_1\kappa_2)p\cos\Delta-(1+4.73\kappa_1\kappa_2)\Big)}{\sqrt{m}(1+p)}t\right),$$

Comparing these two inequalities, we have:

$$T_{\mathrm{hit},-}=T_{\mathrm{hit}}<T_{\mathrm{hit},+},$$

which means

$$\langle\boldsymbol{b}_k(T_{\mathrm{hit}}),\boldsymbol{x}_-\rangle=0,\quad\langle\boldsymbol{b}_k(T_{\mathrm{hit}}),\boldsymbol{x}_+\rangle>0.$$

So the neuron $\boldsymbol{w}_k(T_{\mathrm{hit}})\in\mathcal{M}_+^+\cap\mathcal{M}_-^0$.

$\square$

**Lemma C.8** (Negative, $\mathcal{M}_+^-\cap\mathcal{M}_-^+$)**.**
*For negative neuron $k\in\{k\in[m]-[m/2]:\boldsymbol{w}_k(0)\in\mathcal{M}_+^-\cap\mathcal{M}_-^+\}$, in Phase I ($t\le T_{\mathrm{I}}$), we have:*

*(S1). $\boldsymbol{w}_k(t)\in\mathcal{M}_+^0\cap\mathcal{M}_-^+$ for any $t\le\mathcal{O}\big(\frac{\kappa_1}{\kappa_2}p\Delta\big)$,*

*(S2). It has a small norm : $\rho_k(T_{\mathrm{I}})=\mathcal{O}\big(\frac{\sqrt{\kappa_1\kappa_2}}{\sqrt{m}}(\sqrt{\frac{\kappa_1}{\kappa_2}}+\frac{\Delta}{p})\big)$,*

*(S3). It is aligned with $\boldsymbol{x}_+^\perp$ : $\langle\boldsymbol{w}_k(T_{\mathrm{I}}),\boldsymbol{x}_+^\perp\rangle\ge1-\mathcal{O}\big((\sqrt{\frac{\kappa_1}{\kappa_2}}\frac{p}{\Delta})^{1.6}\big)$.*

*Proof of Lemma C.8.*
We do the following analysis for any $k \in \{k \in [m] - [m/2] : \boldsymbol{w}_k(0) \in \mathcal{M}_+^- \cap \mathcal{M}_-^+\}$, i.e. $s_k = -1$, $\langle \boldsymbol{w}_k(0), \boldsymbol{x}_+ \rangle \leq 0$, and $\langle \boldsymbol{w}_k(0), \boldsymbol{x}_- \rangle > 0$.

**Step I.** The neuron must arrives in $\mathcal{M}_+^0 \cap \mathcal{M}_-^+$ in $\mathcal{O}\left(\frac{\kappa_1 p \Delta}{\kappa_2}\right)$ time.

The case $\langle \boldsymbol{w}_k(0), \boldsymbol{x}_+ \rangle = 0$ is trivial. Then we only need to consider the case $\langle \boldsymbol{w}_k(0), \boldsymbol{x}_+ \rangle < 0$.

First, we define the hitting time

$$T_{\text{hit}} := \inf \left\{ t \in (0, T_{\text{I}}] : \boldsymbol{w}_k(t) \notin \mathcal{M}_+^- \cap \mathcal{M}_-^+ \right\},$$

and we aim to estimate $T_{\text{hit}}$ and prove $\boldsymbol{w}_k(T_{\text{hit}}) \in \mathcal{M}_+^0 \cap \mathcal{M}_-^+$.

We focus on the dynamics of $\langle \boldsymbol{b}_k(t), \boldsymbol{x}_+ \rangle$ and $\langle \boldsymbol{b}_k(t), \boldsymbol{x}_- \rangle$.

From the definition of $T_{\text{hit}}$ and (4), the dynamics of the neuron is:

$$\frac{\mathrm{d}\boldsymbol{b}_k(t)}{\mathrm{d}t} = \frac{\kappa_2}{\sqrt{m}} \frac{1}{1+p} e^{f_-(t)} \boldsymbol{x}_-, \ t \leq T_{\text{hit}}.$$

Then we have

$$\frac{\mathrm{d}\langle \boldsymbol{b}_k(t), \boldsymbol{x}_+ \rangle}{\mathrm{d}t} = \left\langle \frac{\kappa_2}{\sqrt{m}} \frac{1}{1+p} e^{f_-(t)} \boldsymbol{x}_-, \boldsymbol{x}_+ \right\rangle = \frac{\kappa_2 \cos \Delta}{\sqrt{m}(1+p)} e^{f_-(t)},$$

$$\frac{\mathrm{d}\langle \boldsymbol{b}_k(t), \boldsymbol{x}_- \rangle}{\mathrm{d}t} = \left\langle \frac{\kappa_2}{\sqrt{m}} \frac{1}{1+p} e^{f_-(t)} \boldsymbol{x}_-, \boldsymbol{x}_- \right\rangle = \frac{\kappa_2}{\sqrt{m}(1+p)} e^{f_-(t)}.$$

It is clear $\frac{\mathrm{d}\langle \boldsymbol{b}_k(t), \boldsymbol{x}_- \rangle}{\mathrm{d}t} > 0$, so $\langle \boldsymbol{b}_k(t), \boldsymbol{x}_- \rangle > \langle \boldsymbol{b}_k(0), \boldsymbol{x}_- \rangle > 0$ for any $t \leq T_{\text{hit}}$. If we denote

$$T_{\text{hit},+} := \inf \left\{ t \in (0, T_{\text{I}}] : \langle \boldsymbol{w}_k(t), \boldsymbol{x}_+ \rangle \leq 0 \right\},$$

then it holds:
$$T_{\text{hit}} = T_{\text{hit},+}.$$

So we only need to estimate $T_{\text{hit},+}$. Due to $T_{\text{hit}} \leq T_{\text{I}} \leq T_{\text{init}}$ and Lemma C.2, for any $t \leq T_{\text{hit}}$, we have $\left| e^{-y_i f_i(t)} - 1 \right| \leq 0.11$. Then for any $t \leq T_{\text{hit}}$, we have:

$$\frac{\mathrm{d}\langle \boldsymbol{b}_k(t), \boldsymbol{x}_+ \rangle}{\mathrm{d}t} \geq \frac{0.89 \kappa_2 \cos \Delta}{\sqrt{m}(1+p)}.$$

Recalling $\langle \boldsymbol{w}_k(0), \boldsymbol{x}_+ \rangle < 0$ and $\langle \boldsymbol{w}_k(0), \boldsymbol{x}_- \rangle > 0$, with the help of Lemma I.2, we have $\langle \boldsymbol{w}_k(0), \boldsymbol{x}_+ \rangle > -\sin \Delta$ and $\langle \boldsymbol{w}_k(0), \boldsymbol{x}_- \rangle < \sin \Delta$. Combining the two estimate, we have:

$$\langle \boldsymbol{b}_k(t), \boldsymbol{x}_+ \rangle \geq \langle \boldsymbol{b}_k(0), \boldsymbol{x}_+ \rangle + \int_0^t \frac{0.89 \kappa_2 \cos \Delta}{\sqrt{m}(1+p)} \mathrm{d}t$$

$$> -\rho_k(0) \sin \Delta + \frac{0.89 \kappa_2 \cos \Delta}{\sqrt{m}(1+p)} t = -\frac{\kappa_1 \sin \Delta}{\sqrt{m}} + \frac{0.89 \kappa_2 \cos \Delta}{\sqrt{m}(1+p)} t.$$

Hence,

$$T_{\text{hit}} = T_{\text{hit},+} \leq \frac{(1+p) \tan \Delta}{0.89} \frac{\kappa_1}{\kappa_2} \leq 2p\Delta \frac{\kappa_1}{\kappa_2} < T_{\text{I}} = 10\sqrt{\frac{\kappa_1}{\kappa_2}}.$$

Moreover, we can estimate of $\rho_k(T_{\text{hit}})$.

Since $\langle \boldsymbol{w}_k(t), \boldsymbol{x}_+ \rangle < 0$ and $\langle \boldsymbol{w}_k(t), \boldsymbol{x}_- \rangle > 0$ hold for any $t \leq T_{\text{hit}}$, with the help of Lemma I.2, we have $\langle \boldsymbol{w}_k(t), \boldsymbol{x}_- \rangle < \sin \Delta$. Combining (7), for any $t \leq T_{\text{hit}}$, we have

$$\rho_k(t) \leq \rho_k(0) + \int_0^t \frac{\kappa_2}{\sqrt{m}(1+p)} e^{f_-(t)} \langle \boldsymbol{w}_k(t), \boldsymbol{x}_- \rangle \mathrm{d}t$$

$$\leq \frac{\kappa_1}{\sqrt{m}} + \int_0^t \frac{1.11 \kappa_2}{\sqrt{m}(1+p)} \sin \Delta \mathrm{d}t \leq \frac{\kappa_1}{\sqrt{m}} + \frac{1.11 \kappa_2 \sin \Delta}{\sqrt{m}(1+p)} t.$$

**Step II.** Dynamics after arriving in the manifold $\mathcal{M}_+^0 \cap \mathcal{M}_-^+$.

Proceeding as in the proof of **Step II** in the Proof of Theorem C.7, we have:

$\boldsymbol{w}_k(t)$ can not leave $\mathcal{M}_+^0 \cap \mathcal{M}_-^+$ for $T_{\text{hit}} \leq t \leq T_{\text{I}}$. Moreover, the dynamics of $\boldsymbol{w}_k$ on $\mathcal{M}_+^0 \cap \mathcal{M}_-^+$ satisfies:

$$\frac{\mathrm{d}\boldsymbol{w}_k(t)}{\mathrm{d}t} = \frac{\kappa_2 e^{f_-(t)}}{\rho_k(t)\sqrt{m}(1+p)}\Big(\boldsymbol{x}_- - \langle \boldsymbol{w}_k, \boldsymbol{x}_- \rangle \boldsymbol{w}_k - \boldsymbol{x}_+ \cos\Delta\Big),$$

$$\frac{\mathrm{d}\rho_k(t)}{\mathrm{d}t} = \frac{\kappa_2 e^{f_-(t)}}{\sqrt{m}(1+p)}\langle \boldsymbol{w}_k(t), \boldsymbol{x}_- \rangle,$$

$$\frac{\mathrm{d}\boldsymbol{b}_k(t)}{\mathrm{d}t} = \frac{\kappa_2 e^{f_-(t)}}{\sqrt{m}(1+p)}\Big(\boldsymbol{x}_- - \boldsymbol{x}_+ \cos\Delta\Big).$$

Recalling the estimate of $T_{\text{hit}}$ in **Step I**, we have

$$\rho_k(T_{\text{hit}}) \leq \frac{\kappa_1}{\sqrt{m}} + \frac{1.11\kappa_2 \sin\Delta}{\sqrt{m}(1+p)}T_{\text{hit}}.$$

As the proof of **Step II** in the Proof of Theorem C.7, for any $T_{\text{hit}} < t \leq T_{\text{I}}$, we have

$$\rho_k(t) = \rho_k(T_{\text{hit}}) + \int_{T_{\text{hit}}}^t \frac{\mathrm{d}\rho_k(s)}{\mathrm{d}s}\mathrm{d}s \leq \frac{\kappa_1}{\sqrt{m}} + \frac{1.11\kappa_2 \sin\Delta}{\sqrt{m}(1+p)}T_{\text{hit}} + \frac{\kappa_2 e^{f_-(t)} \sin\Delta}{\sqrt{m}(1+p)}(t - T_{\text{hit}})$$

$$\leq \frac{\kappa_1}{\sqrt{m}} + \frac{1.11\kappa_2 \sin\Delta}{\sqrt{m}(1+p)}T_{\text{hit}} + \frac{1.11\kappa_2 \sin\Delta}{\sqrt{m}(1+p)}(t - T_{\text{hit}}) = \frac{\kappa_1}{\sqrt{m}} + \frac{1.11\kappa_2 \sin\Delta}{\sqrt{m}(1+p)}t.$$

Combining the estimate in **Step I**, for any $0 < t \leq T_{\text{I}}$, we have:

$$\rho_k(t) \leq \frac{\kappa_1}{\sqrt{m}} + \frac{1.11\kappa_2 \sin\Delta}{\sqrt{m}(1+p)}t.$$

Specifically, we have:

$$\rho_k(T_{\text{I}}) \leq \frac{\kappa_1}{\sqrt{m}} + \frac{11.1\sqrt{\kappa_1\kappa_2} \sin\Delta}{\sqrt{m}(1+p)} \leq \frac{\sqrt{\kappa_1\kappa_2}}{\sqrt{m}}\Big(\sqrt{\frac{\kappa_1}{\kappa_2}} + 11.1\frac{\Delta}{1+p}\Big).$$

Similar to the proof of **Step II** in the Proof of Theorem C.7, we have the estimate of the dynamics of $\langle \boldsymbol{w}_k(t), \boldsymbol{x}_- \rangle$:

$$\frac{\mathrm{d}\langle \boldsymbol{w}_k(t), \boldsymbol{x}_- \rangle}{\mathrm{d}t} \geq \frac{0.89\kappa_2}{(1+p)\kappa_1 + 1.11\kappa_2 t \sin\Delta}\Big(\sin^2\Delta - \langle \boldsymbol{w}_k(t), \boldsymbol{x}_- \rangle^2\Big), \quad 0 < t \leq T_{\text{I}},$$

$$0 < \langle \boldsymbol{w}_k(0), \boldsymbol{x}_- \rangle < \sin\Delta.$$

In the same way, we can derive

$$\langle \boldsymbol{w}_k(T_{\text{I}}), \boldsymbol{x}_- \rangle > \left(1 - \frac{2}{\Big(1 + 11.1\sqrt{\frac{\kappa_2}{\kappa_1}\frac{\sin\Delta}{1+p}}\Big)^{1.6} + 1}\right)\sin\Delta$$

Hence, we have the estimate of $\langle \boldsymbol{w}_k(t), \boldsymbol{x}_+^{\perp} \rangle$:

$$\langle \boldsymbol{w}_k(T_{\text{I}}), \boldsymbol{x}_+^{\perp} \rangle = \frac{1}{\sin\Delta}\langle \boldsymbol{w}_k(T_{\text{I}}), \boldsymbol{x}_- \rangle > 1 - \frac{2}{\Big(1 + 11.1\sqrt{\frac{\kappa_2}{\kappa_1}\frac{\sin\Delta}{1+p}}\Big)^{1.6} + 1}.$$

$\square$

**Lemma C.9** (Negative, $\mathcal{M}_+^+ \cap \mathcal{M}_-^-$).
*For negative neuron $k \in \{k \in [m] - [m/2] : \boldsymbol{w}_k(0) \in \mathcal{M}_+^+ \cap \mathcal{M}_-^-\}$, it keeps dead:*

$$\boldsymbol{w}_k(t) \in \mathcal{M}_+^- \cap \mathcal{M}_-^-, \text{ for any } t \geq T_{\text{I}} > \mathcal{O}\Big(\frac{\kappa_1\Delta}{\kappa_2}\Big).$$

*Proof of Lemma C.9.*
We do the following analysis for any $k \in \{k \in [m] - [m/2] : \boldsymbol{w}_k(0) \in \mathcal{M}_+^+ \cap \mathcal{M}_-^-\}$, i.e. $\mathrm{s}_k = -1$, $\langle \boldsymbol{w}_k(0), \boldsymbol{x}_+ \rangle > 0$, and $\langle \boldsymbol{w}_k(0), \boldsymbol{x}_- \rangle \leq 0$.

First, we define the hitting time

$$T_{\mathrm{hit}} := \inf \left\{ t \in (0, T_{\mathrm{I}}] : \boldsymbol{w}_k(t) \notin \mathcal{M}_+^+ \cap \mathcal{M}_-^- \right\},$$

and we aim to estimate $T_{\mathrm{hit}}$ and prove $\boldsymbol{w}_k(T_{\mathrm{hit}}) \in \mathcal{M}_+^- \cap \mathcal{M}_-^-$.

From the definition of $T_{\mathrm{hit}}$ and (4), the dynamics of the neuron is:

$$\frac{\mathrm{d}\boldsymbol{b}_k(t)}{\mathrm{d}t} = -\frac{\kappa_2}{\sqrt{m}} \frac{p}{1+p} e^{-f_+(t)} \boldsymbol{x}_+, \ t \leq T_{\mathrm{hit}}.$$

Then we have

$$\frac{\mathrm{d}\langle \boldsymbol{b}_k(t), \boldsymbol{x}_+ \rangle}{\mathrm{d}t} = \left\langle -\frac{\kappa_2}{\sqrt{m}} \frac{p}{1+p} e^{-f_+(t)} \boldsymbol{x}_+, \boldsymbol{x}_+ \right\rangle = -\frac{\kappa_2}{\sqrt{m}} \frac{p}{1+p} e^{-f_+(t)},$$

$$\frac{\mathrm{d}\langle \boldsymbol{b}_k(t), \boldsymbol{x}_- \rangle}{\mathrm{d}t} = \left\langle -\frac{\kappa_2}{\sqrt{m}} \frac{p}{1+p} e^{-f_+(t)} \boldsymbol{x}_+, \boldsymbol{x}_- \right\rangle = -\frac{\kappa_2}{\sqrt{m}} \frac{p}{1+p} e^{-f_+(t)} \cos \Delta.$$

It is clear $\frac{\mathrm{d}\langle \boldsymbol{b}_k(t), \boldsymbol{x}_- \rangle}{\mathrm{d}t} < 0$, so $\langle \boldsymbol{b}_k(t), \boldsymbol{x}_- \rangle < \langle \boldsymbol{b}_k(0), \boldsymbol{x}_- \rangle \leq 0$ for any $t \leq T_{\mathrm{hit}}$. If we denote

$$T_{\mathrm{hit},+} := \inf \left\{ t \in (0, T_{\mathrm{I}}] : \langle \boldsymbol{w}_k(t), \boldsymbol{x}_+ \rangle \leq 0 \right\},$$

then it holds:

$$T_{\mathrm{hit}} = T_{\mathrm{hit},+}.$$

So we only need to estimate $T_{\mathrm{hit},+}$. Due to $T_{\mathrm{hit}} \leq T_{\mathrm{I}} \leq T_{\mathrm{init}}$ and Lemma C.2, for any $t \leq T_{\mathrm{hit}}$, we have $\left| e^{-y_i f_i(t)} - 1 \right| \leq 0.11$. Then for any $t \leq T_{\mathrm{hit}}$, we have:

$$\frac{\mathrm{d}\langle \boldsymbol{b}_k(t), \boldsymbol{x}_+ \rangle}{\mathrm{d}t} \leq -\frac{0.89\kappa_2 p}{\sqrt{m}(1+p)}.$$

Recalling $\langle \boldsymbol{w}_k(0), \boldsymbol{x}_+ \rangle > 0$ and $\langle \boldsymbol{w}_k(0), \boldsymbol{x}_- \rangle \leq 0$, with the help of Lemma I.2, we have $\langle \boldsymbol{w}_k(0), \boldsymbol{x}_+ \rangle \leq \sin \Delta$. Combining the two estimate, we have:

$$\langle \boldsymbol{b}_k(t), \boldsymbol{x}_+ \rangle \leq \langle \boldsymbol{b}_k(0), \boldsymbol{x}_+ \rangle - \int_0^t \frac{0.89\kappa_2 p}{\sqrt{m}(1+p)} \mathrm{d}t$$

$$\leq \rho_k(0) \sin \Delta - \frac{0.89\kappa_2 p}{\sqrt{m}(1+p)} t = \frac{\kappa_1 \sin \Delta}{\sqrt{m}} - \frac{0.89\kappa_2 p}{\sqrt{m}(1+p)} t.$$

Hence,

$$T_{\mathrm{hit}} = T_{\mathrm{hit},+} \leq \frac{(1+p)\sin \Delta}{0.89p} \frac{\kappa_1}{\kappa_2} < T_{\mathrm{I}} = 10\sqrt{\frac{\kappa_1}{\kappa_2}}.$$

Moreover, the analysis gives us $\boldsymbol{w}_k(T_{\mathrm{hit}}) \in \mathcal{M}_+^- \cap \mathcal{M}_-^-$. By Lemma B.2, we obtain:

$$\boldsymbol{w}_k(t) \in \mathcal{M}_+^- \cap \mathcal{M}_-^-, \text{ for any } t \geq T_{\mathrm{hit}}.$$

$\square$

**Lemma C.10** (Negative, $\mathcal{M}_+^- \cap \mathcal{M}_-^-$)**.**
*For negative neuron* $k \in \{k \in [m] - [m/2] : \boldsymbol{w}_k(0) \in \mathcal{M}_+^- \cap \mathcal{M}_-^-\}$*, it keeps dead:* $\boldsymbol{w}_k(t) \in \mathcal{M}_+^- \cap \mathcal{M}_-^-$ *for any* $t \geq 0$.

*Proof of Lemma C.10.* Due to Lemma B.2, this lemma is trivial. $\square$

## C.3 Initialization Estimation and Proof of Theorem 4.1

To get the number of neurons in the eight classes in the subsection above, we also need to estimate the initial positions of these neurons under the random initialization.

**Lemma C.11** (Initialization Estimation).
*If $m = \Omega\big(\log(1/\delta)\big)$, then with probability at least $1 - \delta$, we have:*

$$\left| \#\Big\{ k \in [m/2] : \langle \boldsymbol{w}_k(0), \boldsymbol{x}_+ \rangle > 0 \Big\} - \frac{m}{4} \right| \leq 0.04m,$$

$$\#\Big\{ k \in [m] - [m/2] : \langle \boldsymbol{w}_k(0), \boldsymbol{x}_- \rangle > 0, \langle \boldsymbol{w}_k(0), \boldsymbol{x}_- \rangle > A \langle \boldsymbol{w}_k(0), \boldsymbol{x}_+ \rangle \Big\} \geq 0.075m,$$

$$\#\Big\{ k \in [m] - [m/2] : \langle \boldsymbol{w}_k(0), \boldsymbol{x}_- \rangle > 0, \langle \boldsymbol{w}_k(0), \boldsymbol{x}_+ \rangle \leq B \langle \boldsymbol{w}_k(0), \boldsymbol{x}_- \rangle \Big\} \leq 0.205m.$$

*where $A = \frac{(1+4.73\kappa_1\kappa_2)p\cos\Delta - (1-4.73\kappa_1\kappa_2)}{(1-4.73\kappa_1\kappa_2)p - (1+4.73\kappa_1\kappa_2)\cos\Delta}$ and $B = \frac{(1+4.73\kappa_1\kappa_2)p - (1-4.73\kappa_1\kappa_2)\cos\Delta}{(1-4.73\kappa_1\kappa_2)p\cos\Delta - (1+4.73\kappa_1\kappa_2)}$ (mentioned in Lemma C.7).*

*Proof of Lemma C.11.*
(i) By Hoeffding's Inequality (Lemma I.1), for any $\epsilon > 0$ we have:

$$\mathbb{P}\left( \left| \#\Big\{ k \in [m/2] : \langle \boldsymbol{w}_k(0), \boldsymbol{x}_+ \rangle > 0 \Big\} - \frac{m}{4} \right| \geq \frac{m\epsilon}{2} \right) = \mathbb{P}\left( \left| \frac{2}{m} \sum_{k \in [m/2]} \mathbb{I}\Big\{ \langle \boldsymbol{w}_k(0), \boldsymbol{x}_+ \rangle > 0 \Big\} - \frac{1}{2} \right| \geq \epsilon \right)$$

$$= \mathbb{P}\left( \left| \frac{2}{m} \sum_{k \in [m/2]} \mathbb{I}\Big\{ \langle \boldsymbol{w}_k(0), \boldsymbol{x}_+ \rangle > 0 \Big\} - \mathbb{E}\Big[ \mathbb{I}\Big\{ \langle \boldsymbol{w}_1(0), \boldsymbol{x}_+ \rangle > 0 \Big\} \Big] \right| \geq \epsilon \right)$$

$$\leq 2\exp\left( -\frac{2(\frac{m}{2})^2\epsilon^2}{\frac{m}{2}} \right) = 2\exp(-m\epsilon^2).$$

(ii) From $\boldsymbol{w}_k(0) \sim \mathbb{U}(\mathbb{S}^{d-1})$, without loss of generality, we can let $\boldsymbol{x}_- = \boldsymbol{e}_1$ and $\boldsymbol{x}_+ = \boldsymbol{e}_1 \cos\Delta + \boldsymbol{e}_2 \sin\Delta$.

So we have:

$$\Big\{ k \in [m] - [m/2] : \langle \boldsymbol{w}_k(0), \boldsymbol{x}_- \rangle > 0, \langle \boldsymbol{w}_k(0), \boldsymbol{x}_- \rangle > A \langle \boldsymbol{w}_k(0), \boldsymbol{x}_+ \rangle \Big\}$$

$$= \Big\{ k \in [m] - [m/2] : w_{k,1}(0) > 0, w_{k,1}(0) > A\Big( w_{k,1}(0)\cos\Delta + w_{k,2}(0)\sin\Delta \Big) \Big\}$$

$$= \Big\{ k \in [m] - [m/2] : w_{k,1}(0) > 0, (1 - A\cos\Delta)w_{k,1}(0) > Aw_{k,2}(0)\sin\Delta \Big\}.$$

From (6), we have $A > 0$ and

$$A = 1 + \frac{\Big( (1+4.73\kappa_1\kappa_2)\cos\Delta - (1-4.73\kappa_1\kappa_2) \Big)(p+1)}{(1-4.73\kappa_1\kappa_2)p - (1+4.73\kappa_1\kappa_2)\cos\Delta}$$

$$\leq 1 + \frac{4.73\kappa_1\kappa_2(1+\cos\Delta)}{1-4.73\kappa_1\kappa_2} \frac{p+1}{p-\frac{10}{9}} \leq 1 + \frac{9.46\kappa_1\kappa_2}{1-\frac{1}{19}} \frac{90}{71} \leq 1 + 12.66\kappa_1\kappa_2,$$

$$A = 1 + \frac{\Big( (1+4.73\kappa_1\kappa_2)\cos\Delta - (1-4.73\kappa_1\kappa_2) \Big)(p+1)}{(1-4.73\kappa_1\kappa_2)p - (1+4.73\kappa_1\kappa_2)\cos\Delta}$$

$$\geq 1 - \frac{4.73\kappa_1\kappa_2(1+\cos\Delta) + (1-\cos\Delta)}{1-4.73\kappa_1\kappa_2} \frac{p+1}{p-\frac{10}{9}} \geq 1 - \frac{\frac{2}{19} + \frac{\Delta^2}{2}}{1-\frac{1}{19}} \frac{90}{71}$$

$$\geq 1 - \frac{\frac{2}{19} + \frac{1}{19}}{\frac{18}{19}} \frac{90}{71} \geq 0.78,$$

$$\frac{1 - A\cos\Delta}{A\sin\Delta} \geq \frac{1 - A}{A\sin\Delta} \geq -\frac{12.66\kappa_1\kappa_2}{A\sin\Delta} \geq -\frac{12.66\kappa_1\kappa_2}{0.78\sin\Delta} \geq -\frac{12.66\kappa_1\kappa_2}{0.78\frac{2}{\pi}\Delta} \geq -\frac{25.5\kappa_1\kappa_2}{\Delta} \geq -\frac{1}{100}.$$

For simplicity, we denote the event

$$A_k := \left\{ w_{k,1}(0) > 0, -\frac{1}{100}w_{k,1}(0) > w_{k,2}(0) \right\}, \ k \in [m] - [m/2].$$

Then we have the estimate:

$$\#\left\{ k \in [m] - [m/2] : w_{k,1}(0) > 0, (1 - A\cos\Delta)w_{k,1}(0) > Aw_{k,2}(0)\sin\Delta \right\}$$
$$\geq \#\left\{ k \in [m] - [m/2] : w_{k,1}(0) > 0, -\frac{1}{100}w_{k,1}(0) > w_{k,2}(0) \right\} = \sum_{k \in [m] - [m/2]} \mathbb{I}\{A_k\}.$$

We first estimate the lower bound for $\mathbb{E}[\mathbb{I}\{A_m\}]$:

$$\mathbb{E}\left[\mathbb{I}\{A_m\}\right] = \mathbb{P}(A_m) = \mathbb{P}\left( w_{m,1}(0) > 0, -\frac{1}{100}w_{m,1}(0) > w_{k,2}(0) \right)$$

$$\stackrel{g\sim\mathcal{N}(\mathbf{0},\mathbf{I}_d)}{=} \mathbb{P}\left( \frac{g_1}{\|g\|} > 0, -\frac{1}{100}\frac{g_1}{\|g\|} > \frac{g_2}{\|g\|} \right) = \mathbb{P}\left( g_1 > 0, g_1 < -100g_2 \right) = \mathbb{P}\left( g_1 > 0, g_1 < 100g_2 \right)$$

$$= \mathbb{P}\left( 100g_2 > g_1 > 0 \right) \geq \sup_{t>0}\mathbb{P}\left( g_2 > t, 100t > g_1 > 0 \right) \stackrel{g\sim\mathcal{N}(0,1)}{=} \sup_{t>0}\mathbb{P}\left( g > t \right)\mathbb{P}\left( 100t > g > 0 \right)$$

$$\geq \mathbb{P}\left( g > \frac{1}{10} \right)\mathbb{P}\left( 10 > g > 0 \right) \geq 0.23.$$

Secondly, by Hoeffding's inequality (Lemma I.1), for any $\epsilon > 0$, we have

$$\mathbb{P}\left( \sum_{k\in[m]-[m/2]}\mathbb{I}\{A_k\} - 0.115m \leq -\frac{m}{2}\epsilon \right) \leq \mathbb{P}\left( \sum_{k\in[m]-[m/2]}\mathbb{I}\{A_k\} - \frac{m}{2}\mathbb{E}\left[\mathbb{I}\{A_m\}\right] \leq -\frac{m}{2}\epsilon \right)$$

$$= \mathbb{P}\left( \frac{2}{m}\sum_{k\in[m]-[m/2]}\mathbb{I}\{A_k\} - \mathbb{E}\left[\mathbb{I}\{A_m\}\right] \leq -\epsilon \right) \leq \exp\left( -\frac{2(\frac{m}{2})^2\epsilon^2}{\frac{m}{2}} \right) = \exp(-m\epsilon^2)$$

(iii) This proof is similar to (ii). From $\boldsymbol{w}_k(0) \sim \mathbb{U}(\mathbb{S}^{d-1})$, without loss of generality, we can let $\boldsymbol{x}_- = \boldsymbol{e}_1$ and $\boldsymbol{x}_+ = \boldsymbol{e}_1\cos\Delta + \boldsymbol{e}_2\sin\Delta$.

so we have:

$$\left\{ k \in [m] - [m/2] : \langle \boldsymbol{w}_k(0), \boldsymbol{x}_- \rangle > 0, \langle \boldsymbol{w}_k(0), \boldsymbol{x}_+ \rangle \leq B\langle \boldsymbol{w}_k(0), \boldsymbol{x}_- \rangle \right\}$$
$$= \left\{ k \in [m] - [m/2] : w_{k,1}(0) > 0, w_{k,1}(0)\cos\Delta + w_{k,2}(0)\sin\Delta \leq Bw_{k,1}(0) \right\}$$
$$= \left\{ k \in [m] - [m/2] : w_{k,1}(0) > 0, (B - \cos\Delta)w_{k,1}(0) > w_{k,2}(0)\sin\Delta \right\}.$$

From (6), we have $B > 0$ and

$$B - \cos\Delta = \frac{(1 + 4.73\kappa_1\kappa_2)p - (1 - 4.73\kappa_1\kappa_2)\cos\Delta}{(1 - 4.73\kappa_1\kappa_2)p\cos\Delta - (1 + 4.73\kappa_1\kappa_2)} - \cos\Delta$$

$$= \frac{(1 + 4.73\kappa_1\kappa_2)(p + \cos\Delta) - (1 - 4.73\kappa_1\kappa_2)(1 + p\cos\Delta)\cos\Delta}{(1 - 4.73\kappa_1\kappa_2)p\cos\Delta - (1 + 4.73\kappa_1\kappa_2)}$$

$$= \frac{p\sin^2\Delta + 4.73\kappa_1\kappa_2(p + 2\cos\Delta + p\cos^2\Delta)}{(1 - 4.73\kappa_1\kappa_2)p\cos\Delta - (1 + 4.73\kappa_1\kappa_2)} \leq \frac{\sin^2\Delta + 9.46\kappa_1\kappa_2}{1 - 4.73\kappa_1\kappa_2}\frac{p + 1}{p\cos\Delta - \frac{10}{9}}$$

$$\leq \frac{\sin^2\Delta + 9.46\frac{\Delta}{2550}}{1 - \frac{1}{19}}\frac{p + 1}{\frac{9}{10}p - \frac{10}{9}} \leq \frac{\sin^2\Delta + 9.46\frac{\pi\sin\Delta}{5100}}{1 - \frac{1}{19}}\frac{p + 1}{\frac{9}{10}p - \frac{10}{9}}$$

$$\leq \frac{0.315 + \frac{9.46\pi}{5100}}{\frac{18}{19}}\frac{10}{\frac{81}{10} - \frac{10}{9}}\sin\Delta \leq \frac{\sin\Delta}{2},$$

For simplicity, we denote the event

$$B_k := \left\{ w_{k,1}(0) > 0, \frac{1}{2} w_{k,1}(0) > w_{k,2}(0) \right\}, \; k \in [m] - [m/2].$$

Then we have the estimate:

$$\# \left\{ k \in [m] - [m/2] : w_{k,1}(0) > 0, (B - \cos \Delta) w_{k,1}(0) > w_{k,2}(0) \sin \Delta \right\}$$

$$\leq \# \left\{ k \in [m] - [m/2] : w_{k,1}(0) > 0, \frac{1}{2} w_{k,1}(0) > w_{k,2}(0) \right\} = \sum_{k \in [m] - [m/2]} \mathbb{I}\{B_k\}.$$

We first estimate the lower bound for $\mathbb{E}[\mathbb{I}\{B_m\}]$:

$$\mathbb{E}\left[\mathbb{I}\{B_m\}\right] = \mathbb{P}(B_m) = \mathbb{P}\left(w_{m,1}(0) > 0, \frac{1}{2} w_{m,1}(0) > w_{k,2}(0)\right)$$

$$\overset{\boldsymbol{g} \sim \mathcal{N}(\mathbf{0}, \mathbf{I}_d)}{=} \mathbb{P}\left(\frac{g_1}{\|\boldsymbol{g}\|} > 0, \frac{1}{2} \frac{g_1}{\|\boldsymbol{g}\|} > \frac{g_2}{\|\boldsymbol{g}\|}\right) = \mathbb{P}\left(g_1 > 0, g_1 > 2g_2\right) = \mathbb{P}\left(g_1 > 0, g_2 \leq 0\right) + \mathbb{P}\left(g_1 > 2g_2 > 0\right)$$

$$= \frac{1}{4} + \mathbb{P}\left(g_1 > 2g_2 > 0\right) \leq \frac{1}{4} + \frac{1}{2\pi} \int_0^{+\infty} e^{-\frac{x^2}{2}} \int_0^{\frac{x}{2}} e^{-\frac{y^2}{2}} \mathrm{d}y \mathrm{d}x \leq \frac{1}{4} + \frac{1}{2\pi} \int_0^{+\infty} \frac{x}{2} e^{-\frac{x^2}{2}} \mathrm{d}x$$

$$\leq \frac{1}{4} + \frac{1}{4\pi}.$$

Secondly, by Hoeffding's inequality (Lemma I.1), for any $\epsilon > 0$, we have

$$\mathbb{P}\left(\sum_{k \in [m] - [m/2]} \mathbb{I}\{B_k\} - \left(\frac{1}{8} + \frac{1}{8\pi}\right) m \geq \frac{m}{2} \epsilon\right) \leq \mathbb{P}\left(\sum_{k \in [m] - [m/2]} \mathbb{I}\{B_k\} - \frac{m}{2} \mathbb{E}\left[\mathbb{I}\{B_m\}\right] \geq \frac{m}{2} \epsilon\right)$$

$$= \mathbb{P}\left(\frac{2}{m} \sum_{k \in [m] - [m/2]} \mathbb{I}\{B_k\} - \mathbb{E}\left[\mathbb{I}\{B_m\}\right] \geq \epsilon\right) \leq \exp\left(-\frac{2(\frac{m}{2})^2 \epsilon^2}{\frac{m}{2}}\right) = \exp(-m\epsilon^2).$$

let $\epsilon = 0.08$ and $\delta = 4 \exp(-m\epsilon^2/2)$. Combining the uniform bounds in (i)(ii)(iii), we obtain this theorem:

If $m \geq \frac{2 \log(4/\delta)}{0.08^2}$, then with probability at least $1 - \delta$, we have:

$$\left|\# \left\{ k \in [m/2] : \langle \boldsymbol{w}_k(0), \boldsymbol{x}_+ \rangle > 0 \right\} - \frac{m}{4}\right| \leq 0.04m,$$

$$\# \left\{ k \in [m] - [m/2] : w_{k,1}(0) > 0, (1 - A \cos \Delta) w_{k,1}(0) > A w_{k,2}(0) \sin \Delta \right\} \geq 0.075m,$$

$$\# \left\{ k \in [m] - [m/2] : w_{k,1}(0) > 0, (B - \cos \Delta) w_{k,1}(0) > w_{k,2}(0) \sin \Delta \right\} \leq 0.205m.$$

$\square$

So far, Lemma C.3, C.4, C.5, C.6, C.7, C.8, C.9, C.10 characterize the training dynamics of each neuron in Phase I, and Lemma C.11 estimate the initial positions of the neurons. Now we can prove our main theorem in Phase I.

**Theorem C.12** (Restatement of Theorem 4.1).
*Under the data Assumption 3.1, let the two-layer network trained by Gradient Flow (2) starting from random initialization. Let the width $m = \Omega\left(\log(1/\delta)\right)$, the initialization scales satisfy (6). Then with probability at least $1 - \delta$, the following results (S1)~(S5) hold at the end of Phase I ($T_{\mathrm{I}} = 10\sqrt{\frac{\kappa_1}{\kappa_2}}$):*

**(S1).** *For positive neurons $k \in [m/2]$ ($s_k = 1$), let $\mathcal{K}_+$ be the index set of living neurons, i.e. $\mathcal{K}_+ := \{k \in [m/2] : \boldsymbol{w}_k(T_{\mathrm{I}}) \in \mathcal{M}_+^+ \cup \mathcal{M}_-^+\}$. Then $0.21m \leq |\mathcal{K}_+| \leq 0.29m$. Moreover, for any neuron $k \in \mathcal{K}_+$, it has the following properties (P1)(P2).*

    **(P1).** *Its norm is small but significant :* $\dfrac{4.66\sqrt{\kappa_1 \kappa_2}}{\sqrt{m}} \leq \rho_k(T_{\mathrm{I}}) \leq \dfrac{12\sqrt{\kappa_1 \kappa_2}}{\sqrt{m}}.$

**(P2).** *Its direction is strongly aligned with $\boldsymbol{\mu}$ :*

$$\langle \boldsymbol{w}_k(T_\mathrm{I}), \boldsymbol{\mu}\rangle \geq \left(1 - 4.2\sqrt{\kappa_1\kappa_2}\right)\left(1 - \frac{2}{1 + 0.7\left(1 + 9.9\sqrt{\frac{\kappa_2}{\kappa_1}}\right)^{1.15}}\right).$$

**(S2).** *For negative neurons $k \in [m] - [m/2]$ ($\mathrm{s}_k = -1$), let $\mathcal{K}_-$ be the index set of living neurons, i.e. $\mathcal{K}_- := \{k \in [m] - [m/2] : \boldsymbol{w}_k(T_\mathrm{I}) \in \mathcal{M}_+^+ \cup \mathcal{M}_-^+\}$. Then $0.075m \leq |\mathcal{K}_-| \leq 0.205m$. Moreover, for any neuron $k \in \mathcal{K}_-$, it has the following properties **(N1)(N2)(N3)**.*

**(N1).** *Its norm is tiny : $\rho_k(T_\mathrm{I}) \leq \frac{\sqrt{\kappa_1\kappa_2}}{\sqrt{m}}\left(4.3\sqrt{\frac{\kappa_1}{\kappa_2}} + \frac{11.1\sin\Delta}{1+p}\right).$*

**(N2).** *It lies on a manifold perpendicular to $\boldsymbol{x}_+ : \boldsymbol{w}_k(t) \in \mathcal{M}_+^0 \cap \mathcal{M}_-^+.$*

**(N3).** *Its direction is weakly aligned with $\boldsymbol{x}_+^\perp : \left\langle \boldsymbol{w}_k(T_\mathrm{I}), \boldsymbol{x}_+^\perp\right\rangle > 1 - \dfrac{2}{\left(1 + 2.32\sqrt{\frac{\kappa_2}{\kappa_1}}\frac{\sin\Delta}{1+p}\right)^{1.6} + 1}.$*

**(S3).** *For other neurons $k \notin \mathcal{K}_+ \cup \mathcal{K}_-$, it will remain dead forever:*

$$\boldsymbol{w}_k(T_\mathrm{I}) \in \mathcal{M}_+^- \cap \mathcal{M}_-^-, \quad \boldsymbol{b}_k(t) \equiv \boldsymbol{b}_k(T_\mathrm{I}), \quad \forall t \in [T_\mathrm{I}, +\infty)$$

**(S4).** *The predictions for $\boldsymbol{x}_+$ and $\boldsymbol{x}_-$ have the estimate:*

$$0.978\kappa_2\sqrt{\kappa_1\kappa_2}\left((\frac{p-1}{p+1})^2 - 0.11\right) \leq f_+(T_\mathrm{I}) \leq 3.85\kappa_2\sqrt{\kappa_1\kappa_2},$$

$$0.947\left((\frac{p-1}{p+1})^2\cos\Delta - 0.2\right) \leq f_-(T_\mathrm{I}) \leq 3.85\kappa_2\sqrt{\kappa_1\kappa_2},$$

*and the training accuracy is $\mathrm{Acc}(T_\mathrm{I}) = \frac{p}{1+p}$.*

**(S5).** $0 < 0.258 \leq \frac{0.075}{0.29} \leq \frac{|\mathcal{K}_-|}{|\mathcal{K}_+|} \leq \frac{0.205}{0.21} \leq 0.977 < 1.$

*Proof of Theorem C.12.*
This theorem is a corollary of Lemma C.3, Lemma C.4, Lemma C.5, Lemma C.6, Lemma C.7, Lemma C.8, Lemma C.9, Lemma C.10, and Lemma C.11. We focus on the end of Phase I: $T_\mathrm{I} = 10\sqrt{\frac{\kappa_1}{\kappa_2}}$.

Proof of **(S1)(S2)**. From Lemma C.11, we know that: if $m = \Omega\left(\log(1/\delta)\right)$, then with probability at least $1 - \delta$, we have:

$$\left|\#\left\{k \in [m/2] : \langle \boldsymbol{w}_k(0), \boldsymbol{x}_+\rangle > 0\right\} - \frac{m}{4}\right| \leq 0.04m,$$

$$\#\left\{k \in [m] - [m/2] : \langle \boldsymbol{w}_k(0), \boldsymbol{x}_-\rangle > 0, \langle \boldsymbol{w}_k(0), \boldsymbol{x}_-\rangle > A\langle \boldsymbol{w}_k(0), \boldsymbol{x}_+\rangle\right\} \geq 0.075m,$$

$$\#\left\{k \in [m] - [m/2] : \langle \boldsymbol{w}_k(0), \boldsymbol{x}_-\rangle > 0, \langle \boldsymbol{w}_k(0), \boldsymbol{x}_+\rangle \leq B\langle \boldsymbol{w}_k(0), \boldsymbol{x}_-\rangle\right\} \leq 0.205m.$$

where $A = \frac{(1+4.73\kappa_1\kappa_2)p\cos\Delta - (1-4.73\kappa_1\kappa_2)}{(1-4.73\kappa_1\kappa_2)p - (1+4.73\kappa_1\kappa_2)\cos\Delta}$ and $B = \frac{(1+4.73\kappa_1\kappa_2)p - (1-4.73\kappa_1\kappa_2)\cos\Delta}{(1-4.73\kappa_1\kappa_2)p\cos\Delta - (1+4.73\kappa_1\kappa_2)}$.

Recalling the dynamics analysis in Lemma C.3, C.4, C.5, and C.6, we have:

$$0.21m \leq |\mathcal{K}_+| = \#\left\{k \in [m/2] : \langle \boldsymbol{w}_k(0), \boldsymbol{x}_+\rangle > 0\right\} \leq 0.29m.$$

Recalling the dynamics analysis in Lemma C.7, C.8 C.9, and C.10, we have:

$$|\mathcal{K}_-| \geq \#\left\{k \in [m] - [m/2] : \langle \boldsymbol{w}_k(0), \boldsymbol{x}_-\rangle > 0, \langle \boldsymbol{w}_k(0), \boldsymbol{x}_-\rangle > A\langle \boldsymbol{w}_k(0), \boldsymbol{x}_+\rangle\right\} \geq 0.075m,$$

$$|\mathcal{K}_-| \leq \#\left\{k \in [m] - [m/2] : \langle \boldsymbol{w}_k(0), \boldsymbol{x}_-\rangle > 0, \langle \boldsymbol{w}_k(0), \boldsymbol{x}_+\rangle \leq B\langle \boldsymbol{w}_k(0), \boldsymbol{x}_-\rangle\right\} \leq 0.205m.$$

Moreover, the estimates in Lemma C.3, C.4, C.5, and C.6 ensure that for any $k \in \mathcal{K}_+$, the following results hold:

$$\langle \boldsymbol{w}_k(T_{\mathrm{I}}), \boldsymbol{\mu} \rangle \geq \left(1 - 4.2\sqrt{\kappa_1 \kappa_2}\right)\left(1 - \frac{2}{1 + 0.7\left(1 + 9.9\sqrt{\frac{\kappa_2}{\kappa_1}}\right)^{1.15}}\right);$$

$$\frac{4.66\sqrt{\kappa_1 \kappa_2}}{\sqrt{m}} \leq \rho_k(T_{\mathrm{I}}) \leq \frac{12\sqrt{\kappa_1 \kappa_2}}{\sqrt{m}}.$$

Similarly, the estimates in Lemma C.7, C.8 C.9, and C.10 ensure that for any $k \in \mathcal{K}_-$, the following results hold:

$$\rho_k(T_{\mathrm{I}}) \leq \frac{\sqrt{\kappa_1 \kappa_2}}{\sqrt{m}}\left(4.3\sqrt{\frac{\kappa_1}{\kappa_2}} + \frac{11.1 \sin \Delta}{1 + p}\right);$$

$$\boldsymbol{w}_k(T_{\mathrm{I}}) \in \mathcal{M}_+^0 \cap \mathcal{M}_-^+;$$

$$\langle \boldsymbol{w}_k(T_{\mathrm{I}}), \boldsymbol{x}_+^\perp \rangle > 1 - \frac{2}{\left(1 + 2.32\sqrt{\frac{\kappa_2}{\kappa_1}}\frac{\sin \Delta}{1+p}\right)^{1.6} + 1}.$$

Proof of **(S3)**. A direct corollary of Lemma C.3, C.4, C.5, C.6, C.7, C.8, C.9, C.10.

Proof of **(S4)**. **(S4)** are direct corollaries of **(S1)(S2)**.

For $f_+(T_{\mathrm{I}})$, we have the following estimate:

$$f_+(T_{\mathrm{I}}) = \sum_{k \in \mathcal{K}_+} a_k \sigma\left(\boldsymbol{b}_k(T_{\mathrm{I}})^\top \boldsymbol{x}_+\right) + \sum_{k \in \mathcal{K}_-} a_k \sigma\left(\boldsymbol{b}_k(T_{\mathrm{I}})^\top \boldsymbol{x}_+\right)$$

$$= \sum_{k \in \mathcal{K}_+} a_k \sigma\left(\boldsymbol{b}_k(T_{\mathrm{I}})^\top \boldsymbol{x}_+\right) + 0 = \sum_{k \in \mathcal{K}_+} \frac{\kappa_2}{\sqrt{m}}\rho_k(T_{\mathrm{I}})\sigma\left(\boldsymbol{w}_k(T_{\mathrm{I}})^\top \boldsymbol{x}_+\right)$$

$$\geq \sum_{k \in \mathcal{K}_+} \frac{\kappa_2}{\sqrt{m}}\frac{4.66\sqrt{\kappa_1 \kappa_2}}{\sqrt{m}}\langle \boldsymbol{w}_k(T_{\mathrm{I}}), \boldsymbol{x}_+ \rangle \geq \sum_{k \in \mathcal{K}_+} \frac{\kappa_2}{\sqrt{m}}\frac{4.66\sqrt{\kappa_1 \kappa_2}}{\sqrt{m}}\left(\langle \boldsymbol{\mu}, \boldsymbol{x}_+ \rangle - \|\boldsymbol{w}_k(T_{\mathrm{I}}) - \boldsymbol{\mu}\|\right)$$

$$= \sum_{k \in \mathcal{K}_+} \frac{\kappa_2}{\sqrt{m}}\frac{4.66\sqrt{\kappa_1 \kappa_2}}{\sqrt{m}}\left(\|\boldsymbol{z}\|\langle \boldsymbol{z}, \boldsymbol{x}_+ \rangle - 2 + 2\langle \boldsymbol{w}_k(T_{\mathrm{I}}), \boldsymbol{\mu} \rangle\right)$$

$$\geq |\mathcal{K}_+|\frac{4.66\kappa_2\sqrt{\kappa_1 \kappa_2}}{m}\left(\frac{p-1}{p+1}\frac{p - \cos \Delta}{p+1} - 2 + 2\left(1 - 4.2\sqrt{\kappa_1 \kappa_2} - \frac{2}{1 + 0.7\left(1 + 9.9\sqrt{\frac{\kappa_2}{\kappa_1}}\right)^{1.15}}\right)\right)$$

$$\geq 0.21 \cdot 4.66\kappa_2\sqrt{\kappa_1 \kappa_2}\left(\frac{p-1}{p+1}\frac{p - \cos \Delta}{p+1} - 8.4\sqrt{\kappa_1 \kappa_2} - \frac{4}{1 + 0.7\left(1 + 9.9\sqrt{\frac{\kappa_2}{\kappa_1}}\right)^{1.15}}\right)$$

$$\geq 0.978\kappa_2\sqrt{\kappa_1 \kappa_2}\left(\left(\frac{p-1}{p+1}\right)^2 - 8.4\sqrt{\kappa_1 \kappa_2} - \frac{4}{1 + 0.7\left(1 + 9.9\sqrt{\frac{\kappa_2}{\kappa_1}}\right)^{1.15}}\right)$$

$$\overset{(6)}{\geq} 0.978\kappa_2\sqrt{\kappa_1 \kappa_2}\left((\frac{p-1}{p+1})^2 - 0.11\right);$$

$$f_+(T_{\mathrm{I}}) = \sum_{k \in \mathcal{K}_+} \frac{\kappa_2}{\sqrt{m}}\rho_k(T_{\mathrm{I}})\sigma\left(\boldsymbol{w}_k(T_{\mathrm{I}})^\top \boldsymbol{x}_+\right)$$

$$\leq \sum_{k \in \mathcal{K}_+} \frac{\kappa_2}{\sqrt{m}}\frac{12\sqrt{\kappa_1 \kappa_2}}{\sqrt{m}}\langle \boldsymbol{w}_k(T_{\mathrm{I}}), \boldsymbol{x}_+ \rangle \leq \sum_{k \in \mathcal{K}_+} \frac{\kappa_2}{\sqrt{m}}\frac{12\sqrt{\kappa_1 \kappa_2}}{\sqrt{m}}\left(\langle \boldsymbol{\mu}, \boldsymbol{x}_+ \rangle + \|\boldsymbol{w}_k(T_{\mathrm{I}}) - \boldsymbol{\mu}\|\right)$$

$$= \sum_{k \in \mathcal{K}_+} \frac{\kappa_2}{\sqrt{m}}\frac{12\sqrt{\kappa_1 \kappa_2}}{\sqrt{m}}\left(\|\boldsymbol{z}\|\langle \boldsymbol{z}, \boldsymbol{x}_+ \rangle + 2 - 2\langle \boldsymbol{w}_k(T_{\mathrm{I}}), \boldsymbol{\mu} \rangle\right)$$

$$\le |\mathcal{K}_+| \frac{12\kappa_2 \sqrt{\kappa_1 \kappa_2}}{m} \left( 1 \cdot \frac{p - \cos\Delta}{p+1} + 2 - 2 + 8.4\sqrt{\kappa_1 \kappa_2} + \frac{4}{1 + 0.7\left(1 + 9.9\sqrt{\frac{\kappa_2}{\kappa_1}}\right)^{1.15}} \right)$$

$$\le 0.29 \cdot 12\kappa_2 \sqrt{\kappa_1 \kappa_2} \left( 1 \cdot \frac{p - \cos\Delta}{p+1} + 8.4\sqrt{\kappa_1 \kappa_2} + \frac{4}{1 + 0.7\left(1 + 9.9\sqrt{\frac{\kappa_2}{\kappa_1}}\right)^{1.15}} \right)$$

$$\overset{(6)}{\le} 3.48\kappa_2 \sqrt{\kappa_1 \kappa_2} \left( 1 + 0.084 + \frac{4}{1 + 0.7(1+99)^{1.15}} \right) \le 3.85\kappa_2 \sqrt{\kappa_1 \kappa_2}.$$

Then we have:

$$0.978\kappa_2 \sqrt{\kappa_1 \kappa_2} \left( (\frac{p-1}{p+1})^2 - 0.11 \right) \le f_+(T_{\mathrm{I}}) \le 3.85\kappa_2 \sqrt{\kappa_1 \kappa_2}.$$

In the same way, we can estimate $f_-(T_{\mathrm{I}})$:

$$f_-(T_{\mathrm{I}}) = \sum_{k \in \mathcal{K}_+} a_k \sigma\left(\boldsymbol{b}_k(T_{\mathrm{I}})^\top \boldsymbol{x}_-\right) + \sum_{k \in \mathcal{K}_-} a_k \sigma\left(\boldsymbol{b}_k(T_{\mathrm{I}})^\top \boldsymbol{x}_-\right)$$

$$\ge \sum_{k \in \mathcal{K}_+} \frac{\kappa_2}{\sqrt{m}} \frac{4.66\sqrt{\kappa_1 \kappa_2}}{\sqrt{m}} \langle \boldsymbol{w}_k(T_{\mathrm{I}}), \boldsymbol{x}_- \rangle - \sum_{k \in \mathcal{K}_-} \frac{\kappa_2}{\sqrt{m}} \frac{\sqrt{\kappa_1 \kappa_2}}{\sqrt{m}} \left(4.3\sqrt{\frac{\kappa_1}{\kappa_2}} + \frac{11.1\sin\Delta}{1+p}\right) \langle \boldsymbol{w}_k(T_{\mathrm{I}}), \boldsymbol{x}_- \rangle$$

$$\ge \sum_{k \in \mathcal{K}_+} \frac{\kappa_2}{\sqrt{m}} \frac{4.66\sqrt{\kappa_1 \kappa_2}}{\sqrt{m}} \left( \langle \boldsymbol{\mu}, \boldsymbol{x}_- \rangle - \|\boldsymbol{w}_k(T_{\mathrm{I}}) - \boldsymbol{\mu}\| \right) - \sum_{k \in \mathcal{K}_-} \frac{\kappa_2}{\sqrt{m}} \frac{\sqrt{\kappa_1 \kappa_2}}{\sqrt{m}} \left(0.43 + \frac{11.1}{12}\right) \sin\Delta$$

$$\ge |\mathcal{K}_+| \frac{4.66\kappa_2 \sqrt{\kappa_1 \kappa_2}}{m} \left( \frac{p-1}{p+1} \frac{p\cos\Delta - 1}{p+1} - 2 + 2\left(1 - 4.2\sqrt{\kappa_1 \kappa_2} - \frac{2}{1 + 0.7\left(1 + 9.9\sqrt{\frac{\kappa_2}{\kappa_1}}\right)^{1.15}}\right) \right)$$

$$- |\mathcal{K}_-| \frac{\kappa_2 \sqrt{\kappa_1 \kappa_2}}{m} \cdot 1.355\sin\Delta$$

$$\ge 0.21 \cdot 4.66\kappa_2 \sqrt{\kappa_1 \kappa_2} \left( \frac{p-1}{p+1} \frac{p\cos\Delta - 1}{p+1} - 0.11 \right) - 0.205 \cdot 1.355\sin\Delta \kappa_2 \sqrt{\kappa_1 \kappa_2}$$

$$\ge \kappa_2 \sqrt{\kappa_1 \kappa_2} \left( 0.978(\frac{p-1}{p+1})^2 \cos\Delta - 0.11 - 0.28\sin\Delta \right) \ge \kappa_2 \sqrt{\kappa_1 \kappa_2} \left( 0.947(\frac{p-1}{p+1})^2 \cos\Delta - 0.11 - 0.07 \right)$$

$$\ge 0.947 \left( (\frac{p-1}{p+1})^2 \cos\Delta - 0.2 \right);$$

$$f_-(T_{\mathrm{I}}) = \sum_{k \in \mathcal{K}_+} a_k \sigma\left(\boldsymbol{b}_k(T_{\mathrm{I}})^\top \boldsymbol{x}_-\right) + \sum_{k \in \mathcal{K}_-} a_k \sigma\left(\boldsymbol{b}_k(T_{\mathrm{I}})^\top \boldsymbol{x}_-\right)$$

$$\le \sum_{k \in \mathcal{K}_+} \frac{\kappa_2}{\sqrt{m}} \frac{12\sqrt{\kappa_1 \kappa_2}}{\sqrt{m}} \langle \boldsymbol{w}_k(T_{\mathrm{I}}), \boldsymbol{x}_- \rangle - 0$$

$$= \sum_{k \in \mathcal{K}_+} \frac{\kappa_2}{\sqrt{m}} \frac{12\sqrt{\kappa_1 \kappa_2}}{\sqrt{m}} \left( \|\boldsymbol{z}\| \langle \boldsymbol{z}, \boldsymbol{x}_- \rangle + 2 - 2\langle \boldsymbol{w}_k(T_{\mathrm{I}}), \boldsymbol{\mu} \rangle \right)$$

$$\le |\mathcal{K}_+| \frac{12\kappa_2 \sqrt{\kappa_1 \kappa_2}}{m} \left( 1 \cdot \frac{p\cos\Delta - 1}{p+1} + 2 - 2 + 8.4\sqrt{\kappa_1 \kappa_2} + \frac{4}{1 + 0.7\left(1 + 9.9\sqrt{\frac{\kappa_2}{\kappa_1}}\right)^{1.15}} \right)$$

$$\le 0.29 \cdot 12\kappa_2 \sqrt{\kappa_1 \kappa_2} \left( 1 \cdot \frac{p\cos\Delta - 1}{p+1} + 8.4\sqrt{\kappa_1 \kappa_2} + \frac{4}{1 + 0.7\left(1 + 9.9\sqrt{\frac{\kappa_2}{\kappa_1}}\right)^{1.15}} \right)$$

$$\overset{(6)}{\le} 3.48\kappa_2 \sqrt{\kappa_1 \kappa_2} \left( 1 + 0.084 + \frac{4}{1 + 0.7(1+99)^{1.15}} \right) \le 3.85\kappa_2 \sqrt{\kappa_1 \kappa_2}.$$

Them we have:

$$0.947 \left( (\frac{p-1}{p+1})^2 \cos \Delta - 0.2 \right) \le f_-(T_{\mathrm{I}}) \le 3.85 \kappa_2 \sqrt{\kappa_1 \kappa_2}.$$

Due to $f_+(T_{\mathrm{I}}) > 0$ and $f_-(T_{\mathrm{I}}) > 0$, we obtain $\mathrm{ACC}(T_{\mathrm{I}}) = \frac{p}{1+p}$.

Moreover, from Theorem C.12 (S1)(S2), we have

$$0 < 0.258 \le \frac{0.075m}{0.29m} \le \alpha = \frac{|\mathcal{K}_-|}{|\mathcal{K}_+|} = \frac{m_-}{m_+} \le \frac{0.205m}{0.21m} \le 0.977 < 1. \tag{18}$$

$\square$

**Remark C.13.** The results in the following proofs are all based on the occurrence of the events in Theorem C.12. All of these results use the same settings as Theorem 4.1, except using a stronger condition on the initialization parameters (3) than (6) in Theorem C.12. So they all hold with probability at least $1 - \delta$.

## D  Proofs of Optimization Dynamics in Phase II

In this phase, we study the dynamics before the patterns of living neurons change again after Phase I. Specifically, we define

$$T_{\mathrm{II}} := \inf\{t > T_{\mathrm{I}} : \exists k \in \mathcal{K}_+ \cup \mathcal{K}_-, \mathrm{sgn}_k^+(t) \neq \mathrm{sgn}_k^+(T_{\mathrm{I}}) \text{ or } \mathrm{sgn}_k^-(t) \neq \mathrm{sgn}_k^-(T_{\mathrm{I}})\},$$

and call $t \in (T_{\mathrm{I}}, T_{\mathrm{II}}]$ Phase II.

Recalling the results in Theorem 4.1, during Phase II, the activation patterns do not change with $\mathrm{sgn}_k^+(t) = \mathrm{sgn}_k^-(t) = 1$ for $k \in \mathcal{K}_+$ and $\mathrm{sgn}_k^+(t) = 0, \mathrm{sgn}_k^-(t) = 1$ for $k \in \mathcal{K}_-$. Theorem 4.4 demonstrates that at the end of Phase II, except for one of living positive neuron $k_0 \in \mathcal{K}_+$ precisely changes its pattern on $\boldsymbol{x}_-$, all other activation patterns remain unchanged.

Recall that at the end of Phase I, (i) the neuron $k \notin \mathcal{K}_+ \cup \mathcal{K}_-$ is dead forever; (ii) as for the living neuron $k \in \mathcal{K}_+ \cup \mathcal{K}_-$, the activation patterns are:

$$\mathrm{sgn}_k^+(t) = \mathrm{sgn}_k^-(t) = 1 \text{ for } k \in \mathcal{K}_+;$$
$$\mathrm{sgn}_k^+(t) = 0, \mathrm{sgn}_k^-(t) = 1 \text{ for } k \in \mathcal{K}_-.$$

In this section, we will focus on the Phase when the negative neuron $k \in \mathcal{K}_-$ still stays on the manifold $\mathcal{M}_+^0 \cap \mathcal{M}_-^+$ (hence is still dead for $\boldsymbol{x}_+$) and the positive neuron $k \in \mathcal{K}_+$ is still activated for $\boldsymbol{x}_-$. In general, we aim to estimate the

As stated in the main text, we define as following hitting time:

$$T_{\mathrm{II}} := \inf\left\{t > T_{\mathrm{I}} : \exists k \in \mathcal{K}_+ \cup \mathcal{K}_-, \mathrm{sgn}_k^+(t) \neq \mathrm{sgn}_k^+(T_{\mathrm{I}}) \text{ or } \mathrm{sgn}_k^-(t) \neq \mathrm{sgn}_k^-(T_{\mathrm{I}})\right\}$$

$$= \inf\left\{t > T_{\mathrm{I}} : \exists k \in \mathcal{K}_+, \text{ s.t. } \langle \boldsymbol{w}_k(t), \boldsymbol{x}_+ \rangle \leq 0 \text{ or } \langle \boldsymbol{w}_k(t), \boldsymbol{x}_- \rangle \leq 0, \right. \tag{19}$$

$$\left. \text{or } \exists k \in \mathcal{K}_-, \text{ s.t. } \langle \boldsymbol{w}_k(t), \boldsymbol{x}_+ \rangle \neq 0 \text{ or } \langle \boldsymbol{w}_k(t), \boldsymbol{x}_- \rangle \leq 0\right\},$$

and we call $T_{\mathrm{I}} \leq t \leq T_{\mathrm{II}}$ "Phase II".

First, we define a more relaxed hitting time than $T_{\mathrm{II}}$, only about the change of living positive neurons:

$$T_{\mathrm{II}}^+ := \inf\left\{t > T_{\mathrm{I}} : \exists k \in \mathcal{K}_+, \text{ s.t. } \langle \boldsymbol{w}_k(t), \boldsymbol{x}_+ \rangle \leq 0 \text{ or } \langle \boldsymbol{w}_k(t), \boldsymbol{x}_- \rangle \leq 0\right\}. \tag{20}$$

Noticing that the changes in activation partitions are essentially caused by the change of discontinuous vector fields, we first define the following auxiliary hitting time:

$$T_{\mathrm{II}}^* := T_{\mathrm{II}}^+ \wedge \inf\left\{t > T_{\mathrm{I}} : \langle \boldsymbol{F}_+(t), \boldsymbol{x}_+ \rangle \leq 0\right\},$$

$$\text{where } \boldsymbol{F}_+(t) = \frac{p}{1+p}e^{-f_+(t)}\boldsymbol{x}_+ - \frac{1}{1+p}e^{f_-(t)}\boldsymbol{x}_-. \tag{21}$$

We call $T_{\mathrm{I}} \leq t \leq T_{\mathrm{II}}^*$ "Phase II*".

In the subsequent proof, we first meticulously characterize the optimization dynamics in Phase II* and then prove $T_{\mathrm{II}} = T_{\mathrm{II}}^+ = T_{\mathrm{II}}^*$. The crucial proof technique is fine-grained prior estimations for 2d ODEs on $f_+(t)$ and $f_-(t)$, leading to the vector field estimation.

To begin with, we establish the following lemma about the optimization dynamics of living neurons.
**Lemma D.1** (Dynamics of living neurons in Phase II*)**.**
*In Phase II*, $t \in [T_{\mathrm{I}}, T_{\mathrm{II}}^*]$, we have the following dynamics for each neuron $k \in \mathcal{K}_- \cup \mathcal{K}_+$.*

*(S1) For positive neuron $k \in \mathcal{K}_+$, we have:*

$$\boldsymbol{w}_k(t) \in \mathcal{M}_+^+ \cap \mathcal{M}_-^+,$$

$$\frac{\mathrm{d}\boldsymbol{b}_k(t)}{\mathrm{d}t} = \frac{\kappa_2}{\sqrt{m}}\boldsymbol{F}_+(t) = \frac{\kappa_2}{\sqrt{m}}\left(\frac{p}{1+p}e^{-f_+(t)}\boldsymbol{x}_+ - \frac{1}{1+p}e^{f_-(t)}\boldsymbol{x}_-\right).$$

*(S2). For negative neuron $k \in \mathcal{K}_-$, we have:*

$$\boldsymbol{w}_k(t) \in \mathcal{M}_+^0 \cap \mathcal{M}_-^+,$$

$$\frac{\mathrm{d}\boldsymbol{b}_k(t)}{\mathrm{d}t} = \frac{\kappa_2 e^{f_-(t)}}{\sqrt{m}(1+p)}\left(\boldsymbol{x}_- - \boldsymbol{x}_+ \cos\Delta\right).$$

*Proof of Lemma D.1.*

(S1) Let $k \in \mathcal{K}_+$. Recalling the definition of $T_{\mathrm{II}}^*$, it holds that $\langle \boldsymbol{w}_k(t), \boldsymbol{x}_+ \rangle > 0$ and $\langle \boldsymbol{w}_k(t), \boldsymbol{x}_- \rangle > 0$ for any $T_{\mathrm{I}} \le t \le T_{\mathrm{II}}^*$, so the dynamics holds.

(S2) Let $k \in \mathcal{K}_-$. Recalling the definition of $T_{\mathrm{II}}^*$, it holds $\langle \boldsymbol{F}_+(t), \boldsymbol{x}_+ \rangle > 0$ for any $T_{\mathrm{I}} \le t \le T_{\mathrm{II}}^*$. Due to $\boldsymbol{w}_k(T_{\mathrm{I}}) \in \mathcal{M}_+^0 \cap \mathcal{M}_-^+$, we first analysis the vector field around the manifold $\mathcal{P}_+^0 \cap \mathcal{P}_-^+$ for $T_{\mathrm{I}} \le t \le T_{\mathrm{II}}^*$.

For any $\tilde{\boldsymbol{b}} \in \mathcal{P}_+^0 \cap \mathcal{P}_-^+$ and $0 < \delta_0 \ll 1$, we know that $\mathcal{P}_+^0 \cap \mathcal{P}_-^+$ separates its neighborhood $\mathcal{B}(\tilde{\boldsymbol{b}}, \delta_0)$ into two domains $\mathcal{G}_- = \{\boldsymbol{b} \in \mathcal{B}(\tilde{\boldsymbol{b}}, \delta_0) : \langle \boldsymbol{b}, \boldsymbol{x}_+ \rangle < 0\}$ and $\mathcal{G}_+ = \{\boldsymbol{b} \in \mathcal{B}(\tilde{\boldsymbol{b}}, \delta_0) : \langle \boldsymbol{b}, \boldsymbol{x}_+ \rangle > 0\}$. Following Definition H.1, we calculate the limited vector field on $\tilde{\boldsymbol{b}}$ from $\mathcal{G}_-$ and $\mathcal{G}_+$.

(i) The limited vector field $\boldsymbol{F}^-$ on $\tilde{\boldsymbol{b}}$ (from $\mathcal{G}_-$):
$$\frac{\mathrm{d}\boldsymbol{b}}{\mathrm{d}t} = \boldsymbol{F}^-, \text{ where } \boldsymbol{F}^- = \frac{\kappa_2}{\sqrt{m}} \frac{1}{1+p} e^{f_-(t)} \boldsymbol{x}_-.$$

(ii) The limited vector field $\boldsymbol{F}^+$ on $\tilde{\boldsymbol{b}}$ (from $\mathcal{G}_+$):
$$\frac{\mathrm{d}\boldsymbol{b}}{\mathrm{d}t} = \boldsymbol{F}^+, \text{ where } \boldsymbol{F}^+ = -\frac{\kappa_2}{\sqrt{m}} \left( \frac{pe^{-f_+(t)}}{1+p} \boldsymbol{x}_+ - \frac{e^{f_-(t)}}{1+p} \boldsymbol{x}_- \right).$$

(iii) Then we calculate the projections of $\boldsymbol{F}^-$ and $\boldsymbol{F}^+$ onto $\boldsymbol{x}_+$ (the normal to the surface $\mathcal{P}_+^0 \cap \mathcal{P}_-^+$):
$$F_N^- = \langle \boldsymbol{F}^-, \boldsymbol{x}_+ \rangle = \frac{\kappa_2 e^{f_-(t)}}{\sqrt{m}(1+p)} \cos \Delta,$$
$$F_N^+ = \langle \boldsymbol{F}^+, \boldsymbol{x}_+ \rangle = \frac{\kappa_2 e^{f_-(t)}}{\sqrt{m}(1+p)} \cos \Delta - \frac{\kappa_2 p e^{-f_+(t)}}{\sqrt{m}(1+p)}.$$

We further define the hitting time to check whether $\boldsymbol{w}_k(t) \in \mathcal{M}_+^0 \cap \mathcal{M}_-^+$ for $T_{\mathrm{I}} \le t \le T_{\mathrm{II}}^*$.
$$\tau_-^+ := T_{\mathrm{II}}^* \wedge \inf\{t > T_{\mathrm{I}} : \exists k \in \mathcal{K}_-, \text{ s.t. } \langle \boldsymbol{w}_k(t), \boldsymbol{x}_- \rangle \le 0\}.$$

From the definition of $T_{\mathrm{II}}^*$, we know $\langle \boldsymbol{F}_+(t), \boldsymbol{x}_+ \rangle > 0$ for any $T_{\mathrm{I}} \le t \le T_{\mathrm{II}}^*$, so $F_N^+ < 0$. And it is clear that $F_N^- > 0$. Hence, the dynamics corresponds to Case (I) in Definition H.1 ($F_N^- > 0$ and $F_N^+ < 0$), which means $\boldsymbol{b}_k(t)$ can not leave $\mathcal{P}_+^0$ (i.e., $\langle \boldsymbol{b}_k(t), \boldsymbol{x}_+ \rangle = 0$) for $t \in [T_{\mathrm{I}}, \tau_-^+]$, and the dynamics of $\boldsymbol{b}_k(t)$ satisfies:
$$\frac{\mathrm{d}\boldsymbol{b}}{\mathrm{d}t} = \alpha \boldsymbol{F}^+ + (1-\alpha)\boldsymbol{F}^-, \quad \alpha = \frac{f_N^-}{f_N^- - f_N^+}, \ t \in [T_{\mathrm{I}}, \tau_-^+],$$
which is
$$\frac{\mathrm{d}\boldsymbol{b}_k(t)}{\mathrm{d}t} \frac{\kappa_2 e^{f_-(t)}}{\sqrt{m}(1+p)} \left( \boldsymbol{x}_- - \boldsymbol{x}_+ \cos \Delta \right), \ t \in [T_{\mathrm{I}}, \tau_-^+].$$

By Lemma C.1, we know that the dynamics of $\boldsymbol{w}_k(t)$ on $\mathcal{M}_+^0 \cap \mathcal{M}_-^+$ and the dynamics of $\rho_k(t)$ are:
$$\frac{\mathrm{d}\boldsymbol{w}_k(t)}{\mathrm{d}t} = \frac{\kappa_2 e^{f_-(t)}}{\rho_k(t)\sqrt{m}(1+p)} \left( \boldsymbol{x}_- - \langle \boldsymbol{w}_k(t), \boldsymbol{x}_- \rangle \boldsymbol{w}_k - \boldsymbol{x}_+ \cos \Delta \right).$$
$$\frac{\mathrm{d}\rho_k(t)}{\mathrm{d}t} = \frac{\kappa_2 e^{f_-(t)}}{\sqrt{m}(1+p)} \langle \boldsymbol{w}_k(t), \boldsymbol{x}_- \rangle.$$

From this dynamics, for any $t \in [T_{\mathrm{I}}, \tau_-^+]$, we have
$$\langle \boldsymbol{b}_k(t), \boldsymbol{x}_- \rangle = \langle \boldsymbol{b}_k(0), \boldsymbol{x}_- \rangle + \int_0^t \left\langle \frac{\mathrm{d}\boldsymbol{b}_k(s)}{\mathrm{d}s}, \boldsymbol{x}_- \right\rangle \mathrm{d}s$$
$$= \langle \boldsymbol{b}_k(0), \boldsymbol{x}_- \rangle + \int_0^t \frac{\kappa_2 e^{f_-(s)}}{\sqrt{m}(1+p)} \sin^2 \Delta \mathrm{d}s > \langle \boldsymbol{b}_k(0), \boldsymbol{x}_- \rangle > 0,$$
which means that $T_{\mathrm{II}}^* = \tau_-^+$.

$\square$

Noticing that Lemma D.1 determines the activation patterns for living neurons in Phase II*, the next lemma gives the first-order dynamics of $f_+(t)$ and $f_-(t)$.

**Lemma D.2** (First-order dynamics of predictions in Phase II*).
*In Phase II* ($T_{\mathrm{I}} \leq t \leq T_{\mathrm{II}}^*$), we have the following dynamics for $f_+(t)$ and $f_-(t)$:*

$$\frac{\mathrm{d}f_+(t)}{\mathrm{d}t} = \kappa_2^2 \frac{m_+}{m} \left( \frac{pe^{-f_+(t)}}{1+p} - \frac{e^{f_-(t)}}{1+p} \cos \Delta \right),$$

$$\frac{\mathrm{d}f_-(t)}{\mathrm{d}t} = \kappa_2^2 \frac{m_+}{m} \left( \frac{pe^{-f_+(t)}}{1+p} \cos \Delta - \frac{e^{f_-(t)}}{1+p} \right) - \kappa_2^2 \frac{m_-}{m} \frac{e^{f_-(t)}}{1+p} \sin^2 \Delta.$$

*Proof of Lemma D.2.*
From the definition of $T_{\mathrm{II}}^*$, for any $T_{\mathrm{I}} \leq t \leq T_{\mathrm{II}}^*$, we have

$$f_+(t) = \sum_{k \in \mathcal{K}_+} \frac{\kappa_2}{\sqrt{m}} \sigma(\boldsymbol{b}_k^\top(t) \boldsymbol{x}_+) - \sum_{k \in \mathcal{K}_-} \frac{\kappa_2}{\sqrt{m}} \sigma(\boldsymbol{b}_k^\top(t) \boldsymbol{x}_+) = \sum_{k \in \mathcal{K}_+} \frac{\kappa_2}{\sqrt{m}} \boldsymbol{b}_k^\top(t) \boldsymbol{x}_+,$$

$$f_-(t) = \sum_{k \in \mathcal{K}_+} \frac{\kappa_2}{\sqrt{m}} \sigma(\boldsymbol{b}_k^\top(t) \boldsymbol{x}_-) - \sum_{k \in \mathcal{K}_-} \frac{\kappa_2}{\sqrt{m}} \sigma(\boldsymbol{b}_k^\top(t) \boldsymbol{x}_-) = \sum_{k \in \mathcal{K}_+} \frac{\kappa_2}{\sqrt{m}} \boldsymbol{b}_k^\top(t) \boldsymbol{x}_- - \sum_{k \in \mathcal{K}_-} \frac{\kappa_2}{\sqrt{m}} \boldsymbol{b}_k^\top(t) \boldsymbol{x}_-.$$

With the help of Lemma D.1, we have the dynamics of predictions:

$$\frac{\mathrm{d}f_+(t)}{\mathrm{d}t} = \sum_{k \in \mathcal{K}_+} \frac{\kappa_2}{\sqrt{m}} \left\langle \frac{\mathrm{d}\boldsymbol{b}_k(t)}{\mathrm{d}t}, \boldsymbol{x}_+ \right\rangle = \frac{\kappa_2^2}{m} \sum_{k \in \mathcal{K}_+} \left( \frac{p}{1+p} e^{-f_+(t)} - \frac{1}{1+p} e^{f_-(t)} \cos \Delta \right)$$

$$= \frac{m_+}{m} \kappa_2^2 \left( \frac{pe^{-f_+(t)}}{1+p} - \frac{e^{f_-(t)}}{1+p} \cos \Delta \right).$$

$$\frac{\mathrm{d}f_-(t)}{\mathrm{d}t} = \sum_{k \in \mathcal{K}_+} \frac{\kappa_2}{\sqrt{m}} \left\langle \frac{\mathrm{d}\boldsymbol{b}_k(t)}{\mathrm{d}t}, \boldsymbol{x}_- \right\rangle - \sum_{k \in \mathcal{K}_-} \frac{\kappa_2}{\sqrt{m}} \left\langle \frac{\mathrm{d}\boldsymbol{b}_k(t)}{\mathrm{d}t}, \boldsymbol{x}_- \right\rangle$$

$$= \frac{\kappa_2^2}{m} \sum_{k \in \mathcal{K}_+} \left( \frac{pe^{-f_+(t)}}{1+p} \cos \Delta - \frac{e^{f_-(t)}}{1+p} \right) - \frac{\kappa_2^2}{m} \sum_{k \in \mathcal{K}_-} \frac{e^{f_-(t)}}{1+p} \left( 1 - \cos^2 \Delta \right)$$

$$= \frac{m_+}{m} \kappa_2^2 \left( \frac{pe^{-f_+(t)}}{1+p} \cos \Delta - \frac{e^{f_-(t)}}{1+p} \right) - \frac{m_-}{m} \kappa_2^2 \frac{e^{f_-(t)}}{1+p} \sin^2 \Delta.$$

$$\square$$

Due to the specificity of the first-order dynamics, the following lemma gives an second-order **autonomous** dynamics of predictions, which is is the core dynamics in this phase.

**Lemma D.3** (Second-order Autonomous Dynamics of predictions in Phase II*).
*Consider the following two variables:*

$$\begin{cases} \mathcal{U}(t) := \kappa_2^2 \frac{m_+}{m} \frac{p}{1+p} e^{-f_+(t)}, \\ \mathcal{V}(t) := \kappa_2^2 \frac{m_+}{m} \frac{1}{1+p} e^{f_-(t)}. \end{cases}$$

*Then the following autonomous dynamics of $\mathcal{U}(t)$ and $\mathcal{V}(t)$ hold in Phase II* ($T_{\mathrm{I}} \leq t \leq T_{\mathrm{II}}^*$):*

$$\begin{cases} \frac{\mathrm{d}\mathcal{U}(t)}{\mathrm{d}t} = \mathcal{U}(t)\mathcal{V}(t) \cos \Delta - \mathcal{U}^2(t), \\ \frac{\mathrm{d}\mathcal{V}(t)}{\mathrm{d}t} = \mathcal{U}(t)\mathcal{V}(t) \cos \Delta - \mathcal{V}^2(t) \left( 1 + \alpha \sin^2 \Delta \right). \end{cases}$$

*Proof of Lemma D.3.*
Recall the first-order dynamics in Lemma D.2:

$$\begin{cases} \frac{\mathrm{d}f_+(t)}{\mathrm{d}t} & = \kappa_2^2 \frac{m_+}{m} \left( \frac{pe^{-f_+(t)}}{1+p} - \frac{e^{f_-(t)}}{1+p} \cos \Delta \right), \\ \frac{\mathrm{d}f_-(t)}{\mathrm{d}t} & = \kappa_2^2 \frac{m_+}{m} \left( \frac{pe^{-f_+(t)}}{1+p} \cos \Delta - \frac{e^{f_-(t)}}{1+p} \right) - \kappa_2^2 \frac{m_-}{m} \frac{e^{f_-(t)}}{1+p} \sin^2 \Delta. \end{cases}$$

Then this proof is a straight-forward calculation:

$$
\begin{aligned}
\frac{\mathrm{d}\mathcal{U}(t)}{\mathrm{d}t} &= \kappa_2^2 \frac{m_+}{m} \frac{p}{1+p} \frac{\mathrm{d}e^{-f_+(t)}}{\mathrm{d}t} = -\kappa_2^2 \frac{m_+}{m} \frac{p}{1+p} e^{-f_+(t)} \frac{\mathrm{d}f_+(t)}{\mathrm{d}t} \\
&= -\mathcal{U}(t) \frac{\mathrm{d}f_+(t)}{\mathrm{d}t} = \mathcal{U}(t)\mathcal{V}(t)\cos\Delta - \mathcal{U}^2(t),
\end{aligned}
$$

$$
\begin{aligned}
\frac{\mathrm{d}\mathcal{V}(t)}{\mathrm{d}t} &= \kappa_2^2 \frac{m_+}{m} \frac{1}{1+p} \frac{\mathrm{d}e^{f_-(t)}}{\mathrm{d}t} = \kappa_2^2 \frac{m_+}{m} \frac{1}{1+p} e^{f_-(t)} \frac{\mathrm{d}f_-(t)}{\mathrm{d}t} \\
&= \mathcal{V}(t) \frac{\mathrm{d}f_-(t)}{\mathrm{d}t} = \mathcal{U}(t)\mathcal{V}(t)\cos\Delta - \mathcal{V}^2(t)\left(1 + \alpha\sin^2\Delta\right).
\end{aligned}
$$

$\square$

Lemma D.3 enlighten us that we only need to study the dynamics of $\mathcal{U}(t)$ and $\mathcal{V}(t)$ to study the dynamics in Phase II, where $\mathcal{U}(t), \mathcal{V}(t)$ satisfies the following autonomous dynamics:

$$
\begin{cases}
\frac{\mathrm{d}\mathcal{U}(t)}{\mathrm{d}t} = \mathcal{U}(t)\mathcal{V}(t)\cos\Delta - \mathcal{U}^2(t); \\
\frac{\mathrm{d}\mathcal{V}(t)}{\mathrm{d}t} = \mathcal{U}(t)\mathcal{V}(t)\cos\Delta - \mathcal{V}^2(t)\left(1 + \alpha\sin^2\Delta\right), \qquad t \geq T_{\mathrm{I}}; \\
\mathcal{U}(T_{\mathrm{I}}) = \kappa_2^2 \frac{m_+}{m} \frac{p}{1+p} e^{-f_+(T_{\mathrm{I}})}, \\
\mathcal{V}(T_{\mathrm{I}}) = \kappa_2^2 \frac{m_+}{m} \frac{1}{1+p} e^{f_-(T_{\mathrm{I}})}.
\end{cases}
\tag{22}
$$

The next lemma provides a fine-grained prior estimate of the dynamics (22).

**Lemma D.4** (Fine-grained prior estimate of the dynamics (22))**.**
*For the dynamics* (22)*, then we have the following results:*

**(S1).** $\mathcal{U}(T_{\mathrm{I}}) = \Theta(\kappa_2^2)$ *and* $\mathcal{V}(T_{\mathrm{I}}) = \Theta\left(\frac{\kappa_2^2}{p}\right)$.

**(S2).** *For any* $t \geq T_{\mathrm{I}}$*, we have* $\mathcal{U}(t) > \mathcal{V}(t) > 0$.

**(S3).** *If we define the hitting time* $\tau_1 := \inf\left\{ t \geq T_{\mathrm{I}} : \mathcal{U}(t)\cos\Delta \leq \mathcal{V}(t)\left(1 + \alpha\sin^2\Delta\right) \right\}$*, then*

$$
\mathcal{U}(\tau_1) = \frac{1 + \alpha\sin^2\Delta}{\cos\Delta} \mathcal{V}(\tau_1), \quad \mathcal{V}(\tau_1) = \Theta\left(\kappa_2^2 p^{-\frac{1}{1+\cos\Delta}}\right),
$$

$$
\tau_1 = \mathcal{O}\left(\frac{p^{\frac{1}{1+\cos\Delta}}\log(1/\Delta)}{\kappa_2^2}\right) = \Omega\left(\frac{p^{\frac{1}{1+\cos\Delta}}}{\kappa_2^2}\right) = \tilde{\Theta}\left(\frac{p^{\frac{1}{1+\cos\Delta}}}{\kappa_2^2}\right).
$$

**(S4).** *For any* $t \geq \tau_1$*, we have*

$$
1 + \frac{m_-}{2m_+}\sin^2\Delta < \frac{\mathcal{U}(t)}{\mathcal{V}(t)} < \frac{1 + 2\alpha\sin^2\Delta}{\cos\Delta},
$$

$$
\mathcal{U}(t) = \Theta\left(\frac{1}{\frac{p^{\frac{1}{1+\cos\Delta}}}{\kappa_2^2} + \Delta^2(t - \tau_1)}\right), \quad \mathcal{V}(t) = \Theta\left(\frac{1}{\frac{p^{\frac{1}{1+\cos\Delta}}}{\kappa_2^2} + \Delta^2(t - \tau_1)}\right).
$$

**(S5).** *For any* $t \geq \tau_1$*, we have* $\mathcal{U}(t) - \mathcal{V}(t)\cos\Delta = \Theta\left(\Delta^2\mathcal{V}(t)\right) > 0$.

**(S6).** *For any* $t \geq \tau_2 = \Theta\left(\frac{p^{\frac{1}{1+\cos\Delta}}\log(1/\Delta)}{\kappa_2^2}\right) \geq 2\tau_1$*, we have*

$$
\mathcal{U}(t)\cos\Delta - \mathcal{V}(t) = -\Theta\left(\Delta^2\mathcal{V}(t)\right) < 0.
$$

*Proof of Lemma D.4.*
For simplicity, in this proof, we denote

$$
\epsilon := \alpha\sin^2\Delta.
$$

Step I. Preparation. From Theorem C.12 (S4), we know $0 < f_+(T_{\mathrm{I}}), f_-(T_{\mathrm{I}}) \le 3.85\kappa_2\sqrt{\kappa_1\kappa_2} \le 0.04\log(1.1)$, so

$$1 < e^{f_-(T_{\mathrm{I}})} \le 1 + e^{0.04\log(1.1)}3.85\kappa_2\sqrt{\kappa_1\kappa_2} \le 1 + 1.004 \cdot 3.85\kappa_2\sqrt{\kappa_1\kappa_2} \le 1 + 3.87\kappa_2\sqrt{\kappa_1\kappa_2},$$

$$1 > e^{-f_+(T_{\mathrm{I}})} \ge 1 - e^{0.04\log(1.1)}3.85\kappa_2\sqrt{\kappa_1\kappa_2} \ge 1 - 1.004 \cdot 3.85\kappa_2\sqrt{\kappa_1\kappa_2} \ge 1 - 3.87\kappa_2\sqrt{\kappa_1\kappa_2}.$$

Notice that $\mathcal{U}(T_{\mathrm{I}}) = \kappa_2^2 \frac{m_+}{m} \frac{p}{1+p} e^{-f_+(T_{\mathrm{I}})}$ and $\mathcal{V}(T_{\mathrm{I}}) = \kappa_2^2 \frac{m_+}{m} \frac{1}{1+p} e^{f_-(T_{\mathrm{I}})}$. Then we have the estimate:

$$1 - 3.87\kappa_2\sqrt{\kappa_1\kappa_2} \le \frac{\mathcal{U}(T_{\mathrm{I}})}{\kappa_2^2 \frac{m_+}{m} \frac{p}{1+p}} \le 1,$$

$$1 \le \frac{\mathcal{V}(T_{\mathrm{I}})}{\kappa_2^2 \frac{m_+}{m} \frac{1}{1+p}} \le 1 + 3.87\kappa_2\sqrt{\kappa_1\kappa_2}.$$

From Theorem C.12 (S1)(S2), we have

$$0.258 \le \frac{0.075m}{0.29m} \le \alpha \le \frac{0.205m}{0.21m} \le 0.977. \tag{23}$$

For $t = T_{\mathrm{I}}$, it holds that

$$\mathcal{U}(T_{\mathrm{I}})\mathcal{V}(T_{\mathrm{I}})\cos\Delta - \mathcal{U}^2(T_{\mathrm{I}}) < 0,$$

$$\mathcal{U}(T_{\mathrm{I}})\mathcal{V}(T_{\mathrm{I}})\cos\Delta - \mathcal{V}^2(T_{\mathrm{I}})\,(1+\epsilon) > 0.$$

Step II. A rough estimate on $\mathcal{U}(t)$ and $\mathcal{V}(t)$. In this step, we aim to prove:

$$\mathcal{U}(t) > \mathcal{V}(t) > 0, \quad ,\mathcal{U}(t) + \mathcal{V}(t) \le \mathcal{U}(T_{\mathrm{I}}) + \mathcal{V}(T_{\mathrm{I}}), \quad \forall t \in [T_{\mathrm{I}}, \infty).$$

First, from the definition of $\mathcal{U}(t)$ and $\mathcal{V}(t)$, we have $\mathcal{U}(t) > 0$ and $\mathcal{V}(t) > 0$.

Then we consider the dynamics of $\mathcal{U}(t) + \mathcal{V}(t)$. From

$$\frac{\mathrm{d}}{\mathrm{d}t}\Big(\mathcal{U}(t) + \mathcal{V}(t)\Big) = 2\mathcal{U}(t)\mathcal{V}(t)\cos\Delta - \mathcal{U}^2(t) - \mathcal{V}^2(t)\,(1+\epsilon)$$

$$= -\left(\mathcal{U}(t) - \mathcal{V}(t)\right)^2\cos\Delta - (1-\cos\Delta)\mathcal{U}^2(t) - \mathcal{V}^2(t)\,(1+\epsilon - \cos\Delta) < 0,$$

we have

$$\mathcal{U}(t) + \mathcal{V}(t) \le \mathcal{U}(T_{\mathrm{I}}) + \mathcal{V}(T_{\mathrm{I}}), \quad \forall t \ge T_{\mathrm{I}}.$$

Then we consider the dynamics of $\mathcal{U}(t) - \mathcal{V}(t)$. We define the hitting time

$$\tau_{\mathcal{U}-\mathcal{V}} := \inf\Big\{t \ge T_{\mathrm{I}} : \mathcal{U}(t) \le \mathcal{V}(t)\Big\}.$$

For any $t \in [T_{\mathrm{I}}, \tau_{\mathcal{U}-\mathcal{V}})$, we have:

$$\frac{\mathrm{d}}{\mathrm{d}t}\Big(\mathcal{U}(t) - \mathcal{V}(t)\Big) = -\mathcal{U}^2(t) + \mathcal{V}^2(t)\,(1+\epsilon) = -(\mathcal{U}(t) + \mathcal{V}(t))(\mathcal{U}(t) - \mathcal{V}(t)) + \epsilon\mathcal{V}^2(t)$$

$$> -(\mathcal{U}(t) + \mathcal{V}(t))(\mathcal{U}(t) - \mathcal{V}(t)) \ge -(\mathcal{U}(T_{\mathrm{I}}) + \mathcal{V}(T_{\mathrm{I}}))(\mathcal{U}(t) - \mathcal{V}(t)),$$

We consider the auxiliary ODE: $\frac{d}{\mathrm{d}t}\mathcal{P}(t) = -(\mathcal{U}(T_{\mathrm{I}}) + \mathcal{V}(T_{\mathrm{I}}))\mathcal{P}(t)$, where $\mathcal{P}(T_{\mathrm{I}}) = \mathcal{U}(T_{\mathrm{I}}) - \mathcal{V}(T_{\mathrm{I}}) > 0$. From the Comparison Principle of ODEs, we have:

$$\mathcal{U}(t) - \mathcal{V}(t) \ge \mathcal{P}(t) = (\mathcal{U}(T_{\mathrm{I}}) - \mathcal{V}(T_{\mathrm{I}}))\exp\Big(-(\mathcal{U}(T_{\mathrm{I}}) + \mathcal{V}(T_{\mathrm{I}}))(t - T_{\mathrm{I}})\Big) > 0, \ \forall t \in [T_{\mathrm{I}}, \tau_{\mathcal{U}-\mathcal{V}}).$$

From the definition of $\tau_{\mathcal{U}-\mathcal{V}}$, we have proved

$$\tau_{\mathcal{U}-\mathcal{V}} = +\infty;$$

$$\mathcal{U}(t) > \mathcal{V}(t), \ \forall t \in [T_{\mathrm{I}}, +\infty).$$

Step III. Finer estimate in the early Phase $t \in [T_{\mathrm{I}}, \tau_1]$. Define the following hitting time

$$\tau_1 := \inf\Big\{t \ge T_{\mathrm{I}} : \mathcal{U}(t)\cos\Delta \le \mathcal{V}(t)\,(1+\epsilon)\Big\}.$$

From Step I, we know $\tau_1$ exists and $\tau_1 > T_{\mathrm{I}}$. From (22), we have $\frac{\mathrm{d}\mathcal{U}(t)}{\mathrm{d}t} < 0$ and $\frac{\mathrm{d}\mathcal{V}(t)}{\mathrm{d}t} > 0$ when $t \in [T_{\mathrm{I}}, \tau_1)$. Moreover, we have the following dynamics for $t \in [T_{\mathrm{I}}, \tau_1)$:

$$\frac{\mathrm{d}\mathcal{U}}{\mathrm{d}\mathcal{V}} = \frac{\mathcal{U}\mathcal{V}\cos\Delta - \mathcal{U}^2}{\mathcal{U}\mathcal{V}\cos\Delta - \mathcal{V}^2(1+\epsilon)} = \frac{\frac{\mathcal{U}}{\mathcal{V}}\cos\Delta - \left(\frac{\mathcal{U}}{\mathcal{V}}\right)^2}{\frac{\mathcal{U}}{\mathcal{V}}\cos\Delta - (1+\epsilon)}.$$

If we define $\mathcal{Z}(t) := \frac{\mathcal{U}(t)}{\mathcal{V}(t)}$, then we have $\mathrm{d}\mathcal{U} = \mathcal{Z}\mathrm{d}\mathcal{V} + \mathcal{V}\mathrm{d}\mathcal{Z}$.

The dynamics above can be transformed to:

$$\mathcal{V}\frac{\mathrm{d}\mathcal{Z}}{\mathrm{d}\mathcal{V}} = \frac{\mathcal{Z}\cos\Delta - \mathcal{Z}^2}{\mathcal{Z}\cos\Delta - (1+\epsilon)} - \mathcal{Z},$$

which means

$$\frac{1}{\mathcal{V}}\mathrm{d}\mathcal{V} = -\frac{1}{1+\cos\Delta+\epsilon}\left(\frac{1+\epsilon}{\mathcal{Z}} + \frac{\sin^2\Delta+\epsilon}{(1+\cos\Delta+\epsilon)-\mathcal{Z}(1+\cos\Delta)}\right)\mathrm{d}\mathcal{Z}.$$

Integrating this equation from $T_{\mathrm{I}}$ to $t \in [T_{\mathrm{I}}, \tau_1)$, we have:

$$\log\left(\frac{\mathcal{V}(t)}{\mathcal{V}(T_{\mathrm{I}})}\right) = -\frac{1+\epsilon}{1+\cos\Delta+\epsilon}\log\left(\frac{\mathcal{Z}(t)}{\mathcal{Z}(T_{\mathrm{I}})}\right)$$
$$+ \frac{\sin^2\Delta+\epsilon}{(1+\cos\Delta+\epsilon)(1+\cos\Delta)}\log\left(\frac{(1+\cos\Delta)\mathcal{Z}(t) - (1+\cos\Delta+\epsilon)}{(1+\cos\Delta)\mathcal{Z}(T_{\mathrm{I}}) - (1+\cos\Delta+\epsilon)}\right), \ t \in [T_{\mathrm{I}}, \tau_1).$$
$$(24)$$

From the continuity of $\mathcal{U}(t)$, $\mathcal{V}(t)$ and $\mathcal{Z}(t)$, we have

$$\tau_1 = \inf\left\{t \geq T_{\mathrm{I}} : \mathcal{Z}(t) \leq \frac{1+\epsilon}{\cos\Delta}\right\}. \tag{25}$$

Combining (25) and (24), let $t \to \tau_1^-$. Then we have:

$$\mathcal{Z}(\tau_1) = \frac{1+\epsilon}{\cos\Delta};$$

$$\mathcal{V}(\tau_1) = \mathcal{V}(T_{\mathrm{I}})\left(\frac{\mathcal{Z}(\tau_1)}{\mathcal{Z}(T_{\mathrm{I}})}\right)^{-\frac{1+\epsilon}{1+\cos\Delta+\epsilon}}\left(\frac{(1+\cos\Delta)\mathcal{Z}(\tau_1) - (1+\cos\Delta+\epsilon)}{(1+\cos\Delta)\mathcal{Z}(T_{\mathrm{I}}) - (1+\cos\Delta+\epsilon)}\right)^{\frac{\sin^2\Delta+\epsilon}{(1+\cos\Delta+\epsilon)(1+\cos\Delta)}} > 0,$$

where

$$\left(\frac{1 - 3.87\kappa_2\sqrt{\kappa_1\kappa_2}}{1 + 3.87\kappa_2\sqrt{\kappa_1\kappa_2}}\right)p \leq \mathcal{Z}(T_{\mathrm{I}}) = \frac{\mathcal{U}(T_{\mathrm{I}})}{\mathcal{V}(T_{\mathrm{I}})} \leq p.$$

Therefore,

$$\frac{\mathcal{V}(\tau_1)}{\mathcal{V}(T_{\mathrm{I}})} \leq \left(\frac{p\cos\Delta}{1+\epsilon}\right)^{\frac{1+\epsilon}{1+\cos\Delta+\epsilon}}\left(\frac{\sin^2\Delta+\epsilon}{((1+\cos\Delta)p - (1+\cos\Delta+\epsilon))\cos\Delta}\right)^{\frac{\sin^2\Delta+\epsilon}{(1+\cos\Delta+\epsilon)(1+\cos\Delta)}}$$

$$\frac{\mathcal{V}(\tau_1)}{\mathcal{V}(T_{\mathrm{I}})} \geq \left(\frac{(1-3.87\kappa_2\sqrt{\kappa_1\kappa_2})p\cos\Delta}{(1+3.87\kappa_2\sqrt{\kappa_1\kappa_2})(1+\epsilon)}\right)^{\frac{1+\epsilon}{1+\cos\Delta+\epsilon}}\left(\frac{\sin^2\Delta+\epsilon}{\left(\frac{(1-3.87\kappa_2\sqrt{\kappa_1\kappa_2})(1+\cos\Delta)}{1+3.87\kappa_2\sqrt{\kappa_1\kappa_2}}p - (1+\cos\Delta+\epsilon)\right)\cos\Delta}\right)^{\frac{\sin^2\Delta+\epsilon}{(1+\cos\Delta+\epsilon)(1+\cos\Delta)}}$$

and

$$\mathcal{U}(\tau_1) = \frac{1+\epsilon}{\cos\Delta}\mathcal{V}(\tau_1),$$

where $1 \leq \frac{\mathcal{V}(T_{\mathrm{I}})}{\kappa_2^2 \frac{m_+}{m} \frac{1}{1+p}} \leq 1 + 3.87\kappa_2\sqrt{\kappa_1\kappa_2}$ is estimated in Step I.

Step IV. Nearly tight bounds for $\tau_1$

From the definition of $\tau_1$, we have $\frac{\mathrm{d}\mathcal{V}(t)}{\mathrm{d}t} > 0$ for any $t \in [T_{\mathrm{I}}, \tau_1)$, thus $\mathcal{V}(T_{\mathrm{I}}) < \mathcal{V}(t) < \mathcal{V}(\tau_1)$, $\forall t \in (T_{\mathrm{I}}, \tau_1)$. So we have

$$\mathcal{U}(t)\mathcal{V}(T_{\mathrm{I}})\cos\Delta - \mathcal{U}^2(t) < \frac{\mathrm{d}\mathcal{U}(t)}{\mathrm{d}t} < \mathcal{U}(t)\mathcal{V}(\tau_1)\cos\Delta - \mathcal{U}^2(t), \ \forall t \in (T_{\mathrm{I}}, \tau_1).$$

We first estimate the upper bound for $\tau_1$. Consider the following dynamics and the hitting time

$$\begin{cases} \frac{d\phi(t)}{dt} = \phi(t)\mathcal{V}(\tau_1)\cos\Delta - \phi^2(t), & t \geq T_{\mathrm{I}}, \\ \phi(T_{\mathrm{I}}) = \mathcal{U}(T_{\mathrm{I}}). \end{cases}$$

$$\tau_1^{\mathrm{u}} := \inf\left\{t > T_{\mathrm{I}} : \phi(t) \leq \mathcal{U}(\tau_1)\right\}$$

Then $\tau_1 < \tau_1^{\mathrm{u}}$. From the dynamics of $\phi(t)$, for any $t \in (T_{\mathrm{I}}, \tau_1^{\mathrm{u}}]$, it holds

$$\left. \log\left(\frac{\phi(t)}{\phi(t) - \mathcal{V}(\tau_1)\cos\Delta}\right)\right|_{T_{\mathrm{I}}}^{t} = (t - T_{\mathrm{I}})\mathcal{V}(\tau_1)\cos\Delta.$$

Therefore,

$$\begin{aligned} \tau_1^{\mathrm{u}} - T_{\mathrm{I}} =& \frac{1}{\mathcal{V}(\tau_1)\cos\Delta}\log\left(\frac{\phi(\tau_1^{\mathrm{u}})}{\phi(T_{\mathrm{I}})}\frac{(\phi(T_{\mathrm{I}}) - \mathcal{V}(\tau_1)\cos\Delta)}{(\phi(\tau_1^{\mathrm{u}}) - \mathcal{V}(\tau_1)\cos\Delta)}\right) \\ =& \frac{1}{\mathcal{V}(\tau_1)\cos\Delta}\log\left(\frac{\mathcal{U}(\tau_1)}{\mathcal{U}(T_{\mathrm{I}})}\frac{(\mathcal{U}(T_{\mathrm{I}}) - \mathcal{V}(\tau_1)\cos\Delta)}{(\mathcal{U}(\tau_1) - \mathcal{V}(\tau_1)\cos\Delta)}\right). \end{aligned}$$

With the help of Theorem C.12, we have $\frac{m_+}{m} = \Theta(1)$ and $\frac{m_-}{m} = \Theta(1)$. From Step I, we have $\mathcal{C} = \Theta\left(\frac{\kappa_2^2}{\Delta^2}\right), \mathcal{U}(T_{\mathrm{I}}) = \Theta(\kappa_2^2)$ and $\mathcal{V}(T_{\mathrm{I}}) = \Theta\left(\frac{\kappa_2^2}{p}\right)$. Moreover, it holds

$$\frac{\mathcal{V}(\tau_1)}{\mathcal{V}(T_{\mathrm{I}})} = \Theta\left(p^{\frac{1+\epsilon}{1+\cos\Delta+\epsilon}}\left(\frac{\Delta^2}{p}\right)^{\frac{\sin^2\Delta+\epsilon}{(1+\cos\Delta+\epsilon)(1+\cos\Delta)}}\right) = \Theta\left(p^{\frac{\cos\Delta}{1+\cos\Delta}}\Delta^{\frac{2\sin^2\Delta+\epsilon}{(1+\cos\Delta+\epsilon)(1+\cos\Delta)}}\right) = \Theta\left(p^{\frac{\cos\Delta}{1+\cos\Delta}}\right),$$

$$\mathcal{V}(\tau_1) = \Theta\left(\mathcal{V}(T_{\mathrm{I}})p^{\frac{\cos\Delta}{1+\cos\Delta}}\right) = \Theta\left(\kappa_2^2 p^{-\frac{1}{1+\cos\Delta}}\right).$$

It is easy to verify

$$\frac{\mathcal{U}(\tau_1)}{\mathcal{U}(T_{\mathrm{I}})} = \frac{1+\epsilon}{\cos\Delta}\frac{\mathcal{V}(\tau_1)}{\mathcal{U}(T_{\mathrm{I}})} = \Theta\left(\frac{\mathcal{V}(T_{\mathrm{I}})}{\mathcal{U}(T_{\mathrm{I}})}p^{\frac{\cos\Delta}{1+\cos\Delta}}\right) = \Theta\left(p^{-\frac{1}{1+\cos\Delta}}\right);$$

$$\frac{\mathcal{U}(T_{\mathrm{I}}) - \mathcal{V}(\tau_1)\cos\Delta}{\mathcal{U}(\tau_1) - \mathcal{V}(\tau_1)\cos\Delta} = \Theta\left(\frac{\kappa_2^2\left(1 - p^{-\frac{1}{1+\cos\Delta}}\right)}{\left(\frac{1+\epsilon}{\cos\Delta} - \cos\Delta\right)\kappa_2^2 p^{-\frac{1}{1+\cos\Delta}}}\right) = \Theta\left(\frac{\kappa_2^2\left(1 - p^{-\frac{1}{1+\cos\Delta}}\right)}{\kappa_2^2\Delta^2 p^{-\frac{1}{1+\cos\Delta}}}\right) = \Theta\left(\frac{p^{\frac{1}{1+\cos\Delta}}}{\Delta^2}\right).$$

Hence, we obtain the upper bound for $\tau_1$:

$$\begin{aligned} \tau_1 \leq \tau_1^{\mathrm{u}} = T_{\mathrm{I}} + \Theta\left(\frac{1}{\kappa_2^2 p^{-\frac{1}{1+\cos\Delta}}}\log\left(p^{-\frac{1}{1+\cos\Delta}}\frac{p^{\frac{1}{1+\cos\Delta}}}{\Delta^2}\right)\right) \\ =& \mathcal{O}\left(\sqrt{\frac{\kappa_1}{\kappa_2}}\right) + \Theta\left(\frac{p^{\frac{1}{1+\cos\Delta}}\log(1/\Delta)}{\kappa_2^2}\right) = \Theta\left(\frac{p^{\frac{1}{1+\cos\Delta}}\log(1/\Delta)}{\kappa_2^2}\right). \end{aligned}$$

In a similar way, we can derive the lower bound for $\tau_1$. Consider the following dynamics and the hitting time

$$\begin{cases} \frac{d\psi(t)}{dt} = \psi(t)\mathcal{V}(T_{\mathrm{I}})\cos\Delta - \psi^2(t), & t \geq T_{\mathrm{I}}, \\ \psi(T_{\mathrm{I}}) = \mathcal{U}(T_{\mathrm{I}}). \end{cases}$$

$$\tau_1^{\mathrm{l}} := \inf\left\{t > T_{\mathrm{I}} : \psi(t) \leq \mathcal{U}(\tau_1)\right\}$$

Then $\tau_1 > \tau_1^{\mathrm{l}}$. From the dynamics of $\phi(t)$, for any $t \in (T_{\mathrm{I}}, \tau_1^{\mathrm{l}}]$, it holds

$$\left.\log\left(\frac{\psi(t)}{\psi(t) - \mathcal{V}(T_{\mathrm{I}})\cos\Delta}\right)\right|_{T_{\mathrm{I}}}^{t} = (t - T_{\mathrm{I}})\mathcal{V}(T_{\mathrm{I}})\cos\Delta.$$

Therefore,

$$\tau_1^{\mathrm{l}} - T_{\mathrm{I}} = \frac{1}{\mathcal{V}(T_{\mathrm{I}})\cos\Delta}\log\left(\frac{\phi(\tau_1^{\mathrm{u}})}{\phi(T_{\mathrm{I}})}\frac{(\phi(T_{\mathrm{I}}) - \mathcal{V}(T_{\mathrm{I}})\cos\Delta)}{(\phi(\tau_1^{\mathrm{u}}) - \mathcal{V}(T_{\mathrm{I}})\cos\Delta)}\right)$$

$$= \frac{1}{\mathcal{V}(T_\mathrm{I}) \cos \Delta} \log \left( \frac{\mathcal{U}(\tau_1)}{\mathcal{U}(T_\mathrm{I})} \frac{(\mathcal{U}(T_\mathrm{I}) - \mathcal{V}(T_\mathrm{I}) \cos \Delta)}{(\mathcal{U}(\tau_1) - \mathcal{V}(T_\mathrm{I}) \cos \Delta)} \right)$$

$$= \frac{1}{\mathcal{V}(T_\mathrm{I}) \cos \Delta} \log \left( 1 + \frac{\mathcal{V}(T_\mathrm{I})}{\mathcal{U}(T_\mathrm{I})} \frac{(\mathcal{U}(T_\mathrm{I}) - \mathcal{U}(\tau_1)) \cos \Delta}{(\mathcal{U}(\tau_1) - \mathcal{V}(T_\mathrm{I}) \cos \Delta)} \right).$$

It is easy to verify

$$\frac{\mathcal{V}(T_\mathrm{I})}{\mathcal{U}(T_\mathrm{I})} \frac{(\mathcal{U}(T_\mathrm{I}) - \mathcal{U}(\tau_1)) \cos \Delta}{(\mathcal{U}(\tau_1) - \mathcal{V}(T_\mathrm{I}) \cos \Delta)} = \Theta \left( \frac{1}{p} \frac{\kappa_2^2 (1 - p^{-\frac{1}{1+\cos \Delta}})}{\kappa_2^2 (p^{-\frac{1}{1+\cos \Delta}} - p^{-1})} \right) = \Theta \left( p^{-\frac{\cos \Delta}{1+\cos \Delta}} \right),$$

thus

$$\log \left( 1 + \frac{\mathcal{V}(T_\mathrm{I})}{\mathcal{U}(T_\mathrm{I})} \frac{(\mathcal{U}(T_\mathrm{I}) - \mathcal{U}(\tau_1)) \cos \Delta}{(\mathcal{U}(\tau_1) - \mathcal{V}(T_\mathrm{I}) \cos \Delta)} \right) = \Theta \left( p^{-\frac{\cos \Delta}{1+\cos \Delta}} \right),$$

Hence, we obtain the lower bound for $\tau_1$:

$$\tau_1 \geq \tau_1^{\mathrm{l}} = T_\mathrm{I} + \Theta \left( \frac{1}{\frac{\kappa_2^2}{p}} p^{-\frac{\cos \Delta}{1+\cos \Delta}} \right) = \mathcal{O} \left( \sqrt{\frac{\kappa_1}{\kappa_2}} \right) + \Theta \left( \frac{p^{\frac{1}{1+\cos \Delta}}}{\kappa_2^2} \right) = \Theta \left( \frac{p^{\frac{1}{1+\cos \Delta}}}{\kappa_2^2} \right).$$

Step V. Finer Estimate in the late Phase $t > \tau_1$.

In this step, we focus on the dynamics when $t > \tau_1$.

First, we will prove the following nearly tight bound about the ratio of $\mathcal{U}(t)$ to $\mathcal{V}(t)$:

$$1 + \frac{\epsilon}{2} < \frac{\mathcal{U}(t)}{\mathcal{V}(t)} < \frac{1 + 2\epsilon}{\cos \Delta}, \quad \forall t \in [\tau_1, +\infty).$$

For the right inequality, we define the hitting time

$$\tau_{\mathcal{U}/\mathcal{V}}^r := \inf \left\{ t > \tau_1 : \mathcal{U}(t) \geq \frac{1 + 2\epsilon}{\cos \Delta} \mathcal{V}(t) \right\}.$$

From $\frac{\mathcal{U}(\tau_1)}{\mathcal{V}(\tau_1)} = \frac{1+\epsilon}{\cos \Delta} < \frac{1+2\epsilon}{\cos \Delta}$, we know $\tau_{\mathcal{U}/\mathcal{V}}^r$ exists and $\tau_{\mathcal{U}/\mathcal{V}}^r > \tau_1$.

For any $t \in (\tau_1, \tau_{\mathcal{U}/\mathcal{V}}^r)$, consider

$$\frac{\mathrm{d}}{\mathrm{d}t} \left( \mathcal{U}(t) - \frac{1 + 2\epsilon}{\cos \Delta} \mathcal{V}(t) \right) = \left( 1 - \frac{1 + 2\epsilon}{\cos \Delta} \right) \mathcal{U}(t) \mathcal{V}(t) \cos \Delta - \mathcal{U}^2(t) + \frac{1 + 2\epsilon}{\cos \Delta} (1 + \epsilon) \mathcal{V}^2(t)$$

$$= - \left( \mathcal{U}(t) - \frac{1 + 2\epsilon}{\cos \Delta} \mathcal{V}(t) \right) \left( \mathcal{U}(t) + \left( (1 + 2\epsilon)(1 + \frac{1}{\cos \Delta}) - \cos \Delta \right) \mathcal{V}(t) \right)$$

$$+ \left( \cos \Delta - (1 + 2\epsilon)(1 + \frac{1}{\cos \Delta}) + \frac{1 + 2\epsilon}{\cos \Delta} (1 + \epsilon) \right) \mathcal{V}^2(t)$$

$$= - \left( \mathcal{U}(t) - \frac{1 + 2\epsilon}{\cos \Delta} \mathcal{V}(t) \right) \left( \mathcal{U}(t) + \left( (1 + 2\epsilon)(1 + \frac{1}{\cos \Delta}) - \cos \Delta \right) \mathcal{V}(t) \right)$$

$$+ \left( (\cos \Delta - 1) + (\frac{1 + 2\epsilon}{\cos \Delta} \epsilon - 2\epsilon) \right) \mathcal{V}^2(t)$$

$$< - \left( \mathcal{U}(t) - \frac{1 + 2\epsilon}{\cos \Delta} \mathcal{V}(t) \right) \left( \mathcal{U}(t) + \left( (1 + 2\epsilon)(1 + \frac{1}{\cos \Delta}) - \cos \Delta \right) \mathcal{V}(t) \right)$$

$$\overset{\text{Step II}}{<} - \left( (1 + 2\epsilon)(1 + \frac{1}{\cos \Delta}) - \cos \Delta \right) (\mathcal{U}(t) + \mathcal{V}(t)) \left( \mathcal{U}(t) - \frac{1 + 2\epsilon}{\cos \Delta} \mathcal{V}(t) \right)$$

$$\overset{\text{Step II}}{\leq} - \left( (1 + 2\epsilon)(1 + \frac{1}{\cos \Delta}) - \cos \Delta \right) (\mathcal{U}(T_\mathrm{I}) + \mathcal{V}(T_\mathrm{I})) \left( \mathcal{U}(t) - \frac{1 + 2\epsilon}{\cos \Delta} \mathcal{V}(t) \right).$$

For simplicity, we denote $C_1 := \left( (1 + 2\epsilon)(1 + \frac{1}{\cos \Delta}) - \cos \Delta \right) (\mathcal{U}(T_\mathrm{I}) + \mathcal{V}(T_\mathrm{I})) > 0$. We consider the auxiliary ODE: $\frac{d}{dt} \mathcal{P}(t) = -C_1 \mathcal{P}(t)$, where $\mathcal{P}(\tau_1) = \mathcal{U}(\tau_1) - \frac{1+2\epsilon}{\cos \Delta} \mathcal{V}(\tau_1) < 0$. From the Comparison Principle of ODEs, we have:

$$\mathcal{U}(t) - \frac{1 + 2\epsilon}{\cos \Delta} \mathcal{V}(t) \leq \mathcal{P}(t) = \mathcal{P}(\tau_1) e^{-C_1(t - \tau_1)} < 0, \forall t \in (\tau_1, \tau_{\mathcal{U}/\mathcal{V}}^r).$$

From the definition of $\tau_{\mathcal{U}/\mathcal{V}}^r$, we have proved

$$\tau_{\mathcal{U}/\mathcal{V}}^r = +\infty;$$

$$\mathcal{U}(t) < \frac{1 + 2\epsilon}{\cos\Delta}\mathcal{V}(t), \ \forall t \in [\tau_1, +\infty).$$

In the same way, it can be proved that

$$\mathcal{U}(t) > (1 + \frac{\epsilon}{2})\mathcal{V}(t), \ \forall t \in [\tau_1, +\infty).$$

Moreover, we also need to derive a tight bound for $\mathcal{U}(t)$ and $\mathcal{V}(t)$ when $t > \tau_1$, respectively.

For any $t > \tau_1$, we have

$$\frac{\mathrm{d}}{\mathrm{d}t}\mathcal{V}(t) = \mathcal{V}(t)\Big(\mathcal{U}(t)\cos\Delta - (1+\epsilon)\mathcal{V}(t)\Big)$$
$$> \mathcal{V}(t)\Big((1 + \frac{\epsilon}{2})\mathcal{V}(t)\cos\Delta - (1+\epsilon)\mathcal{V}(t)\Big) = -\Big((1+\epsilon) - (1 + \frac{\epsilon}{2})\cos\Delta\Big)\mathcal{V}^2(t).$$

We consider the auxiliary ODE: $\frac{d}{dt}\mathcal{P}(t) = -\Big((1+\epsilon) - (1 + \frac{\epsilon}{2})\cos\Delta\Big)\mathcal{P}^2(t)$, where $\mathcal{P}(\tau_1) = \mathcal{V}(\tau_1)$. From the Comparison Principle of ODEs, we have the lower bound for $\mathcal{V}(t)$:

$$\mathcal{V}(t) \geq \mathcal{P}(t) = \frac{1}{\frac{1}{\mathcal{V}(\tau_1)} + \Big((1+\epsilon) - (1 + \frac{\epsilon}{2})\cos\Delta\Big)(t - \tau_1)}, \ \forall t \in (\tau_1, +\infty).$$

In the same way, for any $t > \tau_1$, we have

$$\frac{\mathrm{d}}{\mathrm{d}t}\mathcal{U}(t) = \mathcal{U}(t)\Big(\mathcal{V}(t)\cos\Delta - \mathcal{U}(t)\Big)$$
$$< \mathcal{U}(t)\Big(\mathcal{U}(t)\frac{\cos\Delta}{1 + \frac{\epsilon}{2}} - \mathcal{U}(t)\Big) = -\Big(1 - \frac{\cos\Delta}{1 + \frac{\epsilon}{2}}\Big)\mathcal{U}^2(t).$$

We consider the auxiliary ODE: $\frac{d}{dt}\mathcal{P}(t) = -\Big(1 - \frac{\cos\Delta}{1 + \frac{\epsilon}{2}}\Big)\mathcal{P}^2(t)$, where $\mathcal{P}(\tau_1) = \mathcal{U}(\tau_1)$. From the Comparison Principle of ODEs, we have the upper bound for $\mathcal{U}(t)$:

$$\mathcal{U}(t) \leq \mathcal{P}(t) = \frac{1}{\frac{1}{\mathcal{U}(\tau_1)} + \Big(1 - \frac{\cos\Delta}{1 + \frac{\epsilon}{2}}\Big)(t - \tau_1)}, \ \forall t \in (\tau_1, +\infty).$$

The upper bound for $\mathcal{V}(t)$ and the lower bound for $\mathcal{U}(t)$ can be estimated by:

$$\mathcal{V}(t) < \frac{\mathcal{U}(t)}{1 + \frac{\epsilon}{2}} \leq \frac{1}{1 + \frac{\epsilon}{2}}\frac{1}{\frac{1}{\mathcal{U}(\tau_1)} + \Big(1 - \frac{\cos\Delta}{1 + \frac{\epsilon}{2}}\Big)(t - \tau_1)}, \ \forall t \in (\tau_1, +\infty),$$

$$\mathcal{U}(t) > \Big(1 + \frac{\epsilon}{2}\Big)\mathcal{V}(t) \geq \frac{1 + \frac{\epsilon}{2}}{\frac{1}{\mathcal{V}(\tau_1)} + \mathcal{C}\Big((1+\epsilon) - (1 + \frac{\epsilon}{2})\cos\Delta\Big)(t - \tau_1)}, \ \forall t \in (\tau_1, +\infty).$$

Hence, we obtain the tight bound for $\mathcal{U}(t)$ and $\mathcal{V}(t)$:

$$\frac{1}{\frac{1}{(1 + \frac{\epsilon}{2})\mathcal{V}(\tau_1)} + \Big(\frac{1+\epsilon}{1 + \frac{\epsilon}{2}} - \cos\Delta\Big)(t - \tau_1)} < \mathcal{U}(t) \leq \frac{1}{\frac{1}{\mathcal{U}(\tau_1)} + \Big(1 - \frac{\cos\Delta}{1 + \frac{\epsilon}{2}}\Big)(t - \tau_1)}, \ \forall t \in (\tau_1, +\infty);$$

$$\frac{1}{\frac{1}{\mathcal{V}(\tau_1)} + \Big((1+\epsilon) - (1 + \frac{\epsilon}{2})\cos\Delta\Big)(t - \tau_1)} \leq \mathcal{V}(t) < \frac{1}{\frac{1 + \frac{\epsilon}{2}}{\mathcal{U}(\tau_1)} + \Big(1 + \frac{\epsilon}{2} - \cos\Delta\Big)(t - \tau_1)}, \ \forall t \in (\tau_1, +\infty).$$

It means

$$\mathcal{U}(t) = \Theta\left(\frac{1}{\frac{p^{\frac{1}{1+\cos\Delta}}}{\kappa_2^2} + \Delta^2(t - \tau_1)}\right), \ \forall t \geq \tau_1 = \mathcal{O}\left(\frac{p^{\frac{1}{1+\cos\Delta}}\log(1/\Delta)}{\kappa_2^2}\right);$$

$$\mathcal{V}(t) = \Theta\left(\frac{1}{\frac{p^{\frac{1}{1+\cos\Delta}}}{\kappa_2^2} + \Delta^2(t-\tau_1)}\right), \quad \forall t \geq \tau_1 = \mathcal{O}\left(\frac{p^{\frac{1}{1+\cos\Delta}}\log(1/\Delta)}{\kappa_2^2}\right).$$

Step VI. The tight bound for $\mathcal{U}(t) - \mathcal{V}(t)\cos\Delta$.

From $1 + \frac{\epsilon}{2} < \frac{\mathcal{U}(t)}{\mathcal{V}(t)} < \frac{1+2\epsilon}{\cos\Delta}$ proved in Step V, the two-side bound is straight-forward: for any $t \geq \tau_1$,

$$\mathcal{U}(t) - \mathcal{V}(t)\cos\Delta > \left(1 + \frac{\epsilon}{2} - \cos\Delta\right)\mathcal{V}(t) = \Theta\left(\Delta^2\mathcal{V}(t)\right),$$

$$\mathcal{U}(t) - \mathcal{V}(t)\cos\Delta < \left(\frac{1+2\epsilon}{\cos\Delta} - \cos\Delta\right)\mathcal{V}(t) = \Theta\left(\Delta^2\mathcal{V}(t)\right).$$

Then we obtain

$$\mathcal{U}(t) - \mathcal{V}(t)\cos\Delta = \Theta\left(\Delta^2\mathcal{V}(t)\right) = \Theta\left(\frac{1}{\frac{p^{\frac{1}{1+\cos\Delta}}}{\kappa_2^2\Delta^2} + (t-\tau_1)}\right), \quad \forall t \geq \tau_1.$$

Step VII. The tight bound of $\mathcal{U}(t)\cos\Delta - \mathcal{V}(t)$.

If we follow the proof in Step VI, $1 + \frac{\epsilon}{2} < \frac{\mathcal{U}(t)}{\mathcal{V}(t)} < \frac{1+2\epsilon}{\cos\Delta}$ can only gives us a loose two-side bound for $\mathcal{U}(t)\cos\Delta - \mathcal{V}(t)$:

$$\mathcal{U}(t)\cos\Delta - \mathcal{V}(t) > \left(\cos\Delta + \frac{m_-}{2m_+}\cos\Delta - 1\right)\mathcal{V}(t) \overset{(23)}{>} -\Theta\left(\Delta^2\mathcal{V}(t)\right),$$

$$\mathcal{U}(t)\cos\Delta - \mathcal{V}(t)\cos\Delta < \left(1 + 2\epsilon - 1\right)\mathcal{V}(t) = \Theta\left(\Delta^2\mathcal{V}(t)\right).$$

Hence, we need more fine-grained analysis to derive its sharper bounds.

We first focus on its sharper upper bound. From the dynamics (22), for any $t \geq T_{\mathrm{I}}$, we have

$$\frac{\mathrm{d}}{\mathrm{d}t}\left(\mathcal{U}(t)\cos\Delta - \mathcal{V}(t)\right)$$

$$= \left(-\left(\mathcal{U}(t)\cos\Delta - \mathcal{V}(t)\right)\left(\mathcal{U}(t) + (\frac{1}{\cos\Delta} + 1 - \cos\Delta)\mathcal{V}(t)\right) - \left(\frac{1}{\cos\Delta} - \cos\Delta - \epsilon\right)\mathcal{V}^2(t)\right)$$

$$= \left(-\left(\mathcal{U}(t)\cos\Delta - \mathcal{V}(t)\right)\left(\mathcal{U}(t) + (\frac{1}{\cos\Delta} + 1 - \cos\Delta)\mathcal{V}(t)\right) - \mathcal{V}^2(t)\left(\frac{1}{\cos\Delta} - \alpha\right)\sin^2\Delta\right).$$

We define the hitting time

$$\tau_{\mathcal{U}/\mathcal{V}}^+ := \inf\left\{t > \tau_1 : \mathcal{U}(t)\cos\Delta - \mathcal{V}(t) \leq 0\right\}.$$

From $\frac{\mathcal{U}(\tau_1)}{\mathcal{V}(\tau_1)} = \frac{1+\epsilon}{\cos\Delta}$, we know $\tau_{\mathcal{U}/\mathcal{V}}^+$ exists and $\tau_{\mathcal{U}/\mathcal{V}}^+ > \tau_1$.

Then for any $t \in (\tau_1, \tau_{\mathcal{U}/\mathcal{V}}^+)$, we have $\mathcal{U}(t)\cos\Delta - \mathcal{V}(t) > 0$, so

$$\frac{\mathrm{d}}{\mathrm{d}t}\left(\mathcal{U}(t)\cos\Delta - \mathcal{V}(t)\right)$$

$$= \left(-\left(\mathcal{U}(t)\cos\Delta - \mathcal{V}(t)\right)\left(\mathcal{U}(t) + (\frac{1}{\cos\Delta} + 1 - \cos\Delta)\mathcal{V}(t)\right) - \mathcal{V}^2(t)\left(\frac{1}{\cos\Delta} - \alpha\right)\sin^2\Delta\right)$$

$$\overset{\text{Step IV}}{<} \left(-\left(\mathcal{U}(t)\cos\Delta - \mathcal{V}(t)\right)\left(\frac{1}{\cos\Delta} + 2 + \frac{\epsilon}{2} - \cos\Delta\right)\mathcal{V}(t) - \mathcal{V}^2(t)\left(\frac{1}{\cos\Delta} - \alpha\right)\sin^2\Delta\right)$$

$$\overset{\text{Step IV}}{\leq} -\frac{\left(\frac{\sin^2\Delta}{\cos\Delta} + 2 + \frac{\epsilon}{2}\right)\left(\mathcal{U}(t)\cos\Delta - \mathcal{V}(t)\right)}{\frac{1}{\mathcal{V}(\tau_1)} + \left((1+\epsilon) - (1+\frac{\epsilon}{2})\cos\Delta\right)(t-\tau_1)} - \frac{\left(\frac{1}{\cos\Delta} - \alpha\right)\sin^2\Delta}{\left(\frac{1}{\mathcal{V}(\tau_1)} + \left((1+\epsilon) - (1+\frac{\epsilon}{2})\cos\Delta\right)(t-\tau_1)\right)^2}.$$

For simplicity, we denote $A = \frac{1}{\mathcal{V}(\tau_1)}$, $B = (1 + \epsilon) - (1 + \frac{\epsilon}{2}) \cos \Delta$, $C_1 = \frac{\sin^2 \Delta}{\cos \Delta} + 2 + \frac{\epsilon}{2}$, $C_2 = \left( \frac{1}{\cos \Delta} - \alpha \right) \sin^2 \Delta$. And we consider the auxiliary ODE:

$$\frac{\mathrm{d}\mathcal{E}(t)}{\mathrm{d}t} = -\frac{C_1 \mathcal{E}(t)}{A + B(t - \tau_1)} - \frac{C_2}{(A + B(t - \tau_1))^2},$$

$$\text{where } \mathcal{E}(\tau_1) = \mathcal{U}(\tau_1) \cos \Delta - \mathcal{V}(\tau_1) = \epsilon \mathcal{V}(\tau_1).$$

Its solution is

$$\mathcal{E}(t) = \left( 1 + \frac{B}{A}(t - \tau_1) \right)^{-\frac{C_1}{B}} \left( \mathcal{E}(\tau_1) - \frac{C_2}{A(C_1 - B)} \left( \left( 1 + \frac{B}{A}(t - \tau_1) \right)^{\frac{C_1}{B} - 1} - 1 \right) \right).$$

From the Comparison Principle of ODEs, for any $t \in (\tau_1, \tau_{\mathcal{U}/\mathcal{V}}^+)$, we have

$$\mathcal{U}(t) \cos \Delta - \mathcal{V}(t) \leq \mathcal{E}(t).$$

Let $T_{\mathcal{E}} - \tau_1 = \frac{A}{B} \left( \left( 1 + \frac{(C_1 - B)\epsilon}{C_2} \right)^{\frac{1}{\frac{C_1}{B} - 1}} - 1 \right)$, it is easy to verify $\mathcal{E}(T_{\mathcal{E}}) = 0$. Moreover,

$$\frac{A}{B} = \Theta \left( \frac{1}{\mathcal{V}(\tau_1) \left( (1 + \epsilon) - (1 + \frac{\epsilon}{2}) \cos \Delta \right)} \right) = \Theta \left( \frac{1}{\kappa_2^2 p^{-\frac{1}{1 + \cos \Delta}} \Delta^2} \right) = \Theta \left( \frac{p^{\frac{1}{1 + \cos \Delta}}}{\kappa_2^2 \Delta^2} \right);$$

$$\frac{1}{\frac{C_1}{B} - 1} \log \left( 1 + \frac{(C_1 - B)\epsilon}{C_2} \right) = \Theta \left( \Delta^2 \log \left( 1 + \frac{\Theta(\Delta^2)}{\Theta(\Delta^2)} \right) \right) = \Theta(\Delta^2);$$

$$\left( 1 + \frac{(C_1 - B)\epsilon}{C_2} \right)^{\frac{1}{\frac{C_1}{B} - 1}} - 1 = \exp \left( \frac{1}{\frac{C_1}{B} - 1} \log \left( 1 + \frac{(C_1 - B)\epsilon}{C_2} \right) \right) - 1 = \Theta(\Delta^2).$$

Therefore,

$$\tau_{\mathcal{U}/\mathcal{V}}^+ \leq T_{\mathcal{E}} = \tau_1 + \Theta \left( \frac{p^{\frac{1}{1 + \cos \Delta}}}{\kappa_2^4 \Delta^2} \Delta^2 \right)$$

$$= \mathcal{O} \left( \frac{p^{\frac{1}{1 + \cos \Delta}} \log(1/\Delta)}{\kappa_2^2} \right) + \Theta \left( \frac{p^{\frac{1}{1 + \cos \Delta}}}{\kappa_2^2} \right) = \mathcal{O} \left( \frac{p^{\frac{1}{1 + \cos \Delta}} \log(1/\Delta)}{\kappa_2^2} \right).$$

Then we define the next hitting time

$$\tau_{\mathcal{U}/\mathcal{V}}^- := \inf \left\{ t \geq \tau_{\mathcal{U}/\mathcal{V}}^+ : \mathcal{U}(t) \cos \Delta - \mathcal{V}(t) \geq 0 \right\}.$$

From $\mathcal{U}(\tau_{\mathcal{U}/\mathcal{V}}^+) \cos \Delta - \mathcal{V}(\tau_{\mathcal{U}/\mathcal{V}}^+) = 0$ and $\frac{\mathrm{d}}{\mathrm{d}t} \left( \mathcal{U}(t) \cos \Delta - \mathcal{V}(t) \right) \Big|_{t = \tau_{\mathcal{U}/\mathcal{V}}^+} < 0$, we know $\tau_{\mathcal{U}/\mathcal{V}}^-$ exists and $\tau_{\mathcal{U}/\mathcal{V}}^- > \tau_{\mathcal{U}/\mathcal{V}}^+$.

For any $t \in (\tau_{\mathcal{U}/\mathcal{V}}^+, \tau_{\mathcal{U}/\mathcal{V}}^-)$, we have $\mathcal{U}(t) \cos \Delta - \mathcal{V}(t) < 0$, so

$$\frac{\mathrm{d}}{\mathrm{d}t} \left( \mathcal{U}(t) \cos \Delta - \mathcal{V}(t) \right)$$

$$= \left( -\left( \mathcal{U}(t) \cos \Delta - \mathcal{V}(t) \right) \left( \mathcal{U}(t) + \left( \frac{1}{\cos \Delta} + 1 - \cos \Delta \right) \mathcal{V}(t) \right) - \mathcal{V}^2(t) \left( \frac{1}{\cos \Delta} - \alpha \right) \sin^2 \Delta \right)$$

$$\overset{\text{Step IV}}{<} \left( -\left( \mathcal{U}(t) \cos \Delta - \mathcal{V}(t) \right) \left( 1 + \frac{\frac{1}{\cos \Delta} + 1 - \cos \Delta}{1 + \frac{\epsilon}{2}} \right) \mathcal{U}(t) - \mathcal{V}^2(t) \left( \frac{1}{\cos \Delta} - \alpha \right) \sin^2 \Delta \right)$$

$$\overset{\text{Step IV}}{\leq} -\frac{\left( 1 + \frac{\frac{1}{\cos \Delta} + 1 - \cos \Delta}{1 + \frac{\epsilon}{2}} \right) \left( \mathcal{U}(t) \cos \Delta - \mathcal{V}(t) \right)}{\frac{1}{\mathcal{U}(\tau_1)} + \left( 1 - \frac{\cos \Delta}{1 + \frac{\epsilon}{2}} \right)(t - \tau_1)} - \frac{\left( \frac{1}{\cos \Delta} - \alpha \right) \sin^2 \Delta}{\left( \frac{1}{\mathcal{V}(\tau_1)} + \left( (1 + \epsilon) - (1 + \frac{\epsilon}{2}) \cos \Delta \right)(t - \tau_1) \right)^2}$$

$$\leq -\frac{\left(1 + \frac{\frac{1}{\cos\Delta} + 1 - \cos\Delta}{1 + \frac{\epsilon}{2}}\right)\left(\mathcal{U}(t)\cos\Delta - \mathcal{V}(t)\right)}{\frac{\cos\Delta}{1+\epsilon}\frac{1}{\mathcal{V}(\tau_1)} + \frac{1}{3}\left((1+\epsilon) - (1+\frac{\epsilon}{2})\cos\Delta\right)(t-\tau_1)} - \frac{\left(\frac{1}{\cos\Delta} - \alpha\right)\sin^2\Delta}{\left(\frac{1}{\mathcal{V}(\tau_1)} + \left((1+\epsilon) - (1+\frac{\epsilon}{2})\cos\Delta\right)(t-\tau_1)\right)^2}$$

$$\leq -\frac{3\left(1 + \frac{\frac{1}{\cos\Delta} + 1 - \cos\Delta}{1 + \frac{\epsilon}{2}}\right)\left(\mathcal{U}(t)\cos\Delta - \mathcal{V}(t)\right)}{\frac{1}{\mathcal{V}(\tau_1)} + \left((1+\epsilon) - (1+\frac{\epsilon}{2})\cos\Delta\right)(t-\tau_1)} - \frac{\left(\frac{1}{\cos\Delta} - \alpha\right)\sin^2\Delta}{\left(\frac{1}{\mathcal{V}(\tau_1)} + \left((1+\epsilon) - (1+\frac{\epsilon}{2})\cos\Delta\right)(t-\tau_1)\right)^2}.$$

For simplicity, we denote $C_3 = 3\left(1 + \frac{\frac{1}{\cos\Delta} + 1 - \cos\Delta}{1 + \frac{\epsilon}{2}}\right)$. And we consider the auxiliary ODE:

$$\frac{\mathrm{d}\mathcal{F}(t)}{\mathrm{d}t} = -\frac{C_3\mathcal{F}(t)}{A + B(t - \tau_1)} - \frac{C_2}{(A + B(t - \tau_1))^2},$$
$$\text{where } \mathcal{F}(\tau_{\mathcal{U}/\mathcal{V}}^+) = \mathcal{U}(\tau_{\mathcal{U}/\mathcal{V}}^+)\cos\Delta - \mathcal{V}(\tau_{\mathcal{U}/\mathcal{V}}^+) = 0.$$

Its solution is

$$\mathcal{F}(t) = -\frac{C_2}{A(C_3 - B)\left(1 + \frac{B}{A}(t - \tau_1)\right)^{\frac{C_3}{B}}}\left(\left(1 + \frac{B}{A}(t - \tau_1)\right)^{\frac{C_3}{B} - 1} - 1\right)$$

$$\leq -\frac{C_2}{AC_3\left(1 + \frac{B}{A}(t - \tau_1)\right)}\left(1 - \frac{1}{\left(1 + \frac{B}{A}(t - \tau_1)\right)^{\frac{C_3}{B} - 1}}\right).$$

Let $\tau_1' = \tau_1 + \Theta\left(\frac{p^{\frac{1}{1+\cos\Delta}}\log(1/\Delta)}{\kappa_2^2}\right) \geq 2\tau_1$. Then for any $t \geq \tau_1'$, it holds

$$\frac{1}{\left(1 + \frac{B}{A}(t - \tau_1)\right)^{\frac{C_3}{B} - 1}} \leq \frac{1}{\left(1 + \frac{B}{A}(\tau_1' - \tau_1)\right)^{\frac{C_3}{B} - 1}} = \exp\left(-\left(\frac{C_3}{B} - 1\right)\log\left(1 + \frac{B}{A}(\tau_1' - \tau_1)\right)\right)$$

$$= \exp\left(-\Theta\left(\frac{1}{\Delta^2}\right)\log\left(1 + \Theta\left(\frac{\kappa_2^4\Delta^2}{p^{\frac{1}{1+\cos\Delta}}}\frac{p^{\frac{1}{1+\cos\Delta}}\log(1/\Delta)}{\kappa_2^2}\right)\right)\right)$$

$$= \exp\left(-\Theta\left(\frac{1}{\Delta^2}\right)\log\left(1 + \Theta\left(\Delta^2\log(1/\Delta)\right)\right)\right) = \exp\left(-\Theta\left(\frac{1}{\Delta^2}\right)\Theta\left(\Delta^2\log(1/\Delta)\right)\right)$$

$$= \exp\left(-\Theta\left(\log(1/\Delta)\right)\right) \leq \frac{1}{2},$$

thus,

$$\mathcal{F}(t) \leq -\frac{C_2}{AC_3\left(1 + \frac{B}{A}(t - \tau_1)\right)}\left(1 - \frac{1}{2}\right) = -\Theta\left(\frac{\Delta^2}{\frac{p^{\frac{1}{1+\cos\Delta}}}{\kappa_2^2} + \Delta^2(t - \tau_1)}\right),$$
$$\forall t \geq \tau_1' = \tau_1 + \Theta\left(\frac{p^{\frac{1}{1+\cos\Delta}}\log(1/\Delta)}{\kappa_2^2}\right) \geq 2\tau_1.$$

If we let $\tau_2 = \Theta\left(\frac{p^{\frac{1}{1+\cos\Delta}}\log(1/\Delta)}{\kappa_2^2}\right) \geq 2\tau_1'$, then we have:

$$\mathcal{F}(t) \leq -\Theta\left(\frac{\Delta^2}{\frac{p^{\frac{1}{1+\cos\Delta}}}{\kappa_2^2} + \Delta^2(t - \tau_1)}\right), \ \forall t \geq \tau_2.$$

From the Comparison Principle of ODEs, for any $t \in (\tau_{\mathcal{U}/\mathcal{V}}^+, \tau_{\mathcal{U}/\mathcal{V}}^-)$, we have

$$\mathcal{U}(t)\cos\Delta - \mathcal{V}(t) \leq \mathcal{F}(t) < 0.$$

From the definition of $\tau^-_{\mathcal{U}/\mathcal{V}}$, we obtain

$$\tau^-_{\mathcal{U}/\mathcal{V}} = +\infty.$$

Moreover,

$$\mathcal{U}(t)\cos\Delta - \mathcal{V}(t) \leq -\Theta\left(\frac{\Delta^2}{\frac{p^{\frac{1}{1+\cos\Delta}}}{\kappa_2^2} + \Delta^2(t-\tau_1)}\right), \ \forall t \geq \tau_2 = \Theta\left(\frac{p^{\frac{1}{1+\cos\Delta}}\log(1/\Delta)}{\kappa_2^2}\right).$$

Recalling the lower bound at the beginning of Step VII, we obtain the tight bound:

$$\mathcal{U}(t)\cos\Delta - \mathcal{V}(t) = -\Theta\left(\frac{\Delta^2}{\frac{p^{\frac{1}{1+\cos\Delta}}}{\kappa_2^2} + \Delta^2(t-\tau_1)}\right) = -\Theta\left(\Delta^2\mathcal{V}(t)\right), \ \forall t \geq \tau_2 = \Theta\left(\frac{p^{\frac{1}{1+\cos\Delta}}\log(1/\Delta)}{\kappa_2^2}\right).$$

$\square$

**Lemma D.5** (Hitting time relationship).

$$T_{\mathrm{II}}^+ = T_{\mathrm{II}}^* = \inf\left\{t \geq T_{\mathrm{I}} : \exists k \in \mathcal{K}_+, \ \text{s.t.} \ \langle \boldsymbol{b}_k(t), \boldsymbol{x}_-\rangle \leq 0\right\}$$

$$= \inf\left\{t \geq T_{\mathrm{I}} : \exists k \in \mathcal{K}_+, \ \text{s.t.} \ \langle \boldsymbol{b}_k(T_{\mathrm{I}}), \boldsymbol{x}_-\rangle + \int_{T_{\mathrm{I}}}^t \frac{\sqrt{m}}{\kappa_2 m_+}\left(\mathcal{U}(s)\cos\Delta - \mathcal{V}(s)\right)\mathrm{d}s \leq 0\right\},$$

where $T_{\mathrm{II}}^+$ and $T_{\mathrm{II}}^*$ are defined in (20)(21), and $\mathcal{U}(t), \mathcal{V}(t)$ satisfy (22).

*Proof of Lemma D.5.*
Recall the definitions of $T_{\mathrm{II}}^+$ and $T_{\mathrm{II}}^*$:

$$T_{\mathrm{II}}^+ = \inf\left\{t > T_{\mathrm{I}} : \exists k \in \mathcal{K}_+, \ \text{s.t.} \ \langle \boldsymbol{w}_k(t), \boldsymbol{x}_+\rangle \leq 0 \ \text{or} \ \langle \boldsymbol{w}_k(t), \boldsymbol{x}_-\rangle \leq 0\right\}$$

$$= \inf\left\{t > T_{\mathrm{I}} : \exists k \in \mathcal{K}_+, \ \text{s.t.} \ \langle \boldsymbol{b}_k(t), \boldsymbol{x}_+\rangle \leq 0 \ \text{or} \ \langle \boldsymbol{b}_k(t), \boldsymbol{x}_-\rangle \leq 0\right\},$$

$$T_{\mathrm{II}}^* = \inf\left\{t > T_{\mathrm{I}} : \langle \boldsymbol{F}_+(t), \boldsymbol{x}_+\rangle \leq 0 \ \text{or} \ \exists k \in \mathcal{K}_+, \ \text{s.t.} \ \langle \boldsymbol{b}_k(t), \boldsymbol{x}_+\rangle \leq 0 \ \text{or} \ \langle \boldsymbol{b}_k(t), \boldsymbol{x}_-\rangle \leq 0\right\},$$

where $\quad \boldsymbol{F}_+(t) = \frac{p}{1+p}e^{-f_+(t)}\boldsymbol{x}_+ - \frac{1}{1+p}e^{f_-(t)}\boldsymbol{x}_-.$

Notice

$$\langle \boldsymbol{F}_+(t), \boldsymbol{x}_+\rangle = \frac{p}{1+p}e^{-f_+(t)} - \frac{1}{1+p}e^{f_-(t)}\cos\Delta = \frac{m}{\kappa_2^2 m_+}\left(\mathcal{U}(t) - \mathcal{V}(t)\cos\Delta\right).$$

And for any $k \in \mathcal{K}_+$,

$$\langle \boldsymbol{b}_k(t), \boldsymbol{x}_+\rangle = \langle \boldsymbol{b}_k(T_{\mathrm{I}}), \boldsymbol{x}_+\rangle + \int_{T_{\mathrm{I}}}^t \left\langle \frac{\mathrm{d}\boldsymbol{b}_k(s)}{\mathrm{d}s}, \boldsymbol{x}_+\right\rangle \mathrm{d}s$$

$$= \langle \boldsymbol{b}_k(T_{\mathrm{I}}), \boldsymbol{x}_+\rangle + \int_{T_{\mathrm{I}}}^t \frac{\sqrt{m}}{\kappa_2 m_+}\left(\mathcal{U}(s) - \mathcal{V}(s)\cos\Delta\right)\mathrm{d}s;$$

$$\langle \boldsymbol{b}_k(t), \boldsymbol{x}_-\rangle = \langle \boldsymbol{b}_k(T_{\mathrm{I}}), \boldsymbol{x}_-\rangle + \int_{T_{\mathrm{I}}}^t \left\langle \frac{\mathrm{d}\boldsymbol{b}_k(s)}{\mathrm{d}s}, \boldsymbol{x}_-\right\rangle \mathrm{d}s$$

$$= \langle \boldsymbol{b}_k(T_{\mathrm{I}}), \boldsymbol{x}_-\rangle + \int_{T_{\mathrm{I}}}^t \frac{\sqrt{m}}{\kappa_2 m_+}\left(\mathcal{U}(s)\cos\Delta - \mathcal{V}(s)\right)\mathrm{d}s.$$

So we have

$$T_{\mathrm{II}}^* = \sup\left\{t > T_{\mathrm{I}} : \mathcal{U}(t) - \mathcal{V}(t)\cos\Delta > 0;\right.$$

$$\left. \langle \boldsymbol{b}_k(T_{\mathrm{I}}), \boldsymbol{x}_+\rangle + \int_{T_{\mathrm{I}}}^t \frac{\sqrt{m}}{\kappa_2 m_+}\left(\mathcal{U}(s) - \mathcal{V}(s)\cos\Delta\right)\mathrm{d}s > 0, \forall k \in \mathcal{K}_+;\right.$$

$$\langle \boldsymbol{b}_k(T_{\mathrm{I}}), \boldsymbol{x}_- \rangle + \int_{T_{\mathrm{I}}}^t \frac{\sqrt{m}}{\kappa_2 m_+} \Big( \mathcal{U}(s) \cos \Delta - \mathcal{V}(s) \Big) \mathrm{d}s > 0, \forall k \in \mathcal{K}_+ \Big\}.$$

With the help of Lemma D.4, we know that $\mathcal{U}(t) - \mathcal{V}(t) \cos \Delta > 0$ for any $t \geq T_{\mathrm{I}}$. So $\langle \boldsymbol{b}_k(T_{\mathrm{I}}), \boldsymbol{x}_+ \rangle + \int_{T_{\mathrm{I}}}^t \frac{\sqrt{m}}{\kappa_2 m_+ \Delta} \Big( \mathcal{U}(s) - \mathcal{V}(s) \cos \Delta \Big) \mathrm{d}s > \langle \boldsymbol{b}_k(T_{\mathrm{I}}), \boldsymbol{x}_+ \rangle > 0, \forall k \in \mathcal{K}_+, \forall t \geq T_{\mathrm{I}}$. Hence, we have the transformation of the hitting time:

$$T_{\mathrm{II}}^* = \inf \left\{ t \geq T_{\mathrm{I}} : \exists k \in \mathcal{K}_+, \text{ s.t. } \langle \boldsymbol{b}_k(T_{\mathrm{I}}), \boldsymbol{x}_- \rangle + \int_{T_{\mathrm{I}}}^t \frac{\sqrt{m}}{\kappa_2 m_+} \Big( \mathcal{U}(s) \cos \Delta - \mathcal{V}(s) \Big) \mathrm{d}s \leq 0 \right\},$$
$$T_{\mathrm{II}}^+ = T_{\mathrm{II}}^*.$$

$\square$

**Lemma D.6** (Time Estimate of Phase II).

$$T_{\mathrm{II}} = T_{\mathrm{II}}^+ = T_{\mathrm{II}}^* = \Theta \left( \frac{p^{\frac{1}{1 - \alpha \cos \Delta}}}{\kappa_2^2 \Delta^2} \right).$$

*Proof of Lemma D.6.*
With the help of Theorem C.12 (S1), for any $k \in \mathcal{K}_+$, we have:

$$\frac{4.66 \sqrt{\kappa_1 \kappa_2}}{\sqrt{m}} \leq \rho_k(T_{\mathrm{I}}) \leq \frac{12 \sqrt{\kappa_1 \kappa_2}}{\sqrt{m}};$$

$$\langle \boldsymbol{w}_k(T_{\mathrm{I}}), \boldsymbol{\mu} \rangle \geq \Big( 1 - 4.2 \sqrt{\kappa_1 \kappa_2} \Big) \left( 1 - \frac{2}{1 + 0.7 \left( 1 + 9.9 \sqrt{\frac{\kappa_2}{\kappa_1}} \right)^{1.15}} \right)$$
$$> 1 - 4.2 \sqrt{\kappa_1 \kappa_2} - \frac{2}{1 + 0.7 \left( 1 + 9.9 \sqrt{\frac{\kappa_2}{\kappa_1}} \right)^{1.15}}.$$

With the help of Lemma I.4, we have the estimate of $\langle \boldsymbol{w}_k(T_{\mathrm{I}}), \boldsymbol{x}_- \rangle$:

$$-2\sqrt{\epsilon} \sin \Delta - \epsilon \leq \langle \boldsymbol{w}_k(T_{\mathrm{I}}), \boldsymbol{x}_- \rangle - \frac{p \cos \Delta - 1}{\sqrt{p^2 + 1 - 2p \cos \Delta}} \leq 2\sqrt{\epsilon} \sin \Delta,$$

where

$$\epsilon = 4.2 \sqrt{\kappa_1 \kappa_2} + \frac{2}{1 + 0.7 \left( 1 + 9.9 \sqrt{\frac{\kappa_2}{\kappa_1}} \right)^{1.15}}.$$

Then we have:

$$\langle \boldsymbol{b}_k(T_{\mathrm{I}}), \boldsymbol{x}_- \rangle \leq \frac{12 \sqrt{\kappa_1 \kappa_2}}{\sqrt{m}} \left( \frac{p \cos \Delta - 1}{\sqrt{p^2 + 1 - 2p \cos \Delta}} + 2\sqrt{\epsilon} \sin \Delta \right),$$
$$\langle \boldsymbol{b}_k(T_{\mathrm{I}}), \boldsymbol{x}_- \rangle \geq \frac{4.66 \sqrt{\kappa_1 \kappa_2}}{\sqrt{m}} \left( \frac{p \cos \Delta - 1}{\sqrt{p^2 + 1 - 2p \cos \Delta}} - 2\sqrt{\epsilon} \sin \Delta - \epsilon \right),$$

which means

$$\langle \boldsymbol{b}_k(T_{\mathrm{I}}), \boldsymbol{x}_- \rangle = \Theta \left( \frac{\sqrt{\kappa_1 \kappa_2}}{\sqrt{m}} \right).$$

From the dynamics (22), we have

$$\mathcal{U}(t) \cos \Delta - \mathcal{V}(t)$$
$$= \frac{m_- \cos \Delta}{m_- + m_+} \Big( \mathcal{U}(t) - \mathcal{V}(t) \cos \Delta \Big) + \frac{m_+}{m_- + m_+} \Big( \mathcal{U}(t) \cos \Delta - \mathcal{V}(t) \big( 1 + \alpha \sin^2 \Delta \big) \Big)$$

$$= -\frac{m_- \cos \Delta}{m_- + m_+} \frac{\mathrm{d}\mathcal{U}(t)}{\mathcal{U}(t)\mathrm{d}t} + \frac{m_+}{m_- + m_+} \frac{\mathrm{d}\mathcal{V}(t)}{\mathcal{V}(t)\mathrm{d}t},$$

Taking integral, we obtain:

$$\int_{T_{\mathrm{I}}}^{t} \frac{\sqrt{m}}{\kappa_2 m_+} \Big(\mathcal{U}(s) \cos \Delta - \mathcal{V}(s)\Big) \mathrm{d}s$$

$$= \frac{\sqrt{m}}{\kappa_2 m_+} \int_{T_{\mathrm{I}}}^{t} \left( -\frac{m_- \cos \Delta}{m_- + m_+} \frac{\mathrm{d}\mathcal{U}(s)}{\mathcal{U}(s)} + \frac{m_+}{m_- + m_+} \frac{\mathrm{d}\mathcal{V}(s)}{\mathcal{V}(s)} \right)$$

$$= \frac{\sqrt{m}}{\kappa_2 m_+} \left( -\frac{m_- \cos \Delta}{m_- + m_+} \log \left( \frac{\mathcal{U}(t)}{\mathcal{U}(T_{\mathrm{I}})} \right) + \frac{m_+}{m_- + m_+} \log \left( \frac{\mathcal{V}(t)}{\mathcal{V}(T_{\mathrm{I}})} \right) \right)$$

$$= \frac{\sqrt{m}}{\kappa_2 m_+} \log \left( \frac{\mathcal{U}(T_{\mathrm{I}})^{\frac{m_- \cos \Delta}{(m_- + m_+)}}}{\mathcal{U}(t)^{\frac{m_- \cos \Delta}{(m_- + m_+)}}} \cdot \frac{\mathcal{V}(t)^{\frac{m_+}{m_- + m_+}}}{\mathcal{V}(T_{\mathrm{I}})^{\frac{m_+}{m_- + m_+}}} \right).$$

From Lemma D.5, we have:

$$T_{\mathrm{II}}^+ = T_{\mathrm{II}}^* = \inf \left\{ t \geq T_{\mathrm{I}} : \exists k \in \mathcal{K}_+, \text{ s.t. } \langle \boldsymbol{b}_k(T_{\mathrm{I}}), \boldsymbol{x}_- \rangle + \int_{T_{\mathrm{I}}}^{t} \frac{\sqrt{m}}{\kappa_2 m_+} \Big(\mathcal{U}(s) \cos \Delta - \mathcal{V}(s)\Big) \mathrm{d}s \leq 0 \right\}.$$

Recalling the definition of $\tau_1$ in Lemma D.4, we know $\mathcal{U}(s) \cos \Delta - \mathcal{V}(s) > 0$ for any $t \leq \tau_1$, so $T_{\mathrm{II}}^* > \tau_1$.

From Lemma D.4 (S4), we know

$$\mathcal{U}(t) = \Theta \left( \frac{1}{p^{\frac{1}{1+\cos \Delta}}_{\kappa_2^2} + \Delta^2(t - \tau_1)} \right), \quad \mathcal{V}(t) = \Theta \left( \frac{1}{p^{\frac{1}{1+\cos \Delta}}_{\kappa_2^2} + \Delta^2(t - \tau_1)} \right).$$

From the proof of Lemma D.4, we know $\mathcal{U}(T_{\mathrm{I}}) = \Theta(\kappa_2^2)$ and $\mathcal{V}(T_{\mathrm{I}}) = \Theta\left(\frac{\kappa_2^2}{p}\right)$.

Therefore, solving

$$0 = \langle \boldsymbol{b}_k(T_{\mathrm{I}}), \boldsymbol{x}_- \rangle + \int_{T_{\mathrm{I}}}^{T_{\mathrm{II}}^*} \frac{\sqrt{m}}{\kappa_2 m_+} \Big(\mathcal{U}(s) \cos \Delta - \mathcal{V}(s)\Big) \mathrm{d}s$$

$$= \Theta \left( \frac{\sqrt{\kappa_1 \kappa_2}}{\sqrt{m}} \right) + \frac{\sqrt{m}}{\kappa_2 m_+} \log \left( \frac{\mathcal{U}(T_{\mathrm{I}})^{\frac{m_- \cos \Delta}{(m_- + m_+)}}}{\mathcal{U}(T_{\mathrm{II}}^*)^{\frac{m_- \cos \Delta}{(m_- + m_+)}}} \cdot \frac{\mathcal{V}(T_{\mathrm{II}}^*)^{\frac{m_+}{m_- + m_+}}}{\mathcal{V}(T_{\mathrm{I}})^{\frac{m_+}{m_- + m_+}}} \right)$$

$$= \Theta \left( \frac{\sqrt{\kappa_1 \kappa_2}}{\sqrt{m}} \right) + \Theta \left( \frac{1}{\kappa_2 \sqrt{m}} \right) \log \left( \frac{\mathcal{U}(T_{\mathrm{I}})^{\frac{m_- \cos \Delta}{(m_- + m_+)}}}{\mathcal{U}(T_{\mathrm{II}}^*)^{\frac{m_- \cos \Delta}{(m_- + m_+)}}} \cdot \frac{\mathcal{V}(T_{\mathrm{II}}^*)^{\frac{m_+}{m_- + m_+}}}{\mathcal{V}(T_{\mathrm{I}})^{\frac{m_+}{m_- + m_+}}} \right),$$

we obtain

$$\frac{\mathcal{V}(T_{\mathrm{II}}^*)^{\frac{m_+}{m_- + m_+}}}{\mathcal{U}(T_{\mathrm{II}}^*)^{\frac{m_- \cos \Delta}{(m_- + m_+)}}} \cdot \frac{\mathcal{U}(T_{\mathrm{I}})^{\frac{m_- \cos \Delta}{(m_- + m_+)}}}{\mathcal{V}(T_{\mathrm{I}})^{\frac{m_+}{m_- + m_+}}} = \exp\left(-\Theta\left(\kappa_2 \sqrt{\kappa_1 \kappa_2}\right)\right) = \Theta(1).$$

A straight-forward calculation gives us:

$$\frac{\mathcal{V}(T_{\mathrm{II}}^*)^{\frac{m_+}{m_- + m_+}}}{\mathcal{U}(T_{\mathrm{II}}^*)^{\frac{m_- \cos \Delta}{(m_- + m_+)}}} \cdot \frac{\mathcal{U}(T_{\mathrm{I}})^{\frac{m_- \cos \Delta}{(m_- + m_+)}}}{\mathcal{V}(T_{\mathrm{I}})^{\frac{m_+}{m_- + m_+}}}$$

$$= \Theta \left( \frac{1}{\left( p^{\frac{1}{1+\cos \Delta}}_{\kappa_2^2} + \Delta^2(T_{\mathrm{II}}^* - \tau_1) \right)^{\frac{m_+ - m_- \cos \Delta}{m_+ + m_-}}} \cdot \frac{(\kappa_2^2)^{\frac{m_- \cos \Delta}{m_+ + m_-}}}{\left( \frac{\kappa_2^2}{p} \right)^{\frac{m_+}{m_+ + m_-}}} \right)$$

$$=\Theta\left(\frac{p^{\frac{m_+}{m_++m_-}}}{\left(p^{\frac{1}{1+\cos\Delta}}+\kappa_2^2\Delta^2(T_{\mathrm{II}}^*-\tau_1)\right)^{\frac{m_+-m_-\cos\Delta}{m_++m_-}}}\right).$$

Hence, we get

$$T_{\mathrm{II}}^*-\tau_1=\Theta\left(\frac{1}{\kappa_2^2\Delta^2}\left(p^{\frac{m_+}{m_+-m_-\cos\Delta}}-\Theta\left(p^{\frac{1}{1+\cos\Delta}}\right)\right)\right)=\Theta\left(\frac{p^{\frac{1}{1-\alpha\cos\Delta}}}{\kappa_2^2\Delta^2}\right),$$

Combining Lemma D.4 (S3), we obtain

$$T_{\mathrm{II}}^+=T_{\mathrm{II}}^*=\tau_1+\Theta\left(\frac{p^{\frac{1}{1-\alpha\cos\Delta}}}{\kappa_2^2\Delta^2}\right)=\mathcal{O}\left(\frac{p^{\frac{1}{1+\cos\Delta}}\log(1/\Delta)}{\kappa_2^2}\right)+\Theta\left(\frac{p^{\frac{1}{1-\alpha\cos\Delta}}}{\kappa_2^2\Delta^2}\right)=\Theta\left(\frac{p^{\frac{1}{1-\alpha\cos\Delta}}}{\kappa_2^2\Delta^2}\right).$$

Recall the relationship between $T_{\mathrm{II}}$ and $T_{\mathrm{II}}^+$ (19)(20):

$$T_{\mathrm{II}}=T_{\mathrm{II}}^+\wedge\inf\{t>T_{\mathrm{I}}:\exists k\in\mathcal{K}_-,\ \text{s.t. }\langle\boldsymbol{w}_k(t),\boldsymbol{x}_+\rangle\neq 0\text{ or }\langle\boldsymbol{w}_k(t),\boldsymbol{x}_-\rangle\leq 0\}.$$

Then using Lemma D.1 (S2), we obtain:

$$T_{\mathrm{II}}=T_{\mathrm{II}}^+=\Theta\left(\frac{p^{\frac{1}{1-\alpha\cos\Delta}}}{\kappa_2^2\Delta^2}\right).$$

$\square$

**Lemma D.7** (Length of Plateau).
*If we define the hitting time $T_{\mathrm{plat}}:=\inf\left\{t\in[T_{\mathrm{I}},T_{\mathrm{II}}]:\mathrm{Acc}(t)=1\right\}$, then we have:*

**(S1).** $T_{\mathrm{plat}}=\Theta\left(\frac{p}{\kappa_2^2\Delta^2}\right)$.

**(S2).** $\forall t\in[T_{\mathrm{I}},T_{\mathrm{plat}}],\ \mathrm{Acc}(t)\equiv\frac{p}{1+p}$.

**(S3).** $\forall t\in(T_{\mathrm{plat}},T_{\mathrm{II}}],\ \mathrm{Acc}(t)\equiv 1$.

*Proof of Lemma D.7.*
It is easy to verify

$$T_{\mathrm{plat}}=\inf\left\{t\in[T_{\mathrm{I}},T_{\mathrm{II}}]:f_+(t)\leq 0\text{ or }f_-(t)>0\right\}.$$

From Theorem C.12 (S4), we know $f_+(T_{\mathrm{I}})>0$ and $f_-(T_{\mathrm{I}})>0$. From Lemma D.3, we have

$$\mathcal{U}(t)=\kappa_2^2\frac{m_+}{m}\frac{p}{1+p}e^{-f_+(t)},\quad\mathcal{V}(t)=\kappa_2^2\frac{m_+}{m}\frac{1}{1+p}e^{f_-(t)}.$$

From the proof of Lemma D.4, we know $\frac{\mathrm{d}}{\mathrm{d}t}\mathcal{U}(t)<0,\ \forall t\in[T_{\mathrm{I}},T_{\mathrm{II}}]$, so

$$\mathcal{U}(t)\leq\mathcal{U}(T_{\mathrm{I}}),\quad f_+(t)\geq f_+(T_{\mathrm{I}})>0,\ \forall t\in[T_{\mathrm{I}},T_{\mathrm{II}}].$$

Recall the definition of $\tau_1$ and $\tau_2$ in Lemma D.4. From the proof of Lemma D.4, we know $\frac{\mathrm{d}}{\mathrm{d}t}\mathcal{V}(t)>0,\ \forall t\in[T_{\mathrm{I}},\tau_1)$, so

$$\mathcal{V}(t)\geq\mathcal{V}(T_{\mathrm{I}})\quad,f_-(t)\geq f_-(T_{\mathrm{I}})>0,\ \forall t\in[T_{\mathrm{I}},\tau_1].$$

With the help of Lemma D.4 (S4), we know

$$\mathcal{V}(t)=\Theta\left(\frac{1}{\frac{p^{\frac{1}{1+\cos\Delta}}}{\kappa_2^2}+\Delta^2(t-\tau_1)}\right),\ \forall t\in(\tau_1,+\infty).$$

Because $\mathcal{V}(T_{\text{plat}}) = \kappa_2^2 \frac{m_+}{m} \frac{1}{1+p} = \Theta\left(\frac{\kappa_2^2}{p}\right)$, we have

$$\Theta\left(\frac{\kappa_2^2}{p}\right) = \Theta\left(\frac{1}{\frac{p^{\frac{1}{1+\cos\Delta}}}{\kappa_2^2} + \Delta^2(T_{\text{plat}} - \tau_1)}\right), \quad \forall t \in (\tau_1, +\infty).$$

Therefore,

$$T_{\text{plat}} = \tau_1 + \Theta\left(\frac{p}{\kappa_2^2 \Delta^2}\left(1 - \Theta\left(\frac{1}{p^{\frac{\cos\Delta}{1+\cos\Delta}}}\right)\right)\right) = \tau_1 + \Theta\left(\frac{p}{\kappa_2^2\Delta^2}\right)$$

$$= \mathcal{O}\left(\frac{p^{\frac{1}{1+\cos\Delta}}\log(1/\Delta)}{\kappa_2^2}\right) + \Theta\left(\frac{p}{\kappa_2^2\Delta^2}\right) = \Theta\left(\frac{p}{\kappa_2^2\Delta^2}\right).$$

It is easy to verify $T_{\text{plat}} = \Theta\left(\frac{p}{\kappa_2^2\Delta^2}\right) < T_{\text{II}} = \Theta\left(\frac{p^{\frac{1}{1-\alpha\cos\Delta}}}{\kappa_2^2\Delta^2}\right).$

Because $\tau_2 - \tau_1 = \Theta\left(\frac{p^{\frac{1}{1+\cos\Delta}}\log(1/\Delta)}{\kappa_2^2}\right)$, for any $t \in (\tau_1, \tau_2]$, we have

$$\mathcal{V}(t) = \Theta\left(\frac{1}{\frac{p^{\frac{1}{1+\cos\Delta}}}{\kappa_2^2} + \Delta^2(t-\tau_1)}\right) = \Theta\left(\frac{1}{\frac{p^{\frac{1}{1+\cos\Delta}}}{\kappa_2^2} + \Delta^2(\tau_2 - \tau_1)}\right)$$

$$= \Theta\left(\frac{\kappa_2^2}{p^{\frac{1}{1+\cos\Delta}}}\right) \gg \Theta\left(\frac{\kappa_2^2}{p}\right) = \mathcal{V}(T_{\text{plat}}).$$

Thus,

$$f_-(t) > f_-(T_{\text{plat}}) = 0, \quad \forall t \in (\tau_1, \tau_2].$$

From Lemma D.4 (S6), we know that $\mathcal{U}(t)\cos\Delta - \mathcal{V}(t) < 0$, $\forall t \in (\tau_2, T_{\text{II}}]$. Then $\frac{d\mathcal{V}(t)}{dt} < 0$, $\forall t \in (\tau_2, T_{\text{II}}]$. Thus,

$$f_-(t) > f_-(T_{\text{plat}}) = 0, \quad \forall t \in (\tau_2, T_{\text{plat}});$$
$$f_-(t) < f_-(T_{\text{plat}}) = 0, \quad \forall t \in (T_{\text{plat}}, T_{\text{II}}].$$

Hence, we know

$$f_-(t) \geq 0, \quad \forall t \in [T_{\text{I}}, T_{\text{plat}}];$$
$$f_-(t) < 0, \quad \forall t \in (T_{\text{plat}}, T_{\text{II}}].$$

In summary, we have proved (S1)(S2)(S3).

$\square$

**Lemma D.8** (Prediction at end of Phase II).
*(S1) For the predictions, we have:*

$$e^{-f_+(T_{\text{II}})} = \Theta\left(p^{-\frac{1}{1-\alpha\cos\Delta}}\right), \quad e^{f_-(T_{\text{II}})} = \Theta\left(p^{-\frac{\alpha\cos\Delta}{1-\alpha\cos\Delta}}\right), \quad \mathcal{L}(\boldsymbol{\theta}(T_{\text{II}})) = \Theta\left(p^{-\frac{1}{1-\alpha\cos\Delta}}\right);$$

$$\frac{pe^{-f_+(T_{\text{II}})}}{1+p} - \frac{e^{f_-(T_{\text{II}})}}{1+p}\cos\Delta = \Theta\left(\Delta^2 p^{-\frac{1}{1-\alpha\cos\Delta}}\right),$$

$$\frac{pe^{-f_+(T_{\text{II}})}}{1+p}\cos\Delta - \frac{e^{f_-(T_{\text{II}})}}{1+p} = -\Theta\left(\Delta^2 p^{-\frac{1}{1-\alpha\cos\Delta}}\right).$$

*(S2). For any $k \in \mathcal{K}_+$, we have:*

$$\langle \boldsymbol{b}_k(T_{\text{II}}), \boldsymbol{x}_-\rangle = \mathcal{O}\left(\frac{\sqrt{\kappa_1\kappa_2}}{\sqrt{m}}\right).$$

*(S3). For any $k \in \mathcal{K}_-$, we have $\langle \boldsymbol{b}_k(T_{\text{II}}), \boldsymbol{x}_+\rangle = 0$.*

*Proof of Lemma D.8.*
Proof of (S1). Recall the definitions in Lemma D.3:

$$\begin{cases} \mathcal{U}(t) = \kappa_2^2 \frac{m_+}{m} \frac{p}{1+p} e^{-f_+(t)}, \\ \mathcal{V}(t) = \kappa_2^2 \frac{m_+}{m} \frac{1}{1+p} e^{f_-(t)}. \end{cases}$$

From Lemma D.4 (S4) and Lemma D.6, we have

$$\mathcal{U}(t) = \Theta\left( \frac{1}{\frac{p^{\frac{1}{1+\cos\Delta}}}{\kappa_2^2} + \Delta^2(t-\tau_1)} \right), \quad \mathcal{V}(t) = \Theta\left( \frac{1}{\frac{p^{\frac{1}{1+\cos\Delta}}}{\kappa_2^2} + \Delta^2(t-\tau_1)} \right),$$

$$\tau_1 = \mathcal{O}\left( \frac{p^{\frac{1}{1+\cos\Delta}} \log(1/\Delta)}{\kappa_2^2} \right), \quad T_{\mathrm{II}} = \Theta\left( \frac{p^{\frac{1}{1-\alpha\cos\Delta}}}{\kappa_2^2 \Delta^2} \right).$$

Therefore, we obtain the estimate:

$$e^{-f_+(T_{\mathrm{II}})} = \Theta\left( \frac{1}{\kappa_2^2} \frac{1}{\mathcal{U}(T_{\mathrm{II}})} \right) = \Theta\left( \frac{1}{\kappa_2^2} \frac{1}{\frac{p^{\frac{1}{1+\cos\Delta}}}{\kappa_2^2} + \Delta^2 \frac{p^{\frac{1}{1-\alpha\cos\Delta}}}{\kappa_2^2\Delta^2}} \right) = \Theta\left( p^{-\frac{1}{1-\alpha\cos\Delta}} \right),$$

$$e^{f_-(T_{\mathrm{II}})} = \Theta\left( \frac{p}{\kappa_2^2} \frac{1}{\mathcal{V}(T_{\mathrm{II}})} \right) = \Theta\left( \frac{p}{\kappa_2^2} \frac{1}{\frac{p^{\frac{1}{1+\cos\Delta}}}{\kappa_2^2} + \Delta^2 \frac{p^{\frac{1}{1-\alpha\cos\Delta}}}{\kappa_2^2\Delta^2}} \right) = \Theta\left( p^{-\frac{\alpha\cos\Delta}{1-\alpha\cos\Delta}} \right).$$

Moreover, Lemma D.4 (S4)(S5)(S6) give us

$$\mathcal{U}(T_{\mathrm{II}}) - \mathcal{V}(T_{\mathrm{II}})\cos\Delta = \Theta\left( \Delta^2 \mathcal{V}(T_{\mathrm{II}}) \right) = \Theta\left( \kappa_2^2 \Delta^2 p^{-\frac{1}{1-\alpha\cos\Delta}} \right),$$

$$\mathcal{U}(T_{\mathrm{II}})\cos\Delta - \mathcal{V}(T_{\mathrm{II}}) = -\Theta\left( \Delta^2 \mathcal{V}(T_{\mathrm{II}}) \right) = -\Theta\left( \kappa_2^2 \Delta^2 p^{-\frac{1}{1-\alpha\cos\Delta}} \right).$$

Hence,

$$\frac{pe^{-f_+(T_{\mathrm{II}})}}{1+p} - \frac{e^{f_-(T_{\mathrm{II}})}}{1+p}\cos\Delta = \frac{1}{\kappa_2^2}\left( \mathcal{U}(T_{\mathrm{II}}) - \mathcal{V}(T_{\mathrm{II}})\cos\Delta \right) = \Theta\left( \Delta^2 p^{-\frac{1}{1-\alpha\cos\Delta}} \right),$$

$$\frac{pe^{-f_+(T_{\mathrm{II}})}}{1+p}\cos\Delta - \frac{e^{f_-(T_{\mathrm{II}})}}{1+p} = \frac{1}{\kappa_2^2}\left( \mathcal{U}(T_{\mathrm{II}})\cos\Delta - \mathcal{V}(T_{\mathrm{II}}) \right) = -\Theta\left( \Delta^2 p^{-\frac{1}{1-\alpha\cos\Delta}} \right).$$

Proof of (S2). Denote $\mathcal{K}_+^0 := \left\{ k \in \mathcal{K}_+ : \langle \boldsymbol{w}_k(T_{\mathrm{II}}), \boldsymbol{x}_- \rangle = 0 \right\}$. From the definition of $T_{\mathrm{II}}$ and the proof in Phase II, we know that $\langle \boldsymbol{w}_k(T_{\mathrm{II}}), \boldsymbol{x}_- \rangle > 0$ holds for any $k \in \mathcal{K}_+ - \mathcal{K}_+^0$.

From the proof in Lemma D.5, for any $k \in \mathcal{K}_+$, it holds

$$\langle \boldsymbol{b}_k(t), \boldsymbol{x}_- \rangle = \langle \boldsymbol{b}_k(T_{\mathrm{I}}), \boldsymbol{x}_- \rangle + \int_{T_{\mathrm{I}}}^t \left\langle \frac{\mathrm{d}\boldsymbol{b}_k(s)}{\mathrm{d}s}, \boldsymbol{x}_- \right\rangle \mathrm{d}s$$

$$= \langle \boldsymbol{b}_k(T_{\mathrm{I}}), \boldsymbol{x}_- \rangle + \int_{T_{\mathrm{I}}}^t \frac{\sqrt{m}}{\kappa_2 m_+}\left( \mathcal{U}(s)\cos\Delta - \mathcal{V}(s) \right)\mathrm{d}s, \quad \forall t \in [T_{\mathrm{I}}, T_{\mathrm{II}}].$$

Thus for any $k \in \mathcal{K}_+ - \mathcal{K}_+^0$, we have

$$\langle \boldsymbol{b}_k(T_{\mathrm{II}}), \boldsymbol{x}_- \rangle - \langle \boldsymbol{b}_k(T_{\mathrm{I}}), \boldsymbol{x}_- \rangle = \langle \boldsymbol{b}_{k_0}(T_{\mathrm{II}}), \boldsymbol{x}_- \rangle - \langle \boldsymbol{b}_{k_0}(T_{\mathrm{I}}), \boldsymbol{x}_- \rangle = -\langle \boldsymbol{b}_{k_0}(T_{\mathrm{I}}), \boldsymbol{x}_- \rangle,$$

so

$$\langle \boldsymbol{b}_k(T_{\mathrm{II}}), \boldsymbol{x}_- \rangle = \langle \boldsymbol{b}_k(T_{\mathrm{I}}), \boldsymbol{x}_- \rangle - \langle \boldsymbol{b}_{k_0}(T_{\mathrm{I}}), \boldsymbol{x}_- \rangle.$$

From the proof of Lemma D.6, we know

$$\langle \boldsymbol{b}_k(T_{\mathrm{I}}), \boldsymbol{x}_- \rangle \leq \frac{12\sqrt{\kappa_1\kappa_2}}{\sqrt{m}}\left( \frac{p\cos\Delta - 1}{\sqrt{p^2 + 1 - 2p\cos\Delta}} + 2\sqrt{\epsilon}\sin\Delta \right),$$

$$\langle \boldsymbol{b}_k(T_{\mathrm{I}}), \boldsymbol{x}_- \rangle \geq \frac{4.66\sqrt{\kappa_1\kappa_2}}{\sqrt{m}}\left(\frac{p\cos\Delta - 1}{\sqrt{p^2 + 1 - 2p\cos\Delta}} - 2\sqrt{\epsilon}\sin\Delta - \epsilon\right),$$

where $\epsilon = 4.2\sqrt{\kappa_1\kappa_2} + \frac{2}{1+0.7\left(1+9.9\sqrt{\frac{\kappa_2}{\kappa_1}}\right)^{1.15}}$. This means

$$\langle \boldsymbol{b}_k(T_{\mathrm{I}}), \boldsymbol{x}_- \rangle = \Theta\left(\frac{\sqrt{\kappa_1\kappa_2}}{\sqrt{m}}\right).$$

Hence, for any $k \in \mathcal{K}_+ - \mathcal{K}_+^0$,

$$0 < \langle \boldsymbol{b}_k(T_{\mathrm{II}}), \boldsymbol{x}_- \rangle = |\langle \boldsymbol{b}_k(T_{\mathrm{I}}), \boldsymbol{x}_- \rangle - \langle \boldsymbol{b}_{k_0}(T_{\mathrm{I}}), \boldsymbol{x}_- \rangle| \leq |\langle \boldsymbol{b}_k(T_{\mathrm{I}}), \boldsymbol{x}_- \rangle| + |\langle \boldsymbol{b}_{k_0}(T_{\mathrm{I}}), \boldsymbol{x}_- \rangle|$$

$$= \Theta\left(\frac{\sqrt{\kappa_1\kappa_2}}{\sqrt{m}}\right) + \Theta\left(\frac{\sqrt{\kappa_1\kappa_2}}{\sqrt{m}}\right) = \Theta\left(\frac{\sqrt{\kappa_1\kappa_2}}{\sqrt{m}}\right).$$

Proof of (S3). Due to the dynamics of the neuron $k \in \mathcal{K}_-$ in Phase II, this conclusion is clear.

$\square$

As simple corollaries of these lemmas, we can prove two theorems in Phase II.

*Proof of Theorem 4.4 and 4.5.*
Theorem 4.5 is Lemma D.7. Theorem 4.4 (S1) has been proven in Lemma D.6; Theorem 4.4 (S2) has been proven in Lemma D.8. Additionally, combining (i) $T_{\mathrm{II}} = T_{\mathrm{II}}^+$ in Lemma D.6, (ii) the transformation in Lemma D.5, and (iii) the definition of $T_{\mathrm{II}}$, we obtain Theorem 4.4 (S3).

$\square$

# E  Proofs of Optimization Dynamics in Phase III

## E.1  Optimization Dynamics during Phase Transition

Building upon Phase II, we will demonstrate that within a short time, all the living positive neurons $\mathcal{K}_+$ change their activation patterns, corresponding to a "phase transition". After the phase transition, *all* the living positive neurons $k \in \mathcal{K}_+$ undergo deactivation for $\boldsymbol{x}_-$, i.e., $\mathtt{sgn}_k^-(t)$ changes from 1 to 0, while other activation patterns remain unchanged.

Specifically, we define the hitting time

$$
\begin{aligned}
T_{\mathrm{II}}^{\mathrm{PT}} &:= \inf\{t > T_{\mathrm{II}} : \forall k \in \mathcal{K}_+, \mathtt{sgn}_k^-(t) = 0\} \\
&= \inf\big\{t > T_{\mathrm{II}} : \forall k \in \mathcal{K}_+, \langle \boldsymbol{w}_k(t), \boldsymbol{x}_- \rangle = 0\big\},
\end{aligned}
\tag{26}
$$

and we call $t \in (T_{\mathrm{II}}, T_{\mathrm{II}}^{\mathrm{PT}}]$ "Phase Transition" from Phase II to Phase III.

Notice that the dynamics during phase transition is highly nonlinear with $|\mathcal{K}_+| = \Theta(m)$ changes on activation partitions. Fortunately, we can keep the neurons of $\mathcal{K}_+$ and $\mathcal{K}_-$ close enough respectively in Phase I by using sufficiently small initialization $\kappa_1$. Moreover, their differences do not enlarge in Phase II. As a result, the phase transition can be completed quickly without significant changes in the vector field.

In order to analyze the dynamics of neurons and vector fields, we introduce the auxiliary hitting time:

$$
T_{\mathrm{II}}^{\mathrm{PT}*} := T_{\mathrm{II}}^{\mathrm{PT}} \wedge \inf\Big\{t > T_{\mathrm{II}} : \langle \boldsymbol{F}_+(t), \boldsymbol{x}_+ \rangle \le 0 \text{ or } \langle \boldsymbol{F}_+(t), \boldsymbol{x}_- \rangle \ge 0\Big\};
$$
$$
\text{where} \quad \boldsymbol{F}_+(t) = \frac{p}{1+p}e^{-f_+(t)}\boldsymbol{x}_+ - \frac{1}{1+p}e^{f_-(t)}\boldsymbol{x}_-.
\tag{27}
$$

We call $T_{\mathrm{II}} \le t \le T_{\mathrm{II}}^{\mathrm{PT}*}$ "Phase Transition*".

**Lemma E.1** (Dynamics of living neurons during Phase Transition*)**.**
*In Phase Transition*, i.e., $t \in [T_{\mathrm{II}}, T_{\mathrm{II}}^{\mathrm{PT}*}]$, we have the following dynamics for each neuron $k \in \mathcal{K}_- \cup \mathcal{K}_+$.*

*(S1). For living negative neuron $k \in \mathcal{K}_-$, we have:*

$$
\boldsymbol{w}_k(t) \in \mathcal{M}_+^0 \cap \mathcal{M}_-^+,
$$
$$
\frac{\mathrm{d}\boldsymbol{b}_k(t)}{\mathrm{d}t} = \frac{\kappa_2 e^{f_-(t)}}{\sqrt{m}(1+p)}\Big(\boldsymbol{x}_- - \boldsymbol{x}_+ \cos\Delta\Big).
$$

*(S2) For living positive neuron $k \in \mathcal{K}_+$, we define the hitting time:*

$$
T_{\mathrm{II},k}^{\mathrm{PT}*} := \inf\big\{t > T_{\mathrm{II}} : \langle \boldsymbol{w}_k(t), \boldsymbol{x}_- \rangle = 0\big\} \wedge \inf\big\{t > T_{\mathrm{II}} : \langle \boldsymbol{F}_+(t), \boldsymbol{x}_+ \rangle \le 0 \text{ or } \langle \boldsymbol{F}_+(t), \boldsymbol{x}_- \rangle \ge 0\big\}.
$$

*Then it holds that:*

*(P0) $T_{\mathrm{II}}^{\mathrm{PT}*} = \max\limits_{k \in \mathcal{K}_+} T_{\mathrm{II},k}^{\mathrm{PT}*}$;*

*(P1) For any $t \in [T_{\mathrm{II}}, T_{\mathrm{II},k}^{\mathrm{PT}*})$, we have*

$$
\frac{\mathrm{d}\boldsymbol{b}_k(t)}{\mathrm{d}t} = \frac{\kappa_2}{\sqrt{m}}\boldsymbol{F}_+(t);
$$

*(P2) If $T_{\mathrm{II},k}^{\mathrm{PT}*} < T_{\mathrm{II}}^{\mathrm{PT}*}$ strictly, then for any $t \in [T_{\mathrm{II},k}^{\mathrm{PT}*}, T_{\mathrm{II}}^{\mathrm{PT}*}]$, we have $\boldsymbol{w}_k(t) \in \mathcal{M}_+^+ \cap \mathcal{M}_-^0$ and*

$$
\frac{\mathrm{d}\boldsymbol{b}_k(t)}{\mathrm{d}t} = \frac{\kappa_2 p e^{-f_+(t)}}{\sqrt{m}(1+p)}\Big(\boldsymbol{x}_+ - \boldsymbol{x}_- \cos\Delta\Big).
$$

*(P3) Regardless of the relationship between $T_{\mathrm{II},k}^{\mathrm{PT}*}$ and $T_{\mathrm{II}}^{\mathrm{PT}*}$, for any $t \in [T_{\mathrm{II}}, T_{\mathrm{II}}^{\mathrm{PT}*}]$, we have*

$$
\langle \boldsymbol{w}_k(t), \boldsymbol{x}_+ \rangle > 0, \quad \langle \boldsymbol{w}_k(t), \boldsymbol{x}_- \rangle \ge 0.
$$

*Proof of Lemma E.1.*
Proof of (S1). Recalling the definition of $T_{\mathrm{II}}^{\mathrm{PT}*}$, it holds $\langle \boldsymbol{F}_+(t), \boldsymbol{x}_+ \rangle > 0$ for any $T_{\mathrm{II}} \le t \le T_{\mathrm{II}}^{\mathrm{PT}*}$.
So (S1) can be proved in the same way as employed in the proof of Lemma D.1 (S2) and is omitted.

Proof of (S2)(P0) and (S2)(P1). (S2)(P0) is obvious. Moreover, for any $k \in \mathcal{K}_+$ and $t \in [T_{\mathrm{II}}, T_{\mathrm{II},k}^{\mathrm{PT}*})$, we have $\langle \boldsymbol{w}_k(t), \boldsymbol{x}_+ \rangle > 0$ and $\langle \boldsymbol{w}_k(t), \boldsymbol{x}_- \rangle > 0$ for any $T_{\mathrm{I}} \leq t \leq T_{\mathrm{II}}^*$, so (S2)(P1) can be proved in the same method as shown in the proof of Lemma D.1 (S1), and we have the dynamics:

$$\frac{\mathrm{d}\boldsymbol{b}_k(t)}{\mathrm{d}t} = \frac{\kappa_2}{\sqrt{m}} \boldsymbol{F}_+(t), \quad t \in [T_{\mathrm{II}}, T_{\mathrm{II},k}^{\mathrm{PT}*}).$$

Additionally, recalling the definition of $T_{\mathrm{II},k}^{\mathrm{PT}*}$, we know $\langle \boldsymbol{F}_+(t), \boldsymbol{x}_+ \rangle > 0$ holds for any $t \in [T_{\mathrm{II}}, T_{\mathrm{II},k}^{\mathrm{PT}*})$. Combining the dynamics of $\boldsymbol{b}_k(t)$, we further have:

$$\langle \boldsymbol{b}_k(t), \boldsymbol{x}_+ \rangle = \langle \boldsymbol{b}_k(T_{\mathrm{II}}), \boldsymbol{x}_+ \rangle + \frac{\kappa_2}{\sqrt{m}} \int_{T_{\mathrm{II}}}^t \langle \boldsymbol{F}_+(s), \boldsymbol{x}_+ \rangle \, \mathrm{d}s$$

$$> \langle \boldsymbol{b}_k(T_{\mathrm{II}}), \boldsymbol{x}_+ \rangle > 0, \quad \forall t \in [T_{\mathrm{II}}, T_{\mathrm{II},k}^{\mathrm{PT}*}].$$

Proof of (S2)(P2). Let $k \in \mathcal{K}_+$. If $T_{\mathrm{II},k}^{\mathrm{PT}*} < T_{\mathrm{II}}^{\mathrm{PT}*}$, we have the following results:

Step I. $\boldsymbol{w}_k(T_{\mathrm{II},k}^{\mathrm{PT}*}) \in \mathcal{M}_+^+ \cap \mathcal{M}_-^0$.

Recalling the definition of $T_{\mathrm{II},k}^{\mathrm{PT}*}$ and $T_{\mathrm{II}}^{\mathrm{PT}*}$, $T_{\mathrm{II},k}^{\mathrm{PT}*} < T_{\mathrm{II}}^{\mathrm{PT}*}$ implies that $\left\langle \boldsymbol{w}_k(T_{\mathrm{II},k}^{\mathrm{PT}*}), \boldsymbol{x}_- \right\rangle = 0$.

Then recalling our proof of (S2)(P1), we obtain $\left\langle \boldsymbol{b}_k(T_{\mathrm{II},k}^{\mathrm{PT}*}), \boldsymbol{x}_+ \right\rangle > 0$.

Hence, we obtain $\boldsymbol{w}(T_{\mathrm{II},k}^{\mathrm{PT}*}) \in \mathcal{M}_+^+ \cap \mathcal{M}_-^0$.

Step II. Dynamics after $t = T_{\mathrm{II},k}^{\mathrm{PT}*}$.

In this step, we will analyze the training dynamics after $\boldsymbol{w}_k(T_{\mathrm{II},k}^{\mathrm{PT}*}) \in \mathcal{M}_+^+ \cap \mathcal{M}_-^0$, i.e. $\boldsymbol{b}_k(T_{\mathrm{II},k}^{\mathrm{PT}*}) \in \mathcal{P}_+^+ \cap \mathcal{P}_-^0$. We first analysis the vector field around the manifold $\mathcal{P}_+^+ \cap \mathcal{P}_-^0$. For any $\tilde{\boldsymbol{b}} \in \mathcal{P}_+^+ \cap \mathcal{P}_-^0$ and $0 < \delta_0 \ll 1$, we know that $\mathcal{P}_+^0 \cap \mathcal{P}_-^+$ separates its neighborhood $\mathcal{B}(\tilde{\boldsymbol{b}}, \delta_0)$ into two domains $\mathcal{G}_- = \{\boldsymbol{b} \in \mathcal{B}(\tilde{\boldsymbol{b}}, \delta_0) : \langle \boldsymbol{b}, \boldsymbol{x}_- \rangle < 0\}$ and $\mathcal{G}_+ = \{\boldsymbol{b} \in \mathcal{B}(\tilde{\boldsymbol{b}}, \delta_0) : \langle \boldsymbol{b}, \boldsymbol{x}_- \rangle > 0\}$. Following Definition H.1, we calculate the limited vector field on $\tilde{\boldsymbol{b}}$ from $\mathcal{G}_-$ and $\mathcal{G}_+$.

(i) The limited vector field $\boldsymbol{F}^-$ on $\tilde{\boldsymbol{b}}$ (from $\mathcal{G}_-$):

$$\frac{\mathrm{d}\boldsymbol{b}}{\mathrm{d}t} = \boldsymbol{F}^-, \text{ where } \boldsymbol{F}^- = \frac{\kappa_2}{\sqrt{m}} \frac{p}{1+p} e^{-f_+(t)} \boldsymbol{x}_+.$$

(ii) The limited vector field $\boldsymbol{F}^+$ on $\tilde{\boldsymbol{b}}$ (from $\mathcal{G}_+$):

$$\frac{\mathrm{d}\boldsymbol{b}}{\mathrm{d}t} = \boldsymbol{F}^+, \text{ where } \boldsymbol{F}^+ = \frac{\kappa_2}{\sqrt{m}} \left( \frac{pe^{-f_+(t)}}{1+p} \boldsymbol{x}_+ - \frac{e^{f_-(t)}}{1+p} \boldsymbol{x}_- \right).$$

(iii) Then we calculate the projections of $\boldsymbol{F}^-$ and $\boldsymbol{F}^+$ onto $\boldsymbol{x}_-$ (the normal to the surface $\mathcal{P}_+^+ \cap \mathcal{P}_-^0$):

$$F_N^- = \left\langle \boldsymbol{F}^-, \boldsymbol{x}_- \right\rangle = \frac{\kappa_2 p e^{-f_+(t)}}{\sqrt{m}(1+p)} \cos \Delta,$$

$$F_N^+ = \left\langle \boldsymbol{F}^+, \boldsymbol{x}_- \right\rangle = \frac{\kappa_2 e^{f_-(t)}}{\sqrt{m}(1+p)} \cos \Delta - \frac{\kappa_2 p e^{-f_+(t)}}{\sqrt{m}(1+p)}.$$

We further define the hitting time to check whether $\boldsymbol{w}_k(t) \in \mathcal{M}_+^+ \cap \mathcal{M}_-^0$ for $T_{\mathrm{II},k}^{\mathrm{PT}*} \leq t \leq T_{\mathrm{II}}^{\mathrm{PT}*}$.

$$\tau_{+,k}^+ := \inf \left\{ t \in [T_{\mathrm{II},k}^{\mathrm{PT}*}, T_{\mathrm{II}}^{\mathrm{PT}*}] : \langle \boldsymbol{w}_k(t), \boldsymbol{x}_+ \rangle \leq 0 \right\}.$$

From the definition of $T_{\mathrm{II}}^{\mathrm{PT}*}$, we know that $\langle \boldsymbol{F}_+(t), \boldsymbol{x}_- \rangle = \frac{p}{1+p} e^{-f_+(t)} \cos \Delta - \frac{1}{1+p} e^{f_-(t)} < 0$ for any $t \in [T_{\mathrm{II}}, T_{\mathrm{II}}^{\mathrm{PT}*}]$, which means $F_N^+ < 0$. And it is clear that $F_N^- > 0$. Hence, the dynamics

corresponds to Case (I) in Definition H.1 ($F_N^- > 0$ and $F_N^+ < 0$), which means that $\boldsymbol{b}_k(t)$ can not leave $\mathcal{P}_-^0$ for any $t \in [T_{\mathrm{II},k}^{\mathrm{PT}*}, \tau_{+,k}^+]$, and the dynamics of $\boldsymbol{b}_k$ for $t \in [T_{\mathrm{II},k}^{\mathrm{PT}*}, \tau_{+,k}^+]$ satisfies:

$$\frac{\mathrm{d}\boldsymbol{b}}{\mathrm{d}t} = \alpha \boldsymbol{F}^+ + (1-\alpha)\boldsymbol{F}^-, \quad \alpha = \frac{f_N^-}{f_N^- - f_N^+},$$

which is

$$\frac{\mathrm{d}\boldsymbol{b}_k(t)}{\mathrm{d}t} = \frac{\kappa_2 p e^{-f_+(t)}}{\sqrt{m}(1+p)} \left( \boldsymbol{x}_+ - \boldsymbol{x}_- \cos\Delta \right), t \in [T_{\mathrm{II},k}^{\mathrm{PT}*}, \tau_{+,k}^+].$$

By Lemma C.1, we know that the dynamics of $\boldsymbol{w}_k(t)$ on $\mathcal{M}_+^+ \cap \mathcal{M}_-^0$ and the dynamics of $\rho_k(t)$ are:

$$\frac{\mathrm{d}\boldsymbol{w}_k(t)}{\mathrm{d}t} = \frac{\kappa_2 p e^{-f_+(t)}}{\rho_k(t)\sqrt{m}(1+p)} \left( \boldsymbol{x}_+ - \langle \boldsymbol{w}_k, \boldsymbol{x}_+ \rangle \boldsymbol{w}_k - \boldsymbol{x}_- \cos\Delta \right).$$

$$\frac{\mathrm{d}\rho_k(t)}{\mathrm{d}t} = \frac{\kappa_2 p e^{-f_+(t)}}{\sqrt{m}(1+p)} \langle \boldsymbol{w}_k(t), \boldsymbol{x}_+ \rangle.$$

Moreover, The dynamics above also ensures that:

$$\langle \boldsymbol{b}_k(t), \boldsymbol{x}_+ \rangle = \langle \boldsymbol{b}_k(T_{\mathrm{II},k}^{\mathrm{PT}*}), \boldsymbol{x}_+ \rangle + \int_{T_{\mathrm{II},k}^{\mathrm{PT}*}}^t \frac{\kappa_2 p e^{-f_+(s)}}{\sqrt{m}(1+p)} \sin^2\Delta \, \mathrm{d}s$$

$$> \langle \boldsymbol{b}_k(T_{\mathrm{II},k}^{\mathrm{PT}*}), \boldsymbol{x}_+ \rangle > 0, \quad \forall t \in [T_{\mathrm{II},k}^{\mathrm{PT}*}, \tau_{+,k}^+].$$

which means $\tau_{+,k}^+ = T_{\mathrm{II}}^{\mathrm{PT}*}$. Hence, we have proved (S2)(P2).

Proof of (S2)(P3). Our proof for (S2)(P1) and (S2)(P2) imply this result directly.

$\square$

**Lemma E.2** (Evolution of the prediction in Phase III*).
*For any $t \in [T_{\mathrm{II}}, T_{\mathrm{II}}^{\mathrm{PT}*}]$, we have*

$$\frac{e^{-C_1}}{1 + C_0 e^{f_-(T_{\mathrm{II}})}(t - T_{\mathrm{II}})} \leq e^{f_-(t) - f_-(T_{\mathrm{II}})} \leq \frac{1}{1 + C_0 e^{(f_-(T_{\mathrm{II}}) - C_1)}(t - T_{\mathrm{II}})},$$

$$\exp\left( -C_2(t - T_{\mathrm{II}}) \right) \leq e^{f_+(T_{\mathrm{II}}) - f_+(t)} \leq 1,$$

*where*

$$C_0 = \Theta\left( \frac{\kappa_2^2 \Delta^2}{p} \right), \quad C_1 = \mathcal{O}\left( \kappa_2 \sqrt{\kappa_1 \kappa_2} \right), \quad e^{f_-(T_{\mathrm{II}})} = \Theta\left( p^{-\frac{\alpha \cos\Delta}{1 - \alpha \cos\Delta}} \right), \quad C_2 = \Theta\left( \kappa_2^2 p^{-\frac{1}{1 - \alpha \cos\Delta}} \right).$$

*Proof of Lemma E.2.*
Step I. Preparation. With the help of Lemma E.1(S1) and (S2)(P3), we know that

(i) For $k \in \mathcal{K}_-$, we have

$$\langle \boldsymbol{w}_k(t), \boldsymbol{x}_+ \rangle = 0, \quad \langle \boldsymbol{w}_k(t), \boldsymbol{x}_- \rangle > 0, \quad \forall t \in [T_{\mathrm{II}}, T_{\mathrm{II}}^{\mathrm{PT}*}].$$

(ii) For $k \in \mathcal{K}_+$, we have

$$\langle \boldsymbol{w}_k(t), \boldsymbol{x}_+ \rangle > 0, \quad \langle \boldsymbol{w}_k(t), \boldsymbol{x}_- \rangle \geq 0, \quad \forall t \in [T_{\mathrm{II}}, T_{\mathrm{II}}^{\mathrm{PT}*}];$$

So $f_+(t)$ and $f_-(t)$ have the following representation for any $t \in [T_{\mathrm{II}}, T_{\mathrm{II}}^{\mathrm{PT}*}]$:

$$f_+(t) = \sum_{k \in \mathcal{K}_+} \frac{\kappa_2}{\sqrt{m}} \boldsymbol{b}_k^\top(t) \boldsymbol{x}_+,$$

$$f_-(t) = \sum_{k \in \mathcal{K}_+} \frac{\kappa_2}{\sqrt{m}} \boldsymbol{b}_k^\top(t) \boldsymbol{x}_- - \sum_{k \in \mathcal{K}_-} \frac{\kappa_2}{\sqrt{m}} \boldsymbol{b}_k^\top(t) \boldsymbol{x}_-.$$

Step II. Evolution of $f_-(t)$.

To begin with, we need to do a rough estimate of $\sum_{k\in\mathcal{K}_+}\frac{\kappa_2}{\sqrt{m}}\boldsymbol{b}_k^\top(t)\boldsymbol{x}_-$. Let $k\in\mathcal{K}_+$. For any $t\in[T_{\mathrm{II},k}^{\mathrm{PT}*},T_{\mathrm{II}}^{\mathrm{PT}*}]$, we have $\langle\boldsymbol{b}_k(t),\boldsymbol{x}_-\rangle=0$. And for any $t\in[T_{\mathrm{II}},T_{\mathrm{II},k}^{\mathrm{PT}*})$, we have:

$$\frac{\mathrm{d}}{\mathrm{d}t}\langle\boldsymbol{b}_k(t),\boldsymbol{x}_-\rangle \overset{\text{Lemma E.1}}{=} \frac{\kappa_2}{\sqrt{m}}\langle\boldsymbol{F}_k(t),\boldsymbol{x}_-\rangle \overset{(27)}{<} 0.$$

Therefore, for any $t\in[T_{\mathrm{II}},T_{\mathrm{II}}^{\mathrm{PT}*}]$, we have $0\le\langle\boldsymbol{b}_k(t),\boldsymbol{x}_-\rangle\le\langle\boldsymbol{b}_k(T_{\mathrm{II}}),\boldsymbol{x}_-\rangle$, so

$$0\le\sum_{k\in\mathcal{K}_+}\frac{\kappa_2}{\sqrt{m}}\boldsymbol{b}_k^\top(t)\boldsymbol{x}_-\le\sum_{k\in\mathcal{K}_+}\frac{\kappa_2}{\sqrt{m}}\boldsymbol{b}_k^\top(T_{\mathrm{II}})\boldsymbol{x}_-,$$

$$-\sum_{k\in\mathcal{K}_-}\frac{\kappa_2}{\sqrt{m}}\boldsymbol{b}_k^\top(t)\boldsymbol{x}_-\le f_-(t)\le\sum_{k\in\mathcal{K}_+}\frac{\kappa_2}{\sqrt{m}}\boldsymbol{b}_k^\top(T_{\mathrm{II}})\boldsymbol{x}_--\sum_{k\in\mathcal{K}_-}\frac{\kappa_2}{\sqrt{m}}\boldsymbol{b}_k^\top(t)\boldsymbol{x}_-.$$

According to Lemma E.1, it follows that for any $k\in\mathcal{K}_-$, its dynamics is $\frac{\mathrm{d}\boldsymbol{b}_k(t)}{\mathrm{d}t}=\frac{\kappa_2 e^{f_-(t)}}{\sqrt{m}(1+p)}\left(\boldsymbol{x}_--\boldsymbol{x}_+\cos\Delta\right)$, thus

$$\boldsymbol{b}_k^\top(t)\boldsymbol{x}_-=\boldsymbol{b}_k^\top(T_{\mathrm{II}})\boldsymbol{x}_-+\int_{T_{\mathrm{II}}}^t\left\langle\frac{\mathrm{d}\boldsymbol{b}_k(s)}{\mathrm{d}s},\boldsymbol{x}_-\right\rangle\mathrm{d}s=\boldsymbol{b}_k^\top(T_{\mathrm{II}})\boldsymbol{x}_-+\frac{\kappa_2\sin^2\Delta}{\sqrt{m}(1+p)}\int_{T_{\mathrm{II}}}^t e^{f_-(s)}\mathrm{d}s,$$

$$\sum_{k\in\mathcal{K}_-}\frac{\kappa_2}{\sqrt{m}}\boldsymbol{b}_k^\top(t)\boldsymbol{x}_-=\sum_{k\in\mathcal{K}_-}\frac{\kappa_2}{\sqrt{m}}\boldsymbol{b}_k^\top(T_{\mathrm{II}})\boldsymbol{x}_-+\frac{m_-\kappa_2^2\sin^2\Delta}{m(1+p)}\int_{T_{\mathrm{II}}}^t e^{f_-(s)}\mathrm{d}s.$$

Therefore, we have two-side bounds of $f_-(t)$:

$$f_-(t)\le-\frac{m_-\kappa_2^2\sin^2\Delta}{m(1+p)}\int_{T_{\mathrm{II}}}^t e^{f_-(s)}\mathrm{d}s-\sum_{k\in\mathcal{K}_-}\frac{\kappa_2}{\sqrt{m}}\boldsymbol{b}_k^\top(T_{\mathrm{II}})\boldsymbol{x}_-+\sum_{k\in\mathcal{K}_+}\frac{\kappa_2}{\sqrt{m}}\boldsymbol{b}_k^\top(T_{\mathrm{II}})\boldsymbol{x}_-,$$

$$f_-(t)\ge-\frac{m_-\kappa_2^2\sin^2\Delta}{m(1+p)}\int_{T_{\mathrm{II}}}^t e^{f_-(s)}\mathrm{d}s-\sum_{k\in\mathcal{K}_-}\frac{\kappa_2}{\sqrt{m}}\boldsymbol{b}_k^\top(T_{\mathrm{II}})\boldsymbol{x}_-.$$

For simplicity, we denote $C_0:=\frac{m_-\kappa_2^2\sin^2\Delta}{m(1+p)}$, $C_-^-:=\sum_{k\in\mathcal{K}_-}\frac{\kappa_2}{\sqrt{m}}\boldsymbol{b}_k^\top(T_{\mathrm{II}})\boldsymbol{x}_-$ and $C_-^+:=\sum_{k\in\mathcal{K}_+}\frac{\kappa_2}{\sqrt{m}}\boldsymbol{b}_k^\top(T_{\mathrm{II}})\boldsymbol{x}_-$. Then we have:

$$-C_0\int_{T_{\mathrm{II}}}^t e^{f_-(s)}\mathrm{d}s-C_-^-\le f_-(t)\le-C_0\int_{T_{\mathrm{II}}}^t e^{f_-(s)}\mathrm{d}s-C_-^-+C_-^+.$$

Let $\Psi(t):=\int_{T_{\mathrm{II}}}^t e^{f_-(s)}\mathrm{d}s$, then $\frac{\mathrm{d}\Psi(t)}{\mathrm{d}t}=e^{f_-(t)}$. So $\Psi(T_{\mathrm{II}})=0$ and

$$-C_0\Psi(t)-C_-^-\le\log\left(\frac{\mathrm{d}\Psi(t)}{\mathrm{d}t}\right)\le-C_0\Psi(t)-C_-^-+C_-^+,$$

$$e^{-C_-^-}e^{-C_0\Psi(t)}\le\frac{\mathrm{d}\Psi(t)}{\mathrm{d}t}\le e^{-C_-^-+C_-^+}e^{-C_0\Psi(t)}.$$

For the right hand, for any $\epsilon\in(0,1)$, we consider the auxiliary ODE:

$$\begin{cases}\frac{\mathrm{d}\mathcal{P}(t)}{\mathrm{d}t}=e^{-C_-^-+(1+\epsilon)C_-^+}e^{-C_0\mathcal{P}(t)},\\\mathcal{P}(T_{\mathrm{II}})=0.\end{cases}$$

The solution of this ODE is $\mathcal{P}(t)=\frac{1}{C_0}\log\left(1+C_0 e^{-C_-^-+(1+\epsilon)C_-^+}(t-T_{\mathrm{II}})\right)$. From the Comparison Principle of ODEs, we have the upper bound for $\Psi(t)$:

$$\Psi(t)\le\mathcal{P}(t)=\frac{1}{C_0}\log\left(1+C_0 e^{-C_-^-+(1+\epsilon)C_-^+}(t-T_{\mathrm{II}})\right).$$

Taking $\epsilon\to 0$, we obtain

$$\Psi(t)\le\frac{1}{C_0}\log\left(1+C_0 e^{-C_-^-+C_-^+}(t-T_{\mathrm{II}})\right).$$

In the similar way, we can derive the lower bound for $\Psi(t)$:

$$\Psi(t) \geq \frac{1}{C_0} \log \left(1 + C_0 e^{-C_-^-}(t - T_{\mathrm{II}})\right).$$

Consequently, we infer that

$$f_-(t) \leq -C_0 \Psi(t) - C_-^- + C_-^+ \leq -\log\left(1 + C_0 e^{-C_-^-}(t - T_{\mathrm{II}})\right) - C_-^- + C_-^+,$$

$$f_-(t) \geq -C_0 \Psi(t) - C_-^- \geq -\log\left(1 + C_0 e^{-C_-^- + C_-^+}(t - T_{\mathrm{II}})\right) - C_-^-.$$

Noticing $f_-(T_{\mathrm{II}}) = C_-^+ - C_-^-$, we obtain

$$-\log\left(1 + C_0 e^{-C_-^- + C_-^+}(t - T_{\mathrm{II}})\right) - C_-^+ \leq f_-(t) - f_-(T_{\mathrm{II}}) \leq -\log\left(1 + C_0 e^{-C_-^-}(t - T_{\mathrm{II}})\right).$$

Noticing $f_-(T_{\mathrm{II}}) = C_-^+ - C_-^-$, this inequality means

$$\frac{e^{-C_-^+}}{1 + C_0 e^{f_-(T_{\mathrm{II}})}(t - T_{\mathrm{II}})} \leq e^{f_-(t) - f_-(T_{\mathrm{II}})} \leq \frac{1}{1 + C_0 e^{(f_-(T_{\mathrm{II}}) - C_-^+)}(t - T_{\mathrm{II}})}.$$

where $C_0 = \frac{m_- \kappa_2^2 \sin^2 \Delta}{m(1+p)} = \Theta\left(\frac{\kappa_2^2 \Delta^2}{p}\right)$. Moreover, according to Lemma D.8 (S1)(S2), we have

$$e^{f_-(T_{\mathrm{II}})} = \Theta\left(p^{-\frac{\alpha \cos \Delta}{1 - \alpha \cos \Delta}}\right),$$

$$C_-^+ = \sum_{k \in \mathcal{K}_+} \frac{\kappa_2}{\sqrt{m}} \boldsymbol{b}_k^\top(T_{\mathrm{II}}) \boldsymbol{x}_- = \mathcal{O}\left(m_- \frac{\kappa_2}{\sqrt{m}} \frac{\sqrt{\kappa_1 \kappa_2}}{\sqrt{m}}\right) = \mathcal{O}\left(\kappa_2 \sqrt{\kappa_1 \kappa_2}\right).$$

Step III. Evolution of $f_+(t)$.

Let $k \in \mathcal{K}_+$. According to Lemma E.1 (S2)(P2) and (S2)(P3), it follows that for any $k \in \mathcal{K}_+$, its dynamics during $t \in [T_{\mathrm{II}}, T_{\mathrm{II}}^{\mathrm{PT}*}]$ is

$$\frac{\mathrm{d}\boldsymbol{b}_k(t)}{\mathrm{d}t} = \frac{\kappa_2}{\sqrt{m}} \langle \boldsymbol{F}_+(t), \boldsymbol{x}_+ \rangle;$$

$$\text{or } \frac{\mathrm{d}\boldsymbol{b}_k(t)}{\mathrm{d}t} = \frac{\kappa_2 p e^{-f_+(t)}}{\sqrt{m}(1+p)}\left(\boldsymbol{x}_+ - \boldsymbol{x}_- \cos \Delta\right).$$

Notice that

$$f_+(t) = \sum_{k \in \mathcal{K}_+} \frac{\kappa_2}{\sqrt{m}} \boldsymbol{b}_k^\top(t) \boldsymbol{x}_+ = \sum_{k \in \mathcal{K}_+} \frac{\kappa_2}{\sqrt{m}} \boldsymbol{b}_k^\top(T_{\mathrm{II}}) \boldsymbol{x}_+ + \sum_{k \in \mathcal{K}_+} \int_{T_{\mathrm{II}}}^t \frac{\kappa_2}{\sqrt{m}} \left\langle \frac{\mathrm{d}\boldsymbol{b}_k(t)}{\mathrm{d}s}, \boldsymbol{x}_+ \right\rangle \mathrm{d}s$$

$$= f_+(T_{\mathrm{II}}) + \frac{\kappa_2}{\sqrt{m}} \sum_{k \in \mathcal{K}_+} \int_{T_{\mathrm{II}}}^t \left\langle \frac{\mathrm{d}\boldsymbol{b}_k(t)}{\mathrm{d}s}, \boldsymbol{x}_+ \right\rangle \mathrm{d}s.$$

On the one hand, for any $t \in [T_{\mathrm{II}}, T_{\mathrm{II}}^{\mathrm{PT}*}]$, we have the lower bound:

$$f_+(t) \geq f_+(T_{\mathrm{II}}) + \frac{\kappa_2^2}{m} \sum_{k \in \mathcal{K}_+} \int_{T_{\mathrm{II}}}^t \min\left\{\langle \boldsymbol{F}_+(s), \boldsymbol{x}_+ \rangle, \frac{p e^{-f_+(s)}}{1+p} \sin^2 \Delta\right\} \mathrm{d}s \geq f_+(T_{\mathrm{II}}).$$

On the other hand, for any $t \in [T_{\mathrm{II}}, T_{\mathrm{II}}^{\mathrm{PT}*}]$, we can derive an upper bound:

$$f_+(t) \leq f_+(T_{\mathrm{II}}) + \frac{\kappa_2^2}{m} \sum_{k \in \mathcal{K}_+} \int_{T_{\mathrm{II}}}^t \max\left\{\langle \boldsymbol{F}_+(s), \boldsymbol{x}_+ \rangle, \frac{p e^{-f_+(s)}}{1+p} \sin^2 \Delta\right\} \mathrm{d}s$$

$$\leq f_+(T_{\mathrm{II}}) + \frac{\kappa_2^2}{m} \sum_{k \in \mathcal{K}_+} \int_{T_{\mathrm{II}}}^t \max\left\{\frac{p e^{-f_+(s)}}{1+p}, \frac{p e^{-f_+(s)}}{1+p} \sin^2 \Delta\right\} \mathrm{d}s$$

$$\leq f_+(T_{\mathrm{II}}) + \frac{\kappa_2^2}{m} \sum_{k \in \mathcal{K}_+} \int_{T_{\mathrm{II}}}^t \frac{p e^{-f_+(s)}}{1+p} \mathrm{d}s \leq f_+(T_{\mathrm{II}}) + \frac{\kappa_2^2 m_+}{m} \frac{p e^{-f_+(T_{\mathrm{II}})}}{1+p}(t - T_{\mathrm{II}}).$$

Hence, we obtain

$$\exp\left(-\frac{\kappa_2^2 m_+}{m}\frac{pe^{-f_+(T_{\mathrm{II}})}}{1+p}(t-T_{\mathrm{II}})\right) \le e^{f_+(T_{\mathrm{II}})-f_+(t)} \le 1,$$

where

$$\frac{\kappa_2^2 m_+}{m}\frac{pe^{-f_+(T_{\mathrm{II}})}}{1+p} \overset{\text{Lemma D.8}}{=} \Theta\left(\kappa_2^2 e^{-f_+(T_{\mathrm{II}})}\right) = \Theta\left(\kappa_2^2 p^{-\frac{1}{1-\alpha\cos\Delta}}\right).$$

$\square$

**Lemma E.3** (Nearly fixed vector filed in Phase III*)**.**
*There exist absolute constants $Q_1, Q_2 > 0$, such that: For any time $T_{\mathrm{fix}} \in [T_{\mathrm{II}}, +\infty)$, if we choose $\kappa_1, \kappa_2$ s.t.*

$$\kappa_2^2\left(T_{\mathrm{fix}} \wedge T_{\mathrm{II}}^{\mathrm{PT}*} - T_{\mathrm{II}}\right)p^{-\frac{1}{1-\alpha\cos\Delta}} = \mathcal{O}(\Delta^2), \quad \kappa_2^2\sqrt{\frac{\kappa_1}{\kappa_2}} = \mathcal{O}(\Delta^2),$$

*then for any $t \in [T_{\mathrm{II}}, T_{\mathrm{fix}} \wedge T_{\mathrm{II}}^{\mathrm{PT}*}]$, we have*

$$\langle \boldsymbol{F}_+(t), \boldsymbol{x}_+ \rangle \le \frac{Q_1}{2}\Delta^2 p^{-\frac{1}{1-\alpha\cos\Delta}}, \quad \langle \boldsymbol{F}_+(t), \boldsymbol{x}_- \rangle \ge -\frac{Q_2}{2}\Delta^2 p^{-\frac{1}{1-\alpha\cos\Delta}}.$$

*Proof of Lemma E.3.*
For simplicity, we denote $\delta_T := T_{\mathrm{fix}} \wedge T_{\mathrm{II}}^{\mathrm{PT}*} - T_{\mathrm{II}}$ From Lemma E.2, for any $t \in [T_{\mathrm{II}}, T_{\mathrm{fix}} \wedge T_{\mathrm{II}}^{\mathrm{PT}*}]$, we have

$$e^{f_-(t)-f_-(T_{\mathrm{II}})} - 1 \le \frac{1}{1+C_0 e^{(f_-(T_{\mathrm{II}})-C_1)\delta_T}} - 1 \le 0,$$

$$e^{f_-(t)-f_-(T_{\mathrm{II}})} - 1 \ge \frac{e^{-C_1}}{1+C_0 e^{f_-(T_{\mathrm{II}})}\delta_T} - 1 = \frac{e^{-C_1}-1-C_0 e^{f_-(T_{\mathrm{II}})}\delta_T}{1+C_0 e^{f_-(T_{\mathrm{II}})}\delta_T} \ge -\frac{C_1+C_0 e^{f_-(T_{\mathrm{II}})}\delta_T}{1+C_0 e^{f_-(T_{\mathrm{II}})}\delta_T}$$

$$e^{f_+(T_{\mathrm{II}})-f_+(t)} - 1 \le 0,$$

$$e^{f_+(T_{\mathrm{II}})-f_+(t)} - 1 \ge e^{-C_2\delta_T} - 1 \ge -C_2\delta_T.$$

Recalling Lemma D.8 (S1), there exists absolute constants $Q_1, Q_2 > 0$ such that

$$\langle \boldsymbol{F}_+(T_{\mathrm{II}}), \boldsymbol{x}_+ \rangle = \frac{pe^{-f_+(T_{\mathrm{II}})}}{1+p} - \frac{e^{f_-(T_{\mathrm{II}})}}{1+p}\cos\Delta \ge Q_1\Delta^2 p^{-\frac{1}{1-\alpha\cos\Delta}},$$

$$\langle \boldsymbol{F}_+(T_{\mathrm{II}}), \boldsymbol{x}_- \rangle = \frac{pe^{-f_+(T_{\mathrm{II}})}}{1+p}\cos\Delta - \frac{e^{f_-(T_{\mathrm{II}})}}{1+p} \le -Q_2\Delta^2 p^{-\frac{1}{1-\alpha\cos\Delta}}.$$

Step I. Bounding the term $\langle \boldsymbol{F}_+(t), \boldsymbol{x}_+ \rangle$.

$$\left|\langle \boldsymbol{F}_+(t), \boldsymbol{x}_+ \rangle - \langle \boldsymbol{F}_+(T_{\mathrm{II}}), \boldsymbol{x}_+ \rangle\right|$$

$$= \left|\frac{p}{1+p}e^{-f_+(t)} - \frac{p}{1+p}e^{-f_+(T_{\mathrm{II}})} - \frac{e^{f_-(t)}}{1+p}\cos\Delta + \frac{e^{f_-(T_{\mathrm{II}})}}{1+p}\cos\Delta\right|$$

$$\le \left|\frac{p}{1+p}e^{-f_+(t)} - \frac{p}{1+p}e^{-f_+(T_{\mathrm{II}})}\right| + \left|\frac{e^{f_-(t)}}{1+p}\cos\Delta - \frac{e^{f_-(T_{\mathrm{II}})}}{1+p}\cos\Delta\right|$$

$$\le \frac{p}{1+p}e^{-f_+(T_{\mathrm{II}})}\left|e^{f_+(T_{\mathrm{II}})-f_+(t)} - 1\right| + \frac{e^{f_-(T_{\mathrm{II}})}}{1+p}\left|e^{f_-(t)-f_-(T_{\mathrm{II}})} - 1\right|$$

$$\le \frac{p}{1+p}e^{-f_+(T_{\mathrm{II}})}C_2\delta_T + \frac{e^{f_-(T_{\mathrm{II}})}}{1+p}\frac{C_1+C_0 e^{f_-(T_{\mathrm{II}})}\delta_T}{1+C_0 e^{f_-(T_{\mathrm{II}})}\delta_T}$$

To ensure $\left|\langle \boldsymbol{F}_+(t), \boldsymbol{x}_+ \rangle - \langle \boldsymbol{F}_+(T_{\mathrm{II}}), \boldsymbol{x}_+ \rangle\right| \le \frac{1}{2}Q_1\Delta^2 p^{-\frac{1}{1-\alpha\cos\Delta}}$, we need only select parameters such that

$$\frac{p}{1+p}e^{-f_+(T_{\mathrm{II}})}C_2\delta_T + \frac{e^{f_-(T_{\mathrm{II}})}}{1+p}\frac{C_1+C_0 e^{f_-(T_{\mathrm{II}})}\delta_T}{1+C_0 e^{f_-(T_{\mathrm{II}})}\delta_T} \le \frac{1}{2}Q_1\Delta^2 p^{-\frac{1}{1-\alpha\cos\Delta}}.$$

From Lemma E.2 and Lemma D.8, we have:

$$C_0 = \Theta\left(\frac{\kappa_2^2 \Delta^2}{p}\right), \ C_1 = \mathcal{O}\left(\kappa_2\sqrt{\kappa_1\kappa_2}\right), \ C_2 = \Theta\left(\kappa_2^2 p^{-\frac{1}{1-\alpha\cos\Delta}}\right),$$

$$e^{-f_+(T_{\mathrm{II}})} = \Theta\left(p^{-\frac{1}{1-\alpha\cos\Delta}}\right), \quad e^{f_-(T_{\mathrm{II}})} = \Theta\left(p^{-\frac{\alpha\cos\Delta}{1-\alpha\cos\Delta}}\right).$$

Therefore, if we take

$$C_0 e^{f_-(T_{\mathrm{II}})}\delta_T = \Theta\left(\kappa_2^2\Delta^2 p^{-\frac{1}{1-\alpha\cos\Delta}}\delta_T\right) = \mathcal{O}(1),$$

then we have

$$\frac{p}{1+p}e^{-f_+(T_{\mathrm{II}})}C_2\delta_T + \frac{e^{f_-(T_{\mathrm{II}})}}{1+p}\frac{C_1 + C_0 e^{f_-(T_{\mathrm{II}})}\delta_T}{1 + C_0 e^{f_-(T_{\mathrm{II}})}\delta_T}$$

$$=\Theta\left(\kappa_2^2 p^{-\frac{2}{1-\alpha\cos\Delta}}\delta_T\right) + \Theta\left(p^{-\frac{1}{1-\alpha\cos\Delta}}\left(\mathcal{O}(\kappa_2\sqrt{\kappa_1\kappa_2}) + \kappa_2^2\Delta^2 p^{-\frac{1}{1-\alpha\cos\Delta}}\delta_T\right)\right)$$

$$=\Theta\left(\kappa_2^2 p^{-\frac{1}{1-\alpha\cos\Delta}}\left(\delta_T p^{-\frac{1}{1-\alpha\cos\Delta}} + \mathcal{O}(\sqrt{\frac{\kappa_1}{\kappa_2}})\right)\right).$$

If we can take

$$\kappa_2^2\delta_T p^{-\frac{1}{1-\alpha\cos\Delta}} = \mathcal{O}(\Delta^2), \quad \kappa_2^2\sqrt{\frac{\kappa_1}{\kappa_2}} = \mathcal{O}(\Delta^2),$$

then $\kappa_2^2\Delta^2 p^{-\frac{1}{1-\alpha\cos\Delta}}\delta_T = \mathcal{O}(1)$ and

$$|\langle \boldsymbol{F}_+(t), \boldsymbol{x}_+\rangle - \langle \boldsymbol{F}_+(T_{\mathrm{II}}), \boldsymbol{x}_+\rangle| = \mathcal{O}\left(\Delta^2 p^{-\frac{1}{1-\alpha\cos\Delta}}\right) \le \frac{1}{2}Q_1\Delta^2 p^{-\frac{1}{1-\alpha\cos\Delta}},$$

Hence,

$$\langle \boldsymbol{F}_+(t), \boldsymbol{x}_+\rangle \ge \langle \boldsymbol{F}_+(T_{\mathrm{II}}), \boldsymbol{x}_+\rangle - |\langle \boldsymbol{F}_+(t), \boldsymbol{x}_+\rangle - \langle \boldsymbol{F}_+(T_{\mathrm{II}}), \boldsymbol{x}_+\rangle|$$

$$\ge\frac{1}{2}Q_1\Delta^2 p^{-\frac{1}{1-\alpha\cos\Delta}} = \Omega\left(\Delta^2 p^{-\frac{1}{1-\alpha\cos\Delta}}\right), \quad \forall t \in [T_{\mathrm{II}}, T_{\mathrm{fix}}\wedge T_{\mathrm{II}}^{\mathrm{PT}*}].$$

Step II. Bounding the term $\langle \boldsymbol{F}_+(t), \boldsymbol{x}_-\rangle$.

The proof can be completed by the method analogous to that used in Step I, and we omit it. The result is

$$\langle \boldsymbol{F}_+(t), \boldsymbol{x}_-\rangle \ge \langle \boldsymbol{F}_+(T_{\mathrm{II}}), \boldsymbol{x}_-\rangle + |\langle \boldsymbol{F}_+(t), \boldsymbol{x}_+\rangle - \langle \boldsymbol{F}_+(T_{\mathrm{II}}), \boldsymbol{x}_-\rangle|$$

$$\ge -\frac{1}{2}Q_2\Delta^2 p^{-\frac{1}{1-\alpha\cos\Delta}} \equiv -\Omega\left(\Delta^2 p^{-\frac{1}{1-\alpha\cos\Delta}}\right), \quad \forall t \in [T_{\mathrm{II}}, T_{\mathrm{fix}}\wedge T_{\mathrm{II}}^{\mathrm{PT}*}].$$

$\square$

**Lemma E.4** (The end of Phase Transition)**.**
*If we choose $\kappa_1, \kappa_2$ s.t $\kappa_2 = \mathcal{O}(1)$ and $\sqrt{\frac{\kappa_1}{\kappa_2}} = \mathcal{O}\left(\Delta^4\right)$ (3), then it holds that*

*(S1) (Time).*

$$T_{\mathrm{II}}^{\mathrm{PT}} = T_{\mathrm{II}}^{\mathrm{PT}*} = T_{\mathrm{II}} + \mathcal{O}\left(\sqrt{\frac{\kappa_1}{\kappa_2}}\frac{p^{\frac{1}{1-\alpha\cos\Delta}}}{\Delta^2}\right) = \left(1 + \mathcal{O}\left(\sqrt{\kappa_1\kappa_2^3}\right)\right)T_{\mathrm{II}};$$

*(S2) (Prediction).*

$$e^{-f_+(T_{\mathrm{II}}^{\mathrm{PT}})} = \Theta\left(p^{-\frac{1}{1-\alpha\cos\Delta}}\right), \quad e^{f_-(T_{\mathrm{II}}^{\mathrm{PT}})} = \Theta\left(p^{-\frac{\alpha\cos\Delta}{1-\alpha\cos\Delta}}\right);$$

$$\frac{pe^{-f_+(T_{\mathrm{II}}^{\mathrm{PT}})}}{1+p} - \frac{e^{f_-(T_{\mathrm{II}}^{\mathrm{PT}})}}{1+p}\cos\Delta = \Theta\left(\Delta^2 p^{-\frac{1}{1-\alpha\cos\Delta}}\right),$$

$$\frac{pe^{-f_+(T_{\mathrm{II}}^{\mathrm{PT}})}}{1+p}\cos\Delta - \frac{e^{f_-(T_{\mathrm{II}}^{\mathrm{PT}})}}{1+p} = -\Theta\left(\Delta^2 p^{-\frac{1}{1-\alpha\cos\Delta}}\right).$$

*(S3) (Activation patterns).*

$$\langle \boldsymbol{w}_k(T_{\mathrm{II}}^{\mathrm{PT}}), \boldsymbol{x}_+\rangle > 0, \ \langle \boldsymbol{w}_k(T_{\mathrm{II}}^{\mathrm{PT}}), \boldsymbol{x}_-\rangle = 0, \ \forall k \in \mathcal{K}_+;$$

$$\langle \boldsymbol{w}_k(T_{\mathrm{II}}^{\mathrm{PT}}), \boldsymbol{x}_+\rangle = 0, \ \langle \boldsymbol{w}_k(T_{\mathrm{II}}^{\mathrm{PT}}), \boldsymbol{x}_-\rangle > 0, \ \forall k \in \mathcal{K}_-.$$

*Proof of Lemma E.4.*
Step I. Time Estimate. Let $k \in \mathcal{K}_+$.

Recalling the definition of $T_{\mathrm{II},k}^{\mathrm{PT}*}$ in Lemma E.1, Lemma E.1 (S2)(P0) also gives us

$$T_{\mathrm{II}}^{\mathrm{PT}*} = \max_{k \in \mathcal{K}_+} T_{\mathrm{II},k}^{\mathrm{PT}*}.$$

From Lemma D.8 (S2), we know that there exists an absolute constant $Q_3 > 0$, s.t. $0 \le \langle \boldsymbol{b}_k(T_{\mathrm{II}}), \boldsymbol{x}_- \rangle \le Q_3 \frac{\sqrt{\kappa_1 \kappa_2}}{\sqrt{m}}$. And we let $Q_2 > 0$ be the absolute constant $Q_2$ in Lemma E.3.

First, we choose the time

$$T_{\mathrm{fix}} = T_{\mathrm{II}} + \frac{3Q_3}{Q_2 \Delta^2} \sqrt{\frac{\kappa_1}{\kappa_2}} p^{\frac{1}{1-\alpha \cos \Delta}}.$$

then we choose $\kappa_1, \kappa_2$ s.t.

$$\kappa_2 = \mathcal{O}(1), \quad \sqrt{\frac{\kappa_1}{\kappa_2}} = \mathcal{O}\left(\Delta^4\right).$$

It can ensure

$$\kappa_2^2 \left(T_{\mathrm{fix}} \wedge T_{\mathrm{II}}^{\mathrm{PT}*} - T_{\mathrm{II}}\right) p^{-\frac{1}{1-\alpha \cos \Delta}} = \Theta\left(\frac{\kappa_2^2}{\Delta^2} \sqrt{\frac{\kappa_1}{\kappa_2}}\right) = \mathcal{O}(\Delta^2), \quad \kappa_2^2 \sqrt{\frac{\kappa_1}{\kappa_2}} = \mathcal{O}(\Delta^2).$$

Then according to Lemma E.3, it follows that

$$\langle \boldsymbol{F}_+(t), \boldsymbol{x}_- \rangle \le -\frac{Q_2}{2} \Delta^2 p^{-\frac{1}{1-\alpha \cos \Delta}}, \quad \forall t \in [T_{\mathrm{II}}, T_{\mathrm{II},k}^{\mathrm{PT}*} \wedge T_{\mathrm{fix}}).$$

Now we consider the dynamics for $t \in [T_{\mathrm{II}}, T_{\mathrm{II},k}^{\mathrm{PT}*} \wedge T_{\mathrm{fix}})$.

Recalling lemma E.1, we have

$$\left\langle \boldsymbol{b}_k(T_{\mathrm{II},k}^{\mathrm{PT}*} \wedge T_{\mathrm{fix}}), \boldsymbol{x}_- \right\rangle = \langle \boldsymbol{b}_k(T_{\mathrm{II}}), \boldsymbol{x}_- \rangle + \int_{T_{\mathrm{II}}}^{t} \left\langle \frac{\mathrm{d}\boldsymbol{b}_k(s)}{\mathrm{d}s}, \boldsymbol{x}_+ \right\rangle \mathrm{d}s$$

$$= \langle \boldsymbol{b}_k(T_{\mathrm{II}}), \boldsymbol{x}_- \rangle + \frac{\kappa_2}{\sqrt{m}} \int_{T_{\mathrm{II}}}^{T_{\mathrm{II},k}^{\mathrm{PT}*} \wedge T_{\mathrm{fix}}} \langle \boldsymbol{F}_+(s), \boldsymbol{x}_- \rangle \mathrm{d}s$$

$$\le Q_3 \frac{\sqrt{\kappa_1 \kappa_2}}{\sqrt{m}} - \frac{Q_2}{2} \Delta^2 p^{-\frac{1}{1-\alpha \cos \Delta}} \left(T_{\mathrm{II},k}^{\mathrm{PT}*} \wedge T_{\mathrm{fix}} - T_{\mathrm{II}}\right)$$

$$\le Q_3 \frac{\sqrt{\kappa_1 \kappa_2}}{\sqrt{m}} - \frac{Q_2}{2} \Delta^2 p^{-\frac{1}{1-\alpha \cos \Delta}} \left((T_{\mathrm{III},k} - T_{\mathrm{II}}) \wedge \frac{3Q_3}{Q_2 \Delta^2} \sqrt{\frac{\kappa_1}{\kappa_2}} p^{\frac{1}{1-\alpha \cos \Delta}}\right).$$

We claim $T_{\mathrm{II},k}^{\mathrm{PT}*} - T_{\mathrm{II}} \le \frac{2Q_3}{Q_2 \Delta^2} \sqrt{\frac{\kappa_1}{\kappa_2}} p^{\frac{1}{1-\alpha \cos \Delta}}$. If otherwise, then

$$\left\langle \boldsymbol{b}_k(T_{\mathrm{II},k}^{\mathrm{PT}*} \wedge T_{\mathrm{fix}}), \boldsymbol{x}_- \right\rangle < Q_3 \frac{\sqrt{\kappa_1 \kappa_2}}{\sqrt{m}} - Q_3 \frac{\sqrt{\kappa_1 \kappa_2}}{\sqrt{m}} = 0.$$

From the definition of $T_{\mathrm{II},k}^{\mathrm{PT}*}$, we know $T_{\mathrm{II},k}^{\mathrm{PT}*} < T_{\mathrm{II},k}^{\mathrm{PT}*} \wedge T_{\mathrm{fix}}$, which leads to a contradiction.

therefore, we have proved that for any $k \in \mathcal{K}_+$,

$$T_{\mathrm{II},k}^{\mathrm{PT}*} \wedge T_{\mathrm{fix}} = T_{\mathrm{II},k}^{\mathrm{PT}*};$$

$$T_{\mathrm{II},k}^{\mathrm{PT}*} \le T_{\mathrm{II}} + \frac{2Q_3}{Q_2 \Delta^2} \sqrt{\frac{\kappa_1}{\kappa_2}} p^{\frac{1}{1-\alpha \cos \Delta}}.$$

With the help of Lemma E.1 (S2)(P0), we obtain

$$T_{\mathrm{II}}^{\mathrm{PT}*} \wedge T_{\mathrm{fix}} = T_{\mathrm{II}}^{\mathrm{PT}*};$$

$$T_{\mathrm{II}}^{\mathrm{PT}*} = \max_{k \in \mathcal{K}_+} T_{\mathrm{II},k}^{\mathrm{PT}*} \le T_{\mathrm{II}} + \frac{2Q_3}{Q_2 \Delta^2} \sqrt{\frac{\kappa_1}{\kappa_2}} p^{\frac{1}{1-\alpha \cos \Delta}}.$$

Recalling Lemma E.3, $\langle \boldsymbol{F}_+(t), \boldsymbol{x}_+ \rangle > 0$ and $\langle \boldsymbol{F}_+(t), \boldsymbol{x}_- \rangle < 0$ hold for any $t \in [T_{\mathrm{II}}, T_{\mathrm{II}}^{\mathrm{PT}*} \wedge T_{\mathrm{fix}}] = [T_{\mathrm{II}}, T_{\mathrm{II}}^{\mathrm{PT}*}]$. From the definitions of $T_{\mathrm{II}}^{\mathrm{PT}}$ and $T_{\mathrm{II}}^{\mathrm{PT}*}$ (26)(27), we obtain

$$T_{\mathrm{II}}^{\mathrm{PT}} = T_{\mathrm{II}}^{\mathrm{PT}*}.$$

In conclusion, we have proved:

$$T_{\mathrm{II}}^{\mathrm{PT}} = T_{\mathrm{II}}^{\mathrm{PT}*} = T_{\mathrm{II}} + \mathcal{O}\left(\sqrt{\frac{\kappa_1}{\kappa_2}} \frac{p^{\frac{1}{1-\alpha\cos\Delta}}}{\Delta^2}\right) \overset{\text{Lemma D.6}}{=} \left(1 + \mathcal{O}\left(\sqrt{\kappa_1 \kappa_2^3}\right)\right) T_{\mathrm{II}}.$$

Step II. Prediction Estimate. Step I gives us the result:

$$\delta_T := T_{\mathrm{II}}^{\mathrm{PT}} - T_{\mathrm{II}} = \mathcal{O}\left(\sqrt{\frac{\kappa_1}{\kappa_2}} \frac{p^{\frac{1}{1-\alpha\cos\Delta}}}{\Delta^2}\right).$$

Recalling the proof of Lemma E.2, we know

$$-\frac{C_1 + C_0 e^{f_-(T_{\mathrm{II}})} \delta_T}{1 + C_0 e^{f_-(T_{\mathrm{II}})} \delta_T} \le e^{f_-(t) - f_-(T_{\mathrm{II}})} - 1 \le 0,$$

$$-C_2 \delta_T \le e^{f_+(T_{\mathrm{II}}) - f_+(t)} - 1 \le 0.$$

where

$$C_0 = \Theta\left(\frac{\kappa_2^2 \Delta^2}{p}\right), \; C_1 = \mathcal{O}\left(\kappa_2 \sqrt{\kappa_1 \kappa_2}\right), \; C_2 = \Theta\left(\kappa_2^2 p^{-\frac{1}{1-\alpha\cos\Delta}}\right).$$

Then a straightforward calculation gives us:

$$0 \ge e^{f_-(t) - f_-(T_{\mathrm{II}})} - 1 = -\mathcal{O}\left(\kappa_2^2 \sqrt{\frac{\kappa_1}{\kappa_2}}\right) - \mathcal{O}\left(\kappa_2^2 \sqrt{\frac{\kappa_1}{\kappa_2}}\right) = -\mathcal{O}\left(\kappa_2^2 \sqrt{\frac{\kappa_1}{\kappa_2}}\right)$$

$$0 \ge e^{f_+(T_{\mathrm{II}}) - f_+(T_{\mathrm{II}}^{\mathrm{PT}})} - 1 = -\mathcal{O}\left(\kappa_2^2 \sqrt{\frac{\kappa_1}{\kappa_2}} \frac{1}{\Delta^2}\right).$$

With the help of Lemma D.8, we obtain the prediction estimate at the end of Phase III:

$$e^{-f_+(T_{\mathrm{II}}^{\mathrm{PT}})} = e^{-f_+(T_{\mathrm{II}})} e^{f_+(T_{\mathrm{II}}) - f_+(T_{\mathrm{II}}^{\mathrm{PT}})} = \Theta\left(e^{-f_+(T_{\mathrm{II}})}\right) = \Theta\left(p^{-\frac{1}{1-\alpha\cos\Delta}}\right),$$

$$e^{f_-(T_{\mathrm{II}}^{\mathrm{PT}})} = e^{f_-(T_{\mathrm{II}})} e^{f_-(T_{\mathrm{II}}^{\mathrm{PT}}) - f_-(T_{\mathrm{II}})} = \Theta\left(e^{f_-(T_{\mathrm{II}})}\right) = \Theta\left(p^{-\frac{\alpha\cos\Delta}{1-\alpha\cos\Delta}}\right).$$

Moreover,

$$\left|\left(\frac{pe^{-f_+(T_{\mathrm{II}}^{\mathrm{PT}})}}{1+p} - \frac{e^{f_-(T_{\mathrm{II}}^{\mathrm{PT}})}}{1+p} \cos\Delta\right) - \left(\frac{pe^{-f_+(T_{\mathrm{II}})}}{1+p} - \frac{e^{f_-(T_{\mathrm{II}})}}{1+p} \cos\Delta\right)\right|$$

$$\le \left|\frac{pe^{-f_+(T_{\mathrm{II}})}}{1+p}\right| \left|\frac{\frac{pe^{-f_+(T_{\mathrm{II}}^{\mathrm{PT}})}}{1+p}}{\frac{pe^{-f_+(T_{\mathrm{II}})}}{1+p}} - 1\right| + \left|\frac{e^{f_-(T_{\mathrm{II}})} \cos\Delta}{1+p}\right| \left|\frac{\frac{e^{f_-(T_{\mathrm{II}}^{\mathrm{PT}})} \cos\Delta}{1+p}}{\frac{e^{f_-(T_{\mathrm{II}})} \cos\Delta}{1+p}} - 1\right|$$

$$= \mathcal{O}\left(p^{-\frac{1}{1-\alpha\cos\Delta}} \kappa_2^2 \sqrt{\frac{\kappa_1}{\kappa_2}} \frac{1}{\Delta^2}\right) + \mathcal{O}\left(p^{-\frac{1}{1-\alpha\cos\Delta}} \kappa_2^2 \sqrt{\frac{\kappa_1}{\kappa_2}}\right)$$

$$= \mathcal{O}\left(p^{-\frac{1}{1-\alpha\cos\Delta}} \kappa_2^2 \sqrt{\frac{\kappa_1}{\kappa_2}} \frac{1}{\Delta^2}\right) \overset{\kappa_1 \kappa_2^3 = \mathcal{O}(\Delta^8)}{=} \mathcal{O}\left(\Delta^2 p^{-\frac{1}{1-\alpha\cos\Delta}}\right),$$

which means

$$\frac{pe^{-f_+(T_{\mathrm{II}}^{\mathrm{PT}})}}{1+p} - \frac{e^{f_-(T_{\mathrm{II}}^{\mathrm{PT}})}}{1+p} \cos\Delta = \Theta\left(\Delta^2 p^{-\frac{1}{1-\alpha\cos\Delta}}\right).$$

In the same way, we can obtain

$$\frac{pe^{-f_+(T_{\mathrm{II}}^{\mathrm{PT}})}}{1+p} \cos\Delta - \frac{e^{f_-(T_{\mathrm{II}}^{\mathrm{PT}})}}{1+p} = -\Theta\left(\Delta^2 p^{-\frac{1}{1-\alpha\cos\Delta}}\right).$$

Step III. Activation Patterns.

Recall our proofs in Step I, we know that

$$\left\langle \boldsymbol{w}_k(T_{\mathrm{II}}^{\mathrm{PT}}), \boldsymbol{x}_- \right\rangle = 0, \ \forall k \in \mathcal{K}_+.$$

Moreover, from the dynamics in Lemma E.1 (S1) and (S2)(P3), we obtain:

$$\left\langle \boldsymbol{w}_k(T_{\mathrm{II}}^{\mathrm{PT}}), \boldsymbol{x}_+ \right\rangle > 0, \ \forall k \in \mathcal{K}_+;$$
$$\left\langle \boldsymbol{w}_k(T_{\mathrm{II}}^{\mathrm{PT}}), \boldsymbol{x}_+ \right\rangle = 0, \ \left\langle \boldsymbol{w}_k(T_{\mathrm{II}}^{\mathrm{PT}}), \boldsymbol{x}_- \right\rangle > 0, \ \forall k \in \mathcal{K}_-.$$

$\square$

*Proof of Theorem 4.6.*
Theorem 4.6 (S1) has been proven in Lemma E.4 (S1), and Theorem 4.6 (S2) has been proven in Lemma E.4 (S3).

$\square$

## E.2 Optimization Dynamics after Phase Transition

After Phase Transition $(t > T_{\mathrm{II}}^{\mathrm{PT}})$, we study the dynamics before the patterns of living neurons change again. Specifically, we define the following hitting time

$$T_{\mathrm{III}} := \inf \left\{ t > T_{\mathrm{II}}^{\mathrm{PT}} : \exists k \in \mathcal{K}_+ \cup \mathcal{K}_-, \mathtt{sgn}_k^+(t) \neq \mathtt{sgn}_k^+(T_{\mathrm{I}}) \text{ or } \mathtt{sgn}_k^-(t) \neq \mathtt{sgn}_k^-(T_{\mathrm{I}}) \right\}$$

$$= \inf \left\{ t > T_{\mathrm{II}}^{\mathrm{PT}} : \exists k \in \mathcal{K}_+, \text{ s.t. } \langle \boldsymbol{w}_k(t), \boldsymbol{x}_+ \rangle \leq 0 \text{ or } \langle \boldsymbol{w}_k(t), \boldsymbol{x}_- \rangle \neq 0; \right. \tag{28}$$

$$\left. \text{or } \exists k \in \mathcal{K}_-, \text{ s.t. } \langle \boldsymbol{w}_k(t), \boldsymbol{x}_+ \rangle \neq 0 \text{ or } \langle \boldsymbol{w}_k(t), \boldsymbol{x}_- \rangle \leq 0 \right\},$$

and we call $t \in (T_{\mathrm{II}}^{\mathrm{PT}}, T_{\mathrm{III}})$ "L-Phase III".

Moreover, we call $t \in [T_{\mathrm{II}}, T_{\mathrm{III}})$ "Phase III", i.e.. "Phase Transition" + "L-Phase III".

In order to analyze the dynamics of neurons and vector fields, we introduce the auxiliary hitting time:

$$T_{\mathrm{III}}^* := T_{\mathrm{III}} \wedge \inf \left\{ t > T_{\mathrm{II}}^{\mathrm{PT}} : \langle \boldsymbol{F}_+(t), \boldsymbol{x}_+ \rangle \leq 0 \text{ or } \langle \boldsymbol{F}_+(t), \boldsymbol{x}_- \rangle \geq 0 \right\},$$

$$\text{where} \quad \boldsymbol{F}_+(t) = \frac{p}{1+p} e^{-f_+(t)} \boldsymbol{x}_+ - \frac{1}{1+p} e^{f_-(t)} \boldsymbol{x}_-. \tag{29}$$

We call $t \in (T_{\mathrm{II}}^{\mathrm{PT}}, T_{\mathrm{III}}^*)$ "L-Phase III*".

Due to the almost simplest activation patterns, this phase is easier to analyze, and we only need to estimate the time and size of the changes in the vector field. Nevertheless, our challenge is to prove that all living negative neurons simultaneously change their activation patterns at $T_{\mathrm{III}}^*$, which also implies that $T_{\mathrm{III}} = T_{\mathrm{III}}^*$.

**Lemma E.5** (Dynamics of activate neurons during L-Phase III*).
*In L-Phase III\* $(t \in [T_{\mathrm{II}}^{\mathrm{PT}}, T_{\mathrm{III}}^*))$, we have the following dynamics for each neuron $k \in \mathcal{K}_- \cup \mathcal{K}_+$.*

*(S1). For negative neuron $k \in \mathcal{K}_-$, we have:*

$$\boldsymbol{w}_k(t) \in \mathcal{M}_+^0 \cap \mathcal{M}_-^+,$$
$$\frac{\mathrm{d}\boldsymbol{b}_k(t)}{\mathrm{d}t} = \frac{\kappa_2 e^{f_-(t)}}{\sqrt{m}(1+p)} \left( \boldsymbol{x}_- - \boldsymbol{x}_+ \cos\Delta \right).$$

*(S2) For positive neuron $k \in \mathcal{K}_+$, we have:*

$$\boldsymbol{w}_k(t) \in \mathcal{M}_+^+ \cap \mathcal{M}_-^0,$$
$$\frac{\mathrm{d}\boldsymbol{b}_k(t)}{\mathrm{d}t} = \frac{\kappa_2 p e^{-f_+(t)}}{\sqrt{m}(1+p)} \left( \boldsymbol{x}_+ - \boldsymbol{x}_- \cos\Delta \right).$$

*Proof of Lemma E.5.*
From the definition of $T^*_{\mathrm{III}}$, we know that $\langle \boldsymbol{F}_+(t), \boldsymbol{x}_+ \rangle > 0$ and $\langle \boldsymbol{F}_+(t), \boldsymbol{x}_- \rangle < 0$ hold for any $t \in [T^{\mathrm{PT}}_{\mathrm{II}}, T^*_{\mathrm{III}})$. Moreover, Lemma E.4 ensures that for $k \in \mathcal{K}_+$, $\boldsymbol{w}_k(T^{\mathrm{PT}}_{\mathrm{II}}) \in \mathcal{M}^0_+ \cap \mathcal{M}^+_-$; for $k \in \mathcal{K}_-$, $\boldsymbol{w}_k(T^{\mathrm{PT}}_{\mathrm{II}}) \in \mathcal{M}^+_+ \cap \mathcal{M}^0_-$. Hence, this lemma can be proved in the same way as shown in the proof of Lemma E.1 (S1) and (S2)(P2). We do not repeat it here.

$\square$

**Lemma E.6** (Time and prediction estimate at the end of L-Phase III*)**.**
*(S1) (Time).*

$$T^*_{\mathrm{III}} = T^{\mathrm{PT}}_{\mathrm{II}} + \Theta\left( \frac{p^{\frac{1}{1-\alpha \cos \Delta}}}{\kappa_2^2} \right) = \left(1 + \Theta(\Delta^2)\right) T^{\mathrm{PT}}_{\mathrm{II}} = \left(1 + \Theta(\Delta^2)\right) T_{\mathrm{II}};$$

*(S2) (Prediction).*

$$e^{-f_+(T^*_{\mathrm{III}})} = \Theta\left( p^{-\frac{1}{1-\alpha \cos \Delta}} \right), \quad e^{f_-(T^*_{\mathrm{III}})} = \Theta\left( p^{-\frac{\alpha \cos \Delta}{1-\alpha \cos \Delta}} \right);$$

$$\frac{p e^{-f_+(T^*_{\mathrm{III}})}}{1+p} - \frac{e^{f_-(T^*_{\mathrm{III}})}}{1+p} \cos \Delta = 0,$$

$$\frac{p e^{-f_+(T^{\mathrm{PT}}_{\mathrm{II}})}}{1+p} \cos \Delta - \frac{e^{f_-(T^{\mathrm{PT}}_{\mathrm{II}})}}{1+p} = -\Theta\left( \Delta^2 p^{-\frac{1}{1-\alpha \cos \Delta}} \right).$$

*Proof of Lemma E.6.*
Step I. Explicit Solution to $f_+(t)$ and $f_-(t)$.

For any $t \in [T^{\mathrm{PT}}_{\mathrm{II}}, T^*_{\mathrm{III}})$, we have:

$$f_+(t) = \frac{\kappa_2}{\sqrt{m}} \sum_{k \in \mathcal{K}_+} \boldsymbol{b}_k^\top(t) \boldsymbol{x}_+,$$

$$f_-(t) = -\frac{\kappa_2}{\sqrt{m}} \sum_{k \in \mathcal{K}_-} \boldsymbol{b}_k^\top(t) \boldsymbol{x}_-.$$

Let us consider the dynamics of $f_+(t)$ and $f_-(t)$. With the help of Lemma E.5, these two dynamics are nearly independent:

$$\frac{\mathrm{d}f_+(t)}{\mathrm{d}t} = = \frac{\kappa_2}{\sqrt{m}} \sum_{k \in \mathcal{K}_+} \left\langle \frac{\kappa_2 p e^{-f_+(t)}}{\sqrt{m}(1+p)} \left( \boldsymbol{x}_+ - \boldsymbol{x}_- \cos \Delta \right), \boldsymbol{x}_+ \right\rangle = \frac{\kappa_2^2 m_+ p \sin^2 \Delta}{m(1+p)} e^{-f_+(t)},$$

$$\frac{\mathrm{d}f_-(t)}{\mathrm{d}t} = = -\frac{\kappa_2}{\sqrt{m}} \sum_{k \in \mathcal{K}_-} \left\langle \frac{\kappa_2 e^{f_-(t)}}{\sqrt{m}(1+p)} \left( \boldsymbol{x}_- - \boldsymbol{x}_+ \cos \Delta \right), \boldsymbol{x}_- \right\rangle = -\frac{\kappa_2^2 m_- \sin^2 \Delta}{m(1+p)} e^{f_-(t)}.$$

Their solutions are:

$$e^{-f_+(t)} = \frac{e^{-f_+(T^{\mathrm{PT}}_{\mathrm{II}})}}{1 + e^{-f_+(T^{\mathrm{PT}}_{\mathrm{II}})} \frac{\kappa_2^2 m_+ p \sin^2 \Delta}{m(1+p)} (t - T^{\mathrm{PT}}_{\mathrm{II}})},$$

$$e^{f_-(t)} = \frac{e^{f_-(T^{\mathrm{PT}}_{\mathrm{II}})}}{1 + e^{f_-(T^{\mathrm{PT}}_{\mathrm{II}})} \frac{\kappa_2^2 m_- \sin^2 \Delta}{m(1+p)} (t - T^{\mathrm{PT}}_{\mathrm{II}})}.$$

Step II. Time Estimate of $T^*_{\mathrm{III}}$.

For simplicity, we denote $G_+ := \frac{\kappa_2^2 m_+ p \sin^2 \Delta}{m(1+p)}$ and $G_- := \frac{\kappa_2^2 m_- \sin^2 \Delta}{m(1+p)}$.

First, we consider the evolution of the vector field $\langle \boldsymbol{F}_+(t), \boldsymbol{x}_- \rangle$:

$$\langle \boldsymbol{F}_+(t), \boldsymbol{x}_- \rangle = \frac{p e^{-f_+(t)}}{1+p} \cos \Delta - \frac{e^{f_-(t)}}{1+p}$$

$$
= \frac{1}{1+p} \left( \frac{pe^{-f_+(T_{\mathrm{II}}^{\mathrm{PT}})} \cos \Delta}{1 + e^{-f_+(T_{\mathrm{II}}^{\mathrm{PT}})} G_+(t - T_{\mathrm{II}}^{\mathrm{PT}})} - \frac{e^{f_-(T_{\mathrm{II}}^{\mathrm{PT}})}}{1 + e^{f_-(T_{\mathrm{II}}^{\mathrm{PT}})} G_-(t - T_{\mathrm{II}}^{\mathrm{PT}})} \right)
$$

$$
= \frac{(pe^{-f_+(T_{\mathrm{II}}^{\mathrm{PT}})} \cos \Delta - e^{f_-(T_{\mathrm{II}}^{\mathrm{PT}})}) + e^{f_-(T_{\mathrm{II}}^{\mathrm{PT}} - f_+(T_{\mathrm{II}}^{\mathrm{PT}})}(pG_- \cos \Delta - G_+)(t - T_{\mathrm{II}}^{\mathrm{PT}})}{(1+p)(1 + e^{-f_+(T_{\mathrm{II}}^{\mathrm{PT}})} G_+(t - T_{\mathrm{II}}^{\mathrm{PT}}))(1 + e^{f_-(T_{\mathrm{II}}^{\mathrm{PT}})} G_-(t - T_{\mathrm{II}}^{\mathrm{PT}}))}
$$

$$
= \frac{(1+p) \left\langle \boldsymbol{F}_+(T_{\mathrm{II}}^{\mathrm{PT}}), \boldsymbol{x}_- \right\rangle + e^{f_-(T_{\mathrm{II}}^{\mathrm{PT}} - f_+(T_{\mathrm{II}}^{\mathrm{PT}})}(pG_- \cos \Delta - G_+)(t - T_{\mathrm{II}}^{\mathrm{PT}})}{(1+p)(1 + e^{-f_+(T_{\mathrm{II}}^{\mathrm{PT}})} G_+(t - T_{\mathrm{II}}^{\mathrm{PT}}))(1 + e^{f_-(T_{\mathrm{II}}^{\mathrm{PT}})} G_-(t - T_{\mathrm{II}}^{\mathrm{PT}}))} < 0.
$$

Hence, the hitting time $T_{\mathrm{III}}^*$ can be converted to the following $T_{\mathrm{III}}^{**}$:

$$
T_{\mathrm{III}}^* = T_{\mathrm{III}}^{**} := T_{\mathrm{III}} \wedge \inf \left\{ t > T_{\mathrm{II}}^{\mathrm{PT}} : \left\langle \boldsymbol{F}_+(t), \boldsymbol{x}_+ \right\rangle \le 0 \right\}.
$$

Then we consider $\left\langle \boldsymbol{F}_+(t), \boldsymbol{x}_+ \right\rangle$:

$$
\left\langle \boldsymbol{F}_+(t), \boldsymbol{x}_+ \right\rangle = \frac{pe^{-f_+(t)}}{1+p} - \frac{e^{f_-(t)}}{1+p} \cos \Delta
$$

$$
= \frac{1}{1+p} \left( \frac{pe^{-f_+(T_{\mathrm{II}}^{\mathrm{PT}})}}{1 + e^{-f_+(T_{\mathrm{II}}^{\mathrm{PT}})} G_+(t - T_{\mathrm{II}}^{\mathrm{PT}})} - \frac{e^{f_-(T_{\mathrm{II}}^{\mathrm{PT}})} \cos \Delta}{1 + e^{f_-(T_{\mathrm{II}}^{\mathrm{PT}})} G_-(t - T_{\mathrm{II}}^{\mathrm{PT}})} \right)
$$

$$
= \frac{(pe^{-f_+(T_{\mathrm{II}}^{\mathrm{PT}})} - e^{f_-(T_{\mathrm{II}}^{\mathrm{PT}})} \cos \Delta) + e^{f_-(T_{\mathrm{II}}^{\mathrm{PT}} - f_+(T_{\mathrm{II}}^{\mathrm{PT}})}(pG_- - G_+ \cos \Delta)(t - T_{\mathrm{II}}^{\mathrm{PT}})}{(1+p)(1 + e^{-f_+(T_{\mathrm{II}}^{\mathrm{PT}})} G_+(t - T_{\mathrm{II}}^{\mathrm{PT}}))(1 + e^{f_-(T_{\mathrm{II}}^{\mathrm{PT}})} G_-(t - T_{\mathrm{II}}^{\mathrm{PT}}))}
$$

$$
= \frac{(1+p) \left\langle \boldsymbol{F}_+(T_{\mathrm{II}}^{\mathrm{PT}}), \boldsymbol{x}_+ \right\rangle + e^{f_-(T_{\mathrm{II}}^{\mathrm{PT}}) - f_+(T_{\mathrm{II}}^{\mathrm{PT}})}(pG_- - G_+ \cos \Delta)(t - T_{\mathrm{II}}^{\mathrm{PT}})}{(1+p)(1 + e^{-f_+(T_{\mathrm{II}}^{\mathrm{PT}})} G_+(t - T_{\mathrm{II}}^{\mathrm{PT}}))(1 + e^{f_-(T_{\mathrm{II}}^{\mathrm{PT}})} G_-(t - T_{\mathrm{II}}^{\mathrm{PT}}))}.
$$

From Lemma E.4, we know

$$
(1+p) \left\langle \boldsymbol{F}_+(T_{\mathrm{II}}^{\mathrm{PT}}), \boldsymbol{x}_+ \right\rangle = (1+p) \left( \frac{pe^{-f_+(t)}}{1+p} - \frac{e^{f_-(t)}}{1+p} \cos \Delta \right) = \Theta \left( \Delta^2 p^{-\frac{\alpha \cos \Delta}{1 - \alpha \cos \Delta}} \right),
$$

$$
e^{f_-(T_{\mathrm{II}}^{\mathrm{PT}}) - f_+(T_{\mathrm{II}}^{\mathrm{PT}})} = \Theta \left( e^{f_-(T_{\mathrm{I+II}}) - f_+(T_{\mathrm{I+II}})} \right) = \Theta \left( p^{-\frac{1 + \alpha \cos \Delta}{1 - \alpha \cos \Delta}} \right),
$$

$$
pG_- - G_+ \cos \Delta = \frac{\kappa_2^2 p \sin^2 \Delta}{1+p} \frac{(m_- - m_+ \cos \Delta)}{m} = -\Theta \left( \kappa_2^2 \Delta^2 \right).
$$

These imply the hitting time:

$$
T_{\mathrm{III}}^* = T_{\mathrm{III}}^{**} = T_{\mathrm{II}}^{\mathrm{PT}} + \Theta \left( \frac{\Delta^2 p^{-\frac{\alpha \cos \Delta}{1 - \alpha \cos \Delta}}}{p^{-\frac{1 + \alpha \cos \Delta}{1 - \alpha \cos \Delta}} \kappa_2^2 \Delta^2} \right) = T_{\mathrm{II}}^{\mathrm{PT}} + \Theta \left( \frac{p^{\frac{1}{1 - \alpha \cos \Delta}}}{\kappa_2^2} \right).
$$

Step III. Prediction estimate.

From the explicit solution in Step I and the time estimate in Step II, it is easy to verify

$$
e^{-f_+(T_{\mathrm{III}}^*)} = \frac{e^{-f_+(T_{\mathrm{II}}^{\mathrm{PT}})}}{1 + e^{-f_+(T_{\mathrm{II}}^{\mathrm{PT}})} \frac{\kappa_2^2 m_+ p \sin^2 \Delta}{m(1+p)} (T_{\mathrm{III}}^* - T_{\mathrm{II}}^{\mathrm{PT}})} = \Theta \left( p^{-\frac{1}{1 - \alpha \cos \Delta}} \right),
$$

$$
e^{f_-(T_{\mathrm{III}}^*)} = \frac{e^{f_-(T_{\mathrm{II}}^{\mathrm{PT}})}}{1 + e^{f_-(T_{\mathrm{II}}^{\mathrm{PT}})} \frac{\kappa_2^2 m_- \sin^2 \Delta}{m(1+p)} (T_{\mathrm{III}}^* - T_{\mathrm{II}}^{\mathrm{PT}})} = \Theta \left( p^{-\frac{\alpha \cos \Delta}{1 - \alpha \cos \Delta}} \right).
$$

Recalling the calculation in Step II, we have:

$$
\left\langle \boldsymbol{F}_+(T_{\mathrm{III}}^*), \boldsymbol{x}_+ \right\rangle = \frac{pe^{-f_+(T_{\mathrm{III}}^*)}}{1+p} - \frac{e^{f_-(T_{\mathrm{III}}^*)}}{1+p} \cos \Delta = 0,
$$

$$
\left\langle \boldsymbol{F}_+(T_{\mathrm{III}}^*), \boldsymbol{x}_- \right\rangle = \frac{pe^{-f_+(T_{\mathrm{III}}^*)}}{1+p} \cos \Delta - \frac{e^{f_-(T_{\mathrm{III}}^*)}}{1+p}
$$

$$= \frac{(1+p)\left\langle \boldsymbol{F}_+(T_{\mathrm{II}}^{\mathrm{PT}}), \boldsymbol{x}_-\right\rangle + e^{f_-(T_{\mathrm{II}}^{\mathrm{PT}}) - f_+(T_{\mathrm{II}}^{\mathrm{PT}})}(pG_- \cos\Delta - G_+)(T_{\mathrm{III}}^* - T_{\mathrm{II}}^{\mathrm{PT}})}{(1+p)(1 + e^{-f_+(T_{\mathrm{II}}^{\mathrm{PT}})}G_+(T_{\mathrm{III}}^* - T_{\mathrm{II}}^{\mathrm{PT}}))(1 + e^{f_-(T_{\mathrm{II}}^{\mathrm{PT}})}G_-(T_{\mathrm{III}}^* - T_{\mathrm{II}}^{\mathrm{PT}}))}$$

$$= \Theta\left(\frac{-\Delta^2 p^{-\frac{\alpha\cos\Delta}{1 - \alpha\cos\Delta}} - \Delta^2 p^{-\frac{\alpha\cos\Delta}{1 - \alpha\cos\Delta}}}{p\left(1 + \Theta(\Delta^2)\right)\left(1 + \Theta(\Delta^2)\right)}\right) = -\Theta\left(\Delta^2 p^{-\frac{1}{1 - \alpha\cos\Delta}}\right).$$

$\square$

**Lemma E.7** (Hitting time relationship). *If we define the following hitting time:*
$$T_{\mathrm{III}}^{\mathrm{W}} = \inf\left\{t > T_{\mathrm{II}}^{\mathrm{PT}} : \forall k \in \mathcal{K}_-, \langle \boldsymbol{w}_k(t), \boldsymbol{x}_+\rangle > 0\right\},$$
*then it holds that $T_{\mathrm{III}} = T_{\mathrm{III}}^* = T_{\mathrm{III}}^{\mathrm{W}}$.*

*Proof of Lemma E.7.*
We define the following hitting time:
$$T_{\mathrm{III}}^{\mathrm{F}} = \inf\left\{t > T_{\mathrm{II}}^{\mathrm{PT}} : \langle \boldsymbol{F}_+(t), \boldsymbol{x}_+\rangle \leq 0\right\};$$
$$T_{\mathrm{III}}^{\mathrm{N}} = \inf\left\{t > T_{\mathrm{II}}^{\mathrm{PT}} : \exists k \in \mathcal{K}_-, \text{ s.t. } \langle \boldsymbol{w}_k(t), \boldsymbol{x}_+\rangle > 0\right\},$$
$$T_{\mathrm{III}}^{\mathrm{W}} = \inf\left\{t > T_{\mathrm{II}}^{\mathrm{PT}} : \forall k \in \mathcal{K}_-, \langle \boldsymbol{w}_k(t), \boldsymbol{x}_+\rangle > 0\right\},$$

From the proof in Lemma E.4, we know $\langle \boldsymbol{F}_+(T_{\mathrm{III}}^*), \boldsymbol{x}_-\rangle < 0$. From the continuity of $\langle \boldsymbol{F}_+(\cdot), \boldsymbol{x}_-\rangle$, we know that there exists $\tau_1 > 0$, such that $\langle \boldsymbol{F}_+(t), \boldsymbol{x}_-\rangle < 0$ holds for any $t \in [T_{\mathrm{III}}^*, T_{\mathrm{III}}^* + \tau_1)$. Then in the same way as the proof of Lemma E.5 (S2), we know that for $k \in \mathcal{K}_+$, $\boldsymbol{w}(t) \in \mathcal{M}_+^+ \cap \mathcal{M}_-^0$ for any $t \in [T_{\mathrm{III}}^*, T_{\mathrm{III}}^* + \tau_1)$.

Recalling that for any $k \in \mathcal{K}_-$, $\langle \boldsymbol{b}_k(T_{\mathrm{III}}^*), \boldsymbol{x}_-\rangle > 0$, from the continuity, we know that there exists $\tau_2 > 0$ such that $\langle \boldsymbol{b}_k(t), \boldsymbol{x}_-\rangle > 0$ holds for any $t \in [T_{\mathrm{III}}^*, T_{\mathrm{III}}^* + \tau_2)$.

Hence, we have:
$$T_{\mathrm{III}}^* = T_{\mathrm{III}}^{\mathrm{F}} \wedge T_{\mathrm{III}}^{\mathrm{N}} = \inf\left\{t > T_{\mathrm{II}}^{\mathrm{PT}} : \langle \boldsymbol{F}_+(t), \boldsymbol{x}_+\rangle \leq 0 \text{ or } \exists k \in \mathcal{K}_-, \text{ s.t. } \langle \boldsymbol{w}_k(t), \boldsymbol{x}_+\rangle \neq 0\right\}.$$

It is obvious that $T_{\mathrm{III}}^{\mathrm{N}} \geq T_{\mathrm{III}}^{\mathrm{F}} \wedge T_{\mathrm{III}}^{\mathrm{N}} = T_{\mathrm{III}}^*$. Now we prove $T_{\mathrm{III}}^{\mathrm{N}} = T_{\mathrm{III}}^*$.

If we assume $T_{\mathrm{III}}^{\mathrm{N}} > T_{\mathrm{III}}^*$ strictly, then the dynamics about $f_+(t)$ and $f_-(t)$ in the proof (Step I) of Lemma E.6 still hold for any $t \in [T_{\mathrm{III}}^*, T_{\mathrm{III}}^{\mathrm{N}})$. Using the same calculate about $\langle \boldsymbol{F}_+(t), \boldsymbol{x}_+\rangle$ in the proof (Step II, III) of Lemma E.6, we can obtain: $\langle \boldsymbol{F}_+(t), \boldsymbol{x}_+\rangle < 0$, $t \in [T_{\mathrm{III}}^*, T_{\mathrm{III}}^{\mathrm{N}})$.

Then we consider the vector field around the manifold $\mathcal{M}_+^0 \cap \mathcal{M}_-^+$ for $t \in [T_{\mathrm{III}}^*, T_{\mathrm{III}}^{\mathrm{N}})$. In the same way as the proof of Lemma E.1 (S1), we can prove that the two-side projections onto $\boldsymbol{x}_+$ (the normal to the surface $\mathcal{M}_+^+ \cap \mathcal{M}_-^0$) satisfies $f_N^+(t, \tilde{\boldsymbol{w}}), f_N^-(t, \tilde{\boldsymbol{w}}) > 0$ for any $t \in [T_{\mathrm{III}}^*, T_{\mathrm{III}}^* + \tau_1)$, which satisfies (Case II) in Definition H.1. This implies that $\boldsymbol{w}_k(t)$ enter the manifold $\mathcal{M}_+^+$, i.e., $\langle \boldsymbol{w}_k(t), \boldsymbol{x}_+\rangle > 0$ for any $t \in [T_{\mathrm{III}}^*, T_{\mathrm{III}}^{\mathrm{N}})$, which is contradict to the definition of $T_{\mathrm{III}}^{\mathrm{N}}$. Hence, we have proved
$$T_{\mathrm{III}}^{\mathrm{N}} = T_{\mathrm{III}}^{\mathrm{N}} \wedge T_{\mathrm{III}}^{\mathrm{F}} = T_{\mathrm{III}}^*.$$

Noticing that the change of activation patterns of $\mathrm{sgn}_k^+(t)$ ($k \in \mathcal{K}_-$) is due to the change of the vector field $\langle \boldsymbol{F}_+(t), \boldsymbol{x}_+\rangle$, it is easy to verify that $T_{\mathrm{III}}^{\mathrm{F}} = T_{\mathrm{III}}^{\mathrm{N}} \wedge T_{\mathrm{III}}^{\mathrm{F}}*$. Then we have $T_{\mathrm{III}}^{\mathrm{N}} = T_{\mathrm{III}}^{\mathrm{F}} = T_{\mathrm{III}}^{\mathrm{N}} \wedge T_{\mathrm{III}}^{\mathrm{F}} = T_{\mathrm{III}}^*$.

Moreover, noticing that $T_{\mathrm{III}} \leq T_{\mathrm{III}}^{\mathrm{N}}$ and $T_{\mathrm{III}}^* \leq T_{\mathrm{III}}$, we obtain $T_{\mathrm{III}} = T_{\mathrm{III}}^* = T_{\mathrm{III}}^{\mathrm{N}} = T_{\mathrm{III}}^{\mathrm{F}}$.

Lastly, noticing that all living negative neurons ($k \in \mathcal{K}_-$) belong to $\mathcal{M}_+^0 \cap \mathcal{M}_-^+$ at time $T_{\mathrm{III}}$. As discussed above, for each living negative neuron $k \in \mathcal{K}_-$, the vector field near $\boldsymbol{b}_k(T_{\mathrm{III}})$ is the same, with $f_N^- > 0$ and $f_N^+ = 0$ in Definition H.1 (Case II). Hence, each living positive neuron $\boldsymbol{w}_k$ leaves from $\mathcal{M}_+^0$ and enter $\mathcal{M}_+^+$ instantly at $T_{\mathrm{III}}$, which means $T_{\mathrm{III}}^{\mathrm{W}} = T_{\mathrm{III}}^{\mathrm{N}}$.

Hence, we have proved $T_{\mathrm{III}} = T_{\mathrm{III}}^* = T_{\mathrm{III}}^{\mathrm{W}} = T_{\mathrm{III}}^{\mathrm{N}} = T_{\mathrm{III}}^{\mathrm{F}}$. $\square$

*Proof of Theorem 4.7.*
Combining Lemma E.6 and E.7, we obtain $T_{\mathrm{III}} = \left(1 + \Theta(\Delta^2)\right)T_{\mathrm{II}}$. $\square$

# F    Proofs of Optimization Dynamics in Phase IV

*Proof of Theorem 4.8.*
From Lemma E.7, we know that all living negative neuron $k \in \mathcal{K}_-$ simultaneously change their patterns on $\boldsymbol{x}_+$ at $T_{\mathrm{III}}$: $\lim_{t \to T_{\mathrm{III}}^-} \mathbf{sgn}_k^+(t) = 0$, $\lim_{t \to T_{\mathrm{III}}^+} \mathbf{sgn}_k^+(t) = 1$. Moreover, from our proof of Lemma E.7, we know that other activation patterns remain unchanged at $T_{\mathrm{III}}$. $\qquad\square$

In this phase, we study the dynamics before activation patterns change again after the phase transition in Theorem 4.8. We define the hitting time:

$$T_{\mathrm{IV}} := \inf\{t > T_{\mathrm{III}} : \exists k \in \mathcal{K}_+ \cup \mathcal{K}_-, \mathbf{sgn}_k^+(t) \neq \lim_{s \to T_{\mathrm{III}}^+} \mathbf{sgn}_k^+(s) \text{ or } \mathbf{sgn}_k^-(t) \neq \lim_{s \to T_{\mathrm{III}}^+} \mathbf{sgn}_k^-(s)\},$$

and we call $t \in (T_{\mathrm{III}}, T_{\mathrm{IV}})$ Phase IV.

In order to analyze the dynamics of neurons and vector fields, we introduce the auxiliary hitting time:

$$T_{\mathrm{IV}}^* := \inf\left\{t > T_{\mathrm{III}} : \langle \boldsymbol{F}_+(t), \boldsymbol{x}_+ \rangle > 0, \text{ or } \langle \boldsymbol{F}_+(t), \boldsymbol{x}_- \rangle > 0\right\},$$

$$\text{where} \quad \boldsymbol{F}_+(t) = \frac{p}{1+p} e^{-f_+(t)} \boldsymbol{x}_+ - \frac{1}{1+p} e^{f_-(t)} \boldsymbol{x}_-, \tag{30}$$

and we call $t \in (T_{\mathrm{III}}, T_{\mathrm{IV}}^*)$ Phase IV*.

First, we will provide meticulous prior estimations for 2d ODEs on $f_+(t)$ and $f_-(t)$, similar to Phase II, which can imply $T_{\mathrm{IV}}^* = +\infty$. Additionally, we can prove $T_{\mathrm{IV}} = T_{\mathrm{IV}}^*$. Lastly, with the help of our fine-grained analysis for the 2D dynamics and the results in (Lyu and Li, 2019; Ji and Telgarsky, 2020), we can determine the unique convergent direction from numerous KKT directions.

## F.1    Non-asymptotic Analysis of Optimization Dynamics in Phase IV*

**Lemma F.1** (Dynamics of activate neurons in Phase IV*).
*In Phase IV* ($t \in (T_{\mathrm{III}}, T_{\mathrm{IV}}^*)$), we have the following dynamics for each neuron $k \in \mathcal{K}_- \cup \mathcal{K}_+$.*

*(S1). For negative neuron $k \in \mathcal{K}_-$, we have:*

$$\boldsymbol{w}_k(t) \in \mathcal{M}_+^+ \cap \mathcal{M}_-^+,$$

$$\frac{\mathrm{d}\boldsymbol{b}_k(t)}{\mathrm{d}t} = -\frac{\kappa_2}{\sqrt{m}} \boldsymbol{F}_+(t) = -\frac{\kappa_2}{\sqrt{m}}\left(\frac{p}{1+p} e^{-f_+(t)} \boldsymbol{x}_+ - \frac{1}{1+p} e^{f_-(t)} \boldsymbol{x}_-\right).$$

*(S2) For positive neuron $k \in \mathcal{K}_+$, we have:*

$$\boldsymbol{w}_k(t) \in \mathcal{M}_+^+ \cap \mathcal{M}_-^0,$$

$$\frac{\mathrm{d}\boldsymbol{b}_k(t)}{\mathrm{d}t} = \frac{\kappa_2 p e^{-f_+(t)}}{\sqrt{m}(1+p)}\left(\boldsymbol{x}_+ - \boldsymbol{x}_- \cos \Delta\right).$$

*Proof of Lemma F.1.*
Using the definition of $T_{\mathrm{IV}}^*$, this lemma can be proved in the same way as shown in the proof of Lemma D.1, E.1 and E.5. $\qquad\square$

The next lemma gives the first-order dynamics of $f_+(t)$ and $f_-(t)$.

**Lemma F.2** (First-order Dynamics of predictions in Phase IV*).
*In Phase IV* ($T_{\mathrm{III}} \leq t \leq T_{\mathrm{IV}}^*$), we have the following dynamics for $f_+(t)$ and $f_-(t)$:*

$$\frac{\mathrm{d}f_+(t)}{\mathrm{d}t} = \kappa_2^2 \frac{m_+}{m} \frac{p e^{-f_+(t)}}{1+p} \sin^2 \Delta + \kappa_2^2 \frac{m_-}{m}\left(\frac{p e^{-f_+(t)}}{1+p} - \frac{e^{f_-(t)} \cos \Delta}{1+p}\right),$$

$$\frac{\mathrm{d}f_-(t)}{\mathrm{d}t} = \kappa_2^2 \frac{m_-}{m}\left(\frac{p e^{-f_+(t)}}{1+p} \cos \Delta - \frac{e^{f_-(t)}}{1+p}\right).$$

*Proof of Lemma F.2.*
From the definition of $T_{\mathrm{IV}}$, for any $T_{\mathrm{III}} \leq t \leq T_{\mathrm{IV}}$, we have

$$f_+(t) = \sum_{k \in \mathcal{K}_+} \frac{\kappa_2}{\sqrt{m}} \boldsymbol{b}_k^\top(t) \boldsymbol{x}_+ - \sum_{k \in \mathcal{K}_-} \frac{\kappa_2}{\sqrt{m}} \boldsymbol{b}_k^\top(t) \boldsymbol{x}_+,$$

$$f_-(t) = - \sum_{k \in \mathcal{K}_-} \frac{\kappa_2}{\sqrt{m}} \boldsymbol{b}_k^\top(t) \boldsymbol{x}_-.$$

With the help of Lemma F.1, we have the dynamics of predictions:

$$\begin{aligned}
\frac{\mathrm{d}f_+(t)}{\mathrm{d}t} &= \sum_{k \in \mathcal{K}_+} \frac{\kappa_2}{\sqrt{m}} \left\langle \frac{\mathrm{d}\boldsymbol{b}_k(t)}{\mathrm{d}t}, \boldsymbol{x}_+ \right\rangle - \sum_{k \in \mathcal{K}_-} \frac{\kappa_2}{\sqrt{m}} \left\langle \frac{\mathrm{d}\boldsymbol{b}_k(t)}{\mathrm{d}t}, \boldsymbol{x}_+ \right\rangle \\
&= \frac{\kappa_2^2}{m} \sum_{k \in \mathcal{K}_+} \frac{p}{1+p} e^{-f_+(t)} \left(1 - \cos^2 \Delta\right) - \frac{\kappa_2^2}{m} \sum_{k \in \mathcal{K}_-} \left( \frac{\cos \Delta}{1+p} e^{f_-(t)} - \frac{p}{1+p} e^{-f_+(t)} \right) \\
&= \frac{m_+}{m} \kappa_2^2 \frac{p}{1+p} e^{-f_+(t)} \sin^2 \Delta + \frac{m_-}{m} \kappa_2^2 \left( \frac{p e^{-f_+(t)}}{1+p} - \frac{\cos \Delta}{1+p} e^{f_-(t)} \right).
\end{aligned}$$

$$\begin{aligned}
\frac{\mathrm{d}f_-(t)}{\mathrm{d}t} &= - \sum_{k \in \mathcal{K}_-} \frac{\kappa_2}{\sqrt{m}} \left\langle \frac{\mathrm{d}\boldsymbol{b}_k(t)}{\mathrm{d}t}, \boldsymbol{x}_- \right\rangle = \frac{\kappa_2^2}{m} \sum_{k \in \mathcal{K}_-} \left( \frac{p}{1+p} e^{-f_+(t)} \cos \Delta - \frac{1}{1+p} e^{f_-(t)} \right) \\
&= \frac{m_-}{m} \kappa_2^2 \left( \frac{p}{1+p} e^{-f_+(t)} \cos \Delta - \frac{1}{1+p} e^{f_-(t)} \right).
\end{aligned}$$

$\square$

Following the proof in Phase II, we focus on the dynamics about predictions. Due to the specificity of the first-order dynamics, the following lemma gives an second-order **autonomous** dynamics of predictions.

**Lemma F.3** (Second-order Autonomous Dynamics of predictions in Phase IV*)**.**
*If we consider the following two variables:*

$$\begin{cases} \mathcal{I}(t) := \kappa_2^2 \frac{m_-}{m} \frac{p}{1+p} e^{-f_+(t)}, \\ \mathcal{J}(t) := \kappa_2^2 \frac{m_-}{m} \frac{1}{1+p} e^{f_-(t)}, \end{cases}$$

*then the following autonomous dynamics of $\mathcal{U}(t)$ and $\mathcal{V}(t)$ hold in Phase IV* ($T_{\mathrm{III}} \leq t \leq T_{\mathrm{IV}}^*$):*

$$\begin{cases} \frac{\mathrm{d}\mathcal{I}(t)}{\mathrm{d}t} = \mathcal{I}(t)\mathcal{J}(t) \cos \Delta - \mathcal{I}^2(t) \left(1 + \frac{m_+}{m_-} \sin^2 \Delta\right), \\ \frac{\mathrm{d}\mathcal{J}(t)}{\mathrm{d}t} = \mathcal{I}(t)\mathcal{J}(t) \cos \Delta - \mathcal{J}^2(t). \end{cases}$$

*Proof of Lemma F.3.*
With the help of the first-order dynamics in Lemma F.2, the proof is straight-forward. $\square$

Lemma F.3 enlighten us that we only need to study the dynamics of $\mathcal{I}(t)$ and $\mathcal{J}(t)$ to study the dynamics in Phase IV*, where $\mathcal{I}(t), \mathcal{J}(t)$ satisfies the following autonomous dynamics:

$$\begin{cases} \frac{\mathrm{d}\mathcal{I}(t)}{\mathrm{d}t} = \mathcal{I}(t)\mathcal{J}(t) \cos \Delta - \mathcal{I}^2(t) \left(1 + \frac{m_+}{m_-} \sin^2 \Delta\right), \\ \frac{\mathrm{d}\mathcal{J}(t)}{\mathrm{d}t} = \mathcal{I}(t)\mathcal{J}(t) \cos \Delta - \mathcal{J}^2(t), \end{cases} \quad t \geq T_{\mathrm{III}}; \\ \begin{cases} \mathcal{I}(T_{\mathrm{III}}) = \kappa_2^2 \frac{m_-}{m} \frac{p}{1+p} e^{-f_+(T_{\mathrm{III}})}, \\ \mathcal{J}(T_{\mathrm{III}}) = \kappa_2^2 \frac{m_-}{m} \frac{1}{1+p} e^{f_-(T_{\mathrm{III}})}. \end{cases} \tag{31}$$

The next lemma studies the dynamics (31) for any $t \in [T_{\mathrm{III}}, +\infty)$.

**Lemma F.4** (Fine-grained analysis of the dynamics (31)).
*For the dynamics* (31)*, we have the following results:*

**(S1).** *Initialization.*

$$\mathcal{I}(T_{\mathrm{III}}) = \Theta\left(\kappa_2^2 p^{-\frac{1}{1-\alpha\cos\Delta}}\right), \quad \mathcal{J}(T_{\mathrm{III}}) = \Theta\left(\kappa_2^2 p^{-\frac{1}{1-\alpha\cos\Delta}}\right),$$

$$\mathcal{I}(T_{\mathrm{III}}) - \mathcal{J}(T_{\mathrm{III}})\cos\Delta = 0, \quad \mathcal{I}(T_{\mathrm{III}})\cos\Delta - \mathcal{J}(T_{\mathrm{III}}) = -\Theta\left(\kappa_2^2\Delta^2 p^{-\frac{1}{1-\alpha\cos\Delta}}\right).$$

**(S2).** *Fine-grained two-side bound for* $\mathcal{I}(t)/\mathcal{J}(t)$*.*

$$\frac{1+\cos\Delta}{1+\cos\Delta + \frac{m_+}{m_-}\sin^2\Delta} < \frac{\mathcal{I}(t)}{\mathcal{J}(t)} < \cos\Delta, \quad \forall t \in [T_{\mathrm{III}}, +\infty).$$

**(S3).** *The limit of* $\mathcal{I}(t)/\mathcal{J}(t)$*.*

$$\lim_{t\to\infty}\frac{\mathcal{I}(t)}{\mathcal{J}(t)} = \frac{1+\cos\Delta}{1+\cos\Delta + \frac{m_+}{m_-}\sin^2\Delta}.$$

**(S4).** *Tight estimate of* $\mathcal{I}(t)$ *and* $\mathcal{J}(t)$*.*

$$\mathcal{I}(t) = \Theta\left(\frac{1}{\frac{p^{\frac{1}{1-\alpha\cos\Delta}}}{\kappa_2^2} + \Delta^2(t-T_{\mathrm{III}})}\right), \quad \forall t \in [T_{\mathrm{III}}, +\infty);$$

$$\mathcal{J}(t) = \Theta\left(\frac{1}{\frac{p^{\frac{1}{1-\alpha\cos\Delta}}}{\kappa_2^2} + \Delta^2(t-T_{\mathrm{III}})}\right), \quad \forall t \in [T_{\mathrm{III}}, +\infty).$$

*Proof of Lemma F.4.*
For simplicity, in this proof, we denote

$$\epsilon := \frac{m_+}{m_-}\sin^2\Delta.$$

Step I. Preparation. Recalling Lemma E.6, we have:

$$\mathcal{I}(T_{\mathrm{III}}) = \Theta\left(\kappa_2^2 p^{-\frac{1}{1-\alpha\cos\Delta}}\right), \quad \mathcal{J}(T_{\mathrm{III}}) = \Theta\left(\kappa_2^2 p^{-\frac{1}{1-\alpha\cos\Delta}}\right),$$

$$\mathcal{I}(T_{\mathrm{III}}) - \mathcal{J}(T_{\mathrm{III}})\cos\Delta = 0, \quad \mathcal{I}(T_{\mathrm{III}})\cos\Delta - \mathcal{J}(T_{\mathrm{III}}) = -\Theta\left(\kappa_2^2\Delta^2 p^{-\frac{1}{1-\alpha\cos\Delta}}\right).$$

Step II. A rough estimate on $\mathcal{I}(t)$ and $\mathcal{J}(t)$. In this step, we aim to prove:

$$\mathcal{J}(t) > \mathcal{I}(t) > 0, \quad \mathcal{I}(t) + \mathcal{J}(t) \leq \mathcal{I}(T_{\mathrm{III}}) + \mathcal{J}(T_{\mathrm{III}}), \quad \forall t \in [T_{\mathrm{III}}, \infty).$$

First, from the definition of $\mathcal{I}(t)$ and $\mathcal{J}(t)$, we have $\mathcal{I}(t) > 0$ and $\mathcal{J}(t) > 0$.

Then we consider the dynamics of $\mathcal{I}(t) + \mathcal{J}(t)$. From

$$\frac{\mathrm{d}}{\mathrm{d}t}\left(\mathcal{I}(t) + \mathcal{J}(t)\right) = 2\mathcal{I}(t)\mathcal{J}(t)\cos\Delta - \mathcal{I}^2(t)\left(1+\epsilon\right) - \mathcal{J}^2(t)$$

$$= -\left(\mathcal{I}(t) - \mathcal{J}(t)\right)^2\cos\Delta - (1-\cos\Delta)\mathcal{J}^2(t) - \mathcal{I}^2(t)\left(1+\epsilon-\cos\Delta\right) < 0,$$

we have

$$\mathcal{I}(t) + \mathcal{J}(t) \leq \mathcal{I}(T_{\mathrm{III}}) + \mathcal{J}(T_{\mathrm{III}}), \quad \forall t \geq T_{\mathrm{III}}.$$

Then we consider the dynamics of $\mathcal{J}(t) - \mathcal{I}(t)$. We define the hitting time

$$\tau_{\mathcal{J}-\mathcal{I}} := \inf\left\{t \geq T_{\mathrm{III}} : \mathcal{J}(t) \leq \mathcal{I}(t)\right\}.$$

From Step I, we know $\mathcal{J}(T_{\mathrm{III}}) - \mathcal{I}(T_{\mathrm{III}}) = (1-\cos\Delta)\mathcal{J}(T_{\mathrm{III}}) > 0$. From the continuity, $\tau_{\mathcal{J}-\mathcal{I}}$ exists and $\tau_{\mathcal{J}-\mathcal{I}} > T_{\mathrm{III}}$.

For any $t \in [T_{\mathrm{III}}, \tau_{\mathcal{J}-\mathcal{I}})$, we have $\mathcal{J}(t) - \mathcal{I}(t) > 0$ and

$$\frac{\mathrm{d}}{\mathrm{d}t}\Big(\mathcal{J}(t) - \mathcal{I}(t)\Big) = -\mathcal{J}^2(t) + \mathcal{I}^2(t)\,(1+\epsilon) = -(\mathcal{J}(t) + \mathcal{I}(t))(\mathcal{J}(t) - \mathcal{I}(t)) + \epsilon \mathcal{I}^2(t)$$
$$> -(\mathcal{J}(t) + \mathcal{I}(t))(\mathcal{J}(t) - \mathcal{I}(t)) \geq -(\mathcal{J}(T_{\mathrm{III}}) + \mathcal{I}(T_{\mathrm{III}}))(\mathcal{J}(t) - \mathcal{I}(t)),$$

We consider the auxiliary ODE: $\frac{d}{dt}\mathcal{P}(t) = -(\mathcal{J}(T_{\mathrm{III}}) + \mathcal{I}(T_{\mathrm{III}}))\mathcal{P}(t)$, where $\mathcal{P}(T_{\mathrm{III}}) = \mathcal{J}(T_{\mathrm{III}}) - \mathcal{I}(T_{\mathrm{III}}) > 0$. From the Comparison Principle of ODEs, we have:

$$\mathcal{J}(t) - \mathcal{I}(t) \geq \mathcal{P}(t) = (\mathcal{J}(T_{\mathrm{III}}) - \mathcal{I}(T_{\mathrm{III}})) \exp\Big(-(\mathcal{J}(T_{\mathrm{III}}) + \mathcal{I}(T_{\mathrm{III}}))(t - T_{\mathrm{III}})\Big) > 0, \ \forall t \in [T_{\mathrm{I}}, \tau_{\mathcal{U}-\mathcal{V}}).$$

From the definition of $\tau_{\mathcal{J}-\mathcal{I}}$, we have proved

$$\tau_{\mathcal{J}-\mathcal{I}} = +\infty;$$
$$\mathcal{J}(t) > \mathcal{I}(t), \ \forall t \in [T_{\mathrm{III}}, +\infty).$$

Step III. A rough two-side bound for $\mathcal{I}(t)/\mathcal{J}(t)$.

In Step II, we have given a rough upper bound for $\mathcal{I}(t)/\mathcal{J}(t)$: $\mathcal{I}(t)/\mathcal{J}(t) < 1, \forall t \geq T_{\mathrm{III}}$. And we want to derive a lower bound for $\mathcal{I}(t)/\mathcal{J}(t)$ in this step. We aim to prove:

$$\mathcal{I}(t)/\mathcal{J}(t) > \frac{1 + \cos \Delta}{1 + \cos \Delta + \epsilon}, \quad \forall t \in [T_{\mathrm{III}}, +\infty).$$

First, we define the hitting time:

$$\tau^l_{\mathcal{I}/\mathcal{J}} := \inf\Big\{t \geq T_{\mathrm{III}} : \mathcal{I}(t) \leq \frac{1 + \cos \Delta}{1 + \cos \Delta + \epsilon}\mathcal{J}(t)\Big\}.$$

From Step I, we know

$$\mathcal{I}(T_{\mathrm{III}}) - \frac{1 + \cos \Delta}{1 + \cos \Delta + \epsilon}\mathcal{J}(T_{\mathrm{III}}) > \Big(\cos \Delta - \frac{1 + \cos \Delta}{1 + \cos \Delta + \epsilon}\Big)\mathcal{J}(T_{\mathrm{III}})$$

$$= \frac{\big(\frac{m_+}{m_-}\cos \Delta - 1\big)\sin^2 \Delta}{1 + \cos \Delta + \epsilon}\mathcal{J}(T_{\mathrm{III}}) \geq \frac{\big(\frac{\cos \Delta}{0.977} - 1\big)\sin^2 \Delta}{1 + \cos \Delta + \epsilon}\mathcal{J}(T_{\mathrm{III}})$$

$$\geq \frac{\big(\frac{0.980}{0.977} - 1\big)\sin^2 \Delta}{1 + \cos \Delta + \epsilon}\mathcal{J}(T_{\mathrm{III}}) > 0.$$

From the continuity, $\tau^l_{\mathcal{I}/\mathcal{J}}$ exists and $\tau^l_{\mathcal{I}/\mathcal{J}} > T_{\mathrm{III}}$.

For any $t \in [T_{\mathrm{III}}, \tau^l_{\mathcal{I}/\mathcal{J}})$, we have $\mathcal{I}(t) - \frac{1+\cos \Delta}{1+\cos \Delta+\epsilon}\mathcal{J}(t) > 0$ and

$$\frac{\mathrm{d}}{\mathrm{d}t}\Big(\mathcal{I}(t) - \frac{1 + \cos \Delta}{1 + \cos \Delta + \epsilon}\mathcal{J}(t)\Big)$$

$$= \mathcal{I}(t)\mathcal{J}(t)\cos \Delta\Big(1 - \frac{1 + \cos \Delta}{1 + \cos \Delta + \epsilon}\Big) - (1 + \epsilon)\mathcal{I}^2(t) + \frac{1 + \cos \Delta}{1 + \cos \Delta + \epsilon}\mathcal{J}^2(t)$$

$$= -\Big(\mathcal{I}(t) - \frac{1 + \cos \Delta}{1 + \cos \Delta + \epsilon}\mathcal{J}(t)\Big)((1 + \epsilon)\mathcal{I}(t) + \mathcal{J}(t))$$

$$> -(1 + \epsilon)\Big(\mathcal{I}(t) - \frac{1 + \cos \Delta}{1 + \cos \Delta + \epsilon}\mathcal{J}(t)\Big)(\mathcal{I}(t) + \mathcal{J}(t))$$

$$\geq -(1 + \epsilon)(\mathcal{I}(T_{\mathrm{III}}) + \mathcal{J}(T_{\mathrm{III}}))\Big(\mathcal{I}(t) - \frac{1 + \cos \Delta}{1 + \cos \Delta + \epsilon}\mathcal{J}(t)\Big).$$

We consider the auxiliary ODE: $\frac{d}{dt}\mathcal{Q}(t) = -(1 + \epsilon)(\mathcal{I}(T_{\mathrm{III}}) + \mathcal{J}(T_{\mathrm{III}}))\mathcal{Q}(t)$, where $\mathcal{Q}(T_{\mathrm{III}}) = \mathcal{I}(T_{\mathrm{III}}) - \frac{1+\cos \Delta}{1+\cos \Delta+\epsilon}\mathcal{J}(T_{\mathrm{III}}) > 0$. From the Comparison Principle of ODEs, we have:

$$\mathcal{I}(t) - \frac{1 + \cos \Delta}{1 + \cos \Delta + \epsilon}\mathcal{J}(t) \geq \mathcal{Q}(t)$$

$$= \left( \mathcal{I}(T_{\mathrm{III}}) - \frac{1 + \cos \Delta}{1 + \cos \Delta + \epsilon} \mathcal{J}(T_{\mathrm{III}}) \right) \exp \left( - (1 + \epsilon)(\mathcal{I}(T_{\mathrm{III}}) + \mathcal{J}(T_{\mathrm{III}}))(t - T_{\mathrm{III}}) \right) > 0, \quad \forall t \in [T_{\mathrm{III}}, \tau^l_{\mathcal{I}/\mathcal{J}}).$$

From the definition of $\tau_{\mathcal{J} - \mathcal{I}}$, we have proved

$$\tau^l_{\mathcal{I}/\mathcal{J}} = +\infty;$$

$$\mathcal{I}(t)/\mathcal{J}(t) > \frac{1 + \cos \Delta}{1 + \cos \Delta + \epsilon}, \quad \forall t \in [T_{\mathrm{III}}, +\infty).$$

Hence, we obtain the two-side bound for $\mathcal{I}(t)/\mathcal{J}(t)$:

$$\frac{1 + \cos \Delta}{1 + \cos \Delta + \epsilon} < \frac{\mathcal{I}(t)}{\mathcal{J}(t)} < 1, \quad \forall t \in [T_{\mathrm{III}}, +\infty).$$

Step IV. $\mathcal{I}(t) \cos \Delta - \mathcal{J}(t)$ and $\mathcal{I}(t) - \mathcal{J}(t) \cos \Delta$ are both negative.

The estimate on $\mathcal{I}(t) \cos \Delta - \mathcal{J}(t)$ is straight-forward:

$$\mathcal{I}(t) \cos \Delta - \mathcal{J}(t) < \mathcal{I}(t) \cos \Delta - \mathcal{I}(t) < 0.$$

As for $\mathcal{I}(t) - \mathcal{J}(t) \cos \Delta$, we will actually prove a tighter upper bound:

$$\frac{\mathcal{I}(t)}{\mathcal{J}(t)} < \cos \Delta.$$

We need to do finer analysis using the specific dynamics (31). First, we define the following hitting time:

Define the following hitting time

$$\tau^u_{\mathcal{I}/\mathcal{J}} := \inf \left\{ t > T_{\mathrm{I}} : \mathcal{I}(t) - \mathcal{J}(t) \cos \Delta \geq 0 \right\}.$$

From $\mathcal{I}(T_{\mathrm{III}}) - \mathcal{J}(T_{\mathrm{III}}) \cos \Delta = 0$, $\frac{\mathrm{d}}{\mathrm{d}t}(\mathcal{I}(T_{\mathrm{III}}) - \mathcal{J}(T_{\mathrm{III}}) \cos \Delta) < 0$ and the continuity, we know $\tau^u_{\mathcal{I}/\mathcal{J}}$ exists and $\tau^u_{\mathcal{I}/\mathcal{J}} > T_{\mathrm{III}}$.

Recalling $\mathcal{I}(t) \cos \Delta - \mathcal{J}(t) < 0$, we have

$$\frac{\mathrm{d}}{\mathrm{d}t} \mathcal{J}(t) = \mathcal{J}(t) \left( \mathcal{I}(t) \cos \Delta - \mathcal{J}(t) \right) < 0, \quad \forall t \geq T_{\mathrm{III}}.$$

So we can consider the following dynamics for $t \in [T_{\mathrm{III}}, \tau^u_{\mathcal{I}/\mathcal{J}}]$:

$$\frac{\mathrm{d}\mathcal{I}}{\mathrm{d}\mathcal{J}} = \frac{\mathcal{I}\mathcal{J} \cos \Delta - \mathcal{I}^2(1 + \epsilon)}{\mathcal{I}\mathcal{J} \cos \Delta - \mathcal{J}^2} = \frac{\frac{\mathcal{I}}{\mathcal{J}} \cos \Delta - \left( \frac{\mathcal{I}}{\mathcal{J}} \right)^2 (1 + \epsilon)}{\frac{\mathcal{I}}{\mathcal{J}} \cos \Delta - 1}.$$

If we define $\mathcal{Z}(t) := \frac{\mathcal{I}(t)}{\mathcal{J}(t)}$, then we have $\mathrm{d}\mathcal{I} = \mathcal{Z}\mathrm{d}\mathcal{J} + \mathcal{J}\mathrm{d}\mathcal{Z}$.

The dynamics above can be transformed to:

$$\mathcal{J} \frac{\mathrm{d}\mathcal{Z}}{\mathrm{d}\mathcal{J}} = \frac{\mathcal{Z} \cos \Delta - \mathcal{Z}^2(1 + \epsilon)}{\mathcal{Z} \cos \Delta - 1} - \mathcal{Z} = \frac{\mathcal{Z}(1 + \cos \Delta) - \mathcal{Z}^2(1 + \cos \Delta + \epsilon)}{\mathcal{Z} \cos \Delta - 1}.$$

Recalling the result in Step III, we have $(1 + \cos \Delta) - (1 + \cos \Delta + \epsilon)\mathcal{Z}(t) < 0$ holds for any $t \geq T_{\mathrm{III}}$. So the dynamics is equal to:

$$\frac{\mathrm{d}\mathcal{J}}{\mathcal{J}} = \left( \frac{\mathcal{Z} \cos \Delta - 1}{\mathcal{Z}(1 + \cos \Delta) - \mathcal{Z}^2(1 + \cos \Delta + \epsilon)} \right) \mathrm{d}\mathcal{Z}$$

$$= -\frac{1}{1 + \cos \Delta} \left( \frac{1}{\mathcal{Z}} + \frac{\sin^2 \Delta + \epsilon}{1 + \cos \Delta - \mathcal{Z}(1 + \cos \Delta + \epsilon)} \right) \mathrm{d}\mathcal{Z}.$$

Integrating this equation from $T_{\mathrm{III}}$ to $t \in [T_{\mathrm{III}}, \tau^u_{\mathcal{I}/\mathcal{J}})$, we have:

$$\log \left( \frac{\mathcal{J}(t)}{\mathcal{J}(T_{\mathrm{III}})} \right) = -\frac{1}{1 + \cos \Delta} \log \left( \frac{\mathcal{Z}(t)}{\mathcal{Z}(T_{\mathrm{III}})} \right)$$

$$+ \frac{\sin^2 \Delta + \epsilon}{(1 + \cos \Delta + \epsilon)(1 + \cos \Delta)} \log \left( \frac{(1 + \cos \Delta + \epsilon)\mathcal{Z}(t) - (1 + \cos \Delta)}{(1 + \cos \Delta + \epsilon)\mathcal{Z}(T_{\mathrm{I}}) - (1 + \cos \Delta)} \right).$$

$$(32)$$

If we assume $\tau^u_{\mathcal{I}/\mathcal{J}} < +\infty$, the continuity gives us

$$\lim_{t\to\tau^u_{\mathcal{I}/\mathcal{J}}{}^-} \mathcal{Z}(t) = \lim_{t\to\tau^u_{\mathcal{I}/\mathcal{J}}{}^-} \mathcal{I}(t)/\mathcal{J}(t) = \cos\Delta = \mathcal{I}(T_{\mathrm{III}})/\mathcal{J}(T_{\mathrm{III}}) = \mathcal{Z}(T_{\mathrm{III}}).$$

Then letting $t \to \tau^u_{\mathcal{I}/\mathcal{J}}{}^-$ in (32), we have

$$\lim_{t\to\tau^u_{\mathcal{I}/\mathcal{J}}{}^-} \log\left(\frac{\mathcal{J}(t)}{\mathcal{J}(T_{\mathrm{III}})}\right) = 0 + 0 = 0,$$

which means $\mathcal{J}(\tau^u_{\mathcal{I}/\mathcal{J}}) = \mathcal{J}(T_{\mathrm{III}})$.

But on the other hand, we have:

$$\mathcal{J}(\tau^u_{\mathcal{I}/\mathcal{J}}) = \mathcal{J}(T_{\mathrm{III}}) + \int_{T_{\mathrm{III}}}^{\tau^u_{\mathcal{I}/\mathcal{J}}} (\mathcal{I}(t)\mathcal{J}(t)\cos\Delta - \mathcal{J}^2(t))\mathrm{d}t$$

$$=\mathcal{J}(T_{\mathrm{III}}) + \int_{T_{\mathrm{III}}}^{\tau^u_{\mathcal{I}/\mathcal{J}}} \mathcal{J}(t)(\mathcal{I}(t)\cos\Delta - \mathcal{J}(t))\mathrm{d}t$$

$$<\mathcal{J}(T_{\mathrm{III}}) + (\cos\Delta - 1)\int_{T_{\mathrm{III}}}^{\tau^u_{\mathcal{I}/\mathcal{J}}} \mathcal{J}(t)\mathcal{I}(t)\mathrm{d}t < \mathcal{J}(T_{\mathrm{III}}),$$

which leads to a contradiction. Hence, we have proved

$$\tau^u_{\mathcal{I}/\mathcal{J}} = +\infty;$$
$$\mathcal{I}(t) - \mathcal{J}(t)\cos\Delta < 0, \quad \forall t \in [T_{\mathrm{III}}, +\infty).$$

Moreover, we obtain a sharper two-side bound for $\mathcal{I}(t)/\mathcal{J}(t)$:

$$\frac{1+\cos\Delta}{1+\cos\Delta+\epsilon} < \frac{\mathcal{I}(t)}{\mathcal{J}(t)} < \cos\Delta, \quad \forall t \in [T_{\mathrm{III}}, +\infty). \tag{33}$$

Step V. Tight bound for $\mathcal{I}(t)$ and $\mathcal{J}(t)$.

In this step, we aim to give a tight bound for $\mathcal{I}(t) + \mathcal{J}(t)$. With the help of the two-side bound (33), we have

$$\frac{\mathrm{d}}{\mathrm{d}t}\Big(\mathcal{I}(t) + \mathcal{J}(t)\Big) = 2\mathcal{I}(t)\mathcal{J}(t)\cos\Delta - \mathcal{I}^2(t)\,(1+\epsilon) - \mathcal{J}^2(t)$$
$$= -(\mathcal{I}(t) - \mathcal{J}(t))^2\cos\Delta - (1-\cos\Delta)\mathcal{J}^2(t) - \mathcal{I}^2(t)\,(1+\epsilon-\cos\Delta)$$
$$< -(1-\cos\Delta)\left(\mathcal{J}^2(t) + \mathcal{I}^2(t)\right) < -\frac{(1-\cos\Delta)\,(\mathcal{I}(t)+\mathcal{J}(t))^2}{2} < -\frac{\Delta^2}{6}\,(\mathcal{I}(t)+\mathcal{J}(t))^2,$$

$$\frac{\mathrm{d}}{\mathrm{d}t}\Big(\mathcal{I}(t) + \mathcal{J}(t)\Big) = 2\mathcal{I}(t)\mathcal{J}(t)\cos\Delta - \mathcal{I}^2(t)\,(1+\epsilon) - \mathcal{J}^2(t)$$
$$= -(\mathcal{I}(t) - \mathcal{J}(t))^2\cos\Delta - (1-\cos\Delta)\mathcal{J}^2(t) - \mathcal{I}^2(t)\,(1+\epsilon-\cos\Delta)$$
$$> -\left(1 - \frac{1+\cos\Delta}{1+\cos\Delta+\epsilon}\right)^2\mathcal{J}^2(t) - (1-\cos\Delta)\mathcal{J}^2(t) - \mathcal{I}^2(t)\,(1+\epsilon-\cos\Delta)$$
$$> -(1+\epsilon-\cos\Delta)\left(\mathcal{I}^2(t)+\mathcal{J}^2(t)\right) > -\left(\frac{2}{3}+\frac{m_+}{m_-}\right)\Delta^2\,(\mathcal{I}(t)+\mathcal{J}(t))^2 > -2\Delta^2\,(\mathcal{I}(t)+\mathcal{J}(t))^2.$$

For the first inequality, we consider the auxiliary ODE: $\frac{d}{dt}\mathcal{P}(t) = -\frac{\Delta^2}{6}\mathcal{P}^2(t)$, where $\mathcal{P}(T_{\mathrm{III}}) = \mathcal{I}(T_{\mathrm{III}}) + \mathcal{J}(T_{\mathrm{III}}) > 0$. From the Comparison Principle of ODEs, we have:

$$\mathcal{I}(t) + \mathcal{J}(t) \le \mathcal{P}(t) = \frac{1}{\frac{1}{\mathcal{I}(T_{\mathrm{III}})+\mathcal{J}(T_{\mathrm{III}})} + \frac{\Delta^2}{6}(t - T_{\mathrm{III}})}, \quad \forall t \ge T_{\mathrm{III}}.$$

In the same way, we can obtain the lower bound:

$$\mathcal{I}(t) + \mathcal{J}(t) \ge \frac{1}{\frac{1}{\mathcal{I}(T_{\mathrm{III}})+\mathcal{J}(T_{\mathrm{III}})} + 2\Delta^2(t - T_{\mathrm{III}})}, \quad \forall t \ge T_{\mathrm{III}}.$$

Recalling Step I, we have $\frac{1}{\mathcal{I}(T_{\mathrm{III}})+\mathcal{J}(T_{\mathrm{III}})} = \Theta\left(p^{\frac{1}{1-\alpha\cos\Delta}}/\kappa_2^2\right)$. Hence, we obtain the tight bound:

$$\mathcal{I}(t) + \mathcal{J}(t) = \Theta\left(\frac{1}{\frac{p^{\frac{1}{1-\alpha\cos\Delta}}}{\kappa_2^2} + \Delta^2(t-T_{\mathrm{III}})}\right), \quad \forall t \in [T_{\mathrm{III}}, +\infty).$$

Taking (33) into the equation above, we have:

$$\mathcal{I}(t) = \Theta\left(\frac{1}{\frac{p^{\frac{1}{1-\alpha\cos\Delta}}}{\kappa_2^2} + \Delta^2(t-T_{\mathrm{III}})}\right), \quad \forall t \in [T_{\mathrm{III}}, +\infty);$$

$$\mathcal{J}(t) = \Theta\left(\frac{1}{\frac{p^{\frac{1}{1-\alpha\cos\Delta}}}{\kappa_2^2} + \Delta^2(t-T_{\mathrm{III}})}\right), \quad \forall t \in [T_{\mathrm{III}}, +\infty).$$

Step VI. The limit of $\mathcal{I}(t)/\mathcal{J}(t)$.

By Step V and the proof of Step IV, we know $\lim_{t\to\infty}\mathcal{J}(t) = 0$ and $\frac{\mathrm{d}}{\mathrm{d}t}\mathcal{J}(t) < 0$ holds for any $t > T_{\mathrm{III}}$.

Then for any $\epsilon' > 0$, there exists $T' > T_{\mathrm{III}}$ such that

$$\log\left(\frac{\mathcal{J}(t)}{\mathcal{J}(T_{\mathrm{III}})}\right) < \frac{1000}{\Delta^2}\log(1000\epsilon\Delta^2), \quad \forall t > T'.$$

Taking it into (32), we obtain that for any $t > T'$,

$$0 < (1 + \cos\Delta + \epsilon)\mathcal{Z}(t) - (1 + \cos\Delta) < \epsilon'.$$

By the definition of the limit, we get

$$\lim_{t\to\infty}\frac{\mathcal{I}(t)}{\mathcal{J}(t)} = \frac{1+\cos\Delta}{1+\cos\Delta+\epsilon} = \frac{1+\cos\Delta}{1+\cos\Delta+\frac{m_+}{m_-}\sin^2\Delta}.$$

$\square$

**Lemma F.5** (Time and prediction estimate)**.**
**(S1).** *For any $t \in (T_{\mathrm{III}}, +\infty)$*

$$pe^{-f_+(t)} = \Theta\left(\frac{1}{p^{\frac{1}{1-\alpha\cos\Delta}} + \kappa_2^2\Delta^2(t-T_{\mathrm{III}})}\right), \quad e^{f_-(t)} = \Theta\left(\frac{1}{p^{\frac{1}{1-\alpha\cos\Delta}} + \kappa_2^2\Delta^2(t-T_{\mathrm{III}})}\right);$$

$$\mathcal{L}(\boldsymbol{\theta}(t)) = \Theta\left(\frac{1}{p^{\frac{1}{1-\alpha\cos\Delta}} + \kappa_2^2\Delta^2(t-T_{\mathrm{III}})}\right).$$

**(S2).** *For any $t \in (T_{\mathrm{III}}, +\infty)$,*

$$\frac{1+\cos\Delta}{1+\cos\Delta+\frac{m_+}{m_-}\sin^2\Delta} < pe^{-(f_+(t)+f_-(t))} < \cos\Delta.$$

*Moreover, $pe^{-(f_+(T_{\mathrm{III}})+f_-(T_{\mathrm{III}}))} = \cos\Delta$ and $\lim_{t\to\infty} pe^{-(f_+(t)+f_-(t))} = \frac{1+\cos\Delta}{1+\cos\Delta+\frac{m_+}{m_-}\sin^2\Delta}$.*

**(S3).** *For any $t \in (T_{\mathrm{III}}, +\infty)$,*

$$\langle\boldsymbol{b}_k(t), \boldsymbol{x}_+\rangle > 0, \ \langle\boldsymbol{b}_k(t), \boldsymbol{x}_-\rangle = 0, \ k \in \mathcal{K}_+;$$
$$\langle\boldsymbol{b}_k(t), \boldsymbol{x}_+\rangle > 0, \ \langle\boldsymbol{b}_k(t), \boldsymbol{x}_-\rangle > 0, \ k \in \mathcal{K}_-.$$

**(S4) (Time).**

$$T_{\mathrm{IV}} = T_{\mathrm{IV}}^* = +\infty.$$

*Proof of Lemma F.5.*
Notice the relationships: $pe^{-f_+(t)} = \kappa_2^2 \frac{m_-}{m} \frac{\mathcal{I}(t)}{1+p}$, $e^{f_-(t)} = \kappa_2^2 \frac{m_-}{m} \frac{\mathcal{J}(t)}{1+p}$ and $pe^{-(f_+(t)+f_-(t))} = \mathcal{I}(t)/\mathcal{J}(t)$. Then Lemma F.4 implies that $T_{\text{IV}}^* = +\infty$. Recalling the dynamics in Lemma F.1, then lemma (S3)(S4) hold. Then using Lemma F.4 again, we obtain (S1)(S2). $\square$

*Proof of Theorem 4.9.*
Theorem 4.9 (S1), (S2), and (S3) are obtained in Lemma F.5 (S4), (S3), and (S1), respectively. $\square$

## F.2 Asymptotic Directional Convergence

In this section, we will study the final convergence direction in our setting. It mainly depends on our prior fine-grained analysis of the training dynamics in Phase IV and the following result about the final convergence direction at the end of training.

**Lemma F.6.** *Let $f(\cdot; \boldsymbol{\theta})$ be a homogeneous neural network parameterized by $\boldsymbol{\theta}$. Consider minimizing the exponential loss over a binary classification dataset $\{(\boldsymbol{x}_i, y_i)\}_{i=1}^n$ ($\|\boldsymbol{x}_i\|_2 \leq 1, y_i \in \{\pm 1\}$) using Gradient Flow. Assume that there exists time $t_0$ such that $\mathcal{L}(\boldsymbol{\theta}(t_0)) < \frac{1}{n}$. Then,*

*(I) (Paraphrased from (Lyu and Li, 2019; Ji and Telgarsky, 2020)). $\boldsymbol{\theta}(t)$ converges in direction to a KKT point (Definition G.3) of the following maximum margin problem:*

$$\min : \frac{1}{2} \|\boldsymbol{\theta}\|_2^2$$
$$\text{s.t. } y_i f(\boldsymbol{x}_i; \boldsymbol{\theta}) \geq 1.$$

*(II) (Lyu and Li, 2019; Ji and Telgarsky, 2020)). $\|\boldsymbol{\theta}(t)\|_2 \to \infty$ and $\mathcal{L}(\boldsymbol{\theta}(t)) \to 0$.*

*(III) (Ji and Telgarsky (2020)). $-\nabla \mathcal{L}(\boldsymbol{\theta}(t))$ and $\boldsymbol{\theta}(t)$ converge to the same direction, meaning the angle between $\boldsymbol{\theta}(t)$ and $-\nabla \mathcal{L}(\boldsymbol{\theta}(t))$ converges to 0.*

**Lemma F.7** (Final Convergence Direction). *The limit $\lim\limits_{t \to +\infty} \frac{\boldsymbol{\theta}(t)}{\|\boldsymbol{\theta}(t)\|_2}$ exists, and denoted by $\overline{\boldsymbol{\theta}} = (\overline{\boldsymbol{b}}_1^\top, \cdots, \overline{\boldsymbol{b}}_m^\top)^\top \in \mathbb{S}^{md-1}$, then it satisfies*

$$\overline{\boldsymbol{b}}_k = \boldsymbol{0}, \quad \forall k \notin \mathcal{K}_+ \cup \mathcal{K}_-;$$
$$\overline{\boldsymbol{b}}_k = C\Big(\boldsymbol{x}_+ - \boldsymbol{x}_- \cos \Delta\Big), \quad \forall k \in \mathcal{K}_+;$$
$$\overline{\boldsymbol{b}}_k = C\left(\left(1 + \frac{m_+ \sin^2 \Delta}{m_-(1 + \cos \Delta)}\right)\boldsymbol{x}_- - \boldsymbol{x}_+\right), \quad \forall k \in \mathcal{K}_-;$$

*where $C > 0$ is a scaling constant such that $\|\overline{\boldsymbol{\theta}}\|_2 = 1$. Moreover, $f(\boldsymbol{x}_+; \overline{\boldsymbol{\theta}}) = -f(\boldsymbol{x}_-; \overline{\boldsymbol{\theta}}) > 0$.*

*Proof of Lemma F.7.*
Let $\overline{\boldsymbol{\theta}} = (\overline{\boldsymbol{b}}_1^\top, \cdots, \overline{\boldsymbol{b}}_m^\top)^\top \in \mathbb{S}^{md-1}$ be the limits point of $\left\{\frac{\boldsymbol{\theta}(t)}{\|\boldsymbol{\theta}(t)\|_2} : t \geq t_0\right\}$. From Lemma F.6 (I), we know that there exists a scaling factor $\alpha > 0$ such that $\alpha \overline{\boldsymbol{\theta}}$ satisfies KKT conditions (Definition G.3) of the maximum-margin problem

$$\min : \frac{1}{2} \|\boldsymbol{\theta}\|_2^2 \tag{34}$$
$$\text{s.t. } f(\boldsymbol{x}_+; \boldsymbol{\theta}) \geq 1, \ f(\boldsymbol{x}_-; \boldsymbol{\theta}) \leq -1.$$

For simplicity, we denote $\boldsymbol{\theta}^* := \alpha \overline{\boldsymbol{\theta}}$, where

$$\boldsymbol{\theta}^* = (\boldsymbol{b}_1^{*\top}, \cdots, \boldsymbol{b}_m^{*\top})^\top.$$

Moreover, let $\lambda_+^*, \lambda_-^* \geq 0$ be the corresponding Lagrange multipliers (with respect to $\boldsymbol{\theta}^*$) in Definition G.3.

Step I. The rough direction of each neuron.

Recalling the training dynamics about the dead neurons in Theorem 4.1 (S3),

$$\boldsymbol{b}_k(t) \equiv \boldsymbol{b}_k(T_{\text{I}}), \ \langle \boldsymbol{b}_k(t), \boldsymbol{x}_+ \rangle \leq 0, \ \langle \boldsymbol{b}_k(t), \boldsymbol{x}_- \rangle \leq 0, \ k \in [m/2] - \mathcal{K}_+;$$

$$\boldsymbol{b}_k(t) \equiv \boldsymbol{b}_k(T_{\mathrm{I}}), \ \langle \boldsymbol{b}_k(t), \boldsymbol{x}_+ \rangle \le 0, \ \langle \boldsymbol{b}_k(t), \boldsymbol{x}_- \rangle \le 0, \ k \in [m] - [m/2] - \mathcal{K}_-.$$

Noticing Lemma F.6 (II) or Lemma F.5 (S1), $\|\boldsymbol{\theta}(t)\|_2 \to \infty$, so

$$\boldsymbol{b}_k^* = \boldsymbol{0}, \quad \forall k \notin \mathcal{K}_+ \cup \mathcal{K}_-.$$

Then we only need to focus on $\boldsymbol{\theta}_k^*$ for $k \in \mathcal{K}_+ \cup \mathcal{K}_-$.

Recalling in Lemma F.5 (S3), for any $t > T_{\mathrm{IV}}$,

$$\langle \boldsymbol{b}_k(t), \boldsymbol{x}_+ \rangle > 0, \ \langle \boldsymbol{b}_k(t), \boldsymbol{x}_- \rangle = 0, \ k \in \mathcal{K}_+;$$
$$\langle \boldsymbol{b}_k(t), \boldsymbol{x}_+ \rangle > 0, \ \langle \boldsymbol{b}_k(t), \boldsymbol{x}_- \rangle > 0, \ k \in \mathcal{K}_-.$$

then we have

$$\langle \boldsymbol{b}_k^*, \boldsymbol{x}_+ \rangle \ge 0, \ \langle \boldsymbol{b}_k^*, \boldsymbol{x}_- \rangle = 0, \quad k \in \mathcal{K}_+;$$
$$\langle \boldsymbol{b}_k^*, \boldsymbol{x}_+ \rangle \ge 0, \ \langle \boldsymbol{b}_k^*, \boldsymbol{x}_- \rangle \ge 0, \quad k \in \mathcal{K}_-.$$

Moreover,

$$f(\boldsymbol{x}_+; \boldsymbol{\theta}^*) = \frac{\kappa_2}{\sqrt{m}} \Big( \sum_{k \in \mathcal{K}_+} \sigma\left(\langle \boldsymbol{b}_k^*, \boldsymbol{x}_+ \rangle\right) - \sum_{k \in \mathcal{K}_-} \sigma\left(\langle \boldsymbol{b}_k^*, \boldsymbol{x}_+ \rangle\right) \Big),$$

$$f(\boldsymbol{x}_-; \boldsymbol{\theta}^*) = \frac{\kappa_2}{\sqrt{m}} \Big( \sum_{k \in \mathcal{K}_+} \sigma\left(\langle \boldsymbol{b}_k^*, \boldsymbol{x}_- \rangle\right) - \sum_{k \in \mathcal{K}_-} \sigma\left(\langle \boldsymbol{b}_k^*, \boldsymbol{x}_- \rangle\right) \Big) = -\frac{\kappa_2}{\sqrt{m}} \sum_{k \in \mathcal{K}_-} \sigma\left(\langle \boldsymbol{b}_k^*, \boldsymbol{x}_- \rangle\right).$$

Step II. Determine the direction of the neurons $k \in \mathcal{K}_+$.

Since $\boldsymbol{\theta}^*$ is a possible point, $f(\boldsymbol{x}_+; \boldsymbol{\theta}^*) \ge 1$, which gives us

$$\sum_{k \in \mathcal{K}_+} \sigma(\langle \boldsymbol{b}_k^*, \boldsymbol{x}_+ \rangle) \ge \frac{\sqrt{m}}{\kappa_2} > 0.$$

Hence, there exists $k_1 \in \mathcal{K}_+$, s.t. $\langle \boldsymbol{b}_{k_1}^*, \boldsymbol{x}_+ \rangle > 0$ strictly.

Then we study the neuron $k_2 \in \mathcal{K}_+$ ($k_2 \ne k_1$). Lemma F.1 and Lemma F.5 (S3) give use that

$$\langle \boldsymbol{b}_{k_1}(t), \boldsymbol{x}_+ \rangle = \langle \boldsymbol{b}_{k_1}(T_{\mathrm{IV}}), \boldsymbol{x}_+ \rangle + \int_{T_{\mathrm{IV}}}^{t} \frac{\kappa_2 p e^{-f_+(t)}}{\sqrt{m}(1+p)} \sin^2 \Delta \mathrm{d}t$$

$$= \langle \boldsymbol{b}_{k_2}(T_{\mathrm{IV}}), \boldsymbol{x}_+ \rangle + \int_{T_{\mathrm{IV}}}^{t} \frac{\kappa_2 p e^{-f_+(t)}}{\sqrt{m}(1+p)} \sin^2 \Delta \mathrm{d}t + \Big( \langle \boldsymbol{b}_{k_1}(T_{\mathrm{IV}}), \boldsymbol{x}_+ \rangle - \langle \boldsymbol{b}_{k_2}(T_{\mathrm{IV}}), \boldsymbol{x}_+ \rangle \Big)$$

$$= \langle \boldsymbol{b}_{k_2}(t), \boldsymbol{x}_+ \rangle + \Big( \langle \boldsymbol{b}_{k_1}(T_{\mathrm{IV}}), \boldsymbol{x}_+ \rangle - \langle \boldsymbol{b}_{k_2}(T_{\mathrm{IV}}), \boldsymbol{x}_+ \rangle \Big).$$

Multiplying the above formula by $c / \|\boldsymbol{\theta}(t)\|_2$ and taking $t$ go to infinity, we obtain

$$\langle \boldsymbol{b}_{k_1}^*, \boldsymbol{x}_+ \rangle = \langle \boldsymbol{b}_{k_2}^*, \boldsymbol{x}_+ \rangle > 0.$$

Due to the arbitrariness of $k_2$, we know

$$\boldsymbol{b}_k^* \ne \boldsymbol{0}, \quad \langle \boldsymbol{b}_k^*, \boldsymbol{x}_+ \rangle > 0, \quad \langle \boldsymbol{b}_k^*, \boldsymbol{x}_- \rangle = 0, \quad \forall k \in \mathcal{K}_+.$$

Then we can write the KKT condition about the gradient of $\boldsymbol{b}_k^*$ ($k \in \mathcal{K}_+$) of Problem (34):

$$\boldsymbol{0} \in \boldsymbol{b}_k^* - \lambda_+^* \frac{\kappa_2}{\sqrt{m}} \boldsymbol{x}_+ + \lambda_-^* \frac{\kappa_2}{\sqrt{m}} \partial^\circ \sigma(0) \boldsymbol{x}_-.$$

It is clear that $\boldsymbol{b}_k^* \in \mathrm{span}\{\boldsymbol{x}_+, \boldsymbol{x}_-\}$. Then combining two formulations above, we obtain:

$$\boldsymbol{b}_k^* = \lambda_+^* \frac{\kappa_2}{\sqrt{m}} \left( \boldsymbol{x}_+ - \boldsymbol{x}_- \cos \Delta \right), \quad \forall k \in \mathcal{K}_+.$$

Step III. Determine the direction of the neurons $k \in \mathcal{K}_-$.

Since $\boldsymbol{\theta}^*$ is a possible point, $f(\boldsymbol{x}_-; \boldsymbol{\theta}^*) \leq -1$, which gives us

$$\sum_{k \in \mathcal{K}_-} \sigma(\langle \boldsymbol{b}_k^*, \boldsymbol{x}_- \rangle) \geq \frac{\sqrt{m}}{\kappa_2} > 0.$$

Hence, there exists $k_1 \in \mathcal{K}_-$, s.t. $\langle \boldsymbol{b}_{k_1}^*, \boldsymbol{x}_- \rangle > 0$ strictly.

Then we study the neuron $k_2 \in \mathcal{K}_-$ ($k_2 \neq k_1$). Lemma F.1 and Lemma F.5 (S3) give use that

$$\langle \boldsymbol{b}_{k_1}(t), \boldsymbol{x}_- \rangle = \langle \boldsymbol{b}_{k_1}(T_{\mathrm{IV}}), \boldsymbol{x}_+ \rangle + \int_{T_{\mathrm{IV}}}^t \frac{\kappa_2}{\sqrt{m}} \left( \frac{1}{1+p} e^{f_-(t)} - \frac{p}{1+p} e^{-f_+(t)} \cos \Delta \right) \mathrm{d}t$$

$$= \langle \boldsymbol{b}_{k_2}(T_{\mathrm{IV}}), \boldsymbol{x}_- \rangle + \int_{T_{\mathrm{IV}}}^t \frac{\kappa_2}{\sqrt{m}} \left( \frac{1}{1+p} e^{f_-(t)} - \frac{p}{1+p} e^{-f_+(t)} \cos \Delta \right) \mathrm{d}t + \left( \langle \boldsymbol{b}_{k_1}(T_{\mathrm{IV}}), \boldsymbol{x}_- \rangle - \langle \boldsymbol{b}_{k_2}(T_{\mathrm{IV}}), \boldsymbol{x}_- \rangle \right)$$

$$= \langle \boldsymbol{b}_{k_2}(t), \boldsymbol{x}_- \rangle + \left( \langle \boldsymbol{b}_{k_1}(T_{\mathrm{IV}}), \boldsymbol{x}_- \rangle - \langle \boldsymbol{b}_{k_2}(T_{\mathrm{IV}}), \boldsymbol{x}_- \rangle \right).$$

Multiplying the above formula by $c/\|\boldsymbol{\theta}(t)\|_2$ and taking $t$ go to infinity, we obtain

$$\langle \boldsymbol{b}_{k_1}^*, \boldsymbol{x}_- \rangle = \langle \boldsymbol{b}_{k_2}^*, \boldsymbol{x}_- \rangle > 0.$$

Due to the arbitrariness of $k_2$, we know

$$\boldsymbol{b}_k^* \neq \boldsymbol{0}, \quad \langle \boldsymbol{b}_k^*, \boldsymbol{x}_- \rangle > 0, \quad \langle \boldsymbol{b}_k^*, \boldsymbol{x}_+ \rangle \geq 0, \quad \forall k \in \mathcal{K}_-.$$

The next difficulty in this step is to determine whether $\langle \boldsymbol{b}_k^*, \boldsymbol{x}_+ \rangle$ can be 0. To prove this, we will use our fine-grained analysis of training dynamics (Lemma F.5) and Lemma F.6 (III).

Let $k \in \mathcal{K}_-$. Recalling the dynamics of $\boldsymbol{b}_k(t)$ in Lemma F.1, we know

$$-\frac{\partial \mathcal{L}(\boldsymbol{\theta}(t))}{\partial \boldsymbol{b}_k} = -\frac{\kappa_2}{\sqrt{m}} \left( \frac{p}{1+p} e^{-f_+(t)} \boldsymbol{x}_+ - \frac{1}{1+p} e^{f_-(t)} \boldsymbol{x}_- \right)$$

$$= \frac{\kappa_2}{\sqrt{m}} \frac{e^{f_-(t)}}{1+p} \left( \boldsymbol{x}_- - p e^{-(f_+(t)+f_-(t))} \boldsymbol{x}_+ \right).$$

Recalling Lemma Lemma F.5 (S2), $\lim_{t \to \infty} p e^{-(f_+(t)+f_-(t))} = \frac{1+\cos \Delta}{1+\cos \Delta + \frac{m_+}{m_-} \sin^2 \Delta}$. Then using Lemma F.6 (III), there exists $c_1 > 0$, s.t.

$$\boldsymbol{b}_k^* = c_1 \left( \boldsymbol{x}_- - \frac{1+\cos \Delta}{1+\cos \Delta + \frac{m_+}{m_-} \sin^2 \Delta} \boldsymbol{x}_+ \right).$$

Hence, we have proved

$$\langle \boldsymbol{b}_k^*, \boldsymbol{x}_+ \rangle > 0, \quad \forall k \in \mathcal{K}_-.$$

Then writing the KKT condition about the gradient of $\boldsymbol{b}_k^*$ of Problem (34):

$$\boldsymbol{0} = \boldsymbol{b}_k^* + \lambda_+^* \frac{\kappa_2}{\sqrt{m}} \boldsymbol{x}_+ - \lambda_-^* \frac{\kappa_2}{\sqrt{m}} \boldsymbol{x}_-, \quad \forall k \in \mathcal{K}_-.$$

Combining the two equations about $\boldsymbol{b}_k^*$, we obatin

$$\boldsymbol{b}_k^* = \lambda_-^* \frac{\kappa_2}{\sqrt{m}} \left( \boldsymbol{x}_- - \frac{1+\cos \Delta}{1+\cos \Delta + \frac{m_+}{m_-} \sin^2 \Delta} \boldsymbol{x}_+ \right), \quad \forall k \in \mathcal{K}_-;$$

$$\frac{\lambda_+^*}{\lambda_-^*} = \frac{1+\cos \Delta}{1+\cos \Delta + \frac{m_+}{m_-} \sin^2 \Delta}.$$

In summary, we have proved the final convergence direction $\overline{\boldsymbol{\theta}} = (\overline{\boldsymbol{b}}_1^\top, \cdots, \overline{\boldsymbol{b}}_m^\top)^\top \in \mathbb{S}^{md-1}$ satisfies

$$\overline{\boldsymbol{b}}_k = \boldsymbol{0}, \quad \forall k \notin \mathcal{K}_+ \cup \mathcal{K}_-;$$

$$\overline{\boldsymbol{b}}_k = C \left( \boldsymbol{x}_+ - \boldsymbol{x}_- \cos \Delta \right), \quad \forall k \in \mathcal{K}_+;$$

$$\overline{\boldsymbol{b}}_k = C\left(\left(1 + \frac{m_+ \sin^2 \Delta}{m_-(1 + \cos \Delta)}\right)\boldsymbol{x}_- - \boldsymbol{x}_+\right), \quad \forall k \in \mathcal{K}_-;$$

where $C > 0$ is a scaling constant such that $\|\overline{\boldsymbol{\theta}}\|_2 = 1$.

Moreover, a straight-forward calculation gives us that

$$f(\boldsymbol{x}_+; \overline{\boldsymbol{\theta}}) = -f(\boldsymbol{x}_-; \overline{\boldsymbol{\theta}}) > 0.$$

$\square$

*Proof of Theorem 4.10.*
Lemma F.7 implies Theorem 4.10 directly. $\square$

*Proof of Theorem 4.11.*
From Lemma F.7, the final convergence direction $\overline{\boldsymbol{\theta}} = (\overline{\boldsymbol{b}}_1^\top, \cdots, \overline{\boldsymbol{b}}_m^\top)^\top \in \mathbb{S}^{md-1}$ satisfies

$$\begin{aligned}
\overline{\boldsymbol{b}}_k &= \boldsymbol{0}, \quad \forall k \notin \mathcal{K}_+ \cup \mathcal{K}_-; \\
\overline{\boldsymbol{b}}_k &= C\left(\boldsymbol{x}_+ - \boldsymbol{x}_- \cos \Delta\right), \quad \forall k \in \mathcal{K}_+; \\
\overline{\boldsymbol{b}}_k &= C\left(\left(1 + \frac{m_+ \sin^2 \Delta}{m_-(1 + \cos \Delta)}\right)\boldsymbol{x}_- - \boldsymbol{x}_+\right), \quad \forall k \in \mathcal{K}_-;
\end{aligned}$$

where $C > 0$ is a scaling constant such that $\|\overline{\boldsymbol{\theta}}\|_2 = 1$ and $f_+(\overline{\boldsymbol{\theta}}) = -f_-(\overline{\boldsymbol{\theta}}) > 0$.

It is easy to verify that There exists a scaling factor $C_1 > 0$ such that $f_+(\hat{\boldsymbol{\theta}}) = -f_-(\hat{\boldsymbol{\theta}}) = 1$, where $\hat{\boldsymbol{\theta}} = C_1 \overline{\boldsymbol{\theta}}$. For simplicity, we denote $Q := CC_1$, then $\hat{\boldsymbol{\theta}} = (\hat{\boldsymbol{b}}_1^\top, \cdots, \hat{\boldsymbol{b}}_m^\top)^\top \in \mathbb{R}^{md}$ satisfies

$$\begin{aligned}
\hat{\boldsymbol{b}}_k &= \boldsymbol{0}, \quad \forall k \notin \mathcal{K}_+ \cup \mathcal{K}_-; \\
\hat{\boldsymbol{b}}_k &= Q\left(\boldsymbol{x}_+ - \boldsymbol{x}_- \cos \Delta\right), \quad \forall k \in \mathcal{K}_+; \\
\hat{\boldsymbol{b}}_k &= Q\left(\left(1 + \frac{m_+ \sin^2 \Delta}{m_-(1 + \cos \Delta)}\right)\boldsymbol{x}_- - \boldsymbol{x}_+\right), \quad \forall k \in \mathcal{K}_-; \\
f_+(\hat{\boldsymbol{\theta}}) &= -f_-(\hat{\boldsymbol{\theta}}) = 1.
\end{aligned}$$

Therefore, $\hat{\boldsymbol{\theta}}$ is a feasible point of Problem (34). Moreover, from

$$\begin{aligned}
-1 &= -\sum_{k \in \mathcal{K}_-} \frac{\kappa_2}{\sqrt{m}} \left\langle \hat{\boldsymbol{b}}_k, \boldsymbol{x}_- \right\rangle \\
&= -\frac{\kappa_2}{\sqrt{m}} m_- \left\langle Q\left(\left(1 + \frac{m_+ \sin^2 \Delta}{m_-(1 + \cos \Delta)}\right)\boldsymbol{x}_- - \boldsymbol{x}_+\right), \boldsymbol{x}_- \right\rangle \\
&= -Q\frac{\kappa_2}{\sqrt{m}} m_- \left(1 - \cos \Delta + \frac{m_+ \sin^2 \Delta}{m_-(1 + \cos \Delta)}\right) \\
&= -Q\frac{\kappa_2}{\sqrt{m}} \left(m_-(1 - \cos \Delta) + m_+ \frac{\sin^2 \Delta}{1 + \cos \Delta}\right),
\end{aligned}$$

we have

$$\frac{\kappa_2}{\sqrt{m}} Q = \frac{1}{m_-(1 - \cos \Delta) + m_+ \frac{\sin^2 \Delta}{1 + \cos \Delta}}.$$

For any $\epsilon \geq 0$, now we consider another solution $\hat{\boldsymbol{\theta}}(\epsilon)$ near $\hat{\boldsymbol{\theta}}$: $\hat{\boldsymbol{\theta}}(\epsilon) = (\hat{\boldsymbol{b}}_1^\top(\epsilon), \cdots, \hat{\boldsymbol{b}}_m^\top(\epsilon))^\top$, where

$$\begin{aligned}
\hat{\boldsymbol{b}}_k(\epsilon) &= \boldsymbol{0}, \quad \forall k \notin \mathcal{K}_+ \cup \mathcal{K}_-; \\
\hat{\boldsymbol{b}}_k(\epsilon) &= Q\left(\boldsymbol{x}_+ - \boldsymbol{x}_- \cos \Delta\right) - Q\epsilon\left(\boldsymbol{x}_+ - \boldsymbol{x}_- \cos \Delta\right), \quad \forall k \in \mathcal{K}_+;
\end{aligned}$$

$$\hat{\boldsymbol{b}}_k(\epsilon) = Q\left(\left(1 + \frac{m_+ \sin^2 \Delta}{m_-(1 + \cos \Delta)}\right)\boldsymbol{x}_- - \boldsymbol{x}_+\right) + Q\epsilon\left(\boldsymbol{x}_+ - \boldsymbol{x}_- \cos \Delta\right), \quad \forall k \in \mathcal{K}_-.$$

and it holds that $\hat{\boldsymbol{\theta}}(0) = \hat{\boldsymbol{\theta}}$. Moreover, it is easy to verify that $\hat{\boldsymbol{\theta}}(\epsilon)$ is also a feasible point of Problem (34).

$$f_-(\hat{\boldsymbol{\theta}}(\epsilon)) = f_-(\hat{\boldsymbol{\theta}}(0)) = -1;$$
$$f_+(\hat{\boldsymbol{\theta}}(\epsilon)) = f_+(\hat{\boldsymbol{\theta}}(0)) = 1.$$

Then we compare the norm of $\hat{\boldsymbol{\theta}}(\epsilon)$ and $\hat{\boldsymbol{\theta}}(0)$.

$$\left\|\hat{\boldsymbol{\theta}}(\epsilon)\right\|^2 = m_+ \left(Q\left(\boldsymbol{x}_+ - \boldsymbol{x}_- \cos \Delta\right) - Q\epsilon\left(\boldsymbol{x}_+ - \boldsymbol{x}_- \cos \Delta\right)\right)^2$$
$$+ m_- \left(Q\left(\left(1 + \frac{m_+ \sin^2 \Delta}{m_-(1 + \cos \Delta)}\right)\boldsymbol{x}_- - \boldsymbol{x}_+\right) + Q\epsilon\left(\boldsymbol{x}_+ - \boldsymbol{x}_- \cos \Delta\right)\right)^2,$$

At $\epsilon = 0$, we can calculate that

$$\frac{\mathrm{d}\left\|\hat{\boldsymbol{\theta}}(\epsilon)\right\|^2}{2m_+ Q^2 \mathrm{d}\epsilon}\Bigg|_{\epsilon=0}$$

$$= \frac{1}{m_+}\left(-m_+ \|\boldsymbol{x}_+ - \boldsymbol{x}_- \cos \Delta\|^2 + m_- \left\langle\left(1 + \frac{m_+ \sin^2 \Delta}{m_-(1 + \cos \Delta)}\right)\boldsymbol{x}_- - \boldsymbol{x}_+, \boldsymbol{x}_+ - \boldsymbol{x}_- \cos \Delta\right\rangle\right)$$

$$= \left\langle\left(\frac{m_-}{m_+} + \frac{\sin^2 \Delta}{1 + \cos \Delta} + \cos \Delta\right)\boldsymbol{x}_- - (1 + \frac{m_-}{m_+})\boldsymbol{x}_+, \boldsymbol{x}_+ - \boldsymbol{x}_- \cos \Delta\right\rangle$$

$$\overset{\text{Denote } \alpha = \frac{m_-}{m_+}}{=} \left\langle\left(\alpha + \frac{\sin^2 \Delta}{1 + \cos \Delta} + \cos \Delta\right)\boldsymbol{x}_- - (1 + \alpha)\boldsymbol{x}_+, \boldsymbol{x}_+ - \boldsymbol{x}_- \cos \Delta\right\rangle$$

$$= -\left(\alpha + \frac{\sin^2 \Delta}{1 + \cos \Delta} + \cos \Delta\right)\cos \Delta - (1 + \alpha)$$

$$+ \left(\alpha + \frac{\sin^2 \Delta}{1 + \cos \Delta} + \cos \Delta\right)\cos \Delta + (1 + \alpha)\cos^2 \Delta$$

$$= -(1 + \alpha)\sin^2 \Delta < 0$$

Then combining the continuity of $\frac{\mathrm{d}\|\hat{\boldsymbol{\theta}}(\epsilon)\|^2}{\mathrm{d}\epsilon}$, there exists $\delta > 0$ such that the following inequality holds:

$$\left\|\hat{\boldsymbol{\theta}}(\epsilon)\right\|^2 < \left\|\hat{\boldsymbol{\theta}}(0)\right\|^2, \ \forall \epsilon \in (0, \delta).$$

Hence, we have proved that $\overline{\boldsymbol{\theta}}$ is not a local optimal direction of the max-margin problem (34).

$\square$

# G  Clarke Subdifferential and KKT Conditions for Non-smooth Optimization

**Definition G.1** (Clarke's Subdifferential (Clarke et al., 2008))**.** For a locally Lipschitz function $\mathcal{L} : \Omega \to \mathbb{R}$, the Clarke's subdifferential at $\boldsymbol{\theta} \in \Omega$ is the convex set

$$\partial^{\circ} \mathcal{L}(\boldsymbol{\theta}) := \text{conv} \left\{ \lim_{i \to \infty} \nabla \mathcal{L}(\boldsymbol{\theta}_i) : \lim_{i \to \infty} \boldsymbol{\theta}_i = \boldsymbol{\theta}, \mathcal{L} \text{ is differential at } \boldsymbol{\theta}_i \right\}.$$

**Remark G.2.** Notice that if $\mathcal{L}$ is continuously differentiable at $\boldsymbol{\theta}$, then $\partial^{\circ} \mathcal{L}(\boldsymbol{\theta}) = \{\nabla \mathcal{L}(\boldsymbol{\theta})\}$ is unique. However, for discontinuous differentiable points of $\mathcal{L}$, the differential inclusion flow $\frac{\mathrm{d}\boldsymbol{\theta}}{\mathrm{d}t} \in \partial^{\circ} \mathcal{L}(\boldsymbol{\theta})$ defined by Definition G.1 may not be unique. To study a more specific dynamics, we also utilize Definition H.1 to determine GF at some of such points.

Now we review the definition of Karush-Kuhn-Tucker (KKT) conditions for non-smooth optimization problems (Dutta et al., 2013). Consider the following constrained optimization problem (P):

$$\min_{\boldsymbol{x} \in \mathbb{R}^d} : f(\boldsymbol{x})$$
$$\text{s.t. } g_i(\boldsymbol{x}) \leq 0, \quad \forall i \in [N]$$

where $f, g_1, \cdots, g_N : \mathbb{R}^d \to \mathbb{R}$ are locally Lipschitz functions. We say that $\boldsymbol{x} \in \mathbb{R}$ is a feasible point of (P) if $\boldsymbol{x}$ satisfies $g_i(\boldsymbol{x}) \leq 0$ for all $i \in [N]$.

**Definition G.3** (KKT Point for Non-smooth Optimization)**.** We say that a feasible point of (P) is a KKT point if there exists $\lambda_1, \cdots, \lambda_N \geq 0$ such that

$$1. \ \mathbf{0} \in \partial^{\circ} f(\boldsymbol{x}) + \sum_{i \in [N]} \lambda_i \partial^{\circ} g_i(\boldsymbol{x});$$

$$2. \ \forall i \in [N], \lambda_i g_i(\boldsymbol{x}) = 0.$$

# H    Solution of Discontinuous System

In this section, we add some supplements about the definitions of solutions of discontinuous systems, which can overcomes non-uniqueness of GF trajectories (2) to some extent. Many definitions of solutions of differential equations with discontinuous systems have been proposed. In this paper, we adopt a widely used definition of the solutions in Chapter 2.4 in (Filippov, 2013).

**Definition H.1** (Solutions of Discontinuous Systems, Chapter 2.4 in (Filippov, 2013))**.** Consider a $n$-dimensional equation or a system ($\boldsymbol{x} \in \mathbb{R}^n$): $\frac{\mathrm{d}\boldsymbol{x}}{\mathrm{d}t} = \boldsymbol{f}(\boldsymbol{x})$ with a piecewise continuous function $\boldsymbol{f}$ in a domain $G$. We aim to define the dynamics near some discontinuous regions.

Let the function $\boldsymbol{f}$ be discontinuous on a smooth surface $S$ given by the equation $\phi(\boldsymbol{x}) = 0$. Let $\boldsymbol{x}^* \in S$ and the surface $S$ separate the neighborhood of $\boldsymbol{x}^*$ into domains $G^-$ and $G^+$. Let the function $\boldsymbol{f}(\boldsymbol{x})$ have the limit values:

$$\boldsymbol{f}^-(\boldsymbol{x}^*) := \lim_{\boldsymbol{x} \in G^-, \boldsymbol{x} \to \boldsymbol{x}^*} \boldsymbol{f}(\boldsymbol{x}), \quad \boldsymbol{f}^+(\boldsymbol{x}^*) := \lim_{\boldsymbol{x} \in G^+, \boldsymbol{x}^* \to \boldsymbol{x}^*} \boldsymbol{f}(\boldsymbol{x}).$$

Here one should distinguish between two main cases. Let $f_N^-(\boldsymbol{x}^*)$ and $f_N^+(\boldsymbol{x}^*)$ be projections of the vectors $\boldsymbol{f}^-(\boldsymbol{x}^*)$ and $\boldsymbol{f}^+(\boldsymbol{x}^*)$ onto the normal to the surface $S$ at the point $\boldsymbol{x}^*$, where the normal is directed towards the domain $G^+$.

**(Case I)**. If the vectors $\boldsymbol{f}(\boldsymbol{x}^*)$ are directed to the surface $S$ on both sides, i.e. $f_N^-(\boldsymbol{x}^*) > 0$, $f_N^+(\boldsymbol{x}^*) < 0$, then the solution the solution starting from $\boldsymbol{x}^*$ can not leave $S$ for some time. Moreover, its dynamics on $S$ can be defined in the following way:

$$\frac{\mathrm{d}\boldsymbol{x}}{\mathrm{d}t} = \boldsymbol{f}^0(\boldsymbol{x}),$$

$$\text{where } \boldsymbol{f}^0(\boldsymbol{x}) = \alpha \boldsymbol{f}^+(\boldsymbol{x}) + (1 - \alpha) \boldsymbol{f}^-(\boldsymbol{x}), \quad \alpha = \frac{f_N^-(\boldsymbol{x})}{f_N^-(\boldsymbol{x}) - f_N^+(\boldsymbol{x})}.$$

**(Case II)**. If $f_N^-(\boldsymbol{x}^*) \geq 0, f_N^+(\boldsymbol{x}^*) \geq 0$, but $f_N^-(\boldsymbol{x}^*)$ and $f_N^+(\boldsymbol{x}^*)$ are not both $0$, then the solution starting from $\boldsymbol{x}^*$ passes from one side of the surface $S$ to the other instantly.

**(Case III)**. If $f_N^-(\boldsymbol{x}^*) < 0, f_N^+(\boldsymbol{x}^*) > 0$, then the dynamics is defined in the similar way as (Case I).

**(Case IV)**. If $f_N^-(\boldsymbol{x}^*) \leq 0, f_N^+(\boldsymbol{x}^*) \leq 0$, but $f_N^-(\boldsymbol{x}^*)$ and $f_N^+(\boldsymbol{x}^*)$ are not both $0$, then the dynamics is defined in the similar way as (Case II).

**Remark H.2.** Notice that Definition H.1 overcomes non-uniqueness of GF trajectories to some extent. It is worth noting that Definition H.1 and Definition G.1 are compatible and specifically, the dynamics defined in Definition H.1(Case I, III) lie in the convex hull defined in Definition G.1.

**Remark H.3.** In (Lyu et al., 2021), the non-branching starting point Assumption is employed to address the technical challenge of non-uniqueness in GF trajectories. By comparison, in this work, we do not need this assumption. We adopt Definition H.1 to uniquely determine the Gradient Flow trajectories theoretically near some discontinuous differential regions, such as "Ridge", "Valley", and "Refraction edge" discussed in Section I.2 in (Lyu et al., 2021).

# I  Some Basic Inequalities

**Lemma I.1** (Hoeffding's Inequality). *Let $X_1, \cdots, X_n$ are independent random variables, and $X_i \in [a_i, b_i]$ for any $i \in [n]$. Define $\bar{X} = \frac{1}{n} \sum_{i=1}^{n} X_i$. Then for any $\epsilon > 0$, we have the following probability inequalities:*

$$\mathbb{P}\Big(\bar{X} - \mathbb{E}[\bar{X}] \geq \epsilon\Big) \leq \exp\Big(-\frac{2n^2\epsilon^2}{\sum_{i=1}^{n}(b_i - a_i)^2}\Big),$$

$$\mathbb{P}\Big(\bar{X} - \mathbb{E}[\bar{X}] \leq -\epsilon\Big) \leq \exp\Big(-\frac{2n^2\epsilon^2}{\sum_{i=1}^{n}(b_i - a_i)^2}\Big).$$

**Lemma I.2.** *Consider $x_1, x_2, y \in \mathbb{S}^{d-1}$, where $\langle x_1, x_2 \rangle = \cos\Delta$ ($\Delta \in (0, \pi/2)$). If $\langle y, x_1 \rangle \geq 0$ and $\langle y, x_2 \rangle \leq 0$, then we have $0 \leq \langle y, x_1 \rangle \leq \sin\Delta$ and $-\sin\Delta \leq \langle y, x_2 \rangle \leq 0$.*

*Proof of Lemma I.2.*
Denote $\mathcal{M}_x := \text{span}\{x_1, x_2\}$. We can do the orthogonal decomposition of $y$:

$$y = y_{\mathcal{M}} + y_{\mathcal{M}}^{\perp},$$

where $y_{\mathcal{M}} \in \mathcal{M}_x$ and $y_{\mathcal{M}}^{\perp} \perp \mathcal{M}_x$. From $y \in \text{span}\{x_1, x_2\}$, there exist $\alpha, \beta \in \mathbb{R}$, s.t. $y_{\mathcal{M}} = \alpha x_1 + \beta x_2$.

Due to the orthogonal decomposition, we know $\|y_{\mathcal{M}}\| \leq 1$, which means $\alpha^2 + \beta^2 + 2\alpha\beta\cos\Delta \leq 1$. Noticing $\alpha^2 + \beta^2 + 2\alpha\beta\cos\Delta = (\alpha + \beta\cos\Delta)^2 + \alpha^2\sin^2\Delta$, we know $\alpha^2\sin^2\Delta \leq 1$.

Due to $\langle y, x_1 \rangle \geq 0$ and $\langle y, x_2 \rangle \leq 0$, we have $\langle y_{\mathcal{M}}, x_1 \rangle \geq 0$ and $\langle y_{\mathcal{M}}, x_2 \rangle \leq 0$, which means

$$\alpha + \beta\cos\Delta \geq 0, \quad \alpha\cos\Delta + \beta \leq 0.$$

So $\alpha \geq 0$ and $\alpha\sin\Delta \geq 0$. Recalling $\alpha^2\cos^2\Delta \leq 1$, we know $0 \leq \alpha\sin\Delta \leq 1$. Hence, we have $\alpha + \beta\cos\Delta \leq \alpha - \alpha\cos^2\Delta = \alpha\sin^2\Delta \leq \sin\Delta$, i.e. $\langle y, x_1 \rangle \leq \sin\Delta$.

In the same way, we have $-\sin\Delta \leq \langle y, x_2 \rangle \leq 0$.

$\square$

**Lemma I.3.** *If $p \geq 5$, we have $\|z\| \geq \frac{p-1}{p+1} \geq \frac{2}{3}$.*

*Proof of Lemma I.3.*

$$\|z\|^2 = \left\|\frac{1}{n}\sum_{i=1}^{n} y_i x_i\right\|^2 = \left\|\frac{p}{1+p}x_+ - \frac{1}{1+p}x_-\right\|^2 = \left(\frac{p}{1+p}\right)^2 + \left(\frac{1}{1+p}\right)^2 - \frac{2p}{(1+p)^2}\langle x_+, x_- \rangle$$

$$= \left(\frac{p}{1+p} + \frac{1}{1+p}\right)^2 - \frac{2p}{(1+p)^2}(\cos\Delta + 1) \geq 1 - \frac{2p}{(p+1)^2} \cdot 2 = \left(\frac{p-1}{p+1}\right)^2.$$

$\square$

**Lemma I.4.** *Let $w \in \mathbb{S}^{d-1}$. If $\langle w, \mu \rangle \geq 1 - \epsilon$ ($\epsilon \in (0,1)$), $p \geq 5$ and $\cos\Delta \geq 4/5$, then we have*

$$-2\sqrt{\epsilon}\sin\Delta - \epsilon \leq \langle w, x_- \rangle - \frac{p\cos\Delta - 1}{\sqrt{p^2 + 1 - 2p\cos\Delta}} \leq 2\sqrt{\epsilon}\sin\Delta.$$

*Proof of Lemma I.4.*

$$\langle w, x_- \rangle = \langle w, \mu \rangle + \langle w, x_- - \mu \rangle = \langle w, \mu \rangle + \langle \mu, x_- - \mu \rangle + \langle w - \mu, x_- - \mu \rangle.$$

It is easy to verify

$$\langle \mu, x_- - \mu \rangle = \langle \mu, x_- \rangle - 1 = \left\langle \frac{px_+ - x_-}{\|px_+ - x_-\|}, x_- \right\rangle - 1 = \frac{p\cos\Delta - 1}{\sqrt{p^2 + 1 - 2p\cos\Delta}} - 1;$$

$$\|x_- - \mu\| = \sqrt{2 - 2\langle \mu, x_- \rangle} = \sqrt{2 - 2\frac{p\cos\Delta - 1}{\sqrt{p^2 + 1 - 2p\cos\Delta}}};$$

$$\|\boldsymbol{w} - \boldsymbol{\mu}\| = \sqrt{2 - 2\langle \boldsymbol{w}, \boldsymbol{\mu}\rangle}.$$

Thus,

$$|\langle \boldsymbol{w} - \boldsymbol{\mu}, \boldsymbol{x}_- - \boldsymbol{\mu}\rangle| \le \|\boldsymbol{w} - \boldsymbol{\mu}\|\, \|\boldsymbol{x}_- - \boldsymbol{\mu}\|$$

$$= \sqrt{2 - 2\frac{p\cos\Delta - 1}{\sqrt{p^2 + 1 - 2p\cos\Delta}}}\sqrt{2 - 2\langle \boldsymbol{w}, \boldsymbol{\mu}\rangle} \le \sqrt{2 - 2\frac{p\cos\Delta - 1}{\sqrt{p^2 + 1 - 2p\cos\Delta}}}\sqrt{2\epsilon}$$

$$\le \frac{\sqrt{2}}{\sqrt[4]{p^2 + 1 - 2p\cos\Delta}}\sqrt{\frac{p^2\sin^2\Delta}{\sqrt{p^2 + 1 - 2p\cos\Delta} + p\cos\Delta - 1}}\sqrt{2\epsilon}$$

$$\le \frac{2\sqrt{\epsilon}}{\sqrt{p - 1}}\frac{p\sin\Delta}{\sqrt{p - 1 + p\cos\Delta - 1}} \le 2\sqrt{\epsilon}\sin\Delta.$$

Then we have the bound:

$$\langle \boldsymbol{w}, \boldsymbol{x}_-\rangle \le \frac{p\cos\Delta - 1}{\sqrt{p^2 + 1 - 2p\cos\Delta}} - 1 + \langle \boldsymbol{w}, \boldsymbol{\mu}\rangle + |\langle \boldsymbol{w} - \boldsymbol{\mu}, \boldsymbol{x}_- - \boldsymbol{\mu}\rangle|$$

$$\le \frac{p\cos\Delta - 1}{\sqrt{p^2 + 1 - 2p\cos\Delta}} + 2\sqrt{\epsilon}\sin\Delta,$$

$$\langle \boldsymbol{w}, \boldsymbol{x}_-\rangle \ge \frac{p\cos\Delta - 1}{\sqrt{p^2 + 1 - 2p\cos\Delta}} - 1 + \langle \boldsymbol{w}, \boldsymbol{\mu}\rangle - |\langle \boldsymbol{w} - \boldsymbol{\mu}, \boldsymbol{x}_- - \boldsymbol{\mu}\rangle|$$

$$\ge \frac{p\cos\Delta - 1}{\sqrt{p^2 + 1 - 2p\cos\Delta}} - \epsilon - 2\sqrt{\epsilon}\sin\Delta.$$

$\square$

