# OpenReview forum: "Understanding Multi-phase Optimization Dynamics and Rich Nonlinear Behaviors of ReLU Networks"
_NeurIPS.cc/2023/Conference — NeurIPS 2023 spotlight_

### Official Review · Reviewer_w3Qj · 2023-07-04

**Soundness:** 4 excellent
**Presentation:** 3 good
**Contribution:** 4 excellent
**Rating:** 8
**Confidence:** 3

**Summary:**

This paper conducts a comprehensive theoretical analysis of the training dynamics of two-layer ReLU neural networks on a linearly separable data. The authors isolate four discrete phases of the training procedure and describe specific nonlinear behaviors that occur during each phase. Moreover, they derive explicit formulae for the evolution of the network parameters and prove convergence results for the network's output. The paper's contributions include a complete theoretical characterization of the training process, a better understanding of the role of initialization and regularization, and insights into the generalization properties of ReLU networks. In brief, the research provides a valuable addition to the domain of deep learning theory and sheds light on the training process of ReLU.

**Strengths:**

The originality of the paper lies in its complete theoretical characterization of the training process of the two-layer ReLU network. While previous work has focused on local analysis or approximate linear models, this paper provides a more comprehensive understanding of the optimization dynamics of ReLU networks. The authors identify four distinct stages of the training process and describe the specific nonlinear behavior that occurs at each stage. They also derive explicit formulations for the evolution of network parameters and demonstrate convergent results for network outputs. Besides, the authors provide rigorous mathematical proofs and derive explicit formulas that shed light on the complex dynamics of ReLU networks.


**Weaknesses:**

First, a weakness of the paper is that it focuses on the specific setting of two-layer ReLU networks trained on linearly separable data. While this setting is useful for providing a full theoretical characterization of the training process, it may not generalize to more complex networks or datasets.

Second, another weakness of the paper is that it does not provide empirical verification of its theoretical results. While the authors provide rigorous mathematical proofs and derive explicit formulas, it would be useful to validate these results on real-world datasets.

**Questions:**

Can the authors provide empirical validation of their theoretical results? While the authors provide rigorous mathematical proofs and derive explicit formulas, it would be useful to validate these results on real-world datasets.


**Limitations:**

Yes.

---

> ### Author Rebuttal · Authors · 2023-08-08
>
> We appreciate the reviewer's recognition of our work and helpful comments. Below, we offer detailed responses to the reviewer's questions:
>
> **Q1. First, a weakness of the paper is that it focuses on the specific setting of two-layer ReLU networks trained on linearly separable data. While this setting is useful for providing a full theoretical characterization of the training process, it may not generalize to more complex networks or datasets.**
>
> **Response.** As the reviewer commented, under such a relatively strict assumption on data and networks, we can capture the entire training process of the neural network, exhibiting multi-stage optimization dynamics and rich nonlinear behaviors. While our theorems might not hold for more intricate networks and datasets, similar multi-phase dynamics and nonlinear behaviors could still manifest. For a preliminary exploration of the extension of our findings, please refer to our **``Global'' Response to All Reviewers**. Furthermore, we are intrigued by exploring the complete training dynamics of more complex networks and datasets, focusing on characterizing new nonlinear behaviors. We leave this to future work.
>
> **Q2. Second, another weakness of the paper is that it does not provide empirical verification of its theoretical results. While the authors provide rigorous mathematical proofs and derive explicit formulas, it would be useful to validate these results on real-world datasets. Can the authors provide empirical validation of their theoretical results?**
>
> **Response.** We thank the reviewer for reminding us of more empirical validation. We have conducted more experiments to further support our theoretical results. For experimental results, please refer to our **`` Global'' Response to All Reviewers**. Regarding real-world datasets and networks, some previous works indicate similar nonlinear phenomena may still occur; please refer to our **response to Reviewer 79Zb's Q3**. Additionally, we will pursue theoretical investigations into these aspects in our future work.

---

> > ### Comment · Reviewer_w3Qj · 2023-08-11
> >
> > Many thanks to the authors for your careful explanation and detailed rebuttal. Based on the supplementary experiments given, I feel that this paper is of great help in understanding the nonlinear dynamics of neural networks.

---

> > > ### Author Response · Authors · 2023-08-16
> > > **Thanks**
> > >
> > > We thank the reviewer for the positive feedback of our response and the recognition of our work. We appreciate your support very much!

---

### Official Review · Reviewer_yJvt · 2023-07-05

**Soundness:** 3 good
**Presentation:** 3 good
**Contribution:** 3 good
**Rating:** 7
**Confidence:** 3

**Summary:**

The training of ReLU neural networks involves complex nonlinear phenomena, challenging theoretical analysis due to the nonlinearity of models and non-convexity of loss. This study provides a comprehensive theoretical characterization of the training process of a two-layer ReLU network using Gradient Flow on linearly separable data. From random initialization to final convergence, the analysis identifies four different training phases indicating a trend from simplification to complication. Specific nonlinear behaviors, like initial condensation, saddle-to-plateau dynamics, plateau escape, activation pattern changes, and increasing learning complexity, are accurately captured.

**Strengths:**

1. The paper presents a complete theoretical characterization of the training dynamics of 2-layer ReLU networks, which addresses the nonlinearity and non-convexity challenges.
2. The precise characterization of four distinct phases of the training dynamics.

**Weaknesses:**

- A missing related work on network's training dynamics and phases [1]
- Could authors run more hyperparameter settings (\kappa_1, \kappa_2, \delta, etc.) and see if simulated training times (T1 ~ T4) are aligned with estimated ones?

[1] "Neural Networks as Kernel Learners: The Silent Alignment Effect" Atanasov et al. 2021

**Questions:**

- Do these four phases persistently exist across different settings (\kappa_1, \kappa_2, \delta, etc.)?
- For Phase III: “deactivation of positive neurons on $\mathbf{x}-$” — is this a sign of implicit bias of NN during gradient flow?
- Phase IV: “reactivation of negative neurons on $\mathbf{x}+$” — is this a sign of NN's overfitting?
- It is a bit surprising to me that network width ($m$) and learning rate ($\eta$) are not included in the form of T1~T4. Could authors intuitively explain why is it from the viewpoint of the proof strategy in this paper?
- One previous work on NN convergence stated that "strong correlations" (of input patches) "imply faster convergence", which contradicts the conclusions in this work (i.e. smaller $\delta$ indicates slower convergence). Could authors compare their work?

[2] "When is a Convolutional Filter Easy To Learn?" Du et al. 2017.

**Limitations:**

- Assumption 3.1 is oversimplified. Basically, all training data collapse into two signals. How will the results change when we switch to a little bit more realistic setting by adding some stochastic noises on top of $\mathbf{x}+$ and $\mathbf{x}-$ to each data point?

---

> ### Author Rebuttal · Authors · 2023-08-08
>
> We thank the reviewer for the appreciation of our work and helpful suggestions to improve this paper. In the following, we answer the reviewer’s questions in detail.
>
> **Q1. A missing related work on network's training dynamics and phases [1].**
>
> **Response.** We thank the reviewer for pointing out the interesting related work [1]. In [1], the authors also characterize multi-phase dynamics of training NNs and demonstrate that NNs in the rich feature learning regime learn a kernel machine due to the silent alignment phenomenon, similar to initial condensation phenomenon in our work. We will add and discuss this work in the revised version.
>
> **Q2. Could authors run more hyperparameter settings and see if simulated training times are aligned with estimated ones? Do these four phases persistently exist across different settings?**
>
> **Response.** We thank the reviewer's valuable suggestions on new experiments to supplement our theoretical results. To address this, we have conducted additional experiments under more hyperparameter settings (different $\Delta$, $p$, $\kappa_1$, and $\kappa_2$). These experimental results consistently validate the existence of four training phases in our theory and the reliability of our theoretical estimates on training time. For more details, please refer to our **``Global'' Response to All Reviewers**.
>
> **Q3. Explanations for Phase III and IV.**
>
> **Response.** We appreciate the reviewer's insightful perspective. We concur with the reviewer's understanding and will integrate this discussion into our revised version.
>
>   - In Phase III, neurons' deactivation is indeed an implicit bias of GF, which is a ``simplicity bias'' [3] more accurately. Additionally, GF does similar things in Phase I, simplifying the network through initial condensation.
>
>   - In Phase IV, neurons' reactivation is indeed an overfitting behavior because the ReLU activation patterns change from the ``simplest'' scheme to a more complex scheme (as discussed in Table 1).
>
>   - Combining these understandings, the evolution of activation patterns exhibits a simplifying-to-complicating learning trend.
>
> **Q4. Why network width and learning rate are not included in the estimated times?**
>
> **Response.**
>
>   - Learning rate $\eta$. Since our focus lies on GF, the version of GD with infinitesimal learning rate (equ (2) in Line 117), our results are independent of $\eta$.
>
>   - Network width $m$. We would like to provide an intuitive explanation for the influence of $m$ from our proof strategy.
>
>     - $T_I.$ In Phase I, we decompose neurons' dynamics into tangential and radial dynamics. For small initialization, during initial training, radial neuron growth is much slower than tangential velocity, resulting in directional condensation. For tangent velocity $\|d w_k(t)/d t\|$ (equ (7)), note: (i) at initialization, both $\kappa_2/\sqrt{m}\rho _k(0)=\kappa_2/\kappa_1$ and $F_k(0)$ are independent of $m$; (ii) During transient initial training, both the norm $\rho_k(t)$ and $F_k(t)$ stay close to initialization. Thus, tangent velocity is unaffected by $m$ during Phase I.
>
>     - $T_{II}$, $T_{III}$ and $T_{IV}$. In Phase II, III, and IV, we analyze $f( x_+;\theta(t))$ and $f( x_-;\theta(t))$. Notably, both $a_k$ and $b_k$ are normalized by $1/\sqrt{m}$. Thus, in terms of $m$, the prediction $f(x;\theta)=\sum_{k\in[m]}a_k\sigma(b_k^\top x)$ (on data $x$) has magnitude $m\cdot(1/\sqrt{m})\cdot(1/\sqrt{m})=1$, independent of $m$. Moreover, at the end of Phase I, living positive neurons (LPN) and living negative neurons (LNN) align with $\mu$ and $ x_+^\perp$, respectively. After Phase I, LPNs stay close, as do LNNs. So, for prediction, LPN and LNN can be conceptually treated as a single positive neuron and a single negative neuron, respectively, independent of $m$. A similar idea is present in the neuron embedding technique in [3].
>
> **Q5. Comparsion with the previous work [2] ``strong correlations'' (of input patches)**
>
> **Response.** We thank the reviewer for raising this intriguing question, and we provide the following response:
>
>   - In [2], the authors mentioned that when utilizing a CNN to extract a single feature $w^*$, optimization becomes easier with higher correlation between patches $\arg\left<Z_i,Z_j\right>$, and between patch and target $\arg\left<Z_i,w^*\right>$ (as shown in their Figure 1). The underlying reason lies in the context of a regression problem that involves only a ``single class of data'' (single feature).
>
>   - However, our setting is binary classification with two distinct classes of data ($\pm 1$), and the correlation between two classes should be characterized by label-weighted data angle: $\arg\left<1\cdot x_+,-1\cdot x_-\right>=\pi-\Delta$. Thus, smaller $\Delta$ means weaker data correlation. Moreover, our theory and experiments validate that larger $\Delta$ (indicating stronger data correlation) yields easier training.
>
>   - Therefore, the conclusions of our work and [2] are consistent, both implying: ``strong correlations imply faster convergence''.
>
> **Q6. Relax assumption 3.1 to noisy data.**
>
> **Response.** We thank the reviewer for offering this insightful suggestion to generalize our findings to noisy data. We have conducted additional experiments on our dataset with some stochastic noises on top of $x_+$ and $x_-$. Our numerical results illustrate that the same four-phase optimization dynamics and similar nonlinear behaviors persist for noisy data. For more details, please refer to our **``Global'' Response to all Reviewers**.
>
> [1] Atanasov et al. Neural Networks as Kernel Learners: The Silent Alignment Effect. (ICLR 2022)
>
> [2] Du et al. When is a Convolutional Filter Easy To Learn? (ICLR 2018)
>
> [3] Lyu et al. Gradient Descent on Two-layer Nets: Margin Maximization and Simplicity Bias. (NeurIPS 2021)

---

> > ### Comment · Reviewer_yJvt · 2023-08-19
> > **Thanks for the response**
> >
> > I appreciate the authors' response, and thus I would like to raise my score.
> >
> > Another question: in the global response Table 2, why did authors only compare $T_{\text{plat}}$ and $T_{\text{III}}$, instead of $T_\text{I}$ and $T_\text{II}$?

---

> > > ### Author Response · Authors · 2023-08-19
> > > **Thanks**
> > >
> > > We sincerely thank you for your time and effort in reviewing our paper. Moreover, we would like to reiterate our gratitude for your valuable recommendation on additional experiments and thank you for raising the scores!
> > >
> > > **Response to another question.**
> > > - In Table 2 in the global response, we focus on the change of our theoretical bounds under different $p$ and $\Delta$ (data-dependent hyper-parameters). Due to space limitations, we are constrained to showcasing **two most representative** time points.
> > > - The reason for selecting $T_{\rm III}$ and $T_{\rm plat}$ is as follows: According to our theory,
> > > (1) $T_{\rm II}$ and $T_{\rm III}$ have the **same** rate in terms of $p$ and $\Delta$, so we choose only one of them for presentation.
> > > (2) $T_{\rm plat}$ and $T_{\rm III}$ have **different** rates in terms of $p$, and $T_{\rm plat}$ can clearly reflect the "harm" of larger $p$ and smaller $\Delta$ on training accuracy. Hence, we choose to show $T_{\rm plat}$.
> > > (3) $T_{\rm I}$ is extremely short and remains **unaffected** by $p$ and $\Delta$, hence it is not included in the PDF for presentation.
> > > - In our revised version, we will present the **complete** numerical results, i.e. the change of {$T_{\rm I}, T_{\rm II}, T_{\rm plat}, T_{\rm III}$} under different {$p,\Delta,\kappa_1,\kappa_2$}.

---

### Official Review · Reviewer_79Zb · 2023-07-06

**Soundness:** 3 good
**Presentation:** 4 excellent
**Contribution:** 3 good
**Rating:** 7
**Confidence:** 3

**Summary:**

This paper aims to provide a theoretical understanding of the dynamics involved in training neural networks beyond the linear regime. The authors focus on a specific scenario where a two-layer ReLU network is trained using Gradient Flow (GF) on linearly separable data. The analysis encompasses the entire optimization process, starting from random initialization and concluding with final convergence.

Despite the simplicity of the model and data used in the study, the authors uncover multiple phases within the training process. By conducting a meticulous theoretical analysis, they precisely identify four distinct phases that exhibit various nonlinear behaviors.

In Phase I, the initial stage of training, there is a phenomenon of condensation and simplification as active neurons rapidly gather in two different directions. Simultaneously, the GF successfully escapes from the saddle point around the initialization.

Phase II involves a prolonged period where the GF becomes trapped in a plateau of training accuracy. However, it eventually manages to escape from this stagnation.

During Phase III, a significant number of neurons are deactivated, leading to self-simplification of the network. The GF then adapts its learning approach using this simplified network configuration.

In Phase IV, a considerable number of previously deactivated neurons are reactivated, resulting in self-complication of the network. Finally, the GF converges towards an initialization-dependent direction.

Overall, the training process exhibits a remarkable behavior of transitioning from simplification to complication. The detailed analysis of each phase sheds light on the intricate dynamics involved in the learning process beyond the linear regime.

**Strengths:**

The manuscript demonstrates excellent organization, comprehensiveness, and clarity, effectively covering crucial aspects such as a thorough review of related works and a comprehensive discussion of limitations. Moreover, the paper stands out by successfully combining empirical and theoretical approaches.

Overall, this work makes significant contributions that advance the current state of the literature.

**Weaknesses:**

These suggestions are intended to improve the overall clarity and accessibility of the research.

The manuscript does not provide sufficient details regarding the resources and computational aspects involved in the research. Additionally, the work would greatly benefit from a clear delineation of its limitations.

Outlined below are some limitations that underscore the weaknesses of the paper, despite its current status as the best effort in the field.

The paper lacks a significant number of experimental results to demonstrate the robustness of the findings. It would greatly enhance the study's credibility to include additional examples where the predictions have been validated and to provide measures of their robustness.

The submission would be strengthened by the addition of the code ran for the experiments

**Questions:**

Dear authors, I have a few questions regarding your work that would greatly aid my comprehension:

How reliable are the predictions made in this study, and what was the extent of experimentation conducted to support them?

Can general principles be derived from this research that would benefit machine learning practitioners?

Have alternative types of architectures been tested to determine if the findings hold true across different models as well (i.e., more realistic architectures and data sets)?

**Limitations:**

The limitations of the proposed method are summarized as follows:

* Narrow Scope: This study focuses exclusively on ReLU neural networks trained with Gradient Flow (GF) on linearly separable data. Although the analysis comprehensively captures the optimization process and identifies four distinct phases with rich nonlinear behaviors, the generalizability of the findings to other neural network types or more complex datasets may be limited.

* Limited Generalization to Gradient Descent (GD): While the paper provides a detailed analysis of GF dynamics, it acknowledges that Gradient Descent (GD) dynamics are more complex and can exhibit additional nonlinear behaviors like progressive sharpening and the edge of stability. Consequently, a comprehensive understanding of the nonlinear behaviors during GD training is not fully explored in this study, indicating the need for future research in this direction.

* Theoretical Understanding of Neural Network Training: While this work makes significant strides in theoretically understanding the training dynamics of neural networks, it acknowledges that there is still much progress to be made in fully comprehending the entire training process. Theoretical advancements related to nonlinear behaviors in neural network training, beyond the specific focus of this study, present valuable opportunities for future investigations. With this in mind I would like to suggest works like' The Neural Race Reduction: Dynamics of Abstraction in Gated Networks' Saxe 2022, that get analytical solution to 'Relu like' Neural Networks.

---

> ### Author Rebuttal · Authors · 2023-08-08
>
> We thank the reviewer for appreciating our work and for pointing out the relevant papers. We answer the reviewer’s questions in the following.
>
> **Weakness on experimental results and details.**
>
> **Response.** We thank the reviewer's suggestions on experiments to improve our paper.
>
>   - **More experiments.** We have conducted more experiments to verify the robustness of our theoretical results under more hyperparameter settings. Please refer to our **``Global'' Response to All Reviewers** for further details.
>
>   - **Code.** Following the rebuttal rule in NeurIPS 2023, we have sent an anonymized link to the AC in a separate comment encompassing our standard code along with details of computational resources.
>
> **Q1. How reliable are the predictions made in this study, and what was the extent of experimentation conducted to support them?**
>
> **Response.** We have conducted more experiments to verify our theory, please refer to our **``Global'' Response to All Reviewers**.
>
> **Q2. Can general principles be derived from this research that would benefit machine learning practitioners?**
>
> **Response.** Through our theoretical analysis, we suggest that the following insights could offer guidance to practitioners:
>
>   - **Use Balanced data by preprocessing.** Our focus is the binary classification problem a small ``margin'' between two data classes and a slight imbalance ($p=n_+/n_->1$). These factors can lead to training challenges and undesirable behaviors such as plateau. Our theory established that training accuracy platea in Phase II persist for $T_{plat}=\Theta(p/\kappa_2^2\Delta^2)$ (Theorem 4.5), which is proportional to $p$ and inversely proportional to $\Delta^2$. Our experiments support this, revealing that even a slight data imbalance ($p=4$) with $\Delta=\pi/15$ can result in a significantly long plateau. Therefore, for practitioners, although regulating $\Delta$ for multi-class data is complex, the data imbalance can be resolved by simple preprocessing (equalizing the numbers of data in each class), which can reduce the time of plateau and accelerate the rise in training accuracy.
>
>   - **Proper early stopping.** Our theory reveals that neural network training follows a simplifying-to-complicating dynamics learning trend. Initially, the network simplifies itself in Phase I and III, resulting in the ``simplest'' pattern at the end of Phase III (living positive neurons exclusively predict $x_+$, while living negative neurons solely predict $x_-$). However, in Phase IV, living negative neurons revert to predicting $ x_+ $, and the network's pattern becomes more intricate, which can be interpreted as overfitting. Therefore, proper early stopping is necessary to mitigate overfitting in real-world tasks.
>
> **Q3. Have alternative types of architectures been tested to determine if the findings hold true across different models as well (i.e., more realistic architectures and data sets)?**
>
> **Response.** For more practical architectures and datasets beyond our specific settings, while strict theorems might not hold, similar training behaviors persist, as evidenced by the following studies:
>
>   - *Initial condensation* occurs in multi-layer fully-connected neural networks [2];
>
>   - *Directional convergence* is demonstrated in homogeneous deep neural networks [3];
>
>   - *Saddle-to-saddle* dynamics are found in deep linear neural networks [4];
>
>   - *Learning with increasing complexity* is observed in realistic datasets (such as CIFAR-10) and various architectures (FNNs, CNNs, and ResNets) [5].
>
>   These studies provide valuable insights into the generalizability of the training behaviors discussed in our work.
>
> **L1. Narrow Scope.**
>
> **Response.** As the reviewer commented, under such a relatively strong assumption, our work completely characterizes the entire multi-phase optimization dynamics and exhibits rich nonlinear phenomena. For more complex network architectures and datasets, while strict theorems might not be applicable, our **response to Q3** demonstrates the persistence of similar phenomena. Additionally, we will work to relax this assumption conditionally. For example, our new experiments suggest that training accuracy falls into more plateau during training for noisy data. For more details, please refer to our **``Global'' Response to All Reviewers**. For a more in-depth study, we leave it to future work.
>
> **L2. Limited Generalization to Gradient Descent (GD).**
>
> **Response.** As mentioned by the reviewer, GD can exhibit more nonlinear behaviors, such as progressive sharpening and the edge of stability. Technically, analyzing these phenomena is more difficult compared to GF due to the consideration of appropriate learning rate and stability conditions. We leave analyzing GD's training dynamics for future work.
>
> **L3. Theoretical Understanding of Neural Network Training.**
>
> **Response.** As the reviewer commented, there is still a long way to go to study the training dynamics of neural networks. We thank the reviewer for pointing out the interesting related work [1]. The neural race introduces a novel implicit bias of learning dynamics: toward shared representations. This idea and the view of gating networks are very enlightening for extending our two-layer theory to deep ReLU neural networks and understanding the dynamics of deep ReLU networks. In our revised version, we will add and discuss this work.
>
> [1]  Saxe et al. The Neural Race Reduction: Dynamics of Abstraction in Gated Networks. (ICML 2022)
>
> [2] Zhou et al. Towards Understanding the Condensation of Neural Networks at Initial Training. (NeurIPS 2022)
>
> [3] Ji and Telgarsky. Directional convergence and alignment in deep learning. (NeurIPS 2020)
>
> [4] Jacot et al. Saddle-to-Saddle Dynamics in Deep Linear Networks: Small Initialization Training, Symmetry, and Sparsity. (aXiv 2021)
>
> [5] Nakkiran et al. SGD on Neural Networks Learns Functions of Increasing Complexity. (NeurIPS 2019)

---

> ### Comment · Reviewer_79Zb · 2023-08-13
>
> I want to express my gratitude to the authors for their meticulous clarification and thorough response. I raised my evaluation and confidence scores to acknowledge the enhanced clarity and presentation, along with a deeper comprehension of the contribution.

---

> > ### Author Response · Authors · 2023-08-16
> > **Thanks**
> >
> > We would like to express our gratitude to the reviewer for the positive feedback of our response, and thank you for raising the scores!

---

### Official Review · Reviewer_vhgo · 2023-07-07

**Soundness:** 2 fair
**Presentation:** 3 good
**Contribution:** 2 fair
**Rating:** 5
**Confidence:** 3

**Summary:**

In this work, the authors attempt an exact analysis of the training dynamics of 2-layer ReLU networks trained via gradient flow on linearly separable data. Specifically, the authors aim to build on related work (e.g., Boursier et al. [2022] on square loss) to the case of:
- Exponential loss (a more appropriate characteristic of classification problems)
- With data having mild orthogonal separability

Due to the additional non-linearity introduced by this specific loss type, and the complex data structure, the authors aim to characterize a richer non-linear structure.

By considering a 2-layer network, where the second-layer weights are held fixed, the authors demonstrate that under gradient flow, the tunable weights of the network evolve in 4 distinctive phases, indicative of a simple to complex learning phenomena.

**Strengths:**

- The authors extend the analytical formulation established in related previous literature to the case of the harder exponential loss type, which categorizes standard classification loss
- The authors relax the data orthogonality assumption utilized in these previous works by considering a case where the angle of separability between data points is $<90^\circ$
- The authors demonstrate additional data condensation and alignment phases over existent work at the onset and the end of the training, thus attributing to the aim of capturing richer non-linear phenomena.
- The authors establish bounds on the measure of the bounds on the count of the tunable *positive/negative* neurons in each phase along with the same on their norms, the time extent of these phases.

**Weaknesses:**

1. While the authors do draw similarities between their work and that of [Boursier et al. [2022]]*, they refer to the latter as performing substantial simplifications and, therefore, unable to capture a lot of non-linear phenomena. Nevertheless, the authors of this work too adopt an identical model and initialization strategy as in the above-cited work.
1.1 **Most importantly** in Assumption 3.1, are the authors comprising a dataset using **just two data points**? If so, then that is an exceptionally restrictive assumption.
1.2 The reference to the usage of data averaged direction over the Gram-Schmidt type orthogonal direction, between that work and this, as defining a key improvement, is again not entirely justified if 1.1 above is true. In the case of a set of *non-degenerate* data vectors, defining *useful* orthogonal directions is challenging

**Questions:**

- Can you detail how In Fig. 1, the projection to a 2D subspace span was performed?
- If the specific dot product of Line 151 has to be true, then data should be normalized, yet that is not mentioned in Assumption 3.1
- Can you explain the reason behind the assumption on $T_1$ in Line 191?
- Can you at least intuitively explain why the positive neuron has initial condensation/alignment with the label averaged direction $\mu$ and *then* transition to the expected $x_{-}^{\perp}$ direction? On a similar note, is there a reason only the negative neurons eventually *converge toward some specific directions dependent on both data and initialization*? Is it due to $p>1$? If so, I would assume the alignment signal impacts the positive labels over the negative ones, though.

**Limitations:**

See Questions and Weaknesses

---

> ### Author Rebuttal · Authors · 2023-08-08
>
> We thank the reviewer for appreciating our work, as well as the valuable comments. We address the questions in the following.
>
> **Response to Weakness 1.**
> - **Main advantage over [1].**
>   - We acknowledge the similarity in our network and initialization strategy as [1]. However, our analysis takes a step further by delving into more intricate exp-type loss and non-orthogonal data. These aspects empower us to unravel richer nonlinear training behaviors.
>   - While our initial alignment directions ($\mu,x_+^\perp$) differ from [1], we don't consider this point as our main advantage. Initial condensation is broadly observed in reality, manifesting in various directions [2]. As the reviewer mentioned, determining condensation directions for realistic settings, such as useful orthogonal ones, is an important topic for future work.
>   - Our main advantage over [1] is that we capture richer nonlinear behaviors, such as Neuron Reactivation, Staged feature learning, and Initialization-dependent directional convergence. Please refer to our Section 5 for more details.
> - **About Assumption 3.1.**
>   - We acknowledge the importance of this assumption in our theory. However, a complete analysis under such a relatively strong assumption is also of great interest to understanding the neural networks’ training dynamics and nonlinear behaviors.
>   - Our motivation is to illustrate that even in a simple setting, GF could exhibit *numerous nonlinear behaviors* that researchers have speculated about, such as saddle-to-plateau, simplifying-to-complicating, etc.
>   - We will work to relax this assumption conditionally. Inspired by reviewers' comments, we have explored a slight relaxation of this assumption by perturbing $x_+$ and $x_-$ with noise, and our experimental results illustrate that similar four-phase dynamics and nonlinear behaviors still exist. For details, Please refer to our **``Global'' Response to All Reviewers**. For a more in-depth study, we leave it to future work.
>
> **Q1. The projection in Fig 1.**
>
> **Response.** WLOG, we can let $x_+=(\sin(\Delta/2),-\cos(\Delta/2),0_{d-2}^T)^T, x_-=(-\sin(\Delta/2),-\cos(\Delta/2),0_{d-2}^T)^T$. In this case, the subspace D=span{$x_+,x_-$} is span{$e_1,e_2$}. For k-th neuron at time t, denoted as $b=(b_1,\cdots,b_d)^T$, its projection onto D is $(b_1,b_2,0_{d-2}^T)^T$. Hence, its polar coordinates is $(\rho,\theta)=(\sqrt{b_1^2+b_2^2},\arctan(b_2/b_1))$.
>
> **Q2. Data normalization.**
>
> **Response.** Yes, the data is normalized. In Ass 3.1, the requirement $x_+,x_-\in S^{d-1}$ (Line 136) is actually the $l_2$ normalization.
>
> **Q3. Explain $T_I$.**
>
> **Response.** First, we clarify that $T_I=10\sqrt{\kappa_1/\kappa_2}$ is a result derived in our proof rather than an assumption. By our analysis, at $T_I$, initial condensation can be perfectly completed. Second, we can offer an intuitive explanation for the positive correlation between $T_I$ and $\kappa_1$, and the negative correlation between $T_I$ and $\kappa_2$. For small initialization, during initial training, neurons' radial growth is much slower than their tangential speed, leading to directional condensation. For tangent velocity $\|dw_k(t)/dt\|$ (equ(7)), notice that (i) $\kappa_2/\sqrt{m}\rho _k(0)=\kappa_2/\kappa_1$; (ii) during transient initial training, $F_k(t)$ mostly stays constant, independent of $\kappa_1$ and $\kappa_2$. Combining these facts, we see that smaller $\kappa_1$ and larger $\kappa_2$ correlate with greater tangential speed, leading to smaller $T_I$.
>
> **Q4. Intuitive explanations for two dynamics.**
>
> **Response.** We agree with the reviewer that it is important to provide intuitive explanations for these dynamics. We will add this aspect in our revised version.
> - **Some positive neurons (PN) moves from $\mu$ to $x_-^\perp$.** We will provide an intuitive understanding. For rigorous proof,please refer to Lemma C.3 ~ Lemma C.10. For PN s.t. $w_k(0)^Tx_+>0,w_k(0)^Tx_->0$, it holds:
>     - During Phase I, tangential velocity is much larger than radial velocity, and vector field $F_k(t)\approx\mu$. Thus PN rapidly align well with $\mu$. Concurrently, some negative neurons (NN) align with $x_+^\perp$. Additionally, since living PN are much closer to $x_-$ than living NN at $T_I$, the prediction of $x_-$ is incorrect ($f_-(T_I)>0$).
>     - After Phase I, to correct the prediction of $x_-$, living PN gradually move away from $x_-$ and decrease positive $w_k^Tx_-$. Notably, living PN can’t move into {$w:w^Tx_-<0$} because the vector field abruptly changes at boundary {$w:w^Tx_-=0$} (due to ReLU’s non-smoothness), redirecting PN to {$w:w^Tx_->0$}. Thus, living PN eventually satisfy $w_k^Tx_-=0$. Notably, vector field $F_k(t)$ lies in the subspace span{$x_+,x_-$}, causing living PN to reach $x_-^\perp$.
>
> - **Only living negative neurons converge toward directions dependent on initialization.** We will intuitively explain this via "simplicity bias". For rigorous proof, please refer to our proof of Theorem 4.10.
>     - In Remark 4.2, we demonstrated that after Phase I, the number of living positive neurons (LPNs) exceeds living negative neurons (LNNs).
>     - At the end of Phase III, the network has the simplest patterns: LPNs exclusively predict  $x_+$, while LNNs solely predict $x_-$. However, prediction $f_-(t)$ evolutes much slower than $f_+(t)$, disliked by GF. Then GF aims to rectify this speed imbalance through a simple approach -- by adjusting directions of fewer LNNs while preserving LPNs' directions.  Precisely,  for LNNs, GF increases $w_k^Tx_-$ and reactivates on $x_+$.
>     - Hence,  as shown in Thm 4.10, different ratios $\alpha$ between the numbers of LPNs and LNNs can yield different convergent directions of LNNs, without affecting LPNs.
>
> [1] Boursier et al.  Gradient flow dynamics of shallow relu networks for square loss and orthogonal inputs. (NeurIPS 2022)
>
> [2] Zhou et al. Towards Understanding the Condensation of Neural Networks at Initial Training. (NeurIPS 2022)

---

### Author Rebuttal · Authors · 2023-08-08

**``Global'' Response to All Reviewers.**

1. First, we sincerely thank all the reviewers for appreciating our result, i.e., a theoretical analysis of multi-phase optimization dynamics and the rich nonlinear behaviors of ReLU networks. We also thank all the reviewers for their comments and suggestions to improve our paper. In our revised version, we will correct all typos, provide complete experimental settings and results, and incorporate the discussions with the reviewers.

2. **More Experiments.** In response to multiple reviewers' suggestions regarding the need for further numerical validation of our theory, we have included two additional experiments to showcase the effectiveness of our theoretical findings. Below, we summarize the experimental setups, results, and conclusions. As for detailed experimental results, please refer to our updated **one-page PDF**.

    - **Experiment 1: More hyperparameter settings.**
      - **Setup.** We run experiments under more hyperparameter settings (different $\Delta$, $p$, $\kappa_1$, $\kappa_2$). We aim to verify whether the same four training phases persistently exist and evaluate the consistency between our theoretical bounds ($T_{\rm I}$, $T_{\rm plat}$, $T_{\rm II}$, $T_{\rm III}$) and empirical outcomes.

      - **Result.** Due to space constraints, a subset of the results is presented in **Table 2 in one-page PDF**. Precisely, the table displays outcomes under different $\Delta$ and $p$ (data-dependent hyperparameters), and we focus on the changes of $T_{\rm plat}$ (time plateau of training accuracy) and $T_{\rm III}$ (ending time of Phase III).

      - **Conclusion.** From our numerical results in Table 1 in one-page PDF, we have two main conclusions: (i) four training phases in our theory persistently exist; (ii) our theoretical estimates on $T_{\rm plat}$ and $T_{\rm III}$ demonstrate noteworthy alignment with realistic results (in terms of $p$ and $\Delta$).

        In the revised version, we will present the complete results.

    - **Experiment 2: Noisy Data.**

      - **Setup.** We conduct numerical experiments on the setting of adding small stochastic noise on top of $x_+$ and $x_-$, a little bit more realistic setting. Specifically, in span{$x_+,x_-$}, we perturb the angles of data $x_+$ and $x_-$ by noise $\xi\sim{\rm Unif}([0,\Delta/4])$.

      - **Result.** In **Figure 5 in one-page PDF**, we visualize (i) the evolution of each neuron throughout the training process; (ii) some key data directions; (iii) the evolution of training accuracy. To compare these results with noiseless data, the reviewers can refer to our previous Figures 3 and 4 in Appendix A.

      - **Conclusion.** From our numerical results in Figure 1 in the one-page PDF, we have two main conclusions: (1) we ascertain that the same four-phase optimization dynamics and nonlinear behaviors persist, even for our dataset with small stochastic noise; (2) a slight difference is that there is more than one plateau of training accuracy in Phase II. The reason is that for noisy data, GF needs to learn negative data $(y=-1)$ one by one in Phase II. For example, three distinct negative data are employed in this experiment, so three plateaus of training accuracy emerge (12/15, 13/15, and 14/15).

3. We have addressed every concern raised by each reviewer through separate responses provided below.

---

### Decision · Program_Chairs · 2023-09-21

**Decision:**

Accept (spotlight)

**Comment:**

This paper explores the dynamics of learning in ReLu networks. Their theoretical analyses find four stages of learning, expanding on previous efforts that focused on the beginning or end of learning, or on approximations of learning such as linear networks. The contributions of this work are primarily theoretical, although numerical verification of the theory is provided.

The reviewers generally appreciated the work. The main drawbacks were the assumptions chosen to enable the more thorough derivation, including using Gradient Flow over the more generally used Gradient descent, the focus on linearly separable data, and the focus on ReLu networks. Nonetheless the reviewers agreed that the depth of the result compensates for the more narrow assumptions and is a worthy contribution to the NeurIPS conference. I therefore recommend this work be accepted.

Some side commentary, the analysis is reminiscent of the information bottleneck principles in deep learning explored by the late Tali Tishby and others. it would be quite interesting to understand the relationship between the four stages found by the authors and the two stages identified by that analysis.